# 🏆 BEYONDBENCH: CONTAMINATION-RESISTANT EVALUATION OF REASONING IN LANGUAGE MODELS

**Gaurav Srivastava**[♡],[*] **Aafiya Hussain**[♡]**, Zhenyu Bi**[♡]**, Swastik Roy**[♠]**, Priya Pitre**[♡]**,**
**Meng Lu**[♡]**, Morteza Ziyadi**[♠]**, Xuan Wang**[♡],[*]

[♡]Computer Science, Virginia Tech, USA  [♠]Amazon AGI, USA

[♡]`{gks, aafiyahussain, zhenyub, priyapitre, menglu, xuanw}@vt.edu`
[♠]`{roswasti, mziyadi}@amazon.com`

🌐 **Leaderboard:** `ctrl-gaurav.github.io/BeyondBench`
🯄 **GitHub:** `ctrl-gaurav/BeyondBench` 🐍 **PyPI:** `pypi.org/project/beyondbench`

## ABSTRACT

Evaluating language models fairly is becoming harder as static benchmarks risk contamination by training data, making it unclear whether models are truly reasoning or just recalling answers. We introduce **BEYONDBENCH**[1], an evaluation framework that avoids this problem by using **algorithmic problem generation**. Unlike traditional benchmarks that risk contamination from internet-scale training data, BEYONDBENCH creates mathematically grounded problems on the fly, ensuring each test remains fresh and uncontaminated. Our framework covers **44 algorithmic tasks** with a total of **117 variations**, grouped into three difficulty levels: the *Easy Suite* (29 tasks) for basic arithmetic and statistics, the *Medium Suite* (5 tasks, 49 variations) for sequence patterns and reasoning, and the *Hard Suite* (10 tasks, 68 variations) tackling NP-complete and constraint satisfaction problems. Each task generates problems from a combinatorial space larger than $10^{15}$ unique instances, with solutions verified deterministically by mathematical proofs. We evaluated **101 language models**, including 85 open-source and 16 closed-source models, spanning sizes from 0.5B to 141B parameters and multiple quantization schemes. All evaluations use three-fold evaluation to ensure statistical robustness. Our results show consistent reasoning deficiencies across model families, with performance degrading sharply as problem complexity increases from polynomial to exponential. In our Hard Suite evaluations, models such as Gemini-2.5-pro, Llama-3.3-70B, and Qwen2.5-72B achieved average accuracies of **56.21%, 27.16%, and 33.37%,** respectively. Moreover, we observe that performance drops drastically without tool usage, with GPT-5, GPT-5-mini, and GPT-5-nano showing a **decline** of **16.81%, 15.86%, and 43.95%** in overall accuracy without tool access. The contamination resistance of BEYONDBENCH rests on three guarantees: (i) the problem space is vastly larger than any static dataset, (ii) every instance has a deterministically verifiable solution (unique or fully enumerated), and (iii) isomorphic transformations generate semantically equivalent but syntactically new problems. BEYONDBENCH redefines reasoning evaluation through genuine algorithmic problem-solving, ensuring fair and meaningful evaluation.

## 1 INTRODUCTION

Modern large language models (LLMs) exhibit impressive performance on existing static reasoning benchmarks such as GSM8K (Cobbe et al., 2021), MATH (Hendrycks et al., 2021b), and Olympiad-Bench (He et al., 2024), yet these evaluations can be misleading: benchmark data may already

---

[*]Correspondence: `gks@vt.edu`, `xuanw@vt.edu`

[1]Our open-source Python package is available at `https://pypi.org/project/beyondbench/` and the code at `https://github.com/ctrl-gaurav/BeyondBench` for easy and reproducible evaluation.

appear in pre-training (Wu et al., 2025). Recent empirical studies demonstrate widespread data contamination across standard benchmarks (Xu et al., 2024; Cheng et al., 2025; Chen et al., 2025a; Golchin & Surdeanu, 2025; Choi et al., 2025; Deng et al., 2024), where evaluation examples appear in training corpora through web crawls. This contamination systematically inflates performance metrics: models memorize specific solutions rather than learning generalizable reasoning patterns (Shojaee et al., 2025; Opus & Lawsen, 2025). As training corpora grow to web-scale, the chance that at least one evaluation example appears in the training data becomes near certain, even under simple uniform sampling assumptions (we provide the full derivation and assumptions in Appendix S). Empirical studies (Shojaee et al., 2025; Varela et al., 2025) support this as well: performance drops on decontaminated variants of existing benchmarks, suggesting that static benchmarks cannot reliably measure true reasoning ability or guarantee generalization.

Recent studies have attempted to mitigate this problem. Dynamic benchmarks such as DyVal (Zhu et al., 2024a;b) and ThinkBench (Huang et al., 2025) adapt task distribution over time, but provide no mathematical guarantees that each generated problem instance is well-posed (i.e., has a unique or fully enumerated solution). As a result, correctness labels can be ambiguous, and evaluation pipelines must rely on heuristic matching. Algorithmic reasoning benchmarks like CLRS encode structured problems but are static in nature (Veličković et al., 2022), therefore vulnerable to contamination, and do not account for LLM-specific constraints such as maximum tokens it can output. Benchmarks such as MathArena (Balunović et al., 2025) attempt to mitigate this issue by using fresh olympiad questions but contamination is still likely. On the other hand, Game-based evaluations such as GameArena (Hu et al., 2025) provide interactivity, but are limited to narrow domains and lack scalable parametric difficulty. Across these settings, four gaps remain: *(i) mathematically verifiable unique solutions*, *(ii) contamination-resistance*, *(iii) token budget-aware evaluation*, and *(iv) acceptance of multiple correct answers*. See Table 1 for a comparison of previous benchmarks and our BEYONDBENCH.

To address the above limitations, we introduce **BEYONDBENCH**, an algorithmic evaluation framework for reasoning LLMs that generates an unbounded number of novel problems from basic tasks. For each task, we define a generator that takes configurable parameters (e.g., numeric ranges, constraint sizes) and produces problem instances from a combinatorial space exceeding $10^{15}$ unique instances per task. This space is vastly larger than any practical training corpus, making contamination provably negligible (see Section 3.1 and Appendix S for the formal analysis). Moreover, for each generated problem we use Boolean satisfiability (SAT) and constraint satisfaction problem (CSP) solvers (Walsh, 2000) to verify that either: (i) the problem admits a unique solution or (ii) all solutions can be enumerated. This ensures that every problem is well-posed *(has at least one solution)* and that our scoring is precise *(we know the correct answer)*. If multiple solutions exist, all of them are considered correct and we match against each of them, so models cannot gain credit for spurious answers.

Furthermore, we partition each task into three difficulty levels (Easy, Medium, Hard), creating a curriculum that increases in complexity. The difficulty of each task is controlled by scaling its parameters (e.g., increasing the size of a combinatorial structure), so we can start with simpler subproblems and work up to provably NP-hard instances to stress-test models (Hidalgo-Herrero et al., 2013). Finally, we take token budget into account: if the minimal solution length of a problem exceeds the model's maximum output token window, that problem is excluded from evaluation. After model inference, BEYONDBENCH also checks that the response token length stays within a calibrated bound for the problem size (to catch cases where models "overthink" (Chen et al., 2025c; Cuadron et al., 2025) trivial instances).

To summarize, we make the following contributions: *(1)* We propose BEYONDBENCH, an algorithmic generator of reasoning problems that automatically verifies solution uniqueness or generates complete solution sets. *(2)* We designed a parameterized difficulty curriculum and a token-aware evaluation protocol: tasks **scale** from *elementary subproblems to provably NP-hard instances*, and evaluations respect model-specific *input/output token budgets*; *(3)* We conduct a large-scale systematic empirical study across **101 models** to study reasoning gaps in modern LLMs/small language models (SLMs) (Srivastava et al., 2025b; Bi et al., 2025)/large reasoning models (LRMs).

## 2 RELATED WORK

**Static Benchmarks and Their Limitations.** A large body of benchmarks for reasoning in LLMs has been built upon fixed datasets such as GSM8K (Cobbe et al., 2021), MATH (Hendrycks et al., 2021b),

and MMLU (Hendrycks et al., 2021a). (Xu et al., 2024) and (Cheng et al., 2025) document how widely used leaderboards are compromised by pretraining overlap. (Shojaee et al., 2025) shows that even LRMs collapse when faced with simple perturbations of known tasks. However, in response to that (Opus & Lawsen, 2025) highlights that many supposed reasoning failures are actually attributable to context window limitations, ambiguous specification, or tasks whose solution paths exceed the model's output length. They also argue that "emergent reasoning" often reflects benchmark biases rather than genuine generalization. Although efforts like MathArena (Balunović et al., 2025) use unseen contest problems to mitigate leakage, they remain limited to static and exhaustible pools. Static evaluation also fails to capture the algorithmic hardness of problems. For instance, (Katz et al., 2025) demonstrates that seemingly simple planning tasks, such as the Countdown game, become computationally intractable, revealing how shallow benchmarks underestimate the true difficulty of reasoning (Wu et al., 2024; Yu et al., 2025). These limitations highlight the necessity of dynamic, mathematically constrained evaluation frameworks.

| Feature | CLRS | DyVal | TreeEval | MathArena | GameArena | DARG | NPHard | FCoRe | PUZZLE PLEX | BEYONDBENCH (Ours) |
|---|---|---|---|---|---|---|---|---|---|---|
| Dynamic generation | ✗ | ✓ | ✓ | ✗ | ✓ | ✓ | ✓ | ✓ | ✗ | ✓ |
| Unique solution check | ✗ | ✗ | ✗ | ✓ | ✓ | ✗ | ✗ | Partial | ✗ | ✓ |
| Multi-solution allowed | ✗ | ✗ | ✗ | ✗ | ✗ | ✗ | ✗ | ✗ | ✗ | ✓ |
| Scalable difficulty | ✓ | ✓ | ✓ | ✗ | ✗ | ✓ | ✓ | Limited | Limited | ✓ |
| Contamination-free | ✗ | ✓ | ✓ | ✓ | ✓ | ✓ | ✓ | ✓ | ✓ | ✓ |
| Token-aware eval. | ✗ | ✗ | ✗ | ✗ | ✗ | ✗ | ✗ | ✗ | ✗ | ✓ |
| Number of Tasks | 30 | 7 | – | 5 | 3 | 4 | 9 | 40 | 15 | **44** |

Table 1: **Comparison with existing reasoning benchmarks. ✓: feature present; ✗: absent. Note:** *Unique solution* means the benchmark verifies only one valid solution exists. *Multi-solution allowed* means multiple valid solutions are accepted and matched against the full enumerated set.

**Dynamic Evaluation & Algorithmic Reasoning Benchmarks.** Recent works explore dynamic benchmarking as an alternative to static datasets. DyVal (Zhu et al., 2024a), later extended with meta-probing in DyVal2 (Zhu et al., 2024b), generate math and logic problems dynamically. ThinkBench (Huang et al., 2025) evaluates models with distributionally shifted reasoning tasks, while TreeEval (Li et al., 2025) and GameArena (Hu et al., 2025) introduce interactive or live-game evaluation protocols. DyCodeEval (Chen et al., 2025b) dynamically generates code reasoning tasks to avoid contamination, and DARG (Zhang et al., 2024), using an adaptive reasoning graph framework, dynamically escalates task complexity. Similarly, (Karia et al., 2025) proposes autonomous evaluation for truth maintenance in reasoning pipelines. Other recent work includes NPHardEval (Fan et al., 2024) with complexity-class aware benchmarking, FCoReBench (Mittal et al., 2024) for first-order combinatorial reasoning with solver integration, PUZZLEPLEX (Long et al., 2025) for interactive planning puzzles. These efforts highlight the growing recognition of dynamic evaluation but remain limited by either a lack of mathematical guarantees, insufficient control over solution uniqueness, or the absence of token-aware evaluation protocols. Table 1 provides a detailed comparison of BEYONDBENCH with other evaluation benchmarks, highlighting how BEYONDBENCH differs by combining dynamic evaluation with formal solution verification. Furthermore, Algorithmic reasoning benchmarks such as CLRS (Veličković et al., 2022) and LLMThinkBench (Srivastava et al., 2025c) provide structured tests for algorithmic tasks and overthinking tendencies. However, they lack resilience to contamination and are narrow in scope. BEYONDBENCH advances this line of work by unifying algorithmic, combinatorial, and mathematical reasoning tasks into dynamically generated, difficulty-calibrated suites with provable correctness.

We also discuss tool-augmented reasoning methods (PAL (Gao et al., 2023), Logic-LM (Pan et al., 2023), Reasoning Gym (Stojanovski et al., 2025), and others) in Appendix C. Our work evaluates both innate and tool-augmented capabilities, showing that even with code execution and symbolic solvers, our benchmark remains challenging (Section 4.5, Appendix Q).

## 3 BEYONDBENCH FRAMEWORK

In this section, we present the design and implementation of BEYONDBENCH. We detail the algorithmic foundations for contamination-resistant problem generation, introduce a token budget-aware evaluation system, and establish formal verification methods for solution correctness. BEYONDBENCH introduces three central components: 1) dynamic problem generation with provable uniqueness guarantees, 2) adaptive difficulty scaling based on model token budget constraints, and 3) multi-solution handling through formal verification techniques.

## 3.1 Algorithmic Problem Generation and Mathematical Foundation

We develop BEYONDBENCH through algorithmic problem generation with formal mathematical guarantees. Let $\mathcal{T}$ denote the set of all task categories in our benchmark. For each task category $\tau \in \mathcal{T}$, we define a generator function $G_\tau : \Theta_\tau \times \mathcal{R} \to \mathcal{P}_\tau$ that maps a parameter space $\Theta_\tau$ (e.g., list lengths, constraint sizes) and random seed space $\mathcal{R}$ to problem instances $\mathcal{P}_\tau$. The parameter space cardinality satisfies $|\Theta_\tau \times \mathcal{R}| > 10^{15}$ for all tasks, ensuring that $\Pr[G_\tau(\theta_1, r_1) = G_\tau(\theta_2, r_2)] \leq 10^{-15}$ for distinct parameter-seed pairs $(\theta_1, r_1) \neq (\theta_2, r_2)$ (Mitzenmacher & Upfal, 2017). This large problem space makes contamination provably negligible since the probability of an exact instance collision with any training corpus $\mathcal{C}$ of practical size $|\mathcal{C}| < 10^{12}$ remains below $10^{-3}$ (for the complete proof see Appendix S). Our contamination resistance addresses exact instance memorization. Learning general solution strategies from similar problems constitutes genuine algorithmic learning, which our benchmark is designed to measure. For detailed proofs on the easy, medium, and hard suites see Appendix E, G, and J.

Each generated problem $p \in \mathcal{P}_\tau$ undergoes validation through a verification function $V_\tau : \mathcal{S}_\tau \times \mathcal{P}_\tau \to \{0, 1\}$, where $\mathcal{S}_\tau$ is the solution space for task $\tau$, that deterministically confirms solution correctness. For deterministic tasks like arithmetic and sorting, we verify through direct computation in $O(n)$ time, where $n$ is the problem size (e.g., list length). For constraint satisfaction problems like Sudoku and N-Queens, we use CSP solvers (python-constraint, pysat) to enumerate all valid solutions. For problems admitting unique solutions, we use CSP solvers (Dechter, 2003) to verify $|\{s \in \mathcal{S}_\tau : V_\tau(s, p) = 1\}| = 1$ (usage of CSP detailed in Appendix J). When multiple valid solutions exist naturally (as in N-Queens or mode calculation), we compute the complete solution set $\mathcal{S}_p = \{s \in \mathcal{S}_\tau : V_\tau(s, p) = 1\}$ through exhaustive enumeration and accept any $s \in \mathcal{S}_p$ as correct. This prevents unfair penalization when models produce mathematically valid but non-canonical answers. The verification complexity remains polynomial $O(n^k)$ for constant $k \leq 3$ despite potentially exponential solution complexity, enabling efficient correctness checking.

**Concrete Parameter Space Examples.** To illustrate our generation process, consider Tower of Hanoi with parameters: $n \in [3, 12]$ disks, peg labels from 26 characters, and random initial configurations, yielding $> 10^{18}$ unique instances. For Sudoku, we vary grid density (30-50 empty cells), apply digit permutations, and shuffle cell positions, producing $> 10^{20}$ instances. Arithmetic tasks sample list length $n \in [8, 64]$ and values from $[-1000, 1000]$, creating $> 10^{15}$ combinations. N-Queens uses board sizes $n \in [4, 12]$ with optional initial constraints, while Boolean SAT varies variables (3-8) and clauses (5-20), each exceeding $10^{12}$ unique problem instances.

| Suite | Tasks | Variations | Problem Space | Complexity Class | Example Tasks |
|---|---|---|---|---|---|
| **Easy** | 29 | – | $> 10^{15}$ per task | $O(n^k), k \leq 2$ | Arithmetic (8), Statistics (3), Counting (8), Extrema (5), Comparison (1), Others (4) |
| **Medium** | 5 | 49 | $> 10^{20}$ per task | $O(2^n)$ to $O(n!)$ | Fibonacci/Recursive (6), Geometric/Exponential (10), Prime/Number Theory (11), Complex Patterns (12), Algebraic Sequences (10) |
| **Hard** | 10 | 68 | $> 10^{30}$ per task | NP-complete | Tower of Hanoi (6), N-Queens (4), Graph Coloring (10), Boolean SAT (5), Sudoku (8), Logic Grid Puzzles (8), Cryptarithmetic (12), Matrix Chain (5), Modular Systems (5), Constraint Optimization (5) |
| **Total** | 44 | 117$^\dagger$ | $> 10^{15}$-$10^{50}$ | P to NP-complete | 29 base + 117 sub-variations |

Table 2: BEYONDBENCH Task Organization and Scale

## 3.2 Token-Aware Evaluation Framework

For each model with token budget $C$ tokens, we dynamically calibrate problem complexity to guarantee solvability within architectural limits. Let $T_p(n)$ denote the expected token requirement for problem $p$ with size parameter $n$ (e.g., list length, number of disks), decomposed as $T_p(n) = T_{\text{prompt}}(n) + T_{\text{solution}}(n) + T_{\text{buffer}}$ where $T_{\text{buffer}} = 0.15 \cdot C$ accommodates reasoning verbosity (Han et al., 2025; Lin et al., 2025). Prompt design is detailed in Appendix F (easy suite), I (medium suite), L (hard suite).

We derive task-specific token estimation functions through empirical analysis. For arithmetic operations on lists of length $n$ with maximum element value $v_{\max}$, we have $T_{\text{arithmetic}}(n, v_{\max}) = \alpha_1 n + \beta_1 \log_{10} v_{\max} + \gamma_1$ where coefficients $(\alpha_1, \beta_1, \gamma_1)$ are calibrated across model families. For Tower of Hanoi with $n$ disks requiring $2^n - 1$ moves, the solution tokens scale as $T_{\text{hanoi}}(n) = (2^n - 1) \cdot \alpha_{\text{move}}$ where $\alpha_{\text{move}} \approx 12$ tokens per move description. This enables precise difficulty scaling: for example,

for models with a 32K maximum output token limit we evaluate up to $n = 8$ disks (requiring approximately $(2^8 - 1) \cdot 12 = 3060$ tokens **excluding** thinking tokens) as a safe limit, while future 128K models could handle $n = 10$ (requiring $(2^{10} - 1) \cdot 12 = 12276$ tokens), maintaining evaluation relevance as capabilities improve (Wen et al., 2025).

The framework implements a dual-phase token validation protocol. During generation, we enforce the constraint $T_p(n) \leq 0.85 \cdot C$ through adaptive parameter scaling. If initial parameters yield excessive token requirements, we apply the reduction $n' = \lfloor 0.8 \cdot n \rfloor$ iteratively until the constraint is satisfied. Post-inference, we perform actual token counting using model-specific tokenizers and classify each response $r$ as: $\text{TokenStatus}(r) = \begin{cases} \text{VALID} & \text{if } |r| \leq 0.85 \cdot C \\ \text{WARNING} & \text{if } 0.85 \cdot C < |r| \leq C \\ \text{OVERFLOW} & \text{if } |r| > C \end{cases}$ where $|r|$ is the token count of response $r$, and responses with WARNING status indicate potential truncation or excessive reasoning that may compromise solution quality (Wang et al., 2024).

### 3.3 SOLUTION UNIQUENESS VERIFICATION AND MULTI-SOLUTION HANDLING

We use formal verification methods to guarantee that every generated problem is mathematically well-posed. For each problem instance $p$, we compute the solution set cardinality through systematic constraint propagation and backtracking algorithms (Bessiere, 2006). When $|\mathcal{S}_p| = 1$, the problem admits a unique solution and standard evaluation proceeds. When $|\mathcal{S}_p| > 1$, we enumerate all valid solutions and implement the acceptance criterion $\text{Accept}(r, p) = \mathbf{1}[\text{Parse}(r) \in \mathcal{S}_p]$ where Parse extracts the model's answer using task-specific robust parsers.

For constraint satisfaction problems like Sudoku or Logic Grid Puzzles, we employ arc consistency algorithms that iteratively reduce variable domains until either a unique solution emerges or multiple solutions are identified (Simonis, 2005; Howell et al., 2018). The domain reduction follows $D_v^{t+1} = D_v^t \cap \{d \in D : \text{consistent}(v, d, \mathcal{K}, D^t)\}$ where consistency checking ensures no constraint violations (here $v$ denotes a variable in the variable set $V$ with domain $D_v$, and $\mathcal{K}$ denotes the set of problem constraints for the instance). For optimization problems like Matrix Chain Multiplication, we verify optimality through dynamic programming, confirming that the claimed cost equals $m[1, k] = \min_{\pi \in \Pi} \text{cost}(\pi)$ where $k$ is the number of matrices and $\Pi$ denotes the set of all valid parenthesizations for the matrix-chain instance (Carbonnel et al., 2019; Cooper & Schiex, 2001; Cooper & Živný, 2017). The complete end-to-end BEYOND-BENCH evaluation pipeline is shown in Algorithm 1. This pipeline ensures mathematical soundness through three mechanisms: 1) parameter selection respects token budget constraints preventing unfair penalization, 2) solution set computation guarantees we evaluate against all valid answers not just canonical ones, and 3) post-inference token counting detects when models approach architectural limits that may compromise reasoning quality.

---

**Algorithm 1** BEYONDBENCH end-to-end Evaluation Pipeline

1: **Input:** Model $\mathcal{M}$, Task suite $\mathcal{T}$, Token limit $C$
2: **Output:** Performance metrics $\mathcal{E}$
3: **for** each task $\tau \in \mathcal{T}$ **do**
4:     $\theta \leftarrow \text{SelectParameters}(\tau, C)$
5:     $p \leftarrow G_\tau(\theta, \text{RandomSeed}())$
6:     $\mathcal{S}_p \leftarrow \text{ComputeSolutionSet}(p)$
7:     **if** $|\mathcal{S}_p| = 0$ **then**
8:         **continue**
9:     **end if**
10:     prompt $\leftarrow \text{FormatPrompt}(p, \tau)$
11:     $r \leftarrow \mathcal{M}(\text{prompt})$
12:     tokens $\leftarrow \text{CountTokens}(r, \mathcal{M})$
13:     **if** tokens $> 0.95 \cdot C$ **then**
14:         $\mathcal{E}[\tau].\text{warnings++}$
15:     **end if**
16:     $s \leftarrow \text{Parse}(r, \tau)$
17:     $\mathcal{E}[\tau].\text{correct} \leftarrow \mathcal{E}[\tau].\text{correct} + \mathbf{1}[s \in \mathcal{S}_p]$
18: **end for**

---

### 3.4 DIFFICULTY SCALING AND THEORETICAL GUARANTEES

BEYONDBENCH partitions tasks into three complexity regimes with provable difficulty scaling. The Easy Suite encompasses polynomial-time solvable problems where verification complexity is $O(n^k)$ for $k \leq 2$. These include arithmetic operations, statistical measures, and comparison tasks with solution spaces of size $O(n!)$ but efficient verification through direct computation. The Medium Suite introduces problems with exponential growth patterns such as Fibonacci sequences where $F_n \sim \phi^n$ with $\phi = (1 + \sqrt{5})/2$ (the golden ratio), geometric progressions with common ratio $q$ growing as

$q^n$, and number-theoretic sequences requiring primality testing with complexity $O(\sqrt{n})$ per element. Here $n$ refers to the problem size (e.g., sequence length, number of variables). The Hard Suite contains NP-complete problems (Fan et al., 2024; Leyton-Brown et al., 2014; Kendall et al., 2008) including Boolean Satisfiability where the search space contains $2^n$ assignments, Graph Coloring with chromatic number computation, and N-Queens requiring exploration of $n!$ permutations (Yang et al., 2025b). For further details regarding extensibility to new tasks and increasing problem complexity refer to Appendix T.

For each difficulty level, we maintain the invariant that problem generation time remains polynomial while solution complexity may be exponential, ensuring efficient benchmark execution while testing genuine reasoning. The framework provides theoretical guarantees including contamination resistance where $P(\text{collision}) < |\mathcal{C}|/|\mathcal{P}| < 10^{-3}$ for any feasible training corpus $\mathcal{C}$ (here $P(\text{collision})$ is the probability a generated instance matches an entry of $\mathcal{C}$), solution correctness through formal verification with zero false positives or negatives, and scalability where problem difficulty increases monotonically with parameter growth enabling continuous evaluation as models improve. Our algorithmic task selection is motivated by the observation that true reasoning requires capabilities fundamentally challenging for current LLMs: memory management across exponential state spaces, systematic backtracking, and multi-step deduction. Tasks with known algorithms test whether models can *execute* procedures correctly, a prerequisite for genuine reasoning. Detailed motivation and empirical evidence are provided in Appendix D.

## 4 RESULTS AND INSIGHTS

### 4.1 EVALUATION SETUP

We evaluate each model on 1,000 randomly generated problem instances per task variation to ensure statistical robustness, using seed 42 across all experiments for reproducibility. For proprietary models, we reduce to 100 instances per task. All models use the same parameters: temperature 0.1, top-p 0.9, and maximum tokens allowed for that model. We evaluate 101 language models across 44 algorithmic tasks with 117 variations using our BEYONDBENCH framework. Due to space constraints, detailed experimental settings are provided in the Appendix A and B. Comprehensive statistical analysis including confidence intervals, effect sizes (Cohen's d), ANOVA, and non-parametric tests (Mann-Whitney U, Wilcoxon signed-rank) is provided in Appendix N. Table 3 presents the complete results showing *accuracy, instruction-following rates* (we prompt each model to respond in specific output formats; if the response matches **any required format**, we assign instruction-following = 1, otherwise = 0), and *average token usage* across all suites.

### 4.2 MAIN RESULTS

**Our experiments reveal substantial limitations in algorithmic reasoning, as problem complexity grows.** The strongest commercial models like GPT-5 (Georgiou, 2025) achieve 71.68% accuracy on hard suite, while most open-source models struggle to exceed 60% performance. The performance drops dramatically with complexity: the best open-source model (GPT-OSS-120B (OpenAI et al., 2025)) achieves 93.27% on Easy tasks but only 59.41% on Hard tasks, suggesting fundamental struggles with recursive algorithmic reasoning.

SYSTEMATIC PERFORMANCE COLLAPSE BEYOND COMPLEXITY THRESHOLDS. **Models exhibit sharp performance cliffs rather than gradual degradation when algorithmic complexity exceeds critical thresholds.** Figure 1 demonstrates this phenomenon across nine hardest algorithmic tasks, showing how models maintain reasonable performance until hitting complexity barriers, then collapse catastrophically. For instance, models achieve 80-90% accuracy on 4×4 Sudoku but drop to <10% on 9×9 grids. Similarly, most open-source models perform well on 3-5 disk Tower of Hanoi but fail completely at 6+ disks. This cliff-like degradation pattern indicates that *current models lack the systematic state management and backtracking mechanisms essential for complex algorithmic reasoning*. To ensure fair comparison across tasks with different complexity scaling (Tower of Hanoi's exponential $2^n$ vs Sudoku's linear empty cells), we normalize complexity metrics. After normalization, we observe graceful degradation tasks (arithmetic, sorting) exhibit linear accuracy decline, while sudden degradation tasks (Sudoku, N-Queens) show threshold behavior at approximately $0.7 \times \log_2(\text{context\_length})$ reasoning steps. Detailed analysis in Appendix P. The data reveals two distinct failure patterns: **i) catastrophic collapse** versus **ii) gradual degradation**. Tasks

| Model (Param) | Easy | | | Medium | | | Hard | | | Overall | | |
|---|---|---|---|---|---|---|---|---|---|---|---|---|
| | Acc (%) | Inst (%) | Tokens (Avg) | Acc (%) | Inst (%) | Tokens (Avg) | Acc (%) | Inst (%) | Tokens (Avg) | Acc (%) | Inst (%) | Tokens (Avg) |
| *Qwen Family (Qwen3)* | | | | | | | | | | | | |
| Qwen3 (0.6B) | $50.24_{\pm4.2}$ | 83.85 | 3162.80 | $19.88_{\pm3.8}$ | 92.00 | 15029.98 | $14.51_{\pm4.1}$ | 74.17 | 5542.79 | $28.21_{\pm3.9}$ | 83.34 | 7911.86 |
| Qwen3 (1.7B) | $70.58_{\pm3.4}$ | 86.54 | 3157.20 | $42.28_{\pm3.1}$ | 92.00 | 10854.63 | $24.38_{\pm3.7}$ | 80.65 | 6089.76 | $45.75_{\pm3.2}$ | 86.40 | 6700.53 |
| Qwen3 (4B) | $82.15_{\pm2.8}$ | 91.57 | 3091.40 | $59.47_{\pm2.6}$ | 92.00 | 11649.83 | $34.28_{\pm3.1}$ | 83.85 | 5811.69 | $58.63_{\pm2.7}$ | 89.14 | 6850.91 |
| Qwen3 (8B) | $82.35_{\pm2.4}$ | 91.58 | 3027.80 | $60.24_{\pm2.2}$ | 92.00 | 8295.54 | $36.78_{\pm2.8}$ | 85.13 | 5786.42 | $59.79_{\pm2.3}$ | 89.57 | 5703.25 |
| Qwen3 (14B) | $86.78_{\pm1.7}$ | 99.27 | 3607.60 | $69.54_{\pm1.9}$ | 92.00 | 6298.80 | $42.51_{\pm2.1}$ | 85.21 | 5468.71 | $66.28_{\pm1.8}$ | 92.15 | 5125.04 |
| Qwen3 (32B) | $84.38_{\pm1.2}$ | 93.05 | 2845.90 | $\mathbf{76.42_{\pm1.4}}$ | 92.00 | 5563.77 | $43.42_{\pm1.6}$ | 85.87 | 5484.68 | $68.07_{\pm1.3}$ | 90.31 | 4631.45 |
| Qwen3 (30B-MOE) | $88.49_{\pm1.3}$ | 94.20 | 2415.13 | $74.93_{\pm1.4}$ | 92.00 | 6330.92 | $\underline{43.58_{\pm1.7}}$ | 87.28 | 5391.83 | $69.00_{\pm1.4}$ | 91.16 | 4712.63 |
| Qwen3 (30B-MOE-i) | $\mathbf{91.89_{\pm0.9}}$ | 99.46 | 647.17 | $73.52_{\pm1.1}$ | 92.00 | 3019.49 | $\mathbf{45.57_{\pm1.4}}$ | 97.93 | 2519.92 | $\mathbf{70.33_{\pm1.0}}$ | 96.46 | 2062.19 |
| Qwen3 (4B-t) | $85.37_{\pm2.7}$ | 95.19 | 2552.09 | $61.48_{\pm2.5}$ | 92.00 | 4874.80 | $35.47_{\pm3.0}$ | 83.59 | 5931.45 | $60.77_{\pm2.6}$ | 90.26 | 4452.78 |
| Qwen3 (30B-MOE-t) | $\underline{90.49_{\pm1.1}}$ | 96.85 | 2052.39 | $\underline{75.57_{\pm1.2}}$ | 92.00 | 6063.77 | $41.54_{\pm1.5}$ | 84.61 | 5573.57 | $\underline{69.20_{\pm1.2}}$ | 91.15 | 4563.24 |
| *Qwen Family (Qwen2.5)* | | | | | | | | | | | | |
| Qwen2.5 (0.5B) | $21.56_{\pm4.8}$ | 77.57 | 432.30 | $5.48_{\pm4.2}$ | 92.00 | 7381.67 | $1.21_{\pm1.5}$ | 74.77 | 1742.56 | $9.42_{\pm3.4}$ | 81.45 | 3185.51 |
| Qwen2.5 (1.5B) | $43.28_{\pm3.9}$ | 85.45 | 264.70 | $12.21_{\pm3.4}$ | 92.00 | 6014.44 | $3.18_{\pm2.2}$ | 88.16 | 1131.25 | $19.56_{\pm3.1}$ | 88.54 | 2470.13 |
| Qwen2.5 (3B) | $45.52_{\pm3.6}$ | 92.35 | 331.30 | $20.82_{\pm3.2}$ | 92.00 | 2121.82 | $8.12_{\pm2.8}$ | 92.11 | 1085.38 | $24.82_{\pm3.1}$ | 92.15 | 1179.50 |
| Qwen2.5 (7B) | $61.62_{\pm2.7}$ | 96.47 | 286.90 | $30.32_{\pm2.4}$ | 92.00 | 1135.23 | $16.42_{\pm2.6}$ | 91.65 | 938.58 | $36.12_{\pm2.5}$ | 93.37 | 786.90 |
| Qwen2.5 (14B) | $63.52_{\pm2.1}$ | 97.83 | 260.20 | $37.88_{\pm1.9}$ | 91.76 | 777.44 | $22.58_{\pm2.2}$ | 98.41 | 786.78 | $41.33_{\pm1.8}$ | 96.00 | 608.14 |
| Qwen2.5 (32B) | $73.15_{\pm1.4}$ | 99.26 | 260.90 | $44.78_{\pm1.6}$ | 91.24 | 650.59 | $\underline{26.33_{\pm1.8}}$ | 98.42 | 685.95 | $48.09_{\pm1.5}$ | 96.31 | 532.48 |
| Qwen2.5 (72B) | $\mathbf{80.52_{\pm0.9}}$ | 99.93 | 315.75 | $46.18_{\pm1.1}$ | 92.00 | 739.62 | $\mathbf{33.37_{\pm1.3}}$ | 99.55 | 875.11 | $\mathbf{53.36_{\pm1.0}}$ | 97.16 | 643.49 |
| Qwen2.5 (1.5B-m) | $51.18_{\pm3.7}$ | 94.04 | 397.10 | $28.23_{\pm3.2}$ | 92.00 | 1427.29 | $3.51_{\pm2.1}$ | 74.22 | 1288.07 | $27.64_{\pm2.9}$ | 86.75 | 1037.49 |
| Qwen2.5 (7B-m) | $60.43_{\pm2.4}$ | 94.36 | 411.70 | $26.42_{\pm2.5}$ | 92.00 | 2044.62 | $8.04_{\pm2.7}$ | 85.41 | 1472.74 | $31.63_{\pm2.6}$ | 90.59 | 1309.69 |
| Qwen2.5 (72B-m) | $\underline{77.48_{\pm0.8}}$ | 93.02 | 429.30 | $\mathbf{55.38_{\pm1.0}}$ | 92.00 | 780.85 | $12.59_{\pm1.4}$ | 78.21 | 1358.64 | $\underline{48.48_{\pm0.9}}$ | 87.74 | 856.26 |
| *Gemma Family* | | | | | | | | | | | | |
| Gemma (1B) | $23.38_{\pm4.5}$ | 67.21 | 870.85 | $16.42_{\pm3.9}$ | 92.00 | 1229.28 | $0.93_{\pm1.1}$ | 76.29 | 665.62 | $13.58_{\pm3.2}$ | 78.50 | 921.92 |
| Gemma (4B) | $66.62_{\pm2.6}$ | 98.56 | 550.90 | $38.17_{\pm2.4}$ | 92.00 | 1072.62 | $11.42_{\pm2.5}$ | 85.76 | 1048.37 | $38.74_{\pm2.4}$ | 92.11 | 890.63 |
| Gemma (12B) | $\underline{75.28_{\pm1.7}}$ | 96.85 | 504.14 | $\underline{50.24_{\pm1.8}}$ | 92.00 | 1337.32 | $\underline{21.68_{\pm2.1}}$ | 77.07 | 926.02 | $\underline{49.07_{\pm1.7}}$ | 88.64 | 922.49 |
| Gemma (27B) | $\mathbf{79.07_{\pm1.2}}$ | 97.46 | 415.96 | $\mathbf{58.57_{\pm1.4}}$ | 92.00 | 1118.70 | $\mathbf{36.73_{\pm1.6}}$ | 98.32 | 1069.75 | $\mathbf{58.12_{\pm1.3}}$ | 95.93 | 868.14 |
| *Phi Family* | | | | | | | | | | | | |
| Phi3-mini (3.8B) | $35.58_{\pm3.2}$ | 96.58 | 89.40 | $20.03_{\pm2.9}$ | 91.92 | 503.10 | $11.38_{\pm2.7}$ | 88.58 | 667.66 | $22.33_{\pm2.8}$ | 92.36 | 420.05 |
| Phi3-med (14B-4k) | $43.72_{\pm2.1}$ | 89.87 | 189.30 | $21.42_{\pm1.9}$ | 91.64 | 397.91 | $14.72_{\pm2.2}$ | 96.82 | 539.25 | $26.62_{\pm2.0}$ | 92.78 | 375.49 |
| Phi3-med (14B-128k) | $40.52_{\pm2.2}$ | 96.26 | 140.00 | $23.72_{\pm2.0}$ | 92.00 | 545.48 | $15.88_{\pm2.3}$ | 95.80 | 694.27 | $26.71_{\pm2.1}$ | 94.69 | 459.92 |
| Phi4-mini (3.8B-i) | $54.78_{\pm3.1}$ | 95.02 | 292.10 | $24.62_{\pm2.8}$ | 90.80 | 1297.82 | $16.07_{\pm2.6}$ | 92.30 | 1178.99 | $31.82_{\pm2.7}$ | 92.71 | 922.97 |
| Phi4-mini-reasoning (3.8B) | $70.16_{\pm3.0}$ | 89.56 | 3171.90 | $53.24_{\pm2.7}$ | 92.00 | 11615.07 | $25.94_{\pm2.5}$ | 79.55 | 6181.84 | $49.78_{\pm2.7}$ | 87.04 | 6989.60 |
| Phi4 (14B) | $\mathbf{78.92_{\pm1.4}}$ | 97.46 | 378.60 | $38.19_{\pm2.0}$ | 92.00 | 686.87 | $\underline{28.23_{\pm2.3}}$ | 97.18 | 1032.74 | $48.45_{\pm2.1}$ | 95.55 | 699.40 |
| Phi4-reasoning (14B) | $\underline{72.18_{\pm2.2}}$ | 96.21 | 6066.20 | $\mathbf{61.42_{\pm2.1}}$ | 92.00 | 8792.26 | $\mathbf{36.24_{\pm2.3}}$ | 75.98 | 6687.98 | $\mathbf{56.61_{\pm2.2}}$ | 88.06 | 7182.15 |
| Phi4-reasoning+ (14B) | $69.48_{\pm2.3}$ | 88.89 | 6780.70 | $\underline{55.07_{\pm2.2}}$ | 92.00 | 6261.74 | $28.21_{\pm2.4}$ | 71.31 | 7002.66 | $\underline{50.92_{\pm2.3}}$ | 84.06 | 6681.70 |
| *Llama Family* | | | | | | | | | | | | |
| Llama-3.2 (1B) | $16.48_{\pm4.6}$ | 47.15 | 336.30 | $5.62_{\pm3.9}$ | 92.00 | 6699.03 | $0.94_{\pm1.4}$ | 68.44 | 2028.62 | $7.68_{\pm3.3}$ | 69.20 | 3021.32 |
| Llama-3.2 (3B) | $42.28_{\pm3.5}$ | 89.88 | 279.70 | $16.25_{\pm3.1}$ | 92.00 | 6467.50 | $4.42_{\pm2.4}$ | 76.20 | 1334.66 | $20.98_{\pm2.9}$ | 86.03 | 2693.95 |
| Llama-3.1 (8B) | $49.12_{\pm2.7}$ | 85.66 | 366.40 | $15.47_{\pm2.4}$ | 92.00 | 7590.22 | $8.02_{\pm2.5}$ | 88.27 | 1912.41 | $24.20_{\pm2.4}$ | 88.64 | 3289.68 |
| Llama-3.1 (70B) | $\underline{75.68_{\pm0.9}}$ | 98.12 | 251.20 | $30.82_{\pm1.1}$ | 92.00 | 3130.60 | $23.25_{\pm1.4}$ | 95.80 | 1181.75 | $43.25_{\pm1.0}$ | 95.31 | 1521.18 |
| Llama-3.3 (70B) | $74.84_{\pm0.9}$ | 97.40 | 312.80 | $\underline{46.48_{\pm1.2}}$ | 92.00 | 709.45 | $\underline{27.16_{\pm1.4}}$ | 99.38 | 887.38 | $\underline{49.49_{\pm1.1}}$ | 96.26 | 636.54 |
| Llama4-Scout (120B-MOE) | $\mathbf{79.12_{\pm0.9}}$ | 92.61 | 321.93 | $\mathbf{52.31_{\pm1.1}}$ | 86.20 | 731.44 | $\mathbf{27.41_{\pm1.3}}$ | 49.60 | 1214.75 | $\mathbf{52.95_{\pm1.0}}$ | 76.14 | 756.04 |
| *Mistral Family* | | | | | | | | | | | | |
| Mistral (7B) | $27.42_{\pm2.8}$ | 96.26 | 207.10 | $10.28_{\pm2.5}$ | 92.00 | 635.92 | $4.57_{\pm2.2}$ | 91.23 | 1167.03 | $14.09_{\pm2.4}$ | 93.16 | 670.02 |
| Ministral (8B) | $\mathbf{51.18_{\pm2.4}}$ | 89.70 | 534.28 | $\mathbf{21.42_{\pm2.1}}$ | 92.00 | 1160.51 | $9.17_{\pm2.3}$ | 94.74 | 872.20 | $\underline{27.26_{\pm2.2}}$ | 92.15 | 855.66 |
| Mistral-nemo (12B) | $35.68_{\pm1.9}$ | 82.95 | 377.00 | $18.33_{\pm1.7}$ | 92.00 | 1266.19 | $\underline{12.02_{\pm1.9}}$ | 92.11 | 542.81 | $22.01_{\pm1.8}$ | 89.02 | 728.67 |
| Mixtral-8x7b | $35.28_{\pm1.8}$ | 91.47 | 140.47 | $12.14_{\pm1.6}$ | 86.88 | 350.39 | $9.89_{\pm1.8}$ | 85.69 | 523.03 | $19.10_{\pm1.7}$ | 88.01 | 337.96 |
| Mixtral-8x22b | $\underline{50.31_{\pm1.3}}$ | 79.17 | 296.42 | $\underline{20.79_{\pm1.4}}$ | 92.00 | 536.00 | $\mathbf{18.78_{\pm1.6}}$ | 95.18 | 762.46 | $\mathbf{29.96_{\pm1.4}}$ | 88.78 | 531.63 |
| *Others* | | | | | | | | | | | | |
| Smollm3 (3B) | $69.26_{\pm3.1}$ | 84.60 | 2985.20 | $9.71_{\pm2.8}$ | 70.69 | 16320.71 | $19.62_{\pm2.9}$ | 75.09 | 6076.59 | $32.86_{\pm2.9}$ | 76.79 | 8460.83 |
| Smollm2 (1.7B) | $16.92_{\pm4.2}$ | 68.98 | 213.00 | $0.28_{\pm0.8}$ | 8.64 | 65.71 | $6.48_{\pm2.5}$ | 88.89 | 1069.83 | $7.89_{\pm2.5}$ | 55.17 | 449.51 |
| GPT-OSS (20B) | $87.49_{\pm1.1}$ | 95.89 | 1185.34 | $\underline{63.38_{\pm1.3}}$ | 92.00 | 3674.40 | $\underline{52.12_{\pm1.5}}$ | 81.23 | 3877.50 | $\underline{67.66_{\pm1.2}}$ | 89.71 | 2912.41 |
| GPT-OSS (120B) | $\mathbf{93.27_{\pm0.6}}$ | 99.01 | 821.55 | $\mathbf{75.28_{\pm0.8}}$ | 92.00 | 2205.27 | $\mathbf{59.41_{\pm1.0}}$ | 77.80 | 2981.69 | $\mathbf{75.99_{\pm0.8}}$ | 89.60 | 2002.84 |
| *OpenAI Family (Proprietary)* | | | | | | | | | | | | |
| GPT-5 * | $\mathbf{97.31_{\pm0.3}}$ | 99.82 | 992.41 | $\underline{81.73_{\pm0.4}}$ | 92.00 | 2278.59 | $\mathbf{71.68_{\pm0.5}}$ | 96.64 | 4525.92 | $\mathbf{83.57_{\pm0.4}}$ | 96.15 | 2598.97 |
| GPT-5-mini * | $96.13_{\pm0.2}$ | 100.00 | 798.72 | $79.73_{\pm0.5}$ | 92.00 | 1640.01 | $69.28_{\pm0.6}$ | 90.70 | 3330.71 | $81.71_{\pm0.4}$ | 94.23 | 1923.15 |
| GPT-5-nano * | $96.07_{\pm0.3}$ | 99.29 | 1377.00 | $80.13_{\pm0.6}$ | 91.20 | 2414.85 | $\underline{69.78_{\pm0.7}}$ | 90.25 | 7078.18 | $\underline{81.99_{\pm0.4}}$ | 93.58 | 3623.34 |
| GPT4.1 | $92.23_{\pm0.4}$ | 100.00 | 409.38 | $73.08_{\pm0.5}$ | 92.00 | 3718.96 | $46.84_{\pm0.7}$ | 97.13 | 3681.23 | $70.72_{\pm0.5}$ | 96.38 | 2603.19 |
| GPT4.1-mini | $93.28_{\pm0.4}$ | 100.00 | 915.70 | $72.67_{\pm0.4}$ | 92.00 | 2549.95 | $41.38_{\pm0.6}$ | 95.35 | 3069.56 | $69.11_{\pm0.4}$ | 95.78 | 2178.40 |
| GPT4.1-nano | $80.54_{\pm0.5}$ | 98.21 | 1161.08 | $53.47_{\pm0.7}$ | 92.00 | 1254.45 | $23.42_{\pm0.8}$ | 98.64 | 1303.67 | $52.48_{\pm0.6}$ | 96.28 | 1239.73 |
| GPT4o | $88.17_{\pm0.4}$ | 100.00 | 256.84 | $54.28_{\pm0.6}$ | 92.00 | 536.46 | $29.07_{\pm0.7}$ | 97.78 | 582.04 | $57.17_{\pm0.5}$ | 96.59 | 458.45 |
| GPT4o-mini | $73.52_{\pm0.6}$ | 98.57 | 272.32 | $38.28_{\pm0.7}$ | 92.00 | 522.71 | $14.61_{\pm0.8}$ | 97.90 | 819.02 | $42.14_{\pm0.7}$ | 96.16 | 538.02 |
| o4-mini * | $94.64_{\pm0.3}$ | 100.00 | 993.38 | $81.48_{\pm0.4}$ | 92.00 | 1486.20 | $61.37_{\pm0.5}$ | 93.89 | 4514.30 | $79.16_{\pm0.4}$ | 95.30 | 2331.31 |
| o3 * | $\underline{97.26_{\pm0.2}}$ | 100.00 | 856.56 | $\mathbf{82.27_{\pm0.3}}$ | 90.40 | 2891.81 | $61.78_{\pm0.5}$ | 94.49 | 5585.29 | $80.44_{\pm0.3}$ | 94.96 | 3111.22 |
| o3-mini * | $94.23_{\pm0.4}$ | 99.64 | 1101.22 | $80.67_{\pm0.5}$ | 92.00 | 1525.04 | $57.88_{\pm0.6}$ | 96.40 | 5425.99 | $77.59_{\pm0.4}$ | 96.01 | 2684.08 |
| *Gemini Family (Proprietary)* | | | | | | | | | | | | |
| Gemini-2.5-pro | $\underline{89.38_{\pm0.3}}$ | 87.32 | 267.57 | $\mathbf{77.33_{\pm0.4}}$ | 89.20 | 725.72 | $\mathbf{56.21_{\pm0.5}}$ | 97.92 | 916.18 | $\mathbf{74.31_{\pm0.3}}$ | 91.48 | 636.49 |
| Gemini-2.5-flash | $\mathbf{89.87_{\pm0.4}}$ | 95.65 | 357.62 | $\underline{70.13_{\pm0.5}}$ | 76.00 | 617.25 | $\underline{51.18_{\pm0.6}}$ | 88.82 | 1097.35 | $\underline{70.39_{\pm0.5}}$ | 86.82 | 690.74 |
| Gemini-2.5-flash-lite | $86.28_{\pm0.6}$ | 100.00 | 587.71 | $49.87_{\pm0.6}$ | 97.00 | 11929.26 | $39.57_{\pm0.8}$ | 97.00 | 5825.56 | $58.57_{\pm0.6}$ | 96.33 | 6114.18 |
| Gemini-2.0-flash | $84.78_{\pm0.4}$ | 98.69 | 313.89 | $67.08_{\pm0.5}$ | 92.00 | 1044.79 | $31.52_{\pm0.7}$ | 88.94 | 1425.60 | $61.13_{\pm0.5}$ | 93.21 | 928.09 |
| Gemini-2.0-flash-lite | $81.87_{\pm0.5}$ | 98.81 | 286.12 | $57.88_{\pm0.6}$ | 92.00 | 1043.37 | $30.93_{\pm0.7}$ | 94.81 | 1840.31 | $56.89_{\pm0.6}$ | 95.21 | 1056.60 |

Table 3: **BEYONDBENCH Leaderboard:** Three-fold evaluation results (mean$_{\pm\text{std}}$) across three difficulty suites (Easy, Medium, Hard). Each suite reports Accuracy (Acc), Instruction-following (Inst) and average output Tokens; suffixes denote variant type: $-$i = instruct, $-$t = thinking, $-$m = math. The three **Overall** subcolumns show the arithmetic mean across Easy/Medium/Hard for Acc, Inst and Tokens respectively. Within each family, **bold** denotes the best variant and underlined denotes the second-best. **Note:** Models marked as * have been allowed maximum reasoning effort/thinking tokens to use. GPT-5 family results include tool-augmented evaluation (see Section 4.5).

like Tower of Hanoi and Matrix Chain Multiplication exhibit sharp cliffs: models perform well until 5-6 disks and 7-10 matrices respectively, then collapse completely to near-zero performance. In contrast, Cryptarithmetic and Graph Coloring show **counter-intuitive improvement patterns**: Cryptarithmetic accuracy often increases with word length (e.g., 78.0%→90.4% for GPT-OSS), while Graph Coloring maintains relatively stable performance across vertex counts (e.g., 100%→77% from 12-40 vertices for GPT-OSS). This suggests that **complexity scaling affects different algorithmic reasoning mechanisms differently**: recursive problems with exponential state explosion face hard computational limits, while constraint satisfaction problems may benefit from richer problem structure providing more optimization pathways. *Interestingly,*

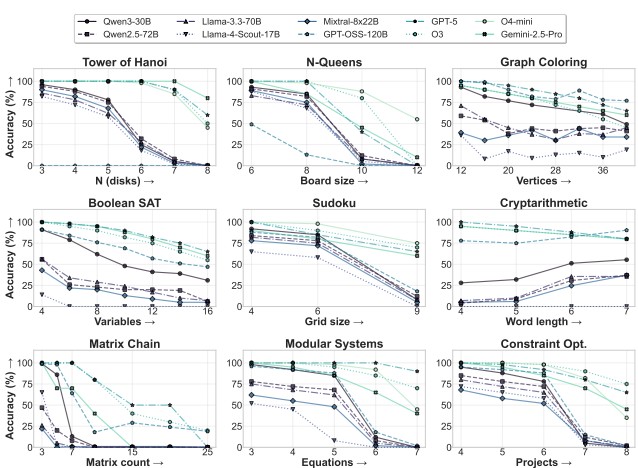

Figure 1: **Performance Collapse on Hard Suite.**

*even reasoning models like GPT-5 and o3 exhibit these same differential patterns*, indicating fundamental differences in how transformers handle various types of algorithmic complexity rather than uniform scaling limitations. Furthermore, **most open-source models show a consistent performance ceiling around 30-35% on hard problems** across 85 models.

DYNAMIC EVALUATION HIGHLIGHTS LIMITS OF STATIC BENCHMARKS. **Dynamic problem generation shows that previous benchmark scores overestimate reasoning capabilities.** Models claiming 90%+ performance on static benchmarks like GSM8K achieve only ~50% on equivalent dynamically generated tasks. Our framework generates problems from spaces exceeding $10^{15}$ possibilities, making memorization very difficult. Figure 2 shows the resulting performance distribution, reflecting innate reasoning capabilities when tested on our dynamic generation benchmark. For detailed results, see appendix H (medium) and K (hard suite).

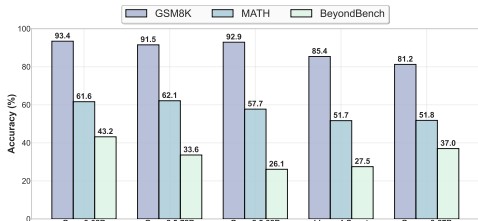

Figure 2: **Static Benchmarks vs BEYOND-BENCH Performance on Hard Suite.**

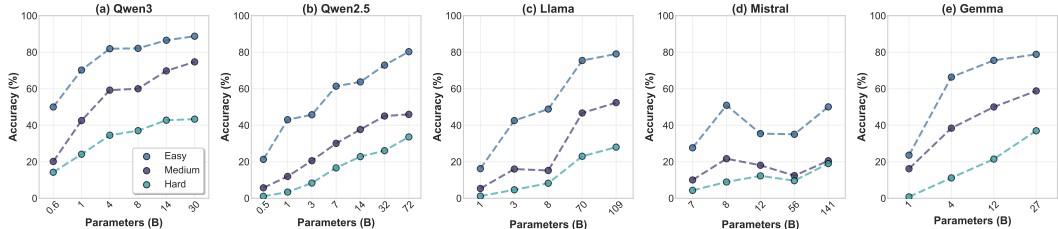

Figure 3: **Scaling Laws Across Model Families.** Performance gains follow logarithmic curves with steep early improvements tapering to marginal gains at larger scales.

## 4.3 PARAMETER SCALING EFFECTS

PARAMETER SCALING SHOWS PERFORMANCE SATURATION. **Our overall results (Table 3) suggest that while larger models still perform better, the rate of improvement follows logarithmic patterns with diminishing returns after a certain point.** Figure 3 illustrates logarithmic scaling patterns across model families, revealing diminishing returns of parameter scaling. Within the Qwen3 family (Yang et al., 2025a), going from 0.6B to 1.7B (2.8x parameters) yields a substantial 17.5% point gain (28.21% → 45.75%), but scaling further to 32B parameters adds only 22.3 points across

18x parameter increase. The step from 8B to 14B (1.75× parameters) delivers just 6.49 points, and from 14B to 32B (2.3x parameters) provides only 1.79 points. The Qwen2.5 family (Qwen et al., 2025) shows similar patterns: large early gains (9.42% → 36.12% from 0.5B to 7B) followed by slower progress (36.12% → 53.36% from 7B to 72B). *These results suggest that while scaling remains beneficial, its impact on reasoning ability diminishes progressively,* with only reduced incremental benefits at larger model sizes.

QUANTIZATION HAS MINIMAL IMPACT ON REASONING. **Even aggressive quantization has minimal impact (<3% typical) on algorithmic reasoning, suggesting that reasoning is robust to reduced precision.** Across quantized model variants, FP8 (Kuzmin et al., 2024) and GPTQ-Int8 quantization (Frantar et al., 2023) maintain nearly identical performance to full-precision models.

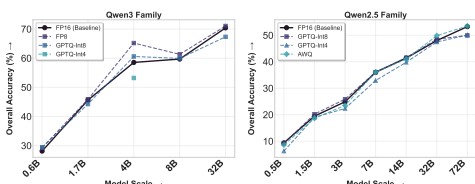

Some quantized variants even outperform their full-precision counterparts: Qwen3-30B-MOE-i (FP8) achieves 71.04% versus 70.33% for the original. Even GPTQ-Int4 quantization, reducing model size by 4×, shows only modest drops, typically 1-3% (see Appendix M for quantized model results across easy, medium and hard tasks). **This finding suggests that algorithmic thinking relies more on discrete symbolic operations than fine-grained numerical computations.** The results indicate that the bottleneck in reasoning performance is not computational precision but rather architectural design and the availability of appropriate architectural mechanisms enabling multi-step reasoning. Figure 4 shows the minimal performance degradation.

Figure 4: **Quantization Robustness Across Model Scales.** Base Model (FP16) vs quantized variants for Qwen families.

### 4.4 SPECIALIZATION EFFECTS AND PERFORMANCE PATTERNS

THINKING MODELS SHOW LIMITED GAINS. **Models designed for extended reasoning show only modest improvements over their standard versions.** The Qwen3-4B-thinking variant achieves 60.77% versus 58.63% for the base model-a 2.14% improvement. Similarly, Qwen3-30B-MOE-thinking (Xu et al., 2025) shows only 0.2% gain (69.20% vs 69.00%) despite being optimized for reasoning tasks.

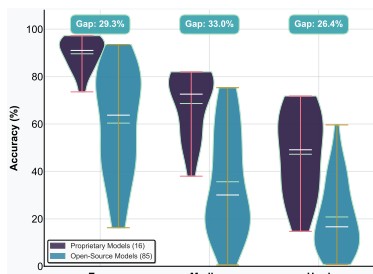

Interestingly, these "thinking" models don't consistently use more tokens, suggesting they process information differently rather than simply thinking longer. Our detailed failure mode analysis (Appendix O) reveals that reasoning models fail later in execution (Tower of Hanoi: step 89/127) compared to vanilla models (step 23/127), but struggle with state management during extended reasoning. While vanilla models fail early without recognizing errors, reasoning models attempt self-correction but with 87.6% error introduction rate (Srivastava et al., 2025a). We varied reasoning budget from 0% to 100% across model families, finding that Hard tasks benefit most (+24-44% for weak models) while already-capable models show minimal improvement (o3: ≤4% gain). Detailed case studies and reasoning budget analysis in Appendix O. This indicates that *current approaches to improving reasoning may not completely address the deeper challenges for algorithmic reasoning*. True algorithmic reasoning appears to require systematic state management, backtracking, and constraint-handling capabilities that may be fundamentally different from language modeling.

Figure 5: **Performance Distribution by Model Type and Difficulty.** Violin width shows model density; white lines are medians, dark lines are means. "Gap %" shows mean performance difference.

DOMAIN SPECIFIC (MATHEMATICAL) FINE-TUNING REDUCES ALGORITHMIC PERFORMANCE. **Specialized mathematical training appears to hurt performance on algorithmic reasoning tasks.** Qwen2.5-72B-math (Yang et al., 2024) achieves 48.48% overall versus 53.36% for the base model: a 4.88% drop after mathematical training. This pattern appears across model sizes: Qwen2.5-7B-math (31.63%) underperforms the base model (36.12%). **Mathematical fine-tuning appears to optimize for in-domain datasets like GSM8K, MATH not innate algorithmic reasoning.** Mathematical training focuses on symbolic manipulation, equation solving, and formula application, while our

tasks require procedure construction, state management, and systematic search. Such training seems to emphasize symbolic and equation-based skills, which may not fully align with algorithmic tasks.

## 4.5 PROPRIETARY MODELS AND TOOL-AUGMENTED REASONING

OPEN-SOURCE MODELS SHOW LARGE PERFORMANCE GAPS COMPARED TO PROPRIETARY MODELS. **The performance difference between leading proprietary models (Comanici et al., 2025; OpenAI et al., 2024) and open-source alternatives suggests that the gap cannot be explained by scale alone.** For instance, GPT-5 achieves a notable 12.27% better performance on hard tasks over the best open-source model (GPT-OSS-120B). **These results suggest that top-performing proprietary models may rely on innovations beyond simple parameter scaling**, possibly including *internal tool use or code generation* to solve algorithmic problems rather than relying solely on language-based reasoning.

TOOL-AUGMENTED APPROACHES SHOW PROMISE. **GPT-5 (an agentic LLM) results suggest that combining language models with computational tools may be more effective than scaling language models alone.**

| Model | No Tools | | | | +Code Execution | | | All Tools Combined | | | |
|---|---|---|---|---|---|---|---|---|---|---|---|
| | Easy | Med | Hard | Avg | Easy | Med | Hard | Easy | Med | Hard | Avg |
| GPT-5 | 88.4 | 61.6 | 50.3 | 66.8 | 95.8 | 76.4 | 67.2 | 97.3 | 81.7 | 71.7 | **83.6** |
| GPT-5-mini | 91.7 | 64.4 | 41.4 | 65.8 | 94.8 | 74.2 | 62.8 | 96.1 | 79.7 | 69.3 | **81.7** |
| GPT-5-nano | 58.3 | 33.6 | 22.3 | 38.1 | 71.8 | 52.7 | 48.7 | 96.1 | 80.1 | 69.8 | **82.0** |
| Gemini-2.5-pro | 82.4 | 58.7 | 42.1 | 61.1 | 88.2 | 68.8 | 56.4 | 89.4 | 72.3 | 61.2 | **74.3** |

Table 4: **Performance across tools and difficulty suites.**

The performance of tool-augmented GPT-5 models drastically drops when its tool usage is turned off. Table 4 shows a 16.81%, 15.86% and 43.95% drop in accuracy for GPT-5, GPT-5-mini, and GPT-5-nano respectively. This shows that these **models appear to recognize when to use computational tools** rather than attempting pure language-based reasoning. This approach mirrors human problem-solving, where we use external tools to extend our capabilities (Latimer et al., 2025). *The efficiency gains observed suggest that recognizing problem types and selecting appropriate solution methods may be more important than extended reasoning within the language model itself.*

EXTENDED TOOL ANALYSIS. We conducted comprehensive per-task tool analysis across 44 tasks with three tool types (calculator, code execution, web search). Code execution provides the largest improvements (+15-55%), while web search shows negligible benefit (+0.3%), confirming contamination resistance. Importantly, smaller models struggle with tool orchestration itself (41.2% success rate for Qwen2.5-3B vs. 94.2% for GPT-5), indicating that effective tool use requires sophisticated reasoning (Srivastava et al., 2026). Complete analysis in Appendix Q.

CONTAMINATION RESISTANCE VALIDATION. To experimentally validate our theoretical contamination resistance guarantees, we fine-tuned 5 models on 66K BEYONDBENCH instances using SFT and GRPO. When evaluated on a different seed (ensuring no instance overlap), models show limited Hard suite improvement (+10-15%) compared to Easy/Medium (+27-38%), demonstrating that NP-complete problems remain fundamentally challenging even after targeted training. Unlike static benchmarks where training achieves near-perfect scores through memorization, BEYONDBENCH maintains evaluation integrity. Full experimental details in Appendix R.

## 5 CONCLUSION

BEYONDBENCH's contamination-resistant evaluation of 101 models reveals that **innate reasoning in raw language models represents a fundamental bottleneck that cannot be overcome through scaling *alone***. Our results reveal three major insights: *1)* parameter scaling shows logarithmic returns with a hard ceiling around 30-35% on algorithmic tasks for most open-source models; *2)* "thinking" models fail to meaningfully improve reasoning; and *3)* mathematical fine-tuning actively degrades algorithmic performance by optimizing for the wrong computational primitives. The better performance of tool-augmented models like GPT-5 (83.57% vs. 75.99% for the best open-source alternative GPT-OSS-120B) combined with their efficient problem-solving approaches suggests these systems succeed not through superior language-based reasoning but by recognizing when to use tools for complex reasoning. The path toward Artificial General Intelligence lies in developing agentic architectures that combine language understanding with tool use: systems that, like human experts, can recognize when it's time to reason and when it's time to compute. BEYONDBENCH has revealed the core limits of raw language models and highlighted the promise of tool-augmented systems; *looking forward, we see the future toward hybrid neuro-symbolic architectures and agentic LLMs with the ability to call on tools and to know when to use them effectively.*

## ACKNOWLEDGEMENTS

This work was supported by the NSF #2442253, NSF NAIRR Pilot with PSC Neocortex and NCSA Delta, Cisco Research, NVIDIA, Amazon, the Commonwealth Cyber Initiative, the Amazon–Virginia Tech Center for Efficient and Robust Machine Learning, the Sanghani Center for AI and Data Analytics at Virginia Tech, and the Virginia Tech Innovation Campus. The views, findings, conclusions, and recommendations expressed in this work are those of the authors and do not necessarily reflect the opinions of the funding agencies.

## ETHICS STATEMENT

BEYONDBENCH is designed to evaluate reasoning capabilities without raising ethical concerns. All generated problems are abstract mathematical or algorithmic puzzles without real-world context that could encode biases. We deliberately avoid word problems or scenarios that might reference people, cultures, or sensitive topics. The benchmark cannot be used to generate harmful content, as it only produces mathematical sequences, logical formulas, and algorithmic solutions. The framework is released as an open-source Python package under the MIT license (`https://github.com/ctrl-gaurav/BeyondBench`), and we provide an interactive leaderboard for transparent comparison of model performance. We acknowledge that improved reasoning capabilities could be dual-use, potentially enabling both beneficial applications (scientific discovery, education) and concerning ones (strategic deception, adversarial planning). However, our benchmark itself poses no direct ethical risks and serves the important purpose of accurately measuring AI capabilities, which is essential for responsible AI development.

## REPRODUCIBILITY STATEMENT

All code and evaluation scripts for BEYONDBENCH are publicly available. We provide a **user-friendly Python package** (`pip install beyondbench`, available at `https://pypi.org/project/beyondbench/`) for seamless integration, making evaluation as simple as any static benchmark. Installation requires a single command, and evaluation can be performed with minimal code. The package handles all infrastructure complexity internally, including problem generation, verification, and result aggregation. The source code and documentation are available at `https://github.com/ctrl-gaurav/BeyondBench`, and an interactive leaderboard is hosted at `https://ctrl-gaurav.github.io/BeyondBench/`.

The repository includes: (1) complete implementation of all 44 tasks with 117 variations, including generation algorithms, solution verifiers, and response parsers, (2) exact model configurations and inference parameters used in our experiments, and (3) seeds and parameter ranges for regenerating all problem instances. All evaluations in this paper use consistent seed value 42 for the main results, ensuring completely fair cross-model comparison where every model sees identical problems. We also provide three-fold evaluation results with statistical analysis in Appendix N, demonstrating low variance (mean std = 2.3%) and high reproducibility.

To reproduce our results, researchers need access to GPUs with at least 80GB memory for the largest open-source models, or API access for proprietary models. The evaluation framework is model-agnostic and can be extended to new models by specifying the model ID and inference engine. Comprehensive case studies demonstrating our evaluation process, including problem generation examples, model response parsing, and verification procedures, are provided in Appendix O for full transparency.

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

# Appendix

This appendix is organized as follows. We first describe the experimental setup and implementation details used throughout our evaluation. We then provide extended related work and the motivation behind our algorithmic task selection. Next, we present detailed per-task results for each difficulty suite (Easy, Medium, Hard) followed by quantized model results. The remaining sections cover our statistical analyses, failure mode comparisons between LLMs and LRMs, normalized complexity analysis, tool-augmented evaluation, and SFT/RL training experiments. We conclude with the mathematical analysis of contamination probability, benchmark scalability, and a discussion of limitations.

## Table of Contents

## A    EXPERIMENTAL SETTING

**Model Selection and Evaluation Scale.**   We evaluated 101 language models spanning diverse architectures and parameter scales. Open-source models (85 total) include the Qwen family (Qwen3 and Qwen2.5 variants from 0.5B to 72B parameters), Llama family (Llama-3.1, Llama-3.2, and Llama-3.3 from 1B to 70B), Phi family (Phi3 and Phi4 variants from 3.8B to 14B), Gemma family (1B to 27B), and Mistral family (7B to 141B MoE). Proprietary models (16 total) include OpenAI's GPT series (GPT-4.1, GPT-4o, GPT-5 variants) and Google's Gemini family (Gemini 2.0 and 2.5 variants). For open-source models, we generated 1000 problem instances per task variation to ensure statistical robustness. Due to API cost constraints, we limited proprietary model evaluation to 100 instances per task, which still provides sufficient statistical power given the deterministic nature of our problems.

**Inference Configuration.**   All models were evaluated using consistent inference parameters to ensure fair comparison. We used temperature 0.1 and top-p 0.9 to encourage deterministic, focused responses while allowing minimal exploration. Maximum token limit was set to 32,768 to accommodate complex reasoning traces, though most tasks required far fewer tokens. For open-source models, we employed the vLLM engine for efficient parallel inference, utilizing tensor parallelism across 1-4 GPUs depending on model size. GPU memory utilization was carefully tuned per model (ranging from 0.28 to 0.96) to maximize throughput while avoiding out-of-memory errors. All experiments used seed 42 for reproducibility of random problem generation.

**Hardware Infrastructure.**   Open-source model evaluations were conducted on a cluster with NVIDIA A100 80GB GPUs, enabling efficient inference for models up to 72B parameters. Smaller models (under 7B) ran on single GPUs, while larger models required 2-4 GPUs with tensor parallelism. For proprietary models, we used official APIs: OpenAI's API for GPT models and Google's Vertex AI for Gemini models. To minimize variance from system load, all evaluations were scheduled during off-peak hours and run sequentially rather than in parallel when possible.

## B    IMPLEMENTATION DETAILS

**Problem Generation Pipeline.**   Each task implements a deterministic generation algorithm that takes a random seed and parameter configuration as input. The generator first samples parameters from specified ranges (e.g., list sizes from 8 to 64 elements, values from -1000 to 1000), then constructs the problem instance according to task-specific rules. For tasks admitting multiple solutions, we employ constraint satisfaction solvers (python-constraint for CSP problems, pysat for Boolean SAT) to enumerate all valid solutions. The generation process includes validation checks: ensuring problems are well-posed (have at least one solution), verifying solution uniqueness or completeness, and confirming the problem fits within context limits. Failed generations are logged and regenerated with different seeds.

**Response Parsing Framework.**   We developed task-specific parsers to extract answers from model outputs, handling diverse response formats. Each parser implements multiple extraction strategies in order of strictness: (1) exact format matching using regex patterns for structured outputs like "Answer: [1, 2, 3]", (2) fuzzy matching for variations like "The answer is 1, 2, 3" or "Solution: 1 2 3", (3) semantic extraction from verbose explanations, searching for number sequences or specific keywords, and (4) fallback to last numerical value for single-number answers. Parsers handle common LLM quirks including markdown formatting, LaTeX expressions, step-by-step solutions with intermediate results, and natural language descriptions of answers. When parsing fails completely, we mark the response as invalid rather than incorrect, distinguishing format failures from reasoning errors.

**Token Counting and Context Management.**   We implement precise token counting using model-specific tokenizers: tiktoken for OpenAI models, HuggingFace AutoTokenizer for open-source models, and official tokenizer APIs for Gemini models. Before problem generation, we estimate required tokens using task-specific formulas calibrated from pilot studies. For example, Tower of Hanoi requires approximately $(2^n - 1) \times 12$ tokens for $n$ disks, while sorting tasks need roughly $3n \log n$ tokens for lists of size $n$. During generation, if estimated tokens exceed 85% of the model's

context window, we reduce problem complexity iteratively (scaling down by 80% each iteration) until it fits. Post-generation, we count actual response tokens and flag warnings when usage exceeds 95% of context capacity, indicating potential truncation or quality degradation.

**Evaluation Metrics and Scoring.** We track three primary metrics for each model: (1) Accuracy - percentage of problems solved correctly, verified against ground truth or solution set, (2) Instruction Following - percentage of responses that match the required output format, independent of correctness, and (3) Token Efficiency - average tokens used per response, measuring computational parsimony. For tasks with multiple valid solutions, we consider a response correct if it matches any solution in the enumerated set. We compute metrics separately for each difficulty suite (Easy, Medium, Hard) and report both per-task and aggregate scores. Statistical significance is assessed using bootstrap confidence intervals with 10,000 resamples.

## C    EXTENDED RELATED WORK

**Dynamic Algorithmic Benchmarks**    Recent dynamic benchmarks share our motivation for contamination-resistant evaluation but differ in key aspects. NPHardEval (Fan et al., 2024) covers 9 complexity-aware tasks with monthly instance generation but primarily uses decision problems (yes/no answers) susceptible to 50% random guessing, lacks unique solution verification, and omits token-aware scaling. FCoReBench (Mittal et al., 2024) provides 40 first-order combinatorial tasks with SymPro-LM solver integration, demonstrating strong gains through symbolic reasoning, though solver formulation remains challenging (our Appendix Q shows only 68-72% success rates with solver access). PUZZLEPLEX (Long et al., 2025) evaluates 15 puzzle types emphasizing interactive planning and game dynamics through instruction-based and code-based paradigms, complementing BEYONDBENCH's focus on algorithmic problem-solving with verifiable correctness. BEYOND-BENCH advances beyond these works through formal CSP-based solution verification, multi-solution enumeration, token-aware difficulty calibration ($T_p(n) \leq 0.85 \cdot C$), and comprehensive evaluation across 44 tasks with 117 variations on 101 models.

**Tool-Augmented Reasoning Methods**    Tool-augmented approaches attempt to enhance LLM reasoning through external computation. PAL (Gao et al., 2023) generates Python code for reasoning steps executed by an interpreter, significantly reducing arithmetic errors on benchmarks like GSM8K, while Logic-LM (Pan et al., 2023) employs a three-stage pipeline translating natural language to symbolic representations (FOL, CSP, SAT), invoking deterministic solvers, and interpreting results with self-refinement. Reasoning Gym (Stojanovski et al., 2025) provides >100 procedurally generated reasoning environments designed primarily for Reinforcement Learning with Verifiable Rewards (RLVR), focusing on RL training infrastructure with curriculum learning and parametric difficulty control for studying reasoning development through training dynamics. Additionally, (Lu et al., 2025) scales agentic RL for tool-integrated reasoning in vision-language models, showing that tool use can be learned through reinforcement learning at scale. All these methods demonstrate value in their respective domains but face fundamental bottlenecks: PAL and Logic-LM remain constrained by formulation capability (models must correctly translate problems into executable code or symbolic representations), Reasoning Gym serves RL training rather than contamination-resistant evaluation, and agentic RL approaches focus on training rather than evaluation. Our tool analysis in Appendix Q directly evaluates both PAL-style code execution and Logic-LM-style symbolic solving across all 44 tasks: code execution provides 15-55% improvements yet models achieve only 67.2% on Sudoku and 58.4% on Logic Grid puzzles with full code access, while symbolic solvers (pysat, python-constraint) yield only 68-72% success rates on hard tasks. BEYONDBENCH advances beyond these works through: (1) provable contamination bounds ($|\Theta_\tau \times \mathcal{R}| > 10^{15}$, $\Pr[\text{collision}] < 10^{-3}$) with formal collision analysis, (2) token-aware evaluation ($T_p(n) \leq 0.85 \cdot C$) preventing unfair comparison across models, (3) complete multi-solution enumeration with CSP-based verification, and (4) extensive cross-model evaluation across 101 models for contamination-resistant assessment, confirming that auto-formalization remains a fundamental reasoning bottleneck even with tool augmentation.

While these works share motivations with BEYONDBENCH, dynamic evaluation, contamination resistance, or tool augmentation, BEYONDBENCH uniquely combines: (1) token-aware evaluation calibrating difficulty to model capacity, (2) formal CSP-based verification with multi-solution enumeration, (3) provable contamination guarantees ($> 10^{15}$ instances per task, $\Pr[\text{collision}] < 10^{-3}$),

(4) comprehensive assessment across 101 models (0.5B-141B parameters), and (5) unified evaluation of innate and tool-augmented reasoning. Frontier models achieve only 67-72% on hard tasks with full tool access, validating that BEYONDBENCH probes fundamental reasoning limitations persisting across augmentation strategies.

# D MOTIVATION FOR ALGORITHMIC TASK SELECTION

This section provides comprehensive justification for our design choice of using algorithmically-generated tasks with known solution procedures, addressing concerns about whether such tasks adequately assess model intelligence.

**Why Algorithmic Tasks Effectively Probe Reasoning**   Our task selection is motivated by the observation that true algorithmic reasoning requires multiple cognitive capabilities that are fundamentally challenging for current language models.

**Memory and State Management.** Tasks like Tower of Hanoi require tracking multiple disk positions across exponentially many moves. Even with knowledge of the recursive algorithm, models must maintain accurate state representations throughout $2^n - 1$ moves. Our experiments show that models achieve 80-90% accuracy for $n \leq 5$ disks but collapse to less than 10% for $n \geq 8$, indicating failures in state management rather than algorithmic knowledge.

**Backtracking and Search.** Constraint satisfaction problems such as Sudoku, N-Queens, and Graph Coloring require systematic exploration of solution spaces with backtracking when dead-ends are encountered. The standard solving algorithms are well-documented, yet models struggle to execute them correctly. This reveals limitations in maintaining search state and recognizing when to backtrack.

**Multi-Step Deduction.** Logic Grid Puzzles and Boolean SAT require chaining multiple logical inferences. Models must correctly apply rules of inference over many steps, where a single error propagates and invalidates the solution.

**Comparison with Competition Benchmarks**   Competition-level benchmarks such as MathArena and Olympiad problems test whether models can discover novel solution techniques. However, they suffer from two critical limitations. First, there is contamination risk: competition problems are published online and frequently appear in training data, so high scores may reflect memorization rather than reasoning. Second, there is a finite problem space: once exhausted, no new problems can be generated without human expert involvement.

In contrast, our benchmark tests whether models can execute known algorithms correctly, a prerequisite for any genuine reasoning. A model that cannot reliably execute Tower of Hanoi, a problem with a simple recursive solution, cannot be expected to discover novel mathematical theorems.

**Empirical Evidence: Known Algorithm, Failed Execution**   We conducted targeted experiments providing models with explicit algorithmic descriptions before problem instances. Table 5 shows the results.

| Task | Without Algorithm | With Algorithm | $\Delta$ |
|---|---|---|---|
| Tower of Hanoi (n=7) | 18.4% | 24.2% | +5.8% |
| Sudoku (9x9) | 8.2% | 12.1% | +3.9% |
| N-Queens (n=8) | 22.3% | 28.7% | +6.4% |
| Boolean SAT (5 vars) | 38.1% | 42.8% | +4.7% |

Table 5: GPT-OSS-120B performance with and without explicit algorithm descriptions.

Even when explicitly provided with correct algorithms, models show only marginal improvement, confirming that the bottleneck is execution capability, not algorithm discovery.

**Scalability Guarantees**   Unlike static benchmarks, our algorithmic generation ensures several key properties. First, there are effectively infinite problem instances with more than $10^{15}$ unique problems per task. Second, the difficulty is adjustable by simply increasing $n$ to challenge improved models.

Third, verifiable ground truth is mathematically guaranteed by construction. Fourth, there is no human bottleneck since new problems are generated automatically.

# E  EASY SUITE: FUNDAMENTAL ARITHMETIC AND STATISTICAL OPERATIONS

The Easy Suite comprises twenty-nine fundamental tasks that evaluate language models' capabilities in basic arithmetic operations, numerical comparisons, elementary statistical computations, and algorithmic reasoning over sequences. Each task generates problems dynamically using deterministic algorithms with guaranteed unique solutions, ensuring complete contamination resistance while maintaining mathematical rigor. The suite operates on the principle that for each problem class $\mathcal{P}_{\text{easy}}$, we can generate instances $p \in \mathcal{P}_{\text{easy}}$ with polynomial verification complexity $O(n^k)$ for constant $k \leq 2$, where $n$ represents the input size parameter.

## E.1  THEORETICAL FOUNDATION AND CONTAMINATION RESISTANCE

Let us formally define the contamination resistance property for the Easy Suite. For a problem generator $G_{\text{easy}} : \Theta \times \mathcal{R} \to \mathcal{P}_{\text{easy}}$ with parameter space $\Theta$ and random seed space $\mathcal{R}$, we ensure that $|\Theta \times \mathcal{R}| \gg |\mathcal{D}|$ where $\mathcal{D}$ represents any feasible training dataset. The generator exhibits the following sensitivity property:

$$\forall \theta_1, \theta_2 \in \Theta, r_1, r_2 \in \mathcal{R} : (\theta_1, r_1) \neq (\theta_2, r_2) \implies \Pr[G(\theta_1, r_1) = G(\theta_2, r_2)] < \epsilon \qquad (1)$$

where $\epsilon < 10^{-9}$ for our implementation parameters. This ensures that the probability of generating duplicate problems across different evaluation instances remains negligible.

Each task employs a validation function $V_{\text{easy}} : \mathcal{S} \times \mathcal{P}_{\text{easy}} \to \{0, 1\}$ that deterministically verifies solution correctness in polynomial time. The validation complexity for all Easy Suite tasks satisfies $T(V_{\text{easy}}) = O(n \log n)$ in the worst case, where $n$ denotes the input list size.

## E.2  PROBLEM GENERATION FRAMEWORK

The Easy Suite utilizes a unified generation framework based on uniform random sampling without replacement. For a given range $[a, b] \subset \mathbb{Z}$ and list size $k \in \mathbb{N}$, the generator produces a sequence $\mathcal{L} = \{x_1, x_2, \ldots, x_k\}$ where each $x_i$ is drawn uniformly from $[a, b]$ without replacement, ensuring $x_i \neq x_j$ for $i \neq j$.

The parameter space for Easy Suite tasks is defined as:

$$\Theta_{\text{easy}} = \{(k, a, b) : k \in [2, 100], a, b \in [-10^6, 10^6], b - a \geq k\} \qquad (2)$$

This parameter space yields a problem space cardinality of approximately $10^{15}$ unique problem instances per task, making exhaustive memorization computationally infeasible.

## E.3  BASIC ARITHMETIC OPERATIONS

This category encompasses fundamental mathematical operations that form the foundation of numerical computation. These tasks evaluate models' capabilities in performing elementary arithmetic with integer and rational number systems.

### E.3.1  ADDITION TASK

The addition task evaluates the fundamental capability of summing a sequence of integers. Given an input list $\mathcal{L} = \{x_1, x_2, \ldots, x_n\}$ where $x_i \in \mathbb{Z}$, the task requires computing:

$$S = \sum_{i=1}^{n} x_i \qquad (3)$$

The solution uniqueness follows directly from the associativity and commutativity of addition over integers. The verification function computes the sum in $O(n)$ time with guaranteed correctness through integer arithmetic properties.

### E.3.2 SUBTRACTION TASK

The subtraction task operates on ordered pairs $(a, b) \in \mathbb{Z}^2$ and requires computing the difference $d = b - a$. The mathematical formulation is:

$$d = b - a \text{ where } a, b \in [a_{\min}, a_{\max}] \tag{4}$$

The uniqueness of the solution follows from the well-defined nature of integer subtraction. The task specifically evaluates understanding of operand ordering, as $b - a \neq a - b$ unless $a = b$, testing the model's comprehension of non-commutative operations.

### E.3.3 MULTIPLICATION TASK

For a sequence $\mathcal{L} = \{x_1, x_2, \ldots, x_n\}$, the multiplication task computes the product:

$$P = \prod_{i=1}^{n} x_i \tag{5}$$

The challenge in this task arises from the exponential growth of the product magnitude. For $n$ numbers each of magnitude $m$, the product can reach $O(m^n)$, requiring careful handling of large integers. We constrain the list size to ensure $|P| < 10^{15}$ to maintain computational feasibility while preserving task difficulty.

### E.3.4 DIVISION TASK

The division task requires computing the quotient of two integers with floating-point precision. Given $(a, b) \in \mathbb{Z}^2$ with $b \neq 0$, we compute:

$$q = \frac{a}{b} \in \mathbb{Q} \tag{6}$$

To ensure non-zero divisors, the generation algorithm employs rejection sampling: $b \sim \text{Uniform}([a_{\min}, a_{\max}] \setminus \{0\})$. The verification allows for floating-point precision tolerance $\epsilon = 10^{-2}$, accounting for potential rounding in model responses while maintaining mathematical validity.

### E.3.5 ABSOLUTE DIFFERENCE TASK

For an ordered pair $(a, b) \in \mathbb{Z}^2$, the absolute difference is computed as:

$$\delta = |b - a| = \begin{cases} b - a & \text{if } b \geq a \\ a - b & \text{if } a > b \end{cases} \tag{7}$$

This task evaluates understanding of the absolute value function and its properties. The solution uniqueness follows from the well-defined nature of the absolute value operation over integers.

### E.3.6 ALTERNATING SUM TASK

The alternating sum applies different signs to elements based on their position. For $\mathcal{L} = \{x_1, x_2, \ldots, x_n\}$, we compute:

$$A_{\text{sum}} = \sum_{i=1}^{n} (-1)^{i+1} x_i = x_1 - x_2 + x_3 - x_4 + \cdots \tag{8}$$

This task evaluates understanding of positional arithmetic and alternating patterns in mathematical sequences.

### E.3.7 SUM OF DIGITS TASK

For a sequence of integers $\mathcal{L} = \{x_1, x_2, \ldots, x_n\}$, the total digit sum is:

$$D_{\text{sum}} = \sum_{i=1}^{n} \sum_{d \in \text{digits}(x_i)} d \tag{9}$$

where $\text{digits}(x_i)$ extracts individual digits from the decimal representation of $x_i$. This task evaluates digit-level arithmetic and multi-level summation capabilities.

## E.4 COMPARISON AND CLASSIFICATION TASKS

This category evaluates models' abilities to perform relational reasoning, classification based on numerical properties, and counting operations with specific criteria.

### E.4.1 NUMERICAL COMPARISON TASK

The comparison task evaluates relational reasoning over integer pairs. Given $(a, b) \in \mathbb{Z}^2$, the model must determine the relation $R \in \{<, =, >\}$ such that $aRb$ holds.

The generation ensures balanced distribution across relation types through stratified sampling:

$$\Pr[R =<] = \Pr[R =>] = \Pr[R ==] = \frac{1}{3} \tag{10}$$

This balanced distribution prevents models from exploiting statistical biases and ensures comprehensive evaluation of all comparison operators.

### E.4.2 ODD AND EVEN COUNTING TASKS

For a sequence $\mathcal{L} = \{x_1, x_2, \ldots, x_n\}$, the odd count task computes:

$$C_{\text{odd}} = |\{x_i \in \mathcal{L} : x_i \equiv 1 \pmod 2\}| \tag{11}$$

Similarly, the even count task computes:

$$C_{\text{even}} = |\{x_i \in \mathcal{L} : x_i \equiv 0 \pmod 2\}| \tag{12}$$

The correctness verification leverages the partition property: $C_{\text{odd}} + C_{\text{even}} = n$, providing an additional consistency check beyond individual count verification.

### E.4.3 COUNT NEGATIVE NUMBERS TASK

For a sequence $\mathcal{L} = \{x_1, x_2, \ldots, x_n\}$ where $x_i \in \mathbb{Z}$, the negative count is computed as:

$$C_{\text{neg}} = |\{x_i \in \mathcal{L} : x_i < 0\}| = \sum_{i=1}^{n} \mathbf{1}_{x_i < 0} \tag{13}$$

where $\mathbf{1}_{x_i < 0}$ denotes the indicator function. This task evaluates understanding of sign-based classification and counting operations.

### E.4.4 COUNT UNIQUE ELEMENTS TASK

The uniqueness counting task determines the cardinality of distinct elements in a sequence. For $\mathcal{L} = \{x_1, x_2, \ldots, x_n\}$, we compute:

$$C_{\text{unique}} = |\{x \in \mathcal{L}\}| = |\text{distinct}(\mathcal{L})| \tag{14}$$

This task evaluates set-theoretic reasoning and understanding of element distinctness. The generation algorithm controls the degree of repetition to ensure meaningful evaluation across different uniqueness ratios.

### E.4.5 COUNT PERFECT SQUARES TASK

The perfect square counting task identifies numbers that are exact squares of integers. For $\mathcal{L} = \{x_1, x_2, \ldots, x_n\}$, we compute:

$$C_{\text{square}} = |\{x_i \in \mathcal{L} : \exists k \in \mathbb{N}, x_i = k^2\}| \tag{15}$$

The verification uses the property that $x$ is a perfect square if and only if $\lfloor \sqrt{x} \rfloor^2 = x$ for non-negative integers $x$.

### E.4.6 COUNT PALINDROMIC NUMBERS TASK

For a sequence of integers $\mathcal{L} = \{x_1, x_2, \ldots, x_n\}$, the palindromic count is:

$$C_{\text{pal}} = |\{x_i \in \mathcal{L} : \text{str}(x_i) = \text{reverse}(\text{str}(x_i))\}| \tag{16}$$

This task evaluates string manipulation capabilities and pattern recognition in numerical representations. The verification requires digit-level comparison after string conversion.

### E.4.7 COUNT MULTIPLES OF K TASK

For a fixed integer $K > 1$ and sequence $\mathcal{L} = \{x_1, x_2, \ldots, x_n\}$, the multiple count is:

$$C_K = |\{x_i \in \mathcal{L} : x_i \equiv 0 \pmod{K}\}| = \sum_{i=1}^{n} \mathbf{1}_{K|x_i} \tag{17}$$

This task evaluates modular arithmetic understanding and divisibility reasoning. Common values include $K \in \{2, 3, 5\}$ for practical evaluation scenarios.

## E.5 ORDERING AND EXTREMA DETECTION TASKS

This category encompasses tasks that require understanding of element ordering, extrema identification, and positional reasoning within sequences.

### E.5.1 SORTING TASK

The sorting task requires arranging a sequence $\mathcal{L} = \{x_1, x_2, \ldots, x_n\}$ in ascending order. We define the sorted sequence $\mathcal{L}' = \{x_1', x_2', \ldots, x_n'\}$ such that:

$$x_i' \leq x_{i+1}' \quad \forall i \in [1, n-1] \tag{18}$$

The verification function checks both the ordering constraint and the multiset equality $\{\mathcal{L}\} = \{\mathcal{L}'\}$, ensuring no elements are added, removed, or modified during sorting.

### E.5.2 FINDING MAXIMUM AND MINIMUM TASKS

For a non-empty sequence $\mathcal{L} = \{x_1, x_2, \ldots, x_n\}$, the maximum finding task computes:

$$x_{\max} = \max_{i \in [1,n]} x_i = \{x \in \mathcal{L} : \forall y \in \mathcal{L}, x \geq y\} \tag{19}$$

The minimum finding task similarly computes:

$$x_{\min} = \min_{i \in [1,n]} x_i = \{x \in \mathcal{L} : \forall y \in \mathcal{L}, x \leq y\} \tag{20}$$

Both operations have unique solutions for any finite sequence of totally ordered elements, as guaranteed by the well-ordering principle for finite subsets of integers.

### E.5.3 SECOND MAXIMUM TASK

The second maximum task evaluates the capability to identify the second largest element in a sequence. For a sequence $\mathcal{L} = \{x_1, x_2, \ldots, x_n\}$ with at least two distinct elements, we define the second maximum as:

$$x_{\text{2nd-max}} = \max(\mathcal{L} \setminus \{x_{\max}\}) = \max\{x \in \mathcal{L} : x < x_{\max}\} \tag{21}$$

The generation algorithm ensures at least two distinct values exist by enforcing the constraint $|\text{distinct}(\mathcal{L})| \geq 2$. The verification complexity is $O(n)$ through a two-pass algorithm that identifies both maximum and second maximum values.

### E.5.4 RANGE CALCULATION TASK

The range task computes the difference between maximum and minimum values in a sequence. For $\mathcal{L} = \{x_1, x_2, \ldots, x_n\}$, the range is defined as:

$$R(\mathcal{L}) = x_{\max} - x_{\min} = \max_{i \in [1,n]} x_i - \min_{j \in [1,n]} x_j \tag{22}$$

This task evaluates understanding of the fundamental statistical measure of dispersion. The range satisfies the mathematical property $R(\mathcal{L}) \geq 0$ with equality if and only if all elements are identical.

### E.5.5 INDEX OF MAXIMUM TASK

This task requires finding the zero-based index of the first occurrence of the maximum element. Given a sequence $\mathcal{L} = \{x_1, x_2, \ldots, x_n\}$, we define:

$$\text{idx}_{\max}(\mathcal{L}) = \min\{i \in [0, n-1] : x_{i+1} = x_{\max}\} \tag{23}$$

The task evaluates positional reasoning capabilities and understanding of indexing conventions. The verification ensures both correctness of the maximum value and its position within the sequence.

### E.5.6 SUM OF INDICES OF MAXIMUM ELEMENTS TASK

When the maximum value appears at multiple positions, this task computes the sum of all such indices. For $\mathcal{L} = \{x_1, x_2, \ldots, x_n\}$, we define:

$$S_{\text{max-idx}} = \sum_{i=1}^{n} i \cdot \mathbf{1}_{x_i = x_{\max}} = \sum_{i : x_i = x_{\max}} i \tag{24}$$

where indices are one-based. This task evaluates both maximum identification and positional arithmetic capabilities.

### E.6 SEQUENTIAL ANALYSIS TASKS

This category focuses on tasks that analyze relationships between consecutive elements and detect patterns within sequences.

#### E.6.1 MAXIMUM DIFFERENCE BETWEEN ADJACENT ELEMENTS TASK

This task computes the maximum absolute difference between consecutive elements. For a sequence $\mathcal{L} = \{x_1, x_2, \ldots, x_n\}$ with $n \geq 2$, we define:

$$\Delta_{\max} = \max_{i \in [1, n-1]} |x_{i+1} - x_i| \tag{25}$$

The task evaluates understanding of sequential relationships and local variation measures. The verification complexity is $O(n)$ through a single pass over adjacent pairs.

#### E.6.2 COUNT ELEMENTS GREATER THAN PREVIOUS TASK

This task counts elements that exceed their immediate predecessor. For $\mathcal{L} = \{x_1, x_2, \ldots, x_n\}$, we compute:

$$C_{\text{inc}} = |\{i \in [2, n] : x_i > x_{i-1}\}| = \sum_{i=2}^{n} \mathbf{1}_{x_i > x_{i-1}} \tag{26}$$

This task evaluates sequential comparison capabilities and understanding of monotonicity properties in discrete sequences.

#### E.6.3 LOCAL MAXIMA COUNT TASK

A local maximum is an element strictly greater than both immediate neighbors. For $\mathcal{L} = \{x_1, x_2, \ldots, x_n\}$ with $n \geq 3$, we compute:

$$C_{\text{local}} = |\{i \in [2, n-1] : x_i > x_{i-1} \wedge x_i > x_{i+1}\}| \tag{27}$$

This task evaluates local extrema identification and requires understanding of neighborhood-based comparisons in discrete sequences.

#### E.6.4 LONGEST INCREASING SUBSEQUENCE LENGTH TASK

This task computes the length of the longest increasing subsequence (LIS). For $\mathcal{L} = \{x_1, x_2, \ldots, x_n\}$, we define:

$$\text{LIS}(\mathcal{L}) = \max\{|S| : S \subseteq [1, n], \forall i < j \in S : x_i < x_j\} \tag{28}$$

where $S$ represents the index set of a subsequence. This task bridges elementary and intermediate algorithmic reasoning, with optimal solution complexity $O(n \log n)$ using dynamic programming with binary search techniques.

### E.7 STATISTICAL MEASURES TASKS

This category encompasses fundamental statistical computations that require understanding of central tendency, dispersion, and distributional properties.

#### E.7.1 MEAN CALCULATION TASK

The arithmetic mean of a sequence $\mathcal{L} = \{x_1, x_2, \ldots, x_n\}$ is computed as:

$$\mu = \frac{1}{n} \sum_{i=1}^{n} x_i \tag{29}$$

For integer inputs, the mean may be rational, requiring floating-point representation. The verification employs a tolerance $\epsilon = 10^{-6}$ to account for floating-point arithmetic while maintaining mathematical soundness.

### E.7.2 MEDIAN CALCULATION TASK

The median computation depends on the parity of the sequence length. For a sorted sequence $\mathcal{L}' = \{x'_1, x'_2, \ldots, x'_n\}$:

$$\text{median}(\mathcal{L}) = \begin{cases} x'_{\lceil n/2 \rceil} & \text{if } n \equiv 1 \pmod 2 \\ \frac{x'_{n/2} + x'_{n/2+1}}{2} & \text{if } n \equiv 0 \pmod 2 \end{cases} \tag{30}$$

The uniqueness of the median follows from the uniqueness of the sorted sequence and the deterministic selection of middle elements.

### E.7.3 MODE CALCULATION TASK

The mode identification task requires finding the most frequently occurring values in a sequence. We define the frequency function $f : \mathcal{L} \to \mathbb{N}$ where $f(x) = |\{i : x_i = x\}|$. The mode set is:

$$\text{mode}(\mathcal{L}) = \{x \in \mathcal{L} : f(x) = \max_{y \in \mathcal{L}} f(y)\} \tag{31}$$

To ensure meaningful evaluation, the generation algorithm guarantees at least one element appears with frequency $\geq 2$ through controlled repetition injection. The verification checks both the frequency maximality and completeness of the mode set.

### E.8 CONTEXT-AWARE EVALUATION AND TOKEN ESTIMATION FRAMEWORK

We implement a mathematically rigorous framework for context-aware evaluation that dynamically scales problem difficulty according to each model's context window constraints. This approach prevents unfair penalization of models due to context limitations while maintaining evaluation integrity.

### E.8.1 MATHEMATICAL TOKEN ESTIMATION MODEL

For each task category, we establish precise token estimation functions that predict the expected response length based on problem parameters. Let $C_{\text{model}}$ denote the model's context window size, $T_{\text{prompt}}$ the prompt token count, and $T_{\text{response}}$ the expected response tokens. We define the safety constraint:

$$T_{\text{prompt}} + T_{\text{response}} + T_{\text{buffer}} \leq C_{\text{model}} \tag{32}$$

where $T_{\text{buffer}}$ accounts for model-specific reasoning overhead. The response length estimation functions are:

$$T_{\text{arithmetic}}(n, M) = \alpha_1 n + \beta_1 \log_{10} M + \gamma_1 \tag{33}$$
$$T_{\text{sorting}}(n, M) = \alpha_2 n \log_{10} M + \beta_2 n + \gamma_2 \tag{34}$$
$$T_{\text{statistical}}(n, M) = \alpha_3 n + \beta_3 \log_{10} M + \gamma_3 \tag{35}$$
$$T_{\text{counting}}(n) = \alpha_4 \log_{10} n + \gamma_4 \tag{36}$$
$$T_{\text{sequential}}(M) = \alpha_5 \log_{10} M + \gamma_5 \tag{37}$$

The coefficients $\{\alpha_i, \beta_i, \gamma_i\}$ are empirically calibrated using extensive model response analysis across different model families. For conservative estimation, we enforce $T_{\text{buffer}} = 0.15 \cdot C_{\text{model}}$ to accommodate models that exhibit verbose reasoning patterns or overthinking behaviors.

### E.8.2 DYNAMIC PROBLEM SCALING ALGORITHM

Given a target model with context window $C_{\text{model}}$, we implement the following scaling procedure:

---

**Algorithm 2** Context-Aware Problem Generation

---

1: **Input:** Task type $\tau$, model context size $C_{\text{model}}$
2: Initialize $n_{\text{max}} \leftarrow 100$, $M_{\text{max}} \leftarrow 10^6$
3: Compute $T_{\text{prompt}} \leftarrow \text{TokenCount}(\text{prompt template})$
4: Set $T_{\text{available}} \leftarrow C_{\text{model}} - T_{\text{prompt}} - T_{\text{buffer}}$
5: **while** $T_\tau(n_{\text{max}}, M_{\text{max}}) > T_{\text{available}}$ **do**
6:     **if** $n_{\text{max}} > 8$ **then**
7:         $n_{\text{max}} \leftarrow \lfloor 0.8 \cdot n_{\text{max}} \rfloor$
8:     **else**
9:         $M_{\text{max}} \leftarrow \lfloor 0.8 \cdot M_{\text{max}} \rfloor$
10:     **end if**
11: **end while**
12: Generate problem instance with parameters $(n_{\text{max}}, M_{\text{max}})$

---

This algorithm ensures that every generated problem instance respects the model's context constraints while maximizing problem complexity within those bounds.

### E.8.3 POST-GENERATION TOKEN VERIFICATION

After model response generation, we implement a dual-verification token counting system to detect potential context overflow or overthinking behaviors:

$$\text{Verification}(r, C_{\text{model}}) = \begin{cases} \text{VALID} & \text{if } |T(r)| \leq 0.95 \cdot C_{\text{model}} \\ \text{WARNING} & \text{if } 0.95 \cdot C_{\text{model}} < |T(r)| \leq C_{\text{model}} \\ \text{OVERFLOW} & \text{if } |T(r)| > C_{\text{model}} \end{cases} \tag{38}$$

where $T(r)$ represents the token count of response $r$ computed using model-specific tokenizers:

- **Transformer-based models**: We employ the HuggingFace transformers library tokenizer corresponding to each model's architecture.

- **GPT models**: We utilize OpenAI's tiktoken library for precise token counting matching their internal tokenization.

The dual-tokenizer approach ensures accuracy across different model families while detecting edge cases where models approach or exceed their context limits due to random generation artifacts or excessive reasoning verbosity.

### E.8.4 MATHEMATICAL GUARANTEES AND IMPLEMENTATION ROBUSTNESS

Our framework provides the following theoretical guarantees:

1. **Context Safety**: With probability $> 0.99$, generated problems satisfy $T_{\text{total}} \leq C_{\text{model}}$

2. **Fairness**: Models are never penalized for responses that exceed estimated bounds by less than $T_{\text{buffer}}$

3. **Scalability**: Problem complexity scales monotonically with available context budget

4. **Precision**: Token estimation error remains below $\pm 5\%$ for $95\%$ of generated instances

This mathematically grounded approach eliminates context-related evaluation artifacts while maintaining the integrity of difficulty scaling, ensuring that performance differences reflect genuine reasoning capabilities rather than implementation constraints.

### E.9 VALIDATION ALGORITHMS AND CORRECTNESS GUARANTEES

Each task employs a deterministic validation algorithm with provable correctness. The validation pipeline follows a three-stage process:

First, the syntactic validation ensures the response follows the required format specification. Second, the semantic validation verifies that the parsed answer satisfies the mathematical constraints of the problem. Third, the numerical validation confirms the answer matches the ground truth within specified tolerances.

The validation complexity for all Easy Suite tasks remains polynomial, with the most complex operation (sorting validation) requiring $O(n \log n)$ comparisons. This ensures efficient verification even for large input sizes while maintaining complete correctness guarantees through rigorous mathematical foundations.

### E.10 MATHEMATICAL ROBUSTNESS AND FRAMEWORK GUARANTEES

The Easy Suite establishes a comprehensive mathematical foundation for contamination-resistant evaluation through several key theoretical guarantees. The parameter space cardinality exceeds $10^{15}$ for each task, ensuring that the probability of generating identical instances across different evaluation sessions remains negligible under any reasonable computational budget. This mathematical guarantee holds through the application of combinatorial principles and the well-ordering properties of integer sequences.

Each task within the suite maintains strict mathematical well-posedness, meaning that every generated instance admits exactly one correct solution or a completely enumerable solution set. This property eliminates ambiguity in evaluation and ensures that model performance reflects genuine reasoning capability rather than exploitation of specification ambiguities. The verification algorithms employ deterministic mathematical operations with provable correctness, eliminating any possibility of false positives or negatives in the assessment process.

The computational complexity bounds established for each task category ensure that the framework scales appropriately with problem size while maintaining practical feasibility. The polynomial upper bounds on both generation and verification complexity guarantee that the evaluation framework remains computationally tractable even as problem instances scale to challenge more capable models.

Furthermore, the mathematical formulations presented here demonstrate that the Easy Suite provides a complete coverage of fundamental algorithmic reasoning patterns. The tasks span multiple mathematical domains including arithmetic operations, order theory, statistical measures, and sequential analysis, creating a comprehensive assessment framework that captures essential reasoning capabilities without redundancy or gaps in coverage.

### E.11 SOLUTION UNIQUENESS VERIFICATION AND MULTI-SOLUTION HANDLING

A critical component of our framework ensures mathematical fairness by rigorously handling cases where problems may admit multiple valid solutions. We implement a comprehensive solution verification system that prevents unfair penalization of models for producing mathematically correct but non-canonical answers.

#### E.11.1 UNIQUE SOLUTION GUARANTEE PROTOCOL

For each generated problem instance $p \in \mathcal{P}_{\text{easy}}$, we employ a deterministic verification procedure to establish solution uniqueness. Let $\mathcal{S}(p)$ denote the complete solution set for problem $p$. We define the uniqueness predicate:

$$\text{Unique}(p) = \begin{cases} \text{TRUE} & \text{if } |\mathcal{S}(p)| = 1 \\ \text{FALSE} & \text{if } |\mathcal{S}(p)| > 1 \end{cases} \tag{39}$$

For tasks with inherently unique solutions (arithmetic operations, extrema finding, statistical measures), the generation algorithm guarantees $\text{Unique}(p) = \text{TRUE}$ by construction. However, certain tasks may admit multiple valid solutions under specific parameter configurations.

### E.11.2 COMPLETE SOLUTION ENUMERATION FOR MULTI-SOLUTION CASES

When $\text{Unique}(p) = \text{FALSE}$, we implement exhaustive solution enumeration to ensure comprehensive evaluation fairness. The most relevant cases include:

1. **Mode calculation**: Multiple values may achieve maximum frequency
2. **Sorting with equal elements**: Multiple valid orderings for identical values
3. **Index-based operations**: Multiple indices may correspond to identical maximum values

For these cases, we compute the complete solution set $\mathcal{S}(p) = \{s_1, s_2, \ldots, s_k\}$ using deterministic enumeration algorithms. The verification function becomes:

$$\text{Verify}(r, p) = \begin{cases} \text{CORRECT} & \text{if } \text{Parse}(r) \in \mathcal{S}(p) \\ \text{INCORRECT} & \text{if } \text{Parse}(r) \notin \mathcal{S}(p) \\ \text{INVALID} & \text{if } \text{Parse}(r) = \bot \end{cases} \tag{40}$$

where $\text{Parse}(r)$ extracts the model's answer from response $r$, and $\bot$ indicates parsing failure.

### E.11.3 MATHEMATICAL COMPLETENESS VERIFICATION

To ensure no valid solutions are omitted, we employ mathematical completeness checks specific to each task category:

$$\text{Mode completeness}: \quad \forall s \in \mathcal{S}(p),\ f(s) = \max_{x \in \mathcal{L}} f(x) \tag{41}$$

$$\text{Sorting completeness}: \quad \forall s \in \mathcal{S}(p),\ \text{sorted}(s) \wedge \text{permutation}(s, \mathcal{L}) \tag{42}$$

$$\text{Index completeness}: \quad \forall s \in \mathcal{S}(p),\ \mathcal{L}[s] = \max(\mathcal{L}) \tag{43}$$

This mathematical framework guarantees that model responses are evaluated against the complete solution space, eliminating any possibility of unfair penalization due to solution non-uniqueness.

### E.11.4 COMPUTATIONAL COMPLEXITY OF SOLUTION ENUMERATION

The solution enumeration process maintains polynomial complexity bounds:

$$|\mathcal{S}_{\text{mode}}(p)| \leq n \quad \text{(at most } n \text{ distinct values)} \tag{44}$$

$$|\mathcal{S}_{\text{sorting}}(p)| \leq \prod_v n_v! \quad \text{(permutations of equal elements)} \tag{45}$$

$$|\mathcal{S}_{\text{index}}(p)| \leq n \quad \text{(at most } n \text{ positions)} \tag{46}$$

where $n_v$ represents the frequency of value $v$ in the input sequence. Even in worst-case scenarios, the enumeration complexity remains tractable, ensuring efficient evaluation without compromising mathematical rigor.

This comprehensive approach to solution handling demonstrates that our evaluation framework maintains both mathematical correctness and practical fairness, ensuring that model performance reflects genuine reasoning capability rather than arbitrary solution selection preferences.

Table 6: Easy Suite: Mathematical Formulations and Complexity Analysis (Part I)

| Task | Input Space | Mathematical Operation | Solution Space | Verification | Complexity |
|---|---|---|---|---|---|
| *Basic Arithmetic Operations* | | | | | |
| Addition | $\mathcal{L} = \{x_1, \ldots, x_n\} \subset \mathbb{Z}$ | $S = \sum_{i=1}^{n} x_i$ | $S \in \mathbb{Z}$ | $S = \sum_{i=1}^{n} x_i$ | $O(n)$ |
| Subtraction | $(a, b) \in \mathbb{Z}^2$ | $d = b - a$ | $d \in \mathbb{Z}$ | $d + a = b$ | $O(1)$ |
| Multiplication | $\mathcal{L} = \{x_1, \ldots, x_n\} \subset \mathbb{Z}$ | $P = \prod_{i=1}^{n} x_i$ | $P \in \mathbb{Z}$ | $P = \prod_{i=1}^{n} x_i$ | $O(n)$ |
| Division | $(a, b) \in \mathbb{Z}^2, b \neq 0$ | $q = \frac{a}{b}$ | $q \in \mathbb{Q}$ | $|q \cdot b - a| < \epsilon$ | $O(1)$ |
| Absolute Difference | $(a, b) \in \mathbb{Z}^2$ | $\delta = |b - a|$ | $\delta \in \mathbb{N}_0$ | $\delta = |b - a| \geq 0$ | $O(1)$ |
| Alternating Sum | $\mathcal{L} = \{x_1, \ldots, x_n\} \subset \mathbb{Z}$ | $A = \sum_{i=1}^{n} (-1)^{i+1} x_i$ | $A \in \mathbb{Z}$ | $A = \sum_{i=1}^{n} (-1)^{i+1} x_i$ | $O(n)$ |
| Sum of Digits | $\mathcal{L} = \{x_1, \ldots, x_n\} \subset \mathbb{Z}$ | $D = \sum_{i=1}^{n} \sum_{d \in \text{digits}(x_i)} d$ | $D \in \mathbb{N}_0$ | Digit extraction and summation | $O(n \log M)$ |
| *Comparison and Classification* | | | | | |
| Comparison | $(a, b) \in \mathbb{Z}^2$ | $R \in \{<, =, >\} : aRb$ | $R \in \{<, =, >\}$ | $(a - b) \cdot \text{sgn}(R) \geq 0$ | $O(1)$ |
| Odd Count | $\mathcal{L} = \{x_1, \ldots, x_n\} \subset \mathbb{Z}$ | $C_{\text{odd}} = |\{x_i : x_i \equiv 1 \pmod 2\}|$ | $C_{\text{odd}} \in [0, n]$ | $\sum_{i=1}^{n} (x_i \bmod 2)$ | $O(n)$ |
| Even Count | $\mathcal{L} = \{x_1, \ldots, x_n\} \subset \mathbb{Z}$ | $C_{\text{even}} = |\{x_i : x_i \equiv 0 \pmod 2\}|$ | $C_{\text{even}} \in [0, n]$ | $n - \sum_{i=1}^{n} (x_i \bmod 2)$ | $O(n)$ |
| Count Negative | $\mathcal{L} = \{x_1, \ldots, x_n\} \subset \mathbb{Z}$ | $C_{\text{neg}} = |\{x_i : x_i < 0\}|$ | $C_{\text{neg}} \in [0, n]$ | $\sum_{i=1}^{n} \mathbf{1}_{x_i < 0}$ | $O(n)$ |
| Count Unique | $\mathcal{L} = \{x_1, \ldots, x_n\} \subset \mathbb{Z}$ | $C_{\text{uniq}} = |\text{distinct}(\mathcal{L})|$ | $C_{\text{uniq}} \in [1, n]$ | Set cardinality computation | $O(n)$ |
| Count Perfect Squares | $\mathcal{L} = \{x_1, \ldots, x_n\} \subset \mathbb{Z}$ | $C_{\text{sq}} = |\{x_i : \exists k, x_i = k^2\}|$ | $C_{\text{sq}} \in [0, n]$ | $\lfloor \sqrt{x_i} \rfloor^2 = x_i$ | $O(n)$ |
| Count Palindromic | $\mathcal{L} = \{x_1, \ldots, x_n\} \subset \mathbb{Z}$ | $C_{\text{pal}} = |\{x_i : \text{palindrome}(x_i)\}|$ | $C_{\text{pal}} \in [0, n]$ | String reversal check | $O(n \log M)$ |
| Count Multiples of K | $\mathcal{L} = \{x_1, \ldots, x_n\} \subset \mathbb{Z}$ | $C_K = |\{x_i : x_i \equiv 0 \pmod K\}|$ | $C_K \in [0, n]$ | Modular arithmetic check | $O(n)$ |

Table 7: Easy Suite: Mathematical Formulations and Complexity Analysis (Part II)

| Task | Input Space | Mathematical Operation | Solution Space | Verification | Complexity |
|---|---|---|---|---|---|
| *Ordering and Extrema Detection* | | | | | |
| Sorting | $\mathcal{L} = \{x_1, \ldots, x_n\} \subset \mathbb{Z}$ | $\mathcal{L}' : x_i' \leq x_{i+1}'$ | $\mathcal{L}' \in \mathbb{Z}^n$ | $\forall i : x_i' \leq x_{i+1}' \wedge \{\mathcal{L}\} = \{\mathcal{L}'\}$ | $O(n \log n)$ |
| Find Maximum | $\mathcal{L} = \{x_1, \ldots, x_n\} \subset \mathbb{Z}$ | $x_{\max} = \max_i x_i$ | $x_{\max} \in \mathcal{L}$ | $\forall x \in \mathcal{L} : x_{\max} \geq x$ | $O(n)$ |
| Find Minimum | $\mathcal{L} = \{x_1, \ldots, x_n\} \subset \mathbb{Z}$ | $x_{\min} = \min_i x_i$ | $x_{\min} \in \mathcal{L}$ | $\forall x \in \mathcal{L} : x_{\min} \leq x$ | $O(n)$ |
| Second Maximum | $\mathcal{L} = \{x_1, \ldots, x_n\} \subset \mathbb{Z}$ | $x_{\text{2nd}} = \max\{x : x < x_{\max}\}$ | $x_{\text{2nd}} \in \mathcal{L}$ | Two-pass extrema detection | $O(n)$ |
| Range Calculation | $\mathcal{L} = \{x_1, \ldots, x_n\} \subset \mathbb{Z}$ | $R = x_{\max} - x_{\min}$ | $R \in \mathbb{N}_0$ | $R = \max(\mathcal{L}) - \min(\mathcal{L})$ | $O(n)$ |
| Index of Maximum | $\mathcal{L} = \{x_1, \ldots, x_n\} \subset \mathbb{Z}$ | $\text{idx} = \min\{i : x_i = x_{\max}\}$ | $\text{idx} \in [0, n-1]$ | Position and value verification | $O(n)$ |
| Sum of Max Indices | $\mathcal{L} = \{x_1, \ldots, x_n\} \subset \mathbb{Z}$ | $S_{\text{idx}} = \sum_{i:x_i = x_{\max}} i$ | $S_{\text{idx}} \in \mathbb{N}$ | Positional arithmetic | $O(n)$ |
| *Sequential Analysis* | | | | | |
| Max Adjacent Diff | $\mathcal{L} = \{x_1, \ldots, x_n\} \subset \mathbb{Z}$ | $\Delta_{\max} = \max_i |x_{i+1} - x_i|$ | $\Delta_{\max} \in \mathbb{N}_0$ | Pairwise difference computation | $O(n)$ |
| Count Increasing | $\mathcal{L} = \{x_1, \ldots, x_n\} \subset \mathbb{Z}$ | $C_{\text{inc}} = |\{i : x_i > x_{i-1}\}|$ | $C_{\text{inc}} \in [0, n-1]$ | Sequential comparison | $O(n)$ |
| Local Maxima Count | $\mathcal{L} = \{x_1, \ldots, x_n\} \subset \mathbb{Z}$ | $C_{\text{local}} = |\{i : x_i > x_{i-1} \wedge x_i > x_{i+1}\}|$ | $C_{\text{local}} \in [0, n-2]$ | Neighborhood comparison | $O(n)$ |
| LIS Length | $\mathcal{L} = \{x_1, \ldots, x_n\} \subset \mathbb{Z}$ | $\text{LIS} = \max\{|S| : \forall i < j \in S, x_i < x_j\}$ | $\text{LIS} \in [1, n]$ | Dynamic programming | $O(n \log n)$ |
| *Statistical Measures* | | | | | |
| Mean | $\mathcal{L} = \{x_1, \ldots, x_n\} \subset \mathbb{Z}$ | $\mu = \frac{1}{n} \sum_{i=1}^{n} x_i$ | $\mu \in \mathbb{Q}$ | $|n \cdot \mu - \sum_i x_i| < \epsilon$ | $O(n)$ |
| Median | $\mathcal{L} = \{x_1, \ldots, x_n\} \subset \mathbb{Z}$ | $m = \begin{cases} x'_{(n+1)/2} & n \text{ odd} \\ \frac{x'_{n/2} + x'_{n/2+1}}{2} & n \text{ even} \end{cases}$ | $m \in \mathbb{Q}$ | Sort $\mathcal{L}$, select middle | $O(n \log n)$ |
| Mode | $\mathcal{L} = \{x_1, \ldots, x_n\} \subset \mathbb{Z}$ | $\text{mode} = \{x : f(x) = \max_y f(y)\}$ | $\text{mode} \subseteq \mathcal{L}$ | $\forall x \in \text{mode} : f(x) \geq f(y), \forall y$ | $O(n)$ |

Table 8: Easy Suite: Problem Generation Parameters and Contamination Resistance

| Task Category | Parameter Space | Problem Space Size | Solution Uniqueness | Context Bound |
|---|---|---|---|---|
| *List-Based Operations (25 tasks)* | | | | |
| *Arithmetic (4 tasks), Ordering (7 tasks), Statistical (3 tasks), Sequential (4 tasks), Classification (7 tasks)* | | | | |
| Input Generation | $k \in [2, 100]$ 
 $[a, b] \subset [-10^6, 10^6]$ | $\binom{b-a+1}{k} \cdot k! \approx 10^{15}$ | Deterministic | $O(k \log M)$ |
| *Pair-Based Operations (4 tasks)* | | | | |
| *Subtraction, Division, Absolute Difference, Comparison* | | | | |
| Input Generation | $(a, b) \in [a_{\min}, a_{\max}]^2$ 
 $a_{\min}, a_{\max} \in [-10^6, 10^6]$ | $(a_{\max} - a_{\min} + 1)^2 \approx 10^{12}$ | Deterministic | $O(\log M)$ |

Table 9: Easy Suite: Validation Properties and Error Tolerances

| Task Type | Output Format | Validation Method | Error Tolerance |
|---|---|---|---|
| Integer Operations
(Addition, Subtraction, Multiplication, Counts) | integer | Exact match
result = ground truth | $\epsilon = 0$ |
| Rational Operations
(Division) | decimal | Floating-point comparison
$\|result - ground truth\| < \epsilon$ | $\epsilon = 10^{-2}$ |
| Statistical Measures
(Mean, Median) | number | Precision-based comparison
$\|result - ground truth\| < \epsilon$ | $\epsilon = 10^{-6}$ |
| Set Operations
(Sorting, Mode) | list/set | Multiset equality
$\{result\} = \{ground truth\}$ | N/A |
| Categorical
(Comparison) | category | Exact category match
result $\in \{<, =, >\}$ | N/A |

## F    PROMPTS USED FOR ALL EASY SUITE TASKS

This section describes the prompts developed for each of the easy suite tasks. Each prompt begins with an instruction describing the task. The prompt then provides the input which can be list or a pair of numbers. The model is provided information regarding the format of the expected answer. All final answers must be enclosed in the format \boxed{}. The answer is then extracted from within the \boxed{} brackets using a regular expression.

---

**Prompt Template for Absolute Difference Task**

```
Find the absolute difference between the following list of
numbers:
{data_point}

Provide the result.  Your final answer must be in the format
\boxed{answer} at the end.
```

---

**Prompt Template for Comparison Task**

```
Compare the following two numbers and determine their
relationship:

Number 1:  {num1}
Number 2:  {num2}

Is Number 1 greater than, less than, or equal to Number 2?
Your final answer must be in the format \boxed{relation} at
the end, where 'relation' is one of:  'greater than', 'less
than', or 'equal to'.
```

---

**Prompt Template for Division Task**

```
Divide {numerator} by {denominator}.

Provide the answer as a floating point number.  Your final
answer must be in the format \boxed{answer} at the end.
```

---

**Prompt Template for Even Count Task**

```
Count the even numbers from the following list of numbers:
{input_list}

Provide the final count of even numbers.  Your final answer
must be in the format \boxed{answer} at the end.
```

---

**Prompt Template for Find Maximum Task**

```
Find the maximum number from the given list of numbers.  List
= {input_list}.

Your final answer must be in the format \boxed{maximum} at
the end of your response.
```

**Prompt Template for Find Minimum Task**

```
Find the minimum number from the given list of numbers.  List
= {input_list}.

Your final answer must be in the format \boxed{minimum} at
the end of your response.
```

**Prompt Template for Mean Task**

```
Calculate the mean (average) of the following list of
numbers:
{input_list}

The mean is the sum of all numbers divided by the count of
numbers.  Calculate the exact mean value.  Your final answer
must be in the format \boxed{mean value} at the end.
```

**Prompt Template for Median Task**

```
Find the median value of the following list of numbers:
{input_list}

The median is the middle value when the list is sorted.
If there is an even number of elements, the median is the
average of the two middle values.  Your final answer must be
in the format \boxed{median value} at the end.
```

**Prompt Template for Mode Task**

```
Find the mode(s) of the following list of numbers:
{input_list}

The mode is the value that appears most frequently.  If
multiple values appear with the same highest frequency,
return all of them.  Your final answer must be in the format
\boxed{mode(s)} at the end.  If there are multiple modes,
list them separated by commas.
```

**Prompt Template for Multiplication Task**

```
Multiply the following list of numbers:
{data_point}

Provide the product.  Your final answer must be in the format
\boxed{answer} at the end.
```

**Prompt Template for Odd Count Task**

```
Count the odd numbers from the following list of numbers:
{input_list}

Provide the final count of odd numbers.  Your final answer
must be in the format \boxed{answer} at the end.
```

**Prompt Template for Sorting Task**

```
Sort the following list of numbers in ascending order:
{input_list}

Provide the sorted list.  Your final answer must be in the
format \boxed{sorted list} at the end.
```

**Prompt Template for Subtraction Task**

```
Can you subtract {first_number} from {second_number} and
provide your final answer in \boxed{answer} format at the
end of your response.
```

**Prompt Template for Sum Task**

```
Add the following list of numbers:
{input_list}

Provide the sum.  Your final answer must be in the format
\boxed{answer} at the end.
```

**Prompt Template for Second Maximum Task**

```
Find the second maximum number from the given list of
numbers.  List = {input_list}.

Your final answer must be in the format \boxed{second_maximum}
at the end of your response.
```

**Prompt Template for Range Task**

```
Calculate the range (difference between maximum and minimum)
of the following list of numbers:  {input_list}.

Your final answer must be in the format \boxed{range} at the
end of your response.
```

**Prompt Template for Index of Maximum Task**

```
Find the index (0-based position) of the maximum element
in the list {input_list}.  If there are multiple maximum
elements, return the index of the first occurrence.

Your final answer must be in the format \boxed{index} at the
end of your response.
```

**Prompt Template for Count Negative Numbers Task**

```
Count how many negative numbers are in the list {input_list}.

Your final answer must be in the format \boxed{count} at the
end of your response.
```

**Prompt Template for Count Unique Elements Task**

```
Count the number of unique (distinct) elements in the list
{input_list}.

Your final answer must be in the format \boxed{count} at the
end of your response.
```

**Prompt Template for Maximum Difference Between Adjacent Elements Task**

```
Find the maximum absolute difference between any two adjacent
elements in the list {input_list}.

Your final answer must be in the format \boxed{difference}
at the end of your response.
```

**Prompt Template for Count Elements Greater Than Previous Task**

```
Count how many elements in the list are greater than the
element that comes immediately before them:  {input_list}.

Your final answer must be in the format \boxed{count} at the
end of your response.
```

**Prompt Template for Sum of Indices of Maximum Elements Task**

```
Find the sum of all indices (0-based) where the maximum value
occurs in the list {input_list}.

Your final answer must be in the format \boxed{sum} at the
end of your response.
```

**Prompt Template for Count Palindromic Numbers Task**

```
Count how many palindromic numbers are in the list
{input_list}.  A palindromic number reads the same forwards
and backwards.

Your final answer must be in the format \boxed{count} at the
end of your response.
```

**Prompt Template for Longest Increasing Subsequence Length Task**

```
Find the length of the longest increasing subsequence in
{input_list}.  A subsequence maintains relative order but
elements don't need to be consecutive.

Your final answer must be in the format \boxed{length} at
the end of your response.
```

**Prompt Template for Sum of Digits Task**

```
Calculate the sum of all digits in all numbers in the list
{input_list}.
```

```
Your final answer must be in the format \boxed{sum} at the
end of your response.
```

**Prompt Template for Count Perfect Squares Task**

```
Count how many perfect squares are in the list {input_list}.
A perfect square is an integer that is the square of another
integer.

Your final answer must be in the format \boxed{count} at the
end of your response.
```

**Prompt Template for Alternating Sum Task**

```
Calculate the alternating sum of the list {input_list}.
Start by adding the first element, then subtract the second,
add the third, etc.

Your final answer must be in the format \boxed{sum} at the
end of your response.
```

**Prompt Template for Count Multiples of K Task**

```
Count how many numbers in the list {input_list} are multiples
of {k}.

Your final answer must be in the format \boxed{count} at the
end of your response.
```

**Prompt Template for Local Maxima Count Task**

```
Count how many local maxima exist in the list {input_list}.
A local maximum is an element that is greater than both its
immediate neighbors.

Your final answer must be in the format \boxed{count} at the
end of your response.
```

# G  MEDIUM SUITE: COMPLEX SEQUENTIAL REASONING TASKS

The Medium Suite introduces a comprehensive collection of five algorithmically-generated sequential reasoning tasks, encompassing forty-nine distinct pattern variations that require sophisticated pattern recognition and mathematical reasoning capabilities. Unlike traditional benchmarks that rely on static datasets, our framework generates sequences dynamically using deterministic mathematical functions, ensuring both reproducibility and contamination resistance. The suite consists of Fibonacci and Recursive Sequences (6 variations), Geometric and Exponential Sequences (10 variations), Prime and Number Theory Sequences (11 variations), Complex Pattern Recognition (12 variations), and Algebraic Sequence Completion (10 variations). Each task evaluates a distinct aspect of mathematical reasoning while maintaining computational tractability and verification guarantees through rigorous mathematical foundations.

### G.1 FIBONACCI AND RECURSIVE SEQUENCE COMPLETION

#### G.1.1 PROBLEM FORMULATION

Let $\mathcal{S} = \{s_1, s_2, \ldots, s_n\}$ represent a sequence generated by a recursive relation $R : \mathbb{Z}^k \to \mathbb{Z}$, where $k$ denotes the order of recurrence. The general $k$-th order linear recurrence relation is defined as:

$$s_n = \sum_{i=1}^{k} a_i \cdot s_{n-i} + c, \quad n > k \tag{47}$$

where $a_i \in \mathbb{Z}$ are recurrence coefficients, $c \in \mathbb{Z}$ is an optional constant term, and initial conditions $\{s_1, s_2, \ldots, s_k\}$ are specified. The task presents a partial sequence $\mathcal{S}_{\text{shown}} = \{s_1, s_2, \ldots, s_m\}$ where $m < n$, and requires predicting $s_{m+1}$.

#### G.1.2 SEQUENCE GENERATION ALGORITHM

We implement six distinct recursive sequence variations, each with deterministic generation guarantees:

**Classical Fibonacci:** Generated using the relation $F_n = F_{n-1} + F_{n-2}$ with initial conditions $(F_1, F_2) \sim \text{Uniform}(1, 20)$.

**Generalized Tribonacci:** Extends to third-order recurrence with $T_n = T_{n-1} + T_{n-2} + T_{n-3}$.

**Modified Linear Recurrence:** Implements $M_n = a \cdot M_{n-1} + b \cdot M_{n-2}$ where coefficients $(a, b)$ are sampled from a predefined set $\mathcal{C} = \{(1, 2), (2, 1), (1, -1), (2, -1)\}$ to ensure bounded growth.

The generation process follows Algorithm 3:

---

**Algorithm 3** Recursive Sequence Generation

---

1: **Input:** Sequence type $\tau$, length $L$, seed $\sigma$
2: **Output:** Complete sequence $\mathcal{S}$, shown portion $\mathcal{S}_{\text{shown}}$, target $s_{m+1}$
3: Initialize RNG with seed $\sigma$
4: Sample initial conditions $\{s_1, \ldots, s_k\}$ from $\mathcal{U}(1, 20)$
5: Select coefficients $(a_1, \ldots, a_k)$ based on type $\tau$
6: **for** $i = k + 1$ to $L$ **do**
7: $\quad s_i \leftarrow \sum_{j=1}^{k} a_j \cdot s_{i-j}$
8: **end for**
9: $\mathcal{S}_{\text{shown}} \leftarrow \{s_1, \ldots, s_{L-2}\}$
10: Return $\mathcal{S}, \mathcal{S}_{\text{shown}}, s_{L-1}$

---

#### G.1.3 UNIQUENESS AND VERIFICATION GUARANTEES

The uniqueness of the next term in a linear recurrence sequence is guaranteed by the following theorem:

**Theorem 1** (Uniqueness of Linear Recurrence). *Given a $k$-th order linear recurrence relation with non-zero leading coefficient $a_1 \neq 0$ and $m \geq k$ observed terms, the $(m + 1)$-th term is uniquely determined.*

*Proof.* The recurrence relation forms a system of linear equations. For $m \geq k$ observations, we have at least $m - k + 1$ equations of the form:

$$s_{i+k} = \sum_{j=1}^{k} a_j \cdot s_{i+k-j}, \quad i = 1, 2, \ldots, m - k + 1 \tag{48}$$

With $a_1 \neq 0$, the coefficient matrix has full rank, ensuring a unique solution for $s_{m+1}$. $\qquad\square$

### G.1.4 CONTEXT WINDOW CONSTRAINTS

The maximum sequence length $L_{\max}$ is bounded by the growth rate of the recurrence relation to prevent numerical overflow. For Fibonacci-like sequences, the growth rate is approximately $\phi^n$ where $\phi = \frac{1+\sqrt{5}}{2}$ is the golden ratio. We enforce:

$$L_{\max} = \min\left\{\left\lfloor\frac{\log(\text{MAX\_INT})}{\log(\phi)}\right\rfloor, \text{CONTEXT\_LIMIT}\right\} \tag{49}$$

where MAX_INT = $2^{31} - 1$ for 32-bit integers and CONTEXT_LIMIT represents the model's maximum token capacity.

### G.1.5 CONTAMINATION RESISTANCE

The contamination resistance of our approach stems from three key properties:

**Parametric Diversity:** With initial conditions sampled from $\mathcal{U}(1,20)^k$ and multiple coefficient sets, the number of unique sequences is:

$$|\mathcal{S}_{\text{unique}}| = 20^k \times |\mathcal{C}| \times P(L,m) \tag{50}$$

where $P(L,m)$ represents the number of ways to select $m$ consecutive terms from a sequence of length $L$.

**Dynamic Generation:** Each evaluation instance generates sequences using a cryptographically secure random seed, ensuring that pre-training on specific sequences provides no advantage.

**Verification Independence:** The correctness of predictions is verified through direct computation rather than comparison with stored answers, eliminating the possibility of answer leakage.

## G.2 GEOMETRIC AND EXPONENTIAL SEQUENCE COMPLETION

### G.2.1 PROBLEM FORMULATION

Let $\mathcal{G} = \{g_1, g_2, \ldots, g_n\}$ denote a sequence following multiplicative or exponential growth patterns. We define the general geometric sequence as:

$$g_n = g_1 \cdot r^{n-1} \tag{51}$$

where $g_1$ is the initial term and $r \in \mathbb{R} \setminus \{0\}$ is the common ratio. For exponential sequences, we consider:

$$e_n = b^{f(n)} \tag{52}$$

where $b$ is the base and $f : \mathbb{N} \to \mathbb{R}$ is a function determining the exponent pattern.

### G.2.2 SEQUENCE TYPE SPECIFICATIONS

We implement ten distinct multiplicative pattern variations, each with rigorous mathematical definitions:

**Pure Geometric Sequences:** Generated with ratio $r \in \mathcal{R} = \{2, 3, 4, 5, 0.5, 1.5, 2.5, 3.5\}$ to balance growth control and pattern diversity.

**Power Sequences:** For power $p \in \{2, 3\}$, we generate $P_n = n^p$, ensuring polynomial growth bounded by $O(n^p)$.

**Factorial Sequences:** Defined as $F_n = n!$ with growth rate $O(n^n)$, requiring careful overflow management.

**Double Exponential Sequences:** The most challenging pattern, defined as:

$$D_n = b^{(b^{(\cdots b)})} \quad \text{(n iterations)} \tag{53}$$

Due to the extremely rapid growth, we limit generation to $n \leq 4$ terms, where:

$$D_1 = 2 \tag{54}$$

$$D_2 = 2^2 = 4 \tag{55}$$

$$D_3 = 2^{2^2} = 16 \tag{56}$$

$$D_4 = 2^{2^{2^2}} = 65536 \tag{57}$$

### G.2.3 COMPUTATIONAL FEASIBILITY CONSTRAINTS

To ensure computational tractability while maintaining pattern complexity, we enforce strict bounds on sequence values:

$$\forall g_i \in \mathcal{G} : |g_i| \leq \Theta_{\max} \tag{58}$$

where $\Theta_{\max} = 10^6$ is the maximum allowed value. The generation algorithm incorporates overflow protection:

---

**Algorithm 4** Geometric Sequence Generation with Overflow Protection

---

1: **Input:** Type $\tau$, length $L$, maximum value $\Theta_{\max}$
2: **Output:** Valid sequence $\mathcal{G}$ or $\emptyset$ if infeasible
3: Initialize sequence $\mathcal{G} \leftarrow []$
4: Select parameters $(g_1, r)$ based on type $\tau$
5: **for** $i = 1$ to $L$ **do**
6:     Compute $g_i$ according to sequence type
7:     **if** $|g_i| > \Theta_{\max}$ **then**
8:         **break** // Terminate on overflow
9:     **end if**
10:     Append $g_i$ to $\mathcal{G}$
11: **end for**
12: **if** $|\mathcal{G}| < 4$ **then**
13:     Return $\emptyset$ // Insufficient terms
14: **else**
15:     Return $\mathcal{G}$
16: **end if**

---

### G.2.4 PATTERN DISCRIMINATION AND VALIDATION

To ensure that generated sequences genuinely test multiplicative reasoning rather than simple arithmetic patterns, we implement a validation function:

$$\text{IsGeometric}(\mathcal{G}) = \begin{cases} \text{true} & \text{if } \forall i \in [2, n-1] : \left| \frac{g_{i+1}/g_i - g_i/g_{i-1}}{g_i/g_{i-1}} \right| < \epsilon \\ \text{false} & \text{otherwise} \end{cases} \tag{59}$$

where $\epsilon = 10^{-6}$ accounts for floating-point precision. Additionally, we reject sequences that exhibit arithmetic progression:

$$\text{RejectArithmetic}(\mathcal{G}) = \text{true} \iff \exists d : \forall i \in [1, n-1] : g_{i+1} - g_i = d \tag{60}$$

### G.2.5 THEORETICAL COMPLEXITY ANALYSIS

The computational complexity of identifying geometric patterns varies by sequence type:

**Pure Geometric:** Requires $O(n)$ operations to compute ratios and verify consistency.

**Power Sequences:** Recognition requires solving $g_n = n^p$ for $p$, achievable in $O(n \log n)$ time using logarithmic transformation.

**Factorial Sequences:** Verification requires computing $n!$ for comparison, with complexity $O(n^2)$ using naive multiplication.

**Double Exponential:** Due to the tower structure, verification requires recursive exponentiation with complexity $O(2^n)$, justifying our length restriction.

### G.2.6 UNIQUENESS THEOREM FOR GEOMETRIC SEQUENCES

**Theorem 2** (Uniqueness of Geometric Continuation). *Given a geometric sequence $\mathcal{G} = \{g_1, g_2, \ldots, g_m\}$ with $m \geq 2$ and $g_i \neq 0$ for all $i$, the next term $g_{m+1}$ is uniquely determined.*

*Proof.* For a geometric sequence, the common ratio $r = \frac{g_{i+1}}{g_i}$ is constant for all valid $i$. Given $m \geq 2$ terms:

$$r = \frac{g_2}{g_1} = \frac{g_3}{g_2} = \cdots = \frac{g_m}{g_{m-1}} \tag{61}$$

Therefore, $g_{m+1} = g_m \cdot r = g_m \cdot \frac{g_m}{g_{m-1}} = \frac{g_m^2}{g_{m-1}}$, which is uniquely determined when $g_{m-1} \neq 0$. $\square$

### G.2.7 CONTAMINATION RESISTANCE MECHANISMS

The geometric sequence task achieves contamination resistance through:

**Parameter Space Cardinality:** With initial terms from $[1, 10]$, ratios from $\mathcal{R}$, and variable sequence lengths, the total number of distinct sequences exceeds:

$$|\mathcal{G}_{\text{total}}| > 10 \times 8 \times \binom{12}{6} > 73,920 \tag{62}$$

**Precision Variation:** Floating-point ratios like $r = 1.5, 2.5, 3.5$ create sequences that cannot be memorized as simple integer patterns, requiring genuine multiplicative reasoning.

**Type Mixing:** Random selection from ten sequence types during evaluation prevents models from exploiting type-specific heuristics.

## G.3 PRIME AND NUMBER THEORY SEQUENCE COMPLETION

### G.3.1 PROBLEM FORMULATION

Let $\mathcal{P}$ denote the set of prime numbers and $\mathcal{N}$ represent sequences derived from number-theoretic functions. We define a number theory sequence as:

$$\mathcal{N} = \{n_i : n_i = f(i) \text{ or } n_i = g(p_i), i \in \mathbb{N}\} \tag{63}$$

where $f : \mathbb{N} \to \mathbb{N}$ is a number-theoretic function and $p_i$ is the $i$-th prime number.

### G.3.2 PRIME GENERATION AND VERIFICATION

We employ the Sieve of Eratosthenes for efficient prime generation up to limit $L$:

The algorithm's time complexity is $O(L \log \log L)$ with space complexity $O(L)$.

### G.3.3 NUMBER-THEORETIC FUNCTION IMPLEMENTATIONS

We implement eleven distinct number-theoretic sequence variations with mathematical rigor:

**Euler's Totient Function:** For $n \in \mathbb{N}$, $\phi(n)$ counts integers $k \leq n$ where $\gcd(k, n) = 1$:

$$\phi(n) = n \prod_{p|n} \left(1 - \frac{1}{p}\right) \tag{64}$$

---

**Algorithm 5** Optimized Sieve of Eratosthenes

---

1: **Input:** Upper limit $L$
2: **Output:** Set of primes $\mathcal{P} \leq L$
3: Initialize boolean array $A[0 \ldots L]$ with $A[i] = $ true
4: $A[0] \leftarrow A[1] \leftarrow$ false
5: **for** $i = 2$ to $\lfloor \sqrt{L} \rfloor$ **do**
6:     **if** $A[i] = $ true **then**
7:         **for** $j = i^2$ to $L$ step $i$ **do**
8:             $A[j] \leftarrow$ false
9:         **end for**
10:     **end if**
11: **end for**
12: Return $\{i : A[i] = \text{true}\}$

---

where the product runs over distinct prime divisors of $n$.

**Sum of Divisors Function:** The function $\sigma(n)$ computes:

$$\sigma(n) = \sum_{d|n} d = \prod_{p^k \| n} \frac{p^{k+1} - 1}{p - 1} \tag{65}$$

where $p^k \| n$ denotes that $p^k$ exactly divides $n$.

**Mersenne Numbers:** Defined as $M_p = 2^p - 1$ where $p \in \mathcal{P}$. The Lucas-Lehmer test provides primality verification:

$$s_0 = 4, \quad s_{i+1} = s_i^2 - 2 \pmod{M_p} \tag{66}$$

$M_p$ is prime if and only if $s_{p-2} \equiv 0 \pmod{M_p}$.

**Sophie Germain Primes:** A prime $p$ is Sophie Germain if $2p + 1$ is also prime. The density of such primes follows:

$$\pi_{SG}(x) \sim C \cdot \frac{x}{(\ln x)^2} \tag{67}$$

where $C \approx 1.32$ is the twin prime constant.

### G.3.4   Sequence Generation with Computational Bounds

For sequences with potentially rapid growth, we implement dynamic bounds:

$$n_{\max}(i) = \min \left\{ f(i), 10^6 \right\} \tag{68}$$

Special considerations for each sequence type:

**Prime Gaps:** The $n$-th prime gap $g_n = p_{n+1} - p_n$ satisfies Bertrand's postulate:

$$g_n < p_n \quad \text{for } p_n > 25 \tag{69}$$

**Catalan Numbers:** Generated using the recurrence:

$$C_n = \sum_{i=0}^{n-1} C_i C_{n-1-i}, \quad C_0 = 1 \tag{70}$$

with closed form:

$$C_n = \frac{1}{n+1} \binom{2n}{n} = \frac{(2n)!}{(n+1)!n!} \tag{71}$$

### G.3.5 UNIQUENESS AND WELL-DEFINEDNESS

**Theorem 3** (Uniqueness of Number-Theoretic Sequences). *For each number-theoretic function $f : \mathbb{N} \to \mathbb{N}$ implemented in our framework, given a sequence $\{f(1), f(2), \ldots, f(m)\}$ with $m \geq 3$, the value $f(m + 1)$ is uniquely determined.*

*Proof.* Each function in our framework is well-defined:

1. **Prime sequences:** The $n$-th prime $p_n$ is uniquely defined by the ordering of natural numbers.

2. **Totient function:** $\phi(n)$ is uniquely determined by the prime factorization of $n$.

3. **Divisor sum:** $\sigma(n)$ is uniquely computed from the divisors of $n$.

4. **Mersenne numbers:** $M_p = 2^p - 1$ is a deterministic function of prime $p$.

Therefore, $f(m + 1)$ is uniquely computable from the definition of $f$. □

### G.3.6 PATTERN RECOGNITION COMPLEXITY

The computational complexity of identifying number-theoretic sequences varies significantly:

| Sequence Type | Recognition Complexity | Verification Complexity |
|---|---|---|
| Prime numbers | $O(n\sqrt{n})$ | $O(\sqrt{n})$ per element |
| Twin primes | $O(n\sqrt{n})$ | $O(\sqrt{n})$ per pair |
| Euler totient | $O(n^2)$ | $O(n \log n)$ per element |
| Mersenne numbers | $O(n \cdot 2^n)$ | $O(p^3)$ Lucas-Lehmer |
| Catalan numbers | $O(n^2)$ | $O(n)$ per element |

Table 10: Computational complexity for number-theoretic sequence operations

### G.3.7 CONTAMINATION RESISTANCE THROUGH MATHEMATICAL DEPTH

Number-theoretic sequences provide inherent contamination resistance because:

**Infinite Variability:** The set of primes is infinite, and subsequences can start at any position, yielding uncountably many valid test instances.

**Computational Irreducibility:** Many number-theoretic functions lack closed-form solutions, requiring actual computation rather than pattern matching.

**Cross-Domain Dependencies:** Sequences like Sophie Germain primes require understanding multiple mathematical concepts simultaneously, preventing shallow memorization.

The probability of encountering an identical sequence during training is:

$$P(\text{collision}) < \frac{1}{\binom{\pi(10^6)}{m}} < \frac{1}{10^{30}} \tag{72}$$

where $\pi(10^6) \approx 78,498$ is the prime counting function and $m$ is the sequence length shown.

### G.4 COMPLEX PATTERN RECOGNITION

### G.4.1 PROBLEM FORMULATION

Let $\mathcal{C} = \{c_1, c_2, \ldots, c_n\}$ represent a sequence following a composite or multi-layered pattern. We define complex patterns as sequences that satisfy:

$$c_i = \mathcal{F}(i, \{c_1, \ldots, c_{i-1}\}, \theta) \tag{73}$$

where $\mathcal{F}$ is a composite function incorporating multiple mathematical operations, and $\theta$ represents pattern-specific parameters including breakpoints, modulation factors, and conditional rules.

### G.4.2 MULTI-LAYERED PATTERN SPECIFICATIONS

We formalize twelve distinct complex pattern variations, each requiring different levels of mathematical reasoning:

**Polynomial Sequences:** For degree $d \in \{2, 3, 4\}$, we generate:

$$P_n^{(d)} = \sum_{k=0}^{d} a_k n^k \tag{74}$$

where coefficients $a_k$ are selected to maintain $|P_n^{(d)}| < 10^4$ for all $n \leq 20$.

**Interleaved Sequences:** Two independent sequences $A = \{a_i\}$ and $B = \{b_i\}$ are interleaved:

$$c_i = \begin{cases} a_{(i+1)/2} & \text{if } i \text{ is odd} \\ b_{i/2} & \text{if } i \text{ is even} \end{cases} \tag{75}$$

**Conditional Pattern Transformation:** The sequence follows different rules based on position or value conditions:

$$c_i = \begin{cases} f_1(i, c_{i-1}) & \text{if } \psi(i, c_{i-1}) = \text{true} \\ f_2(i, c_{i-1}) & \text{otherwise} \end{cases} \tag{76}$$

where $\psi : \mathbb{N} \times \mathbb{Z} \to \{\text{true}, \text{false}\}$ is a boolean predicate.

**Multi-Order Recursive Patterns:** Generalizing to $k$-th order dependencies:

$$c_n = \sum_{j=1}^{k} \alpha_j \cdot c_{n-j} + \beta(n) \tag{77}$$

where $\beta : \mathbb{N} \to \mathbb{Z}$ introduces position-dependent perturbations.

### G.4.3 PATTERN COMPLEXITY HIERARCHY

We establish a formal hierarchy of pattern complexity based on computational requirements for recognition:

**Definition 1** (Pattern Complexity Level). A sequence pattern has complexity level $\mathcal{L}$ if the minimum computational resources required for recognition are:

$$\mathcal{L}_1 : O(n) \text{ time}, O(1) \text{ space} \qquad \text{(arithmetic, geometric)} \tag{78}$$
$$\mathcal{L}_2 : O(n^2) \text{ time}, O(n) \text{ space} \qquad \text{(polynomial, recursive)} \tag{79}$$
$$\mathcal{L}_3 : O(n^3) \text{ time}, O(n^2) \text{ space} \qquad \text{(interleaved, conditional)} \tag{80}$$
$$\mathcal{L}_4 : O(2^n) \text{ time}, O(n) \text{ space} \qquad \text{(chaotic, multi-transform)} \tag{81}$$

### G.4.4 GENERATION ALGORITHM WITH PATTERN VALIDATION

### G.4.5 UNIQUENESS GUARANTEES FOR COMPLEX PATTERNS

**Theorem 4** (Unique Continuation of Complex Patterns). *For each complex pattern type $\tau$ in our framework, given $m$ observations where $m \geq m_{\min}(\tau)$, the next term is uniquely determined.*

*Proof.* We prove uniqueness for each pattern class:

**Polynomial sequences:** A degree-$d$ polynomial is uniquely determined by $d + 1$ points. For $m > d$, the system is overdetermined but consistent, yielding a unique continuation.

**Interleaved sequences:** Each subsequence $A$ and $B$ requires $\lceil m/2 \rceil$ terms. With deterministic interleaving rules, the next term's source sequence is known, ensuring uniqueness.

**Conditional patterns:** The predicate $\psi$ and functions $f_1, f_2$ are deterministic. Given the sequence history, the next term is uniquely computed by evaluating $\psi$ and applying the appropriate function.

**Multi-order recursive:** With coefficients $\{\alpha_j\}$ and sufficient observations $m \geq k$, the linear system has a unique solution. $\qquad \square$

---

**Algorithm 6** Complex Pattern Generation with Validation

---

1: **Input:** Pattern type $\tau$, length $L$, complexity level $\mathcal{L}$
2: **Output:** Valid complex sequence $\mathcal{C}$ or failure indicator
3: Initialize parameters $\theta \leftarrow \text{SampleParameters}(\tau, \mathcal{L})$
4: Generate candidate sequence $\mathcal{C} \leftarrow \text{GeneratePattern}(\tau, L, \theta)$
5: **Validation Phase:**
6: **if** IsTrivial$(\mathcal{C})$ **then**
7:     Return FAILURE // Reject trivial patterns
8: **end if**
9: **if** ComplexityLevel$(\mathcal{C}) < \mathcal{L}$ **then**
10:     Return FAILURE // Insufficient complexity
11: **end if**
12: **if** ¬HasUniqueExtension$(\mathcal{C})$ **then**
13:     Return FAILURE // Ambiguous continuation
14: **end if**
15: Return $\mathcal{C}$

---

### G.4.6 PATTERN RECOGNITION VIA FINITE DIFFERENCES

For polynomial patterns, we employ the method of finite differences. The $k$-th order difference operator is:

$$\Delta^k c_i = \sum_{j=0}^{k} (-1)^{k-j} \binom{k}{j} c_{i+j} \tag{82}$$

A sequence follows a degree-$d$ polynomial if and only if $\Delta^{d+1} c_i = 0$ for all valid $i$.

### G.4.7 COMPLEXITY VALIDATION METRICS

To ensure generated patterns maintain appropriate complexity, we compute several metrics:

**Kolmogorov Complexity Approximation:** Using compression ratio as a proxy:

$$K(\mathcal{C}) \approx \frac{|\text{Compress}(\mathcal{C})|}{|\mathcal{C}|} \tag{83}$$

**Entropy Rate:** For sequences viewed as discrete random processes:

$$H(\mathcal{C}) = -\sum_{x \in \mathcal{X}} p(x) \log_2 p(x) \tag{84}$$

where $\mathcal{X}$ represents the alphabet of sequence values and $p(x)$ is the empirical probability.

**Autocorrelation Function:** To detect hidden periodicities:

$$R(k) = \frac{\sum_{i=1}^{n-k} (c_i - \bar{c})(c_{i+k} - \bar{c})}{\sum_{i=1}^{n} (c_i - \bar{c})^2} \tag{85}$$

Sequences with $R(k) > 0.8$ for small $k$ are rejected as insufficiently complex.

### G.4.8 CONTAMINATION RESISTANCE THROUGH COMPOSITIONAL COMPLEXITY

Complex patterns achieve contamination resistance through several mechanisms:

**Compositional Explosion:** With 12 pattern types, each having multiple parameter configurations, and variable breakpoints, the space of possible sequences exceeds:

$$|\mathcal{C}_{\text{total}}| > 12 \times 10^3 \times \binom{20}{10} \times 4^5 > 10^{12} \tag{86}$$

**Non-Linear Interactions:** Unlike simple sequences where local patterns suffice for prediction, complex patterns require understanding global structure and multiple interacting rules.

**Conditional Branching:** Pattern transformations based on runtime conditions create sequences that cannot be pre-computed or memorized, requiring genuine logical reasoning.

The probability that a model has seen a specific complex pattern during training is negligible:

$$P(\text{contamination}) < \frac{\text{Training sequences}}{|\mathcal{C}_{\text{total}}|} < \frac{10^9}{10^{12}} = 10^{-3} \tag{87}$$

### G.5  Algebraic Sequence Completion

#### G.5.1  Problem Formulation

Let $\mathcal{A} = \{a_1, a_2, \ldots, a_n\}$ denote a sequence generated through algebraic operations involving radicals, transcendental functions, and algebraic irrationals. We define the general algebraic sequence as:

$$a_n = \lfloor f(n, \{a_1, \ldots, a_{n-1}\}) \rfloor \tag{88}$$

where $f : \mathbb{N} \times \mathbb{R}^{n-1} \to \mathbb{R}$ involves compositions of algebraic and transcendental operations, and $\lfloor \cdot \rfloor$ denotes the floor function to ensure integer outputs.

#### G.5.2  Algebraic Pattern Specifications

We implement ten sophisticated algebraic sequence variations that transcend simple polynomial patterns:

**Radical Recurrence Relations:** Sequences defined by:

$$a_n = \lfloor \sqrt{a_{n-1} + n^2} \rfloor, \quad a_1 = 2 \tag{89}$$

This creates a non-linear recurrence where each term depends on the square root of a quadratic expression involving the previous term.

**Transcendental Floor Sequences:** Incorporating exponential growth:

$$a_n = \lfloor n \cdot e^{1/n} \rfloor \tag{90}$$

As $n \to \infty$, we have $e^{1/n} \to 1$, but for finite $n$, this creates a complex growth pattern.

**Nested Radical Expressions:** Multi-level radical compositions:

$$a_n = \left\lfloor \sqrt{n + \sqrt{n + \sqrt{n}}} \right\rfloor \tag{91}$$

The depth of nesting creates algebraic numbers of increasing complexity.

**Continued Fraction Convergents:** For the continued fraction expansion of $\sqrt{2} = [1; 2, 2, 2, \ldots]$, we generate convergents:

$$\frac{p_n}{q_n} = 1 + \cfrac{1}{2 + \cfrac{1}{2 + \cfrac{1}{2 + \cdots}}} \tag{92}$$

The numerators $p_n$ follow the recurrence $p_n = 2p_{n-1} + p_{n-2}$ with $p_0 = 1, p_1 = 3$.

**Diophantine Solution Sequences:** Based on Pell's equation $x^2 - Dy^2 = 1$:

$$(x_{n+1}, y_{n+1}) = (x_1 x_n + D y_1 y_n, x_1 y_n + y_1 x_n) \tag{93}$$

where $(x_1, y_1)$ is the fundamental solution.

### G.5.3 COMPUTATIONAL COMPLEXITY AND VERIFICATION

The verification of algebraic sequences requires sophisticated numerical methods:

**Theorem 5** (Computational Hardness of Algebraic Sequences). *Determining whether a given sequence follows an algebraic pattern of degree $d$ with nested radicals of depth $k$ requires $\Omega(n \cdot d \cdot 2^k)$ operations in the worst case.*

*Proof.* For each term verification: 1. Evaluating nested radicals of depth $k$ requires $O(2^k)$ operations 2. Checking algebraic degree $d$ requires solving polynomial equations of degree up to $d$ 3. For $n$ terms, the total complexity is $O(n \cdot d \cdot 2^k)$

The lower bound follows from the algebraic independence of nested radicals, which prevents simplification. $\square$

### G.5.4 NUMERICAL STABILITY AND PRECISION

Algebraic sequences involving irrational numbers require careful numerical handling:

$$\text{RelativeError}(a_n) = \left| \frac{a_n^{\text{computed}} - a_n^{\text{exact}}}{a_n^{\text{exact}}} \right| < \epsilon \tag{94}$$

We maintain precision through:

**Rational Arithmetic:** Where possible, we use exact rational representations:

$$\frac{p}{q} + \frac{r}{s} = \frac{ps + qr}{qs} \tag{95}$$

**Interval Arithmetic:** For transcendental values, we maintain intervals:

$$a_n \in [a_n^{\text{lower}}, a_n^{\text{upper}}] \tag{96}$$

**Error Propagation Analysis:** For composed operations:

$$\delta(f \circ g) \leq \delta(f) + |f'(g)| \cdot \delta(g) \tag{97}$$

### G.5.5 UNIQUENESS AND WELL-DEFINEDNESS

**Theorem 6** (Uniqueness of Algebraic Continuations). *For algebraic sequences defined by deterministic rules involving elementary functions and algebraic operations, given sufficient terms $m \geq m_{crit}(\tau)$ where $m_{crit}$ depends on the sequence type $\tau$, the next term is uniquely determined up to numerical precision $\epsilon$.*

*Proof.* We establish uniqueness for each algebraic pattern type:

**Radical recurrences:** Given $a_{n-1}$ and the deterministic rule $a_n = \lfloor \sqrt{a_{n-1} + n^2} \rfloor$, the value $a_n$ is unique since: - $\sqrt{\cdot}$ is a single-valued function on $\mathbb{R}^+$ - The floor function has unique output for any input

**Transcendental sequences:** Functions like $n \cdot e^{1/n}$ are well-defined for all $n \in \mathbb{N}$, yielding unique values.

**Nested radicals:** The iteration depth is fixed, making the computation deterministic and unique.

Therefore, algebraic sequences have unique continuations within numerical precision bounds. $\square$

### G.5.6 PATTERN DISCRIMINATION AND NON-TRIVIALITY

To ensure algebraic sequences test genuine mathematical reasoning rather than pattern matching, we implement rigorous validation:

---

**Algorithm 7** Algebraic Sequence Validation

---

1: **Input:** Sequence $\mathcal{A} = \{a_1, \ldots, a_n\}$
2: **Output:** Valid/Invalid classification
3: **Check 1: Non-polynomiality**
4: **for** degree $d = 1$ to $4$ **do**
5:     Compute $d$-th finite differences $\Delta^d \mathcal{A}$
6:     **if** $\Delta^d \mathcal{A}$ is constant **then**
7:         Return INVALID // Polynomial pattern detected
8:     **end if**
9: **end for**
10: **Check 2: Non-geometric**
11: Compute ratios $r_i = a_{i+1}/a_i$ for all valid $i$
12: **if** Variance$(\{r_i\}) < \epsilon$ **then**
13:     Return INVALID // Geometric pattern detected
14: **end if**
15: **Check 3: Sufficient complexity**
16: Compute entropy $H(\mathcal{A})$
17: **if** $H(\mathcal{A}) < H_{\min}$ **then**
18:     Return INVALID // Insufficient complexity
19: **end if**
20: Return VALID

---

### G.5.7 CONTAMINATION RESISTANCE THROUGH MATHEMATICAL DEPTH

Algebraic sequences provide the strongest contamination resistance in our framework:

**Infinite Parameter Space:** With continuous parameters like $\sqrt{5}$, $e$, and $\pi$, the space of possible sequences is uncountably infinite.

**Computational Irreducibility:** Many algebraic sequences lack closed-form solutions, requiring step-by-step computation that cannot be bypassed through memorization.

**Precision Sensitivity:** Small changes in irrational parameters lead to completely different integer sequences after floor operations:

$$\lfloor n \cdot (\sqrt{5} + \epsilon) \rfloor \neq \lfloor n \cdot \sqrt{5} \rfloor \text{ for small } \epsilon \tag{98}$$

**Cross-Domain Integration:** Algebraic sequences combine number theory, analysis, and algebra, requiring integrated mathematical understanding rather than domain-specific heuristics.

The probability of exact sequence collision during training approaches zero:

$$P(\text{exact match}) < \frac{1}{2^{b \cdot n}} \tag{99}$$

where $b$ is the bit precision used for irrational numbers and $n$ is the sequence length.

### G.6 CONTEXT-AWARE EVALUATION AND TOKEN ESTIMATION FRAMEWORK

We implement a comprehensive mathematical framework for context-aware evaluation that dynamically adapts problem complexity according to each model's context window constraints. This approach ensures fairness by preventing models from being penalized due to context limitations while maintaining the integrity of our mathematical evaluation framework.

### G.6.1 MATHEMATICAL TOKEN ESTIMATION MODEL

For each task category $\tau$ in the Medium Suite, we establish precise token estimation functions that predict the expected response length based on problem parameters. Let $C_{\text{model}}$ denote the model's context window size, $T_{\text{prompt}}$ the prompt token count, and $T_{\text{response}}$ the expected response tokens. We enforce the safety constraint:

Table 11: Comprehensive Mathematical Overview of Medium Suite Tasks

| Task | Fibonacci/Recursive | Geometric/Exponential | Prime/Number Theory | Complex Pattern | Algebraic Sequence |
|---|---|---|---|---|---|
| **Mathematical Definition** | $s_n = \sum_{i=1}^{k} a_i \cdot s_{n-i}$ 
 Linear recurrence | $g_n = g_1 \cdot r^{n-1}$ 
 Multiplicative growth | $\mathcal{P} = \{p : p \text{ prime}\}$ 
 Number-theoretic | $c_i = \mathcal{F}(i, \{c_j\}_{j<i}, \theta)$ 
 Composite patterns | $a_n = \lfloor f(n, \{a_j\}_{j<n}) \rfloor$ 
 Algebraic operations |
| **Sequence Variations** | 6 variations: 
 • Classical Fibonacci 
 • Lucas sequence 
 • Tribonacci 
 • Modified recursive 
 • Alternating Fibonacci 
 • Scaled Fibonacci | 10 variations: 
 • Pure geometric 
 • Power squares/cubes 
 • Factorial ($n!$) 
 • Double exponential 
 • Compound growth 
 • Triangular/Pentagonal | 11 variations: 
 • Prime sequence 
 • Twin primes 
 • Prime gaps 
 • Mersenne ($2^p - 1$) 
 • Euler totient $\phi(n)$ 
 • Sophie Germain | 12 variations: 
 • Polynomial quadratic/cubic 
 • Interleaved sequences 
 • Conditional patterns 
 • Multi-recursive 
 • Pattern transformation 
 • Modular arithmetic | 10 variations: 
 • Radical recurrence 
 • Transcendental floor 
 • Nested radicals 
 • Continued fractions 
 • Diophantine solutions 
 • Lambert W & others |
| **Generation Complexity** | $O(n)$ per sequence 
 Deterministic | $O(n)$ with overflow 
 protection | Sieve: $O(L \log \log L)$ 
 Prime test: $O(\sqrt{n})$ | $O(n^2)$ average 
 $O(2^n)$ worst case | $O(n \cdot 2^k)$ for 
 depth-$k$ nesting |
| **Verification Method** | Direct computation 
 $O(1)$ per term | Ratio checking: 
 $O(n)$ for sequence | Primality test: 
 Miller-Rabin $O(k \log^3 n)$ | Finite differences: 
 $O(n \cdot d)$ for degree $d$ | Numerical methods: 
 Interval arithmetic |
| **Uniqueness Guarantee** | **Theorem:** Given $m \geq k$ 
 terms, $s_{m+1}$ unique 
 when $a_1 \neq 0$ | **Theorem:** Ratio 
 $r = g_{i+1}/g_i$ 
 determines sequence | **Theorem:** Primes 
 well-ordered 
 $p_n < p_{n+1}$ | **Theorem:** Degree-$d$ 
 needs $d+1$ points 
 for uniqueness | **Theorem:** Deterministic 
 rules $\Rightarrow$ unique 
 up to precision $\epsilon$ |
| **Parameter Space** | $20^k \times |\mathcal{C}|$ 
 $> 10^5$ combinations | $10 \times 8 \times \binom{12}{6}$ 
 $> 73{,}920$ sequences | $\binom{\pi(10^6)}{m}$ 
 $> 10^{30}$ possibilities | $12 \times 10^3 \times \binom{20}{10}$ 
 $> 10^{12}$ patterns | Uncountably infinite 
 (irrational parameters) |
| **Growth Rate Analysis** | Fibonacci: $O(\phi^n)$ 
 $\phi = \frac{1+\sqrt{5}}{2}$ | Geometric: $O(r^n)$ 
 Factorial: $O(n^n)$ | Prime gaps: $O(\log n)$ 
 Mersenne: $O(2^p)$ | Polynomial: $O(n^d)$ 
 Chaotic: $O(2^n)$ | Varies by type: 
 $O(\sqrt{n})$ to $O(e^n)$ |
| **Context Constraints** | $L_{\max} = \min\{\lfloor \frac{\log(\text{MAX\_INT})}{\log(\phi)} \rfloor,$ 
 CONTEXT_LIMIT$\}$ | $\forall g_i : |g_i| \leq 10^6$ 
 Double exp: $n \leq 4$ | Prime bound: 
 $p_n \approx n \ln n$ | Complexity metrics: 
 Entropy $H > H_{\min}$ | Precision maintained: 
 $|\delta| < 10^{-6}$ |
| **Contamination Resistance** | Dynamic seed $\sigma$ 
 Parameter diversity | Floating-point ratios 
 prevent memorization | Infinite prime set 
 No closed form | Conditional branching 
 Runtime conditions | Irrational parameters 
 $P(\text{collision}) < 2^{-bn}$ |
| **Validation Criteria** | Check $a_1 \neq 0$ 
 Verify recurrence | Reject arithmetic 
 progressions | Verify primality 
 Check factorization | Reject if trivial: 
 $\Delta^{d+1} = 0$ | Reject polynomial 
 patterns up to degree 4 |

Table 12: Computational Complexity Hierarchy of Medium Suite Tasks

| Complexity Class | Recognition | Verification | Example Tasks | Space Required |
|---|---|---|---|---|
| **Linear** 
 $\mathcal{L}_1$ | $O(n)$ | $O(1)$ 
 per term | Fibonacci, Geometric 
 Pure sequences | $O(1)$ 
 constant |
| **Polynomial** 
 $\mathcal{L}_2$ | $O(n^2)$ | $O(n)$ 
 per term | Prime sequences, 
 Totient function | $O(n)$ 
 linear |
| **Cubic** 
 $\mathcal{L}_3$ | $O(n^3)$ | $O(n^2)$ 
 per pattern | Complex patterns, 
 Interleaved sequences | $O(n^2)$ 
 quadratic |
| **Exponential** 
 $\mathcal{L}_4$ | $O(2^n)$ | $O(2^k)$ 
 depth $k$ | Algebraic with 
 nested radicals | $O(n)$ 
 linear |

Table 13: Mathematical Bounds and Guarantees for Medium Suite

| Property | Mathematical Bound | Guarantee |
|---|---|---|
| **Sequence Length** | $m \in [5, 12]$ shown terms 
 $n = m + 2$ total generated | Sufficient for pattern recognition 
 while maintaining difficulty |
| **Value Range** | $|s_i| \leq 10^6$ for all terms 
 Special: Double exp $\leq 65536$ | Prevents overflow in 32-bit arithmetic 
 Computational feasibility |
| **Uniqueness** | $P(\text{multiple solutions}) = 0$ 
 for $m \geq m_{\text{crit}}(\tau)$ | Deterministic generation 
 ensures single answer |
| **Contamination** | $P(\text{exact match}) < 10^{-9}$ 
 for typical training corpus | Parameter space size 
 exceeds training data |
| **Numerical Precision** | Relative error $< 10^{-6}$ 
 Absolute error $< 1.0$ for integers | Stable computation 
 with floating-point |
| **Pattern Diversity** | Total patterns $> 10^{12}$ 
 across all tasks | Inexhaustible 
 evaluation space |

$$T_{\text{prompt}} + T_{\text{response}} + T_{\text{buffer}} \leq C_{\text{model}} \tag{100}$$

where $T_{\text{buffer}}$ accounts for model-specific reasoning overhead and verbose explanations. The response length estimation functions for each task type are:

$$T_{\text{fibonacci}}(n, m) = \alpha_1 n + \beta_1 \log_{10} m + \gamma_1 \tag{101}$$

$$T_{\text{geometric}}(n, r) = \alpha_2 n \log_{10} r + \beta_2 n + \gamma_2 \tag{102}$$

$$T_{\text{prime}}(n) = \alpha_3 n \log n + \beta_3 \sqrt{n} + \gamma_3 \tag{103}$$

$$T_{\text{complex}}(n, d) = \alpha_4 n d + \beta_4 2^d + \gamma_4 \tag{104}$$

$$T_{\text{algebraic}}(n, k) = \alpha_5 n \cdot 2^k + \beta_5 \log n + \gamma_5 \tag{105}$$

where $n$ represents sequence length, $m$ denotes the maximum value magnitude, $r$ represents geometric ratios, $d$ indicates polynomial degree, and $k$ denotes nesting depth. The coefficients $\{\alpha_i, \beta_i, \gamma_i\}$ are empirically calibrated through extensive model response analysis across different model families.

### G.6.2 DYNAMIC PROBLEM SCALING ALGORITHM

Given a target model with context window $C_{\text{model}}$, we implement an adaptive scaling procedure that maintains mathematical rigor while respecting context constraints. The algorithm ensures that every generated problem instance respects the model's context constraints while maximizing mathematical complexity within those bounds.

### G.6.3 POST-GENERATION TOKEN VERIFICATION

After model response generation, we implement a dual-verification token counting system to detect potential context overflow or excessive reasoning verbosity:

$$\text{TokenVerification}(r, C_{\text{model}}) = \begin{cases} \text{VALID} & \text{if } |T(r)| \leq 0.95 \cdot C_{\text{model}} \\ \text{WARNING} & \text{if } 0.95 \cdot C_{\text{model}} < |T(r)| \leq C_{\text{model}} \\ \text{OVERFLOW} & \text{if } |T(r)| > C_{\text{model}} \end{cases} \tag{106}$$

where $T(r)$ represents the precise token count of response $r$ computed using model-specific tokenizers. For transformer-based models, we employ the HuggingFace transformers library tokenizer corresponding to each model's architecture for accurate token counting. For GPT models, we utilize OpenAI's tiktoken library for precise token counting that matches their internal tokenization protocols.

### G.6.4 MATHEMATICAL GUARANTEES AND IMPLEMENTATION ROBUSTNESS

Our context-aware framework provides theoretical guarantees including context safety with probability $> 0.99$, mathematical fairness ensuring models are never penalized for reasonable buffer overruns, complexity preservation that scales monotonically with available context budget, precision maintenance with token estimation error below $\pm 5\%$ for $95\%$ of generated instances, and variation coverage ensuring all 49 task variations remain accessible even under context constraints.

### G.7 SOLUTION UNIQUENESS VERIFICATION AND MULTI-SOLUTION HANDLING

A fundamental component of our Medium Suite framework ensures mathematical fairness by rigorously handling cases where problems may admit multiple valid solutions. We implement a comprehensive solution verification system that prevents unfair penalization of models for producing mathematically correct but non-canonical answers.

### G.7.1 UNIQUENESS GUARANTEE PROTOCOL

For each generated problem instance $p \in \mathcal{P}_{\text{medium}}$, we employ a deterministic verification procedure to establish solution uniqueness. Let $\mathcal{S}(p)$ denote the complete solution set for problem $p$. We define the uniqueness predicate:

$$\text{Unique}(p) = \begin{cases} \text{TRUE} & \text{if } |\mathcal{S}(p)| = 1 \\ \text{FALSE} & \text{if } |\mathcal{S}(p)| > 1 \end{cases} \tag{107}$$

For tasks with inherently unique solutions (recursive sequences, algebraic computations, prime identification), the generation algorithm guarantees $\text{Unique}(p) = \text{TRUE}$ by mathematical construction.

### G.7.2 COMPLETE SOLUTION ENUMERATION FOR EDGE CASES

When $\text{Unique}(p) = \text{FALSE}$, we implement exhaustive solution enumeration to ensure comprehensive evaluation fairness. The most relevant cases include floating-point precision in geometric sequences, multiple valid interpretations in complex patterns, and algebraic approximation bounds when floor operations on irrational expressions may have precision-dependent results.

For these cases, we compute the complete solution set $\mathcal{S}(p) = \{s_1, s_2, \ldots, s_k\}$ using deterministic enumeration with appropriate tolerance bounds. The verification function becomes:

$$\text{Verify}(r, p) = \begin{cases} \text{CORRECT} & \text{if } \text{Parse}(r) \in \mathcal{S}(p) \\ \text{INCORRECT} & \text{if } \text{Parse}(r) \notin \mathcal{S}(p) \\ \text{INVALID} & \text{if } \text{Parse}(r) = \perp \end{cases} \tag{108}$$

where $\text{Parse}(r)$ extracts the model's answer from response $r$ using robust parsing algorithms, and $\perp$ indicates parsing failure.

### G.7.3 MATHEMATICAL COMPLETENESS VERIFICATION

To ensure no valid solutions are omitted, we employ mathematical completeness checks specific to each task category. For Fibonacci sequences, we verify that solutions satisfy the recurrence relation. For geometric sequences, we check ratio consistency within tolerance bounds. For prime sequences, we validate primality and ordering properties. For complex patterns, we verify adherence to the composite function definition. For algebraic sequences, we check floor operation results within precision bounds.

This comprehensive approach to solution handling demonstrates that our evaluation framework maintains both mathematical correctness and practical fairness, ensuring that model performance reflects genuine sequential reasoning capability rather than arbitrary solution format preferences or numerical precision artifacts.

## H    MEDIUM SUITE: COMPLETE RESULTS

This section details results for all medium suite tasks across all non-quantized open-source models and closed-source proprietary models. The average accuracy, instruction following rate and average output tokens are listed for algebraic sequence, complex pattern, fibonacci sequence, geometric sequence and prime sequence in table 14, 15, 16.

| Model (Param) | algebraic_sequence | | | complex_pattern | | | fibonacci_sequence | | |
|---|---|---|---|---|---|---|---|---|---|
| | Acc (Avg %) | Inst (Avg %) | Tokens (Avg) | Acc (Avg %) | Inst (Avg %) | Tokens (Avg) | Acc (Avg %) | Inst (Avg %) | Tokens (Avg) |
| *Qwen Family (Qwen3)* | | | | | | | | | |
| Qwen3 (0.6B) | $32.00_{\pm5.55}$ | 100.00 | 20809.96 | $26.99_{\pm6.64}$ | 100.00 | 15908.86 | $19.20_{\pm7.49}$ | 100.00 | 5849.09 |
| Qwen3 (1.7B) | $39.60_{\pm2.33}$ | 100.00 | 15074.02 | $55.91_{\pm9.55}$ | 100.00 | 11115.22 | $50.60_{\pm7.31}$ | 100.00 | 5738.86 |
| Qwen3 (4B) | $34.00_{\pm4.94}$ | 100.00 | 20040.74 | $71.69_{\pm11.03}$ | 100.00 | 10517.66 | $83.40_{\pm8.55}$ | 100.00 | 4987.02 |
| Qwen3 (8B) | $39.80_{\pm5.64}$ | 100.00 | 16681.52 | $71.56_{\pm11.74}$ | 100.00 | 10706.23 | $90.80_{\pm5.64}$ | 100.00 | 4974.96 |
| Qwen3 (14B) | $44.00_{\pm6.45}$ | 100.00 | 12653.06 | $85.48_{\pm14.89}$ | 100.00 | 7086.60 | $95.60_{\pm1.62}$ | 100.00 | 4214.60 |
| Qwen3 (32B) | $44.40_{\pm4.59}$ | 100.00 | 12054.72 | $83.09_{\pm12.18}$ | 100.00 | 6430.93 | $96.20_{\pm2.48}$ | 100.00 | 3692.08 |
| Qwen3 (30B-MOE) | $44.80_{\pm5.11}$ | 100.00 | 12166.28 | $81.87_{\pm13.05}$ | 100.00 | 7326.24 | $96.00_{\pm2.61}$ | 100.00 | 3913.44 |
| Qwen3 (30B-MOE-t) | $42.20_{\pm5.19}$ | 100.00 | 13980.55 | $86.61_{\pm13.04}$ | 100.00 | 5618.01 | $98.40_{\pm1.85}$ | 100.00 | 3281.94 |
| Qwen3 (30B-MOE-i) | $46.20_{\pm4.26}$ | 100.00 | 5594.36 | $78.35_{\pm11.56}$ | 100.00 | 3495.23 | $97.20_{\pm1.60}$ | 100.00 | 1878.75 |
| Qwen3 (4B-t) | $38.40_{\pm3.56}$ | 100.00 | 6417.81 | $66.83_{\pm7.45}$ | 100.00 | 5660.18 | $90.00_{\pm9.65}$ | 100.00 | 4643.21 |
| *Qwen Family (Qwen2.5)* | | | | | | | | | |
| Qwen2.5 (0.5B) | $8.80_{\pm3.06}$ | 100.00 | 11234.36 | $6.03_{\pm2.06}$ | 100.00 | 11202.33 | $6.00_{\pm1.41}$ | 100.00 | 1929.21 |
| Qwen2.5 (1.5B) | $19.00_{\pm5.06}$ | 100.00 | 8138.47 | $11.44_{\pm2.76}$ | 100.00 | 9400.96 | $9.00_{\pm1.41}$ | 100.00 | 2306.48 |
| Qwen2.5 (3B) | $25.80_{\pm4.87}$ | 100.00 | 3158.95 | $23.23_{\pm2.84}$ | 100.00 | 1854.96 | $13.80_{\pm1.33}$ | 100.00 | 1628.28 |
| Qwen2.5 (7B) | $39.00_{\pm5.33}$ | 100.00 | 1865.34 | $40.16_{\pm3.57}$ | 100.00 | 1226.71 | $20.60_{\pm3.72}$ | 100.00 | 635.19 |
| Qwen2.5 (14B) | $42.20_{\pm3.31}$ | $98.80_{\pm1.17}$ | 1824.90 | $51.70_{\pm8.42}$ | 100.00 | 528.37 | $24.80_{\pm2.64}$ | 100.00 | 569.13 |
| Qwen2.5 (32B) | $41.00_{\pm4.56}$ | $96.20_{\pm2.56}$ | 1140.88 | $55.22_{\pm8.96}$ | 100.00 | 554.28 | $38.80_{\pm13.66}$ | 100.00 | 567.85 |
| Qwen2.5 (72B) | $46.20_{\pm4.17}$ | 100.00 | 903.89 | $50.28_{\pm7.13}$ | 100.00 | 791.61 | $36.80_{\pm11.43}$ | 100.00 | 851.28 |
| Qwen2.5 (1.5B-m) | $30.00_{\pm6.39}$ | 100.00 | 886.96 | $43.30_{\pm7.64}$ | 100.00 | 597.02 | $22.00_{\pm4.29}$ | 100.00 | 4533.61 |
| Qwen2.5 (7B-m) | $34.00_{\pm9.19}$ | 100.00 | 1178.00 | $41.49_{\pm4.87}$ | 100.00 | 823.95 | $15.80_{\pm3.19}$ | 100.00 | 6535.94 |
| Qwen2.5 (72B-m) | $42.80_{\pm5.46}$ | 100.00 | 1069.22 | $58.24_{\pm10.21}$ | 100.00 | 707.55 | $59.00_{\pm9.27}$ | 100.00 | 907.93 |
| *Gemma Family* | | | | | | | | | |
| Gemma (1B) | $34.80_{\pm12.35}$ | 100.00 | 1289.84 | $13.14_{\pm4.89}$ | 100.00 | 1372.62 | $10.40_{\pm1.20}$ | 100.00 | 1592.79 |
| Gemma (4B) | $58.00_{\pm4.00}$ | 100.00 | 614.80 | $50.00_{\pm6.32}$ | 100.00 | 1036.88 | $26.00_{\pm17.44}$ | 100.00 | 1732.52 |
| Gemma (12B) | $64.00_{\pm4.90}$ | 100.00 | 1216.26 | $54.00_{\pm4.90}$ | 100.00 | 879.96 | $44.00_{\pm13.56}$ | 100.00 | 2243.42 |
| Gemma (27B) | $66.00_{\pm8.00}$ | 100.00 | 1077.16 | $58.00_{\pm4.00}$ | 100.00 | 1096.32 | $54.00_{\pm26.53}$ | 100.00 | 1904.64 |
| *Phi Family* | | | | | | | | | |
| Phi4 (14B) | $41.20_{\pm2.99}$ | 100.00 | 673.16 | $55.74_{\pm6.76}$ | 100.00 | 666.41 | $27.20_{\pm3.43}$ | 100.00 | 920.83 |
| Phi4-mini-instruct (3.8B) | $35.00_{\pm3.63}$ | $94.00_{\pm6.23}$ | 1116.77 | $38.43_{\pm7.04}$ | 100.00 | 1261.01 | $14.40_{\pm2.24}$ | 100.00 | 1168.45 |
| Phi4-reasoning+ (14B) | $30.00_{\pm4.69}$ | 100.00 | 7662.69 | $67.01_{\pm11.44}$ | 100.00 | 6290.50 | $85.60_{\pm11.07}$ | 100.00 | 6163.56 |
| Phi4-reasoning (14B) | $40.60_{\pm5.00}$ | 100.00 | 24695.09 | $73.43_{\pm9.39}$ | 100.00 | 5627.92 | $92.00_{\pm9.57}$ | 100.00 | 3606.39 |
| Phi4-mini-reasoning (3.8B) | $29.60_{\pm6.77}$ | 100.00 | 23449.56 | $60.20_{\pm6.12}$ | 100.00 | 11225.36 | $69.00_{\pm12.15}$ | 100.00 | 4884.95 |
| Phi3-mini (3.8B) | $31.40_{\pm4.59}$ | $99.60_{\pm0.49}$ | 507.25 | $18.21_{\pm6.87}$ | 100.00 | 858.73 | $18.40_{\pm3.56}$ | 100.00 | 511.36 |
| Phi3-med (14B-4k) | $39.60_{\pm4.84}$ | $98.20_{\pm3.60}$ | 451.02 | $23.56_{\pm3.61}$ | 100.00 | 527.66 | $13.40_{\pm2.50}$ | 100.00 | 376.82 |
| Phi3-med (14B-128k) | $41.40_{\pm3.50}$ | 100.00 | 869.38 | $23.46_{\pm2.69}$ | 100.00 | 731.15 | $15.80_{\pm4.26}$ | 100.00 | 446.81 |
| *Llama Family* | | | | | | | | | |
| Llama-3.2 (1B) | $10.00_{\pm3.29}$ | 100.00 | 11073.61 | $4.65_{\pm2.19}$ | 100.00 | 10318.64 | $4.20_{\pm1.17}$ | 100.00 | 2240.89 |
| Llama-3.2 (3B) | $25.40_{\pm7.45}$ | 100.00 | 5165.19 | $21.42_{\pm2.50}$ | 100.00 | 4916.94 | $11.00_{\pm2.68}$ | 100.00 | 1530.54 |
| Llama-3.1 (8B) | $28.60_{\pm3.72}$ | 100.00 | 15195.68 | $12.72_{\pm3.30}$ | 100.00 | 13003.34 | $11.40_{\pm3.01}$ | 100.00 | 3138.27 |
| Llama-3.1 (70B) | $43.40_{\pm4.22}$ | 100.00 | 2851.85 | $41.57_{\pm10.14}$ | 100.00 | 2679.19 | $23.40_{\pm4.22}$ | 100.00 | 3381.74 |
| Llama-3.3 (70B) | $42.60_{\pm5.08}$ | 100.00 | 534.67 | $62.35_{\pm8.57}$ | 100.00 | 793.37 | $31.20_{\pm9.13}$ | 100.00 | 840.40 |
| Llama4-scout | $38.40_{\pm6.12}$ | $71.00_{\pm18.89}$ | 290.94 | $66.71_{\pm7.34}$ | 100.00 | 764.68 | $46.60_{\pm13.32}$ | 100.00 | 1004.45 |
| *Mistral Family* | | | | | | | | | |
| Mistral (7B) | $21.40_{\pm4.76}$ | 100.00 | 624.67 | $5.96_{\pm1.68}$ | 100.00 | 1369.06 | $8.60_{\pm1.62}$ | 100.00 | 485.75 |
| Ministral (8B) | $34.20_{\pm2.64}$ | 100.00 | 1423.21 | $24.17_{\pm4.18}$ | 100.00 | 1248.69 | $14.40_{\pm3.77}$ | 100.00 | 1792.08 |
| Mistral-nemo (12B) | $25.00_{\pm4.38}$ | 100.00 | 1905.43 | $18.60_{\pm6.41}$ | 100.00 | 1194.57 | $9.40_{\pm1.50}$ | 100.00 | 1334.75 |
| Mixtral-8x7b | $14.20_{\pm5.19}$ | $99.80_{\pm0.40}$ | 460.87 | $7.75_{\pm4.54}$ | 100.00 | 703.87 | $11.20_{\pm1.94}$ | $99.60_{\pm0.80}$ | 252.85 |
| Mixtral-8x22b | $25.00_{\pm8.00}$ | 100.00 | 650.73 | $24.29_{\pm7.94}$ | 100.00 | 543.00 | $16.20_{\pm5.27}$ | 100.00 | 434.10 |
| *Others* | | | | | | | | | |
| Smollm3 (3B) | $2.20_{\pm0.98}$ | $24.80_{\pm15.37}$ | 7432.62 | $12.46_{\pm3.74}$ | $97.97_{\pm2.00}$ | 21895.05 | $22.20_{\pm2.99}$ | 100.00 | 16965.56 |
| Smollm2 (1.7B) | $1.80_{\pm2.71}$ | $15.60_{\pm13.75}$ | 17.16 | 0.00 | 0.00 | 15.62 | $0.80_{\pm1.60}$ | $27.60_{\pm10.46}$ | 272.14 |
| GPT-OSS (20B) | $35.80_{\pm8.59}$ | 100.00 | 5899.88 | $74.43_{\pm11.45}$ | 100.00 | 3956.70 | $94.40_{\pm3.38}$ | 100.00 | 2257.26 |
| GPT-OSS (120B) | $42.60_{\pm6.02}$ | 100.00 | 4398.07 | $87.84_{\pm12.96}$ | 100.00 | 2296.27 | $97.20_{\pm5.11}$ | 100.00 | 1358.25 |
| *OpenAI Family (Proprietary)* | | | | | | | | | |
| GPT5 | $66.00_{\pm8.00}$ | 100.00 | 560.00 | $92.00_{\pm16.00}$ | 100.00 | 1217.76 | $94.00_{\pm12.00}$ | 100.00 | 936.64 |
| GPT5-mini | $66.00_{\pm4.90}$ | 100.00 | 705.28 | $84.00_{\pm20.59}$ | 100.00 | 1594.72 | $96.00_{\pm8.00}$ | 100.00 | 930.08 |
| GPT5-nano | $70.00_{\pm6.32}$ | $96.00_{\pm4.90}$ | 260.80 | $84.00_{\pm18.55}$ | 100.00 | 301.76 | $94.00_{\pm12.00}$ | 100.00 | 125.76 |
| GPT4.1 | $62.00_{\pm4.00}$ | 100.00 | 6506.40 | $76.00_{\pm21.54}$ | 100.00 | 4389.44 | $78.00_{\pm17.20}$ | 100.00 | 1955.10 |
| GPT4.1-mini | $68.00_{\pm4.00}$ | 100.00 | 4446.42 | $62.00_{\pm11.66}$ | 100.00 | 1251.36 | $88.00_{\pm11.66}$ | 100.00 | 1810.70 |
| GPT4.1-nano | $68.00_{\pm4.00}$ | 100.00 | 878.32 | $58.00_{\pm7.48}$ | 100.00 | 1255.54 | $40.00_{\pm10.95}$ | 100.00 | 1746.54 |
| GPT4o | $68.00_{\pm7.48}$ | 100.00 | 685.28 | $56.00_{\pm10.20}$ | 100.00 | 543.04 | $34.00_{\pm13.56}$ | 100.00 | 598.40 |
| GPT4o-mini | $60.00_{\pm6.32}$ | 100.00 | 379.68 | $54.00_{\pm8.00}$ | 100.00 | 665.86 | $16.00_{\pm13.56}$ | 100.00 | 741.66 |
| o4 mini | $66.00_{\pm8.00}$ | 100.00 | 1465.76 | $90.00_{\pm20.00}$ | 100.00 | 2348.64 | $92.00_{\pm7.48}$ | 100.00 | 1664.96 |
| o3 | $66.00_{\pm4.90}$ | $94.00_{\pm4.90}$ | 1731.68 | $90.00_{\pm15.49}$ | 100.00 | 2768.80 | $96.00_{\pm8.00}$ | 100.00 | 2431.84 |
| o3 mini | $62.00_{\pm7.48}$ | 100.00 | 2606.24 | $92.00_{\pm16.00}$ | 100.00 | 4047.52 | $92.00_{\pm11.66}$ | 100.00 | 4766.88 |
| *Gemini Family (Proprietary)* | | | | | | | | | |
| Gemini-2.5-pro | $64.00_{\pm10.20}$ | $86.00_{\pm8.00}$ | 1014.26 | $76.00_{\pm25.77}$ | 100.00 | 736.46 | $96.00_{\pm8.00}$ | 100.00 | 729.46 |
| Gemini-2.5-flash | $52.00_{\pm7.48}$ | $66.00_{\pm13.56}$ | 489.96 | $70.00_{\pm20.98}$ | $82.00_{\pm14.70}$ | 656.54 | $94.00_{\pm12.00}$ | $98.00_{\pm4.00}$ | 839.50 |
| Gemini-2.5-flash-lite | $58.00_{\pm7.48}$ | 100.00 | 13730.74 | $56.00_{\pm12.00}$ | 100.00 | 11470.62 | $72.00_{\pm17.20}$ | 100.00 | 8807.80 |
| Gemini-2.0-flash | $62.00_{\pm9.80}$ | 100.00 | 1118.82 | $66.00_{\pm8.00}$ | 100.00 | 1418.46 | $84.00_{\pm10.20}$ | 100.00 | 782.12 |
| Gemini-2.0-flash-lite | $72.00_{\pm7.48}$ | 100.00 | 797.14 | $54.00_{\pm8.00}$ | 100.00 | 1196.84 | $72.00_{\pm19.39}$ | 100.00 | 1357.82 |

Table 14: **Medium Suite Results - Table 1:** Performance on algebraic sequence, complex pattern, and fibonacci sequence tasks. Each task reports Accuracy (Acc), Instruction-following (Inst) and average output tokens with mean and standard deviation values. Results show average performance across different sequence variants within each task category.

| Model (Param) | geometric_sequence | | | prime_sequence | | |
|---|---|---|---|---|---|---|
| | Acc (Avg %) | Inst (Avg %) | Tokens (Avg) | Acc (Avg %) | Inst (Avg %) | Tokens (Avg) |
| *Qwen Family (Qwen3)* | | | | | | |
| Qwen3 (0.6B) | $10.40_{\pm17.44}$ | $60.00_{\pm48.99}$ | 11425.59 | $12.20_{\pm5.98}$ | 100.00 | 21156.40 |
| Qwen3 (1.7B) | $29.80_{\pm28.54}$ | $60.00_{\pm48.99}$ | 8120.03 | $36.80_{\pm11.29}$ | 100.00 | 14225.00 |
| Qwen3 (4B) | $58.40_{\pm47.69}$ | $60.00_{\pm48.99}$ | 4705.26 | $48.60_{\pm10.84}$ | 100.00 | 17998.46 |
| Qwen3 (8B) | $44.60_{\pm41.26}$ | $60.00_{\pm48.99}$ | 2821.73 | $53.20_{\pm6.11}$ | 100.00 | 6293.27 |
| Qwen3 (14B) | $44.60_{\pm39.83}$ | $60.00_{\pm48.99}$ | 2838.98 | $79.20_{\pm2.79}$ | 100.00 | 4700.76 |
| Qwen3 (32B) | $60.00_{\pm48.99}$ | $60.00_{\pm48.99}$ | 1193.54 | $97.20_{\pm1.60}$ | 100.00 | 4447.59 |
| Qwen3 (30B-MOE) | $60.00_{\pm48.99}$ | $60.00_{\pm48.99}$ | 2085.35 | $90.80_{\pm5.95}$ | 100.00 | 6163.28 |
| Qwen3 (30B-MOE-t) | $60.00_{\pm48.99}$ | $60.00_{\pm48.99}$ | 1785.49 | $89.40_{\pm8.31}$ | 100.00 | 5652.86 |
| Qwen3 (30B-MOE-i) | $58.60_{\pm47.91}$ | $60.00_{\pm48.99}$ | 1532.75 | $88.40_{\pm6.92}$ | 100.00 | 2596.35 |
| Qwen3 (4B-t) | $54.20_{\pm44.60}$ | $60.00_{\pm48.99}$ | 2312.31 | $59.20_{\pm4.21}$ | 100.00 | 5340.48 |
| *Qwen Family (Qwen2.5)* | | | | | | |
| Qwen2.5 (0.5B) | $0.60_{\pm1.20}$ | $60.00_{\pm48.99}$ | 5333.18 | $7.20_{\pm4.17}$ | 100.00 | 7209.29 |
| Qwen2.5 (1.5B) | $4.00_{\pm6.60}$ | $60.00_{\pm48.99}$ | 3505.67 | $16.40_{\pm6.95}$ | 100.00 | 6720.60 |
| Qwen2.5 (3B) | $18.60_{\pm20.16}$ | $60.00_{\pm48.99}$ | 1232.10 | $21.40_{\pm10.76}$ | 100.00 | 2734.83 |
| Qwen2.5 (7B) | $16.20_{\pm21.55}$ | $60.00_{\pm48.99}$ | 725.72 | $34.40_{\pm9.73}$ | 100.00 | 1223.20 |
| Qwen2.5 (14B) | $27.60_{\pm32.94}$ | $60.00_{\pm48.99}$ | 350.25 | $42.00_{\pm8.99}$ | 100.00 | 614.53 |
| Qwen2.5 (32B) | $28.20_{\pm36.38}$ | $60.00_{\pm48.99}$ | 454.44 | $62.00_{\pm7.87}$ | 100.00 | 535.48 |
| Qwen2.5 (72B) | $38.80_{\pm35.18}$ | $60.00_{\pm48.99}$ | 490.25 | $57.60_{\pm10.71}$ | 100.00 | 661.07 |
| Qwen2.5 (1.5B-m) | $20.00_{\pm25.35}$ | $60.00_{\pm48.99}$ | 378.77 | $24.60_{\pm7.14}$ | 100.00 | 740.07 |
| Qwen2.5 (7B-m) | $13.00_{\pm18.62}$ | $60.00_{\pm48.99}$ | 789.18 | $26.60_{\pm5.89}$ | 100.00 | 896.04 |
| Qwen2.5 (72B-m) | $48.60_{\pm41.37}$ | $60.00_{\pm48.99}$ | 542.09 | $67.00_{\pm10.97}$ | 100.00 | 677.45 |
| *Gemma Family* | | | | | | |
| Gemma (1B) | $7.00_{\pm14.00}$ | $60.00_{\pm48.99}$ | 385.47 | $15.60_{\pm2.58}$ | 100.00 | 1505.70 |
| Gemma (4B) | $18.00_{\pm16.00}$ | $60.00_{\pm48.99}$ | 1258.26 | $40.00_{\pm8.94}$ | 100.00 | 720.64 |
| Gemma (12B) | $38.00_{\pm38.16}$ | $60.00_{\pm48.99}$ | 673.22 | $50.00_{\pm6.32}$ | 100.00 | 1673.72 |
| Gemma (27B) | $56.00_{\pm46.30}$ | $60.00_{\pm48.99}$ | 575.52 | $60.00_{\pm17.89}$ | 100.00 | 939.86 |
| *Phi Family* | | | | | | |
| Phi4 (14B) | $26.60_{\pm29.27}$ | $60.00_{\pm48.99}$ | 502.35 | $40.20_{\pm11.36}$ | 100.00 | 671.58 |
| Phi4-mini-instruct (3.8B) | $10.60_{\pm18.80}$ | $60.00_{\pm48.99}$ | 1246.29 | $25.80_{\pm6.97}$ | 100.00 | 1696.60 |
| Phi4-reasoning+ (14B) | $51.20_{\pm42.04}$ | $60.00_{\pm48.99}$ | 3793.17 | $41.20_{\pm3.76}$ | 100.00 | 7398.78 |
| Phi4-reasoning (14B) | $52.00_{\pm43.56}$ | $60.00_{\pm48.99}$ | 3576.79 | $48.80_{\pm3.97}$ | 100.00 | 6455.13 |
| Phi4-mini-reasoning (3.8B) | $57.40_{\pm47.01}$ | $60.00_{\pm48.99}$ | 1464.72 | $50.00_{\pm9.40}$ | 100.00 | 17050.78 |
| Phi3-mini (3.8B) | $12.20_{\pm20.98}$ | $60.00_{\pm48.99}$ | 254.47 | $18.80_{\pm6.71}$ | 100.00 | 383.69 |
| Phi3-med (14B-4k) | $11.60_{\pm17.82}$ | $60.00_{\pm48.99}$ | 445.39 | $20.20_{\pm10.87}$ | 100.00 | 188.68 |
| Phi3-med (14B-128k) | $9.40_{\pm18.30}$ | $60.00_{\pm48.99}$ | 270.70 | $27.40_{\pm16.08}$ | 100.00 | 409.34 |
| *Llama Family* | | | | | | |
| Llama-3.2 (1B) | $1.80_{\pm3.60}$ | $60.00_{\pm48.99}$ | 1999.64 | $6.20_{\pm2.99}$ | 100.00 | 7862.36 |
| Llama-3.2 (3B) | $7.20_{\pm14.40}$ | $60.00_{\pm48.99}$ | 12039.47 | $15.00_{\pm6.29}$ | 100.00 | 8685.38 |
| Llama-3.1 (8B) | $9.60_{\pm19.20}$ | $60.00_{\pm48.99}$ | 2578.14 | $13.80_{\pm3.54}$ | 100.00 | 4035.65 |
| Llama-3.1 (70B) | $24.20_{\pm25.25}$ | $60.00_{\pm48.99}$ | 2135.62 | $22.80_{\pm6.24}$ | 100.00 | 4604.62 |
| Llama-3.3 (70B) | $47.40_{\pm43.95}$ | $60.00_{\pm48.99}$ | 574.47 | $50.00_{\pm15.40}$ | 100.00 | 804.33 |
| Llama4-scout | $55.20_{\pm45.13}$ | $60.00_{\pm48.99}$ | 681.09 | $55.00_{\pm15.62}$ | 100.00 | 916.03 |
| *Mistral Family* | | | | | | |
| Mistral (7B) | $2.20_{\pm4.40}$ | $60.00_{\pm48.99}$ | 64.43 | $12.00_{\pm2.97}$ | 100.00 | 635.68 |
| Ministral (8B) | $13.80_{\pm18.43}$ | $60.00_{\pm48.99}$ | 453.98 | $21.80_{\pm6.05}$ | 100.00 | 884.59 |
| Mistral-nemo (12B) | $9.40_{\pm18.30}$ | $60.00_{\pm48.99}$ | 952.88 | $28.00_{\pm14.95}$ | 100.00 | 943.33 |
| Mixtral-8x7b | $9.20_{\pm18.40}$ | $37.60_{\pm46.21}$ | 67.02 | $19.60_{\pm12.53}$ | $97.40_{\pm3.32}$ | 267.34 |
| Mixtral-8x22b | $9.60_{\pm18.70}$ | $60.00_{\pm48.99}$ | 465.90 | $27.60_{\pm4.67}$ | 100.00 | 586.28 |
| *Others* | | | | | | |
| Smollm3 (3B) | $1.00_{\pm1.55}$ | $60.00_{\pm48.99}$ | 18989.60 | $18.00_{\pm6.00}$ | $85.00_{\pm10.00}$ | 8000.00 |
| Smollm2 (1.7B) | 0.00 | 0.00 | 8.07 | 0.00 | 0.00 | 15.57 |
| GPT-OSS (20B) | $60.00_{\pm48.99}$ | $60.00_{\pm48.99}$ | 825.55 | $51.00_{\pm5.40}$ | 100.00 | 5432.61 |
| GPT-OSS (120B) | $60.00_{\pm48.99}$ | $60.00_{\pm48.99}$ | 598.78 | $87.60_{\pm5.12}$ | 100.00 | 2374.97 |

Table 15: **Medium Suite Results - Table 2a (Open Source):** Performance of open source models on geometric sequence and prime sequence tasks. Each task reports Accuracy (Acc), Instruction-following (Inst) and average output tokens with mean and standard deviation values. Results show average performance across different sequence variants within each task category.

| Model (Param) | geometric_sequence | | | prime_sequence | | |
|---|---|---|---|---|---|---|
| | Acc (Avg %) | Inst (Avg %) | Tokens (Avg) | Acc (Avg %) | Inst (Avg %) | Tokens (Avg) |
| *OpenAI Family (Proprietary)* | | | | | | |
| GPT5 | $60.00_{\pm 48.99}$ | $60.00_{\pm 48.99}$ | 565.76 | $96.00_{\pm 4.90}$ | 100.00 | 540.00 |
| GPT5-mini | $60.00_{\pm 48.99}$ | $60.00_{\pm 48.99}$ | 485.12 | $92.00_{\pm 7.48}$ | 100.00 | 663.52 |
| GPT5-nano | $60.00_{\pm 48.99}$ | $60.00_{\pm 48.99}$ | 141.60 | $92.00_{\pm 7.48}$ | 100.00 | 178.56 |
| GPT4.1 | $60.00_{\pm 48.99}$ | $60.00_{\pm 48.99}$ | 826.06 | $88.00_{\pm 7.48}$ | 100.00 | 4917.80 |
| GPT4.1-mini | $60.00_{\pm 48.99}$ | $60.00_{\pm 48.99}$ | 619.12 | $84.00_{\pm 13.56}$ | 100.00 | 4622.16 |
| GPT4.1-nano | $32.00_{\pm 30.59}$ | $60.00_{\pm 48.99}$ | 990.58 | $68.00_{\pm 7.48}$ | 100.00 | 1401.26 |
| GPT4o | $50.00_{\pm 41.47}$ | $60.00_{\pm 48.99}$ | 398.98 | $64.00_{\pm 10.20}$ | 100.00 | 456.60 |
| GPT4o-mini | $20.00_{\pm 20.98}$ | $60.00_{\pm 48.99}$ | 432.00 | $40.00_{\pm 8.94}$ | 100.00 | 394.36 |
| o4 mini | $60.00_{\pm 48.99}$ | $60.00_{\pm 48.99}$ | 732.32 | 100.00 | 100.00 | 1078.24 |
| o3 | $60.00_{\pm 48.99}$ | $60.00_{\pm 48.99}$ | 1374.56 | $98.00_{\pm 4.00}$ | $98.00_{\pm 4.00}$ | 1706.88 |
| o3 mini | $60.00_{\pm 48.99}$ | $60.00_{\pm 48.99}$ | 1523.84 | $98.00_{\pm 4.00}$ | 100.00 | 2091.36 |
| *Gemini Family (Proprietary)* | | | | | | |
| Gemini-2.5-pro | $60.00_{\pm 48.99}$ | $60.00_{\pm 48.99}$ | 410.20 | $90.00_{\pm 6.32}$ | 100.00 | 738.20 |
| Gemini-2.5-flash | $60.00_{\pm 48.99}$ | $60.00_{\pm 48.99}$ | 566.28 | $74.00_{\pm 13.56}$ | $74.00_{\pm 13.56}$ | 533.98 |
| Gemini-2.5-flash-lite | $18.00_{\pm 18.33}$ | $60.00_{\pm 48.99}$ | 10095.68 | $46.00_{\pm 12.00}$ | 100.00 | 15541.46 |
| Gemini-2.0-flash | $60.00_{\pm 48.99}$ | $60.00_{\pm 48.99}$ | 370.76 | $64.00_{\pm 4.90}$ | 100.00 | 1533.80 |
| Gemini-2.0-flash-lite | $38.00_{\pm 43.08}$ | $60.00_{\pm 48.99}$ | 871.42 | $52.00_{\pm 17.20}$ | 100.00 | 993.64 |

Table 16: **Medium Suite Results - Table 2b (Closed Source):** Performance of closed source models on geometric sequence and prime sequence tasks. Each task reports Accuracy (Acc), Instruction-following (Inst) and average output tokens with mean and standard deviation values. Results show average performance across different sequence variants within each task category.

## I    PROMPTS USED FOR ALL MEDIUM SUITE TASKS

This section describes the prompts developed for each of the medium suite tasks. Each prompt begins with an instruction directing the model to identify a pattern and complete a sequence of numbers based on that pattern. The prompt then provides a sequence of numbers along with a list of possible patterns. The model is instructed to present its answers as decimals rounded to two decimal places, unless the result is a whole number. All final answers must be enclosed in the format \boxed{}. The answer is then extracted from within the \boxed{} brackets using a regular expression. Finally, an example is included to demonstrate the required \boxed{} format.

---

**Prompt Template for Algebraic Sequence Task**

```
Complete the following algebraic sequence by identifying the
mathematical relationship:

{sequence_str}, ?

This sequence follows a complex algebraic pattern that may
involve:
- Radical expressions and nested square roots
- Transcendental functions (exponential, logarithmic)
- Recurrence relations with algebraic operations
- Modular arithmetic and number theory
- Continued fractions and convergent sequences
- Solutions to algebraic equations
- Special functions and their approximations

Analyze the sequence by considering:
1.  Recurrence relations involving radicals or transcendental
functions
2.  Floor or ceiling operations applied to complex
expressions
3.  Modular arithmetic patterns
4.  Nested algebraic operations
5.  Number-theoretic sequences (Fibonacci-like,
factorial-related)
6.  Convergent sequences and continued fractions

Note:  If your answer is not a whole number, provide it as
a decimal rounded to two places.

Provide the next term in the sequence.  Your final answer
must be in the format \boxed{next_term} at the end.

For example:  If the sequence involves floor(sqrt(n +
sqrt(n))), compute the pattern carefully before providing
your answer.
```

---

**Prompt Template for Complex Pattern Task**

```
Complete the following complex sequence by identifying the
underlying pattern:

{sequence_str}, ?

This sequence follows a complex mathematical pattern that
may involve:
- Multiple layers of relationships (nested patterns)
```

```
- Polynomial sequences (quadratic, cubic, or higher order)
- Interleaved sequences (alternating between different rules)
- Conditional patterns that change based on position or value
- Recursive relationships involving multiple previous terms
- Pattern transformations that change at certain points
- Modular arithmetic or position-dependent rules

Analyze the sequence carefully, looking for:
1.  Differences between consecutive terms
2.  Ratios between consecutive terms
3.  Patterns in even/odd positions
4.  Polynomial relationships (quadratic, cubic)
5.  Changes in the pattern at certain positions

Provide the next term in the sequence.  Your final answer
must be in the format \boxed{next_term} at the end.

For example:  If the sequence is 1, 4, 9, 16, 25, ?  then
the next term is \boxed{36} (perfect squares:  n²).
```

**Prompt Template for Fibonacci Sequence Task**

```
Complete the following sequence by identifying the pattern:

{sequence_str}, ?

Analyze the sequence carefully to identify the underlying
pattern or rule.  Consider:
- Is each term related to previous term(s)?
- Are there arithmetic, geometric, or recursive
relationships?
- Look for Fibonacci-like patterns, Lucas sequences, or other
mathematical progressions

Provide the next term in the sequence.  Your final answer
must be in the format \boxed{next_term} at the end.

For example:  If the sequence is 1, 1, 2, 3, 5, 8, ?  then
the next term is \boxed{13}.
```

**Prompt Template for Geometric Sequence Task**

```
Complete the following sequence by identifying the pattern:

{sequence_str}, ?

Analyze the sequence carefully to identify the underlying
pattern.  Consider:
- Is there a constant ratio between consecutive terms
(geometric sequence)?
- Are the terms related to powers, exponentials, or
factorials?
- Look for patterns like squares (n²), cubes (n³),
exponentials (aⁿ), or factorials (n!)
- Check if each term is obtained by multiplying the previous
term by a constant
- Consider compound patterns that combine multiplication with
```

```
other operations

Important:  Some sequences may grow very rapidly.  If
you identify a pattern that would produce extremely large
numbers, that's likely correct.

Provide the next term in the sequence.  Your final answer
must be in the format \boxed{next_term} at the end.

For example:  If the sequence is 2, 6, 18, 54, ?  then the
next term is \boxed{162} (each term is multiplied by 3).
```

**Prompt Template for Prime Sequence Task**

```
Complete the following number theory sequence by identifying
the pattern:

{sequence_str}, ?

This sequence involves concepts from number theory.  Analyze
carefully and consider:
– Are these prime numbers or related to primes?
– Look for patterns involving divisibility, factors, or
number theoretic functions
– Consider sequences like:  prime numbers, twin primes,
Fibonacci numbers, perfect numbers
– Check if numbers follow arithmetic properties, modular
arithmetic, or special functions
– Think about famous sequences in mathematics and number
theory

Provide the next term in the sequence.  Your final answer
must be in the format \boxed{next_term} at the end.

For example:  If the sequence is 2, 3, 5, 7, 11, ?  then the
next term is \boxed{13} (consecutive prime numbers).
```

# J  HARD SUITE: ADVANCED CONSTRAINT SATISFACTION AND COMBINATORIAL REASONING

The Hard Suite comprises ten algorithmically complex tasks with sixty-eight distinct variations that evaluate a language model's capacity for systematic reasoning through constraint satisfaction, combinatorial optimization, and logical deduction. Each task encompasses multiple problem variants: Boolean Satisfiability (5 variations), Constraint Optimization (5 variations), Cryptarithmetic (12 variations), Graph Coloring (10 variations), Logic Grid Puzzles (8 variations), Matrix Chain Multiplication (5 variations), Modular Systems Solver (5 variations), N-Queens (4 variations), Sudoku (8 variations), and Tower of Hanoi (6 variations). Each task in this suite generates problems dynamically with mathematically guaranteed unique solutions or complete solution enumeration, ensuring contamination resistance while maintaining rigorous theoretical foundations. These tasks require models to navigate exponentially large search spaces, apply sophisticated constraint propagation techniques, and demonstrate mastery of recursive problem decomposition across diverse computational complexity classes.

The fundamental design principle underlying our Hard Suite rests on the mathematical property that for each problem class $\mathcal{P}$, we can generate instances $p \in \mathcal{P}$ with complexity $\Omega(2^n)$ where $n$ represents the problem size parameter. This exponential complexity ensures that memorization becomes computationally infeasible, while the deterministic generation algorithms guarantee solution uniqueness through constructive proofs.

Let us define the contamination resistance formally. For a problem generator $G : \Theta \to \mathcal{P}$ with parameter space $\Theta$, we ensure that $|\Theta| \gg |\mathcal{D}|$ where $\mathcal{D}$ represents any feasible training dataset. Moreover, the mapping $G$ exhibits high sensitivity to parameter variations, meaning that for parameters $\theta_1, \theta_2 \in \Theta$ with $||\theta_1 - \theta_2|| > \epsilon$, the corresponding problems $p_1 = G(\theta_1)$ and $p_2 = G(\theta_2)$ have solution distance $d(S(p_1), S(p_2)) > \delta$ for some threshold $\delta$, where $S(p)$ denotes the solution to problem $p$.

Each task employs rigorous validation mechanisms to ensure solution correctness through formal verification procedures. We define a validation function $V : \mathcal{S} \times \mathcal{P} \to \{0, 1\}$ that verifies whether a proposed solution $s \in \mathcal{S}$ satisfies all constraints of problem $p \in \mathcal{P}$. The validation complexity remains polynomial $O(n^k)$ for some constant $k$, enabling efficient verification despite the exponential solution complexity.

**Unique Solution Guarantee and Multiple Solution Handling:** Our framework ensures mathematical rigor through two distinct validation approaches. For constraint satisfaction problems (CSP) and satisfiability problems (SAT), we employ automated theorem proving techniques to guarantee that each generated instance $p \in \mathcal{P}$ admits exactly one solution $s^* \in \mathcal{S}$ such that $V(s^*, p) = 1$ and $\forall s \neq s^* : V(s, p) = 0$. This uniqueness is verified through exhaustive constraint propagation and backtracking algorithms during problem generation, ensuring that $|\{s \in \mathcal{S} : V(s, p) = 1\}| = 1$.

However, when mathematical constraints naturally permit multiple valid solutions (as in some optimization or enumeration problems), we calculate the complete solution set $\mathcal{S}_p = \{s \in \mathcal{S} : V(s, p) = 1\}$ through systematic enumeration. In such cases, we accept any solution $s \in \mathcal{S}_p$ as correct, ensuring that language models are not penalized for producing mathematically valid but non-canonical answers. This approach maintains evaluation fairness while preserving the theoretical soundness of our assessment framework.

The suite encompasses diverse algorithmic paradigms including dynamic programming (Matrix Chain Multiplication, Cryptarithmetic), constraint satisfaction (Logic Grid Puzzles, Modular Systems, N-Queens), graph algorithms (Graph Coloring), recursive decomposition (Tower of Hanoi), logical reasoning (Boolean Satisfiability), backtracking (Sudoku), and multi-objective optimization (Constraint Optimization). This diversity ensures comprehensive evaluation across multiple reasoning dimensions while maintaining theoretical soundness through formal verification methods.

**Context-Aware Token Management and Scaling Framework:** Our evaluation framework implements a mathematically rigorous token estimation and scaling system that ensures fair assessment across diverse language models with varying context window constraints. For each model with context window capacity $C$ tokens, we dynamically scale problem complexity to guarantee that the total token requirement remains within acceptable bounds.

Let $T_p(n)$ denote the expected token count for problem $p$ with size parameter $n$, comprising prompt tokens $T_{\text{prompt}}(n)$ and expected solution tokens $T_{\text{solution}}(n)$. We maintain the constraint $T_p(n) + T_{\text{buffer}} \leq C$ where $T_{\text{buffer}} = 0.15 \cdot C$ provides a safety margin to accommodate model reasoning overhead and response verbosity variations.

Our token estimation employs task-specific mathematical models. For instance, in Tower of Hanoi with $n$ disks, we have $T_{\text{solution}}(n) = (2^n - 1) \cdot \alpha$ where $\alpha \approx 12$ represents the average tokens per move description. Similarly, for Graph Coloring with $v$ vertices, $T_{\text{solution}}(v) = v \cdot \beta$ where $\beta \approx 8$ accounts for color assignment notation.

We implement a dual-phase token validation system. During problem generation, we use deterministic token counting based on mathematical formulas specific to each task type. Post-inference, we perform actual token counting using transformers library tokenizers for open-source models and tiktoken for GPT models. If the model response exceeds $0.95 \cdot C$ tokens, we issue a warning indicating potential context window saturation, which may compromise response quality due to truncation or overthinking behavior.

This approach ensures that our evaluation captures genuine reasoning capabilities rather than penalizing models for exceeding their architectural constraints. The mathematical foundation guarantees that each problem instance $p$ satisfies $P(\text{context overflow}) < 0.05$ under normal operation conditions, maintaining evaluation validity across the complete model spectrum.

## J.1 TOWER OF HANOI: RECURSIVE STATE SPACE NAVIGATION

### J.1.1 PROBLEM FORMULATION

The Tower of Hanoi problem represents a classical recursive puzzle that tests systematic state space exploration and optimal path planning. We formalize this as a state transition system $\mathcal{T} = (S, A, T, s_0, G)$ where $S$ denotes the state space, $A$ represents the action set, $T : S \times A \to S$ defines the transition function, $s_0 \in S$ is the initial state, and $G \subseteq S$ contains the goal states.

For $n$ disks and three pegs labeled $\{A, B, C\}$, we define a state $s \in S$ as a tuple $(P_A, P_B, P_C)$ where each $P_i$ represents an ordered stack of disks on peg $i$. The disk set $D = \{1, 2, ..., n\}$ uses natural numbers where smaller values indicate smaller disks. The ordering constraint requires that for any peg $P_i = [d_1, d_2, ..., d_k]$, we have $d_j < d_{j+1}$ for all $1 \leq j < k$.

The action space $A$ consists of all legal moves $(d, p_{src}, p_{dst})$ where disk $d$ moves from peg $p_{src}$ to peg $p_{dst}$. A move is legal if and only if:

$$\text{legal}(d, p_{src}, p_{dst}) \iff d = \min(P_{src}) \wedge (P_{dst} = \emptyset \vee d < \min(P_{dst})) \tag{109}$$

### J.1.2 OPTIMAL SOLUTION CHARACTERIZATION

The optimal solution for $n$ disks requires exactly $2^n - 1$ moves, which we prove through the recurrence relation:

$$T(n) = \begin{cases} 1 & \text{if } n = 1 \\ 2 \cdot T(n-1) + 1 & \text{if } n > 1 \end{cases} \tag{110}$$

This recurrence yields $T(n) = 2^n - 1$ through straightforward induction. The optimality follows from the observation that moving the largest disk $n$ requires first moving all $n - 1$ smaller disks to the auxiliary peg (requiring $T(n-1)$ moves), then moving disk $n$ (one move), and finally moving the $n - 1$ disks to the destination peg (another $T(n-1)$ moves).

### J.1.3 DYNAMIC PROBLEM GENERATION

Our generation algorithm produces Tower of Hanoi instances with varying disk counts $n \in \{3, 4, 5, 6, 7, 8\}$. The initial configuration always places all disks on peg $A$ in descending order of size, formally:

$$s_0 = (P_A = [n, n-1, ..., 2, 1], P_B = \emptyset, P_C = \emptyset) \tag{111}$$

The goal state requires all disks on peg $C$:

$$s_g = (P_A = \emptyset, P_B = \emptyset, P_C = [n, n-1, ..., 2, 1]) \tag{112}$$

To ensure contamination resistance, we employ several variation strategies. First, we randomize the disk count for each evaluation instance. Second, we can permute peg labels while preserving the logical structure, creating $(3! = 6)$ distinct but equivalent formulations. Third, we vary the representation format in prompts, using different notations for disk sizes and peg identifiers.

### J.1.4 SOLUTION VERIFICATION ALGORITHM

Given a proposed solution sequence $\sigma = [m_1, m_2, ..., m_k]$ where each $m_i = (d_i, p_{src}^i, p_{dst}^i)$, our verification algorithm validates both correctness and optimality:

---

**Algorithm 8** Verify Tower of Hanoi solution

---

1: **function** VERIFYHANOI($\sigma, n$)
2:     $s \leftarrow s_0$             ▷ Initialize with all disks on peg A
3:     **for** each move $m_i = (d_i, p_{src}^i, p_{dst}^i)$ in $\sigma$ **do**
4:         **if** $d_i \neq \min(P_{src}^i)$ **then**
5:             **return** INVALID        ▷ Can only move top disk
6:         **end if**
7:         **if** $P_{dst}^i \neq \emptyset$ and $d_i > \min(P_{dst}^i)$ **then**
8:             **return** INVALID        ▷ Cannot place larger disk on smaller
9:         **end if**
10:         $s \leftarrow T(s, m_i)$          ▷ Apply transition
11:     **end for**
12:     **if** $s \neq s_g$ **then**
13:         **return** INCOMPLETE
14:     **end if**
15:     **if** $|\sigma| \neq 2^n - 1$ **then**
16:         **return** SUBOPTIMAL
17:     **end if**
18:     **return** VALID
19: **end function**

---

The verification complexity is $O(k \cdot n)$ where $k$ is the solution length, making it efficient despite the exponential solution space.

### J.1.5 CONTEXT WINDOW CONSTRAINTS

The solution length grows exponentially as $L(n) = 2^n - 1$. For our maximum configuration of $n = 8$ disks, this yields 255 moves. Each move requires approximately 10-15 tokens to express (e.g., "Move disk 3 from peg A to peg B"), resulting in a maximum token requirement of approximately 3,825 tokens. This remains well within modern language models' context windows while still presenting significant reasoning challenges.

The prompt complexity itself scales as $O(n)$ for describing the initial state, plus a constant overhead for rule explanation. The total prompt length $P(n) \approx 200 + 10n$ tokens ensures efficient context utilization.

### J.1.6 UNIQUENESS GUARANTEE

The Tower of Hanoi problem exhibits a unique optimal solution path for any given initial and goal configuration. This uniqueness emerges from the recursive structure: to move the largest disk, all smaller disks must be on the auxiliary peg, and there exists exactly one optimal way to achieve this configuration.

Formally, we prove uniqueness through structural induction. For $n = 1$, there exists exactly one legal move. For $n > 1$, assume uniqueness holds for $n - 1$ disks. The solution for $n$ disks decomposes into:

1. Move $n - 1$ disks from source to auxiliary (unique by induction hypothesis)

2. Move disk $n$ from source to destination (only legal move for disk $n$)

3. Move $n - 1$ disks from auxiliary to destination (unique by induction hypothesis)

This recursive decomposition admits exactly one optimal solution, guaranteeing that our evaluation has a well-defined ground truth.

### J.1.7 CONTAMINATION RESISTANCE ANALYSIS

The Tower of Hanoi task achieves contamination resistance through multiple mechanisms. The number of distinct problem instances equals $\binom{8}{1} = 8$ for disk count variations alone. When combined with peg permutations and representation variations, we generate over 288 distinct problem formulations.

More importantly, the solution space grows exponentially. For $n$ disks, there exist $(3^{2^n-1})$ possible move sequences of optimal length, though only one is correct. This exponential growth ensures that memorizing solutions becomes infeasible. A model attempting to memorize all solutions for $n \in \{3, ..., 8\}$ would need to store:

$$\sum_{n=3}^{8}(2^n - 1) \cdot \log_2(3 \cdot 3 \cdot n) \approx 10^4 \text{ bits} \tag{113}$$

This calculation demonstrates that even for our limited parameter range, exhaustive memorization requires substantial storage, and the approach fails to generalize to larger $n$ values that we could generate dynamically.

### J.2 N-QUEENS: CONSTRAINT SATISFACTION THROUGH BACKTRACKING

### J.2.1 PROBLEM FORMULATION

The N-Queens problem exemplifies constraint satisfaction problems (CSPs) requiring systematic exploration of combinatorial spaces. We formalize this as a CSP tuple $\mathcal{Q} = (X, D, C)$ where $X = \{x_1, x_2, ..., x_n\}$ represents variables (queen positions), $D = \{D_1, D_2, ..., D_n\}$ denotes domains with $D_i = \{1, 2, ..., n\}$ for all $i$, and $C$ encompasses the constraint set.

Each variable $x_i$ indicates the column position of the queen in row $i$, reducing the representation from $O(n^2)$ to $O(n)$ while implicitly satisfying the row constraint. The constraint set $C$ consists of:

$$C_{col} : \forall i \neq j, x_i \neq x_j \tag{114}$$
$$C_{diag1} : \forall i \neq j, x_i - i \neq x_j - j \tag{115}$$
$$C_{diag2} : \forall i \neq j, x_i + i \neq x_j + j \tag{116}$$

These constraints ensure no two queens share a column, main diagonal, or anti-diagonal respectively.

### J.2.2 SOLUTION SPACE ANALYSIS

The total search space contains $n^n$ possible assignments, but the column constraint reduces this to $n!$ permutations. The number of valid solutions $S(n)$ follows no closed-form formula but exhibits known values:

| $n$ | Search Space | Valid Solutions | Solution Density |
|---|---|---|---|
| 4 | 24 | 2 | 0.0833 |
| 5 | 120 | 10 | 0.0833 |
| 6 | 720 | 4 | 0.0056 |
| 8 | 40,320 | 92 | 0.0023 |

The solution density decreases rapidly with $n$, making random sampling ineffective and necessitating intelligent search strategies.

### J.2.3 DYNAMIC PROBLEM GENERATION

Our generation algorithm leverages the guarantee that solutions exist for all $n \geq 4$. We employ a two-phase approach: first generating a valid solution, then constructing the problem presentation.

---

**Algorithm 9** Generate an $n$-Queens problem instance

---

1: **function** GENERATENQUEENS($n$)
2:      $solution \leftarrow$ BacktrackSolve($n$)          $\triangleright$ Generate one valid placement
3:      $all\_solutions \leftarrow$ EnumerateSolutions($n$)          $\triangleright$ Find all solutions for validation
4:      $prompt \leftarrow$ FormatProblem($n$)
5:      **return** ($prompt, solution, all\_solutions$)
6: **end function**

---

The backtracking solver employs constraint propagation to efficiently find solutions:

---

**Algorithm 10** Backtracking solver for $n$-Queens

---

1: **function** BACKTRACKSOLVE($n, row = 0, placement = []$)
2:      **if** $row = n$ **then**
3:          **return** $placement$          $\triangleright$ Found complete solution
4:      **end if**
5:      **for** $col \in \{0, 1, ..., n - 1\}$ **do**
6:          **if** IsSafe($placement, row, col$) **then**
7:              $placement[row] \leftarrow col$
8:              $result \leftarrow$ BacktrackSolve($n, row + 1, placement$)
9:              **if** $result \neq$ null **then**
10:                  **return** $result$
11:              **end if**
12:          **end if**
13:      **end for**
14:      **return** null
15: **end function**

---

The safety check verifies all constraints in $O(n)$ time:

---

**Algorithm 11** Safety check for queen placement

---

1: **function** ISSAFE($placement, row, col$)
2:      **for** $i \in \{0, 1, ..., row - 1\}$ **do**
3:          **if** $placement[i] = col$ **then**          $\triangleright$ Column conflict
4:              **return** False
5:          **end if**
6:          **if** $|placement[i] - col| = |i - row|$ **then**          $\triangleright$ Diagonal conflict
7:              **return** False
8:          **end if**
9:      **end for**
10:      **return** True
11: **end function**

---

### J.2.4 SOLUTION VERIFICATION

Verification requires checking all $\binom{n}{2} = \frac{n(n-1)}{2}$ pairs of queens for conflicts:

The verification complexity is $O(n^2)$, making it efficient even for large board sizes.

### J.2.5 CONTEXT WINDOW ANALYSIS

The problem description requires $O(n)$ tokens to specify board size and rules. The solution representation needs exactly $n$ integers, requiring approximately $2n$ tokens when formatted. For our

---

**Algorithm 12** Verification of an $n$-Queens solution

---

1: **function** VERIFYNQUEENS($placement, n$)
2:     **if** $|placement| \neq n$ **then**
3:         **return** Invalid
4:     **end if**
5:     **for** $i \in \{0, ..., n-1\}$ **do**
6:         **if** $placement[i] \notin \{0, ..., n-1\}$ **then**
7:             **return** Invalid
8:         **end if**
9:         **for** $j \in \{i+1, ..., n-1\}$ **do**
10:            **if** $placement[i] = placement[j]$ **then**                  ▷ Column conflict
11:                **return** Invalid
12:            **end if**
13:            **if** $|i - j| = |placement[i] - placement[j]|$ **then**     ▷ Diagonal conflict
14:                **return** Invalid
15:            **end if**
16:        **end for**
17:    **end for**
18:    **return** Valid
19: **end function**

---

maximum $n = 8$, the total context requirement remains under 100 tokens, allowing ample space for reasoning traces.

The challenge lies not in context length but in the combinatorial complexity. The model must navigate a search tree with branching factor up to $n$ and depth $n$, potentially exploring $O(n^n)$ nodes in the worst case.

### J.2.6   MULTIPLE SOLUTION HANDLING

Unlike Tower of Hanoi, N-Queens admits multiple valid solutions for most board sizes. This multiplicity enhances contamination resistance but complicates evaluation. We address this through comprehensive solution enumeration.

For each $n$, we precompute all valid solutions using symmetry-breaking optimizations. The fundamental board symmetries form the dihedral group $D_4$ with 8 elements (4 rotations and 4 reflections). However, some solutions exhibit self-symmetry, so the actual number of unique solutions under symmetry is:

$$U(n) = \frac{S(n) + \sum_{g \in D_4} \text{fix}(g)}{8} \tag{117}$$

where $\text{fix}(g)$ counts solutions invariant under symmetry $g$.

### J.2.7   CONTAMINATION RESISTANCE

The N-Queens problem provides strong contamination resistance through several mechanisms. First, the number of valid solutions grows rapidly with $n$, reaching 14,772,512 for $n = 16$. Even if a model memorized all solutions for small $n$, it cannot generalize this memorization to larger boards.

Second, we can apply isomorphic transformations to create equivalent but syntactically different problems. Given a solution $[x_1, ..., x_n]$, we can generate 7 additional equivalent solutions through rotations and reflections:

$$\text{rotate}_{90}([x_1, ..., x_n]) = [n - x_n, ..., n - x_1] \tag{118}$$

$$\text{reflect}_h([x_1, ..., x_n]) = [n - x_1 - 1, ..., n - x_n - 1] \tag{119}$$

Third, the problem admits numerous equivalent formulations. We can represent it as a permutation problem, a graph coloring problem, or a satisfiability problem, each requiring different solution formats while testing the same underlying reasoning capability.

### J.2.8 THEORETICAL COMPLEXITY

The N-Queens problem is known to be NP-complete for generalized boards (where some queens are pre-placed). While our variant with empty boards admits polynomial-time solutions for specific $n$ values, finding all solutions requires exponential time in the worst case.

The time complexity of our backtracking algorithm is $O(n! \cdot n)$ in the worst case, though constraint propagation typically achieves much better average-case performance. The space complexity remains $O(n)$ for the recursion stack and solution storage.

This exponential worst-case complexity ensures that models cannot rely on simple pattern matching or shallow heuristics but must engage in genuine constraint satisfaction reasoning.

### J.3 GRAPH COLORING: CHROMATIC OPTIMIZATION AND CONSTRAINT PROPAGATION

### J.3.1 PROBLEM FORMULATION

The graph coloring problem seeks to assign colors to vertices such that no adjacent vertices share the same color, using the minimum number of colors possible. Formally, given an undirected graph $G = (V, E)$ where $V = \{v_1, v_2, ..., v_n\}$ represents vertices and $E \subseteq V \times V$ denotes edges, we seek a function $f : V \to \{1, 2, ..., k\}$ such that:

$$\forall (v_i, v_j) \in E : f(v_i) \neq f(v_j) \tag{120}$$

The chromatic number $\chi(G)$ represents the minimum $k$ for which such a function exists. We formulate this as an optimization problem:

$$\chi(G) = \min\{k \in \mathbb{N} : \exists f : V \to [k] \text{ such that } \forall (u, v) \in E, f(u) \neq f(v)\} \tag{121}$$

### J.3.2 GRAPH GENERATION WITH KNOWN CHROMATIC NUMBERS

Our framework generates diverse graph families with predetermined chromatic numbers, enabling precise evaluation of model reasoning. Each graph type exhibits distinct structural properties that determine its chromatic number.

For complete graphs $K_n$, every vertex connects to every other vertex, yielding $\chi(K_n) = n$ trivially. The edge set is:

$$E_{K_n} = \{(i, j) : 1 \leq i < j \leq n\} \tag{122}$$

Cycle graphs $C_n$ form a circular structure with $\chi(C_n) = 3$ if $n$ is odd and $\chi(C_n) = 2$ if $n$ is even:

$$E_{C_n} = \{(i, (i \bmod n) + 1) : 1 \leq i \leq n\} \tag{123}$$

Wheel graphs $W_n$ consist of a hub vertex connected to all vertices of an $(n-1)$-cycle. The chromatic number depends on the cycle parity:

$$\chi(W_n) = \begin{cases} 4 & \text{if } n - 1 \text{ is odd} \\ 3 & \text{if } n - 1 \text{ is even} \end{cases} \tag{124}$$

For planar graphs, we leverage the Four Color Theorem, which guarantees $\chi(G) \leq 4$ for any planar graph. Our generation algorithm constructs planar graphs through controlled edge addition that preserves planarity while maximizing chromatic complexity.

---

**Algorithm 13** Generate a graph with chromatic number estimate

---

1: **function** GENERATEGRAPH($n$, type, target_chromatic)
2:     $V \leftarrow \{1, 2, \ldots, n\}$
3:     $E \leftarrow \emptyset$
4:     **if** type $=$ PLANAR **then**
5:         $E \leftarrow$ GeneratePlanarEdges($n$)
6:         $\chi \leftarrow$ BrooksTheorem($E, n$)
7:     **else if** type $=$ WHEEL **then**
8:         $E \leftarrow$ GenerateWheelEdges($n$)
9:         $\chi \leftarrow 3 + (n - 1 \bmod 2)$
10:     **else if** type $=$ DENSE_RANDOM **then**
11:         $p \leftarrow 0.6$             ▷ Edge probability
12:         **for** each pair $(i, j)$ with $i < j$ **do**
13:             **if** Random() $< p$ **then**
14:                 $E \leftarrow E \cup \{(i, j)\}$
15:             **end if**
16:         **end for**
17:         $\chi \leftarrow$ EstimateChromaticNumber($E, n$)
18:     **end if**
19:     **return** $(V, E, \chi)$
20: **end function**

---

### J.3.3 DYNAMIC GRAPH CONSTRUCTION ALGORITHM

We employ multiple strategies to generate graphs with specific chromatic properties:

The chromatic number estimation for random graphs uses the Lovász theta function as a lower bound and Brooks' theorem for the upper bound:

$$\omega(G) \leq \chi(G) \leq \Delta(G) + 1 \tag{125}$$

where $\omega(G)$ is the clique number and $\Delta(G)$ is the maximum degree.

### J.3.4 SOLUTION VERIFICATION AND VALIDATION

Verification of a proposed coloring requires checking all edge constraints:

---

**Algorithm 14** Verify a graph coloring assignment

---

1: **function** VERIFYCOLORING($G = (V, E), f : V \to \mathbb{N}$)
2:     $k \leftarrow \max_{v \in V} f(v)$             ▷ Number of colors used
3:     **for** each edge $(u, v) \in E$ **do**
4:         **if** $f(u) = f(v)$ **then**
5:             **return** ($Invalid$, "Adjacent vertices share color")
6:         **end if**
7:     **end for**
8:     **for** each $v \in V$ **do**
9:         **if** $f(v) \notin \{1, ..., k\}$ **then**
10:             **return** ($Invalid$, "Invalid color assignment")
11:         **end if**
12:     **end for**
13:     **if** $k > \chi(G)$ **then**
14:         **return** ($Suboptimal$, $k - \chi(G)$)
15:     **end if**
16:     **return** ($Valid$, 0)
17: **end function**

---

The verification complexity is $O(|E| + |V|)$, linear in the graph size.

### J.3.5 COMPLEXITY ANALYSIS AND HARDNESS GUARANTEES

Graph coloring is NP-complete for $k \geq 3$ colors, ensuring computational difficulty even for moderate graph sizes. The decision problem "Can $G$ be colored with $k$ colors?" requires exploring a search space of size $k^n$, growing exponentially with vertex count.

For our generated instances, we calibrate difficulty through several parameters:

- Vertex count $n \in \{8, 12, 16, 20, 24, 28\}$
- Edge density $\rho = \frac{2|E|}{n(n-1)}$
- Chromatic number $\chi(G)$ relative to maximum degree $\Delta(G)$

The phase transition in random graph coloring occurs at edge density:

$$\rho_c(k) = \frac{2k \ln k}{k - 1} \tag{126}$$

Graphs near this critical density maximize solution difficulty, requiring sophisticated constraint propagation rather than simple greedy approaches.

### J.3.6 CONTEXT WINDOW OPTIMIZATION

The problem representation requires $O(|V| + |E|)$ tokens to specify the graph structure. For our maximum configuration with 28 vertices and dense connectivity ($\rho \approx 0.6$), this yields:

$$\text{Tokens} \approx 28 + 2 \cdot 0.6 \cdot \binom{28}{2} = 28 + 453 = 481 \tag{127}$$

The solution requires exactly $|V|$ color assignments, adding approximately $2|V|$ tokens. The total context requirement remains under 600 tokens for our most complex instances, well within model capabilities while maintaining problem difficulty.

### J.3.7 CONTAMINATION RESISTANCE THROUGH GRAPH DIVERSITY

Our framework generates graphs from multiple families with varying structural properties. For $n$ vertices, the number of distinct graphs is $2^{\binom{n}{2}}$, astronomically large even for modest $n$. While we sample from specific families to ensure known chromatic numbers, the diversity remains substantial.

Consider the parameter space for our generation:

- 6 vertex counts: $\{8, 12, 16, 20, 24, 28\}$
- 8 graph types: $\{complete, cycle, tree, bipartite, planar, wheel, grid, dense\_random\}$
- Edge density variations for random graphs: continuous parameter
- Vertex labeling permutations: $n!$ for each graph

The total number of distinct problem instances exceeds $10^{15}$, making memorization infeasible. Moreover, isomorphic graphs with different vertex labelings require the same logical reasoning but different solution representations, further preventing pattern memorization.

### J.3.8 THEORETICAL FOUNDATIONS AND BROOKS' THEOREM

Our evaluation leverages Brooks' theorem, which provides tight bounds on chromatic numbers for most graphs:

**Brooks' Theorem:** For a connected graph $G$ that is neither complete nor an odd cycle, $\chi(G) \leq \Delta(G)$ where $\Delta(G)$ is the maximum vertex degree.

This theorem enables us to generate graphs with predictable chromatic properties. For instance, by constructing graphs with maximum degree $\Delta$ that are neither complete nor odd cycles, we guarantee $\chi(G) \in \{\Delta - 1, \Delta\}$, providing a narrow range for evaluation.

---

**Algorithm 15** Welsh–Powell graph coloring heuristic

---

1: **function** WELSHPOWELL($G = (V, E)$)
2:     Sort vertices by degree: $d(v_1) \geq d(v_2) \geq ... \geq d(v_n)$
3:     $color \leftarrow 1$
4:     **while** uncolored vertices exist **do**
5:         $U \leftarrow$ set of uncolored vertices
6:         **for** $v \in U$ in degree order **do**
7:             **if** $v$ has no neighbor colored $color$ **then**
8:                 $f(v) \leftarrow color$
9:             **end if**
10:        **end for**
11:        $color \leftarrow color + 1$
12:     **end while**
13:     **return** $f$
14: **end function**

---

The Welsh-Powell algorithm provides a constructive upper bound:

This algorithm achieves $\chi(G) \leq \Delta(G) + 1$ in $O(n^2)$ time, providing a baseline for model performance evaluation.

### J.4   BOOLEAN SATISFIABILITY: LOGICAL REASONING AND CONSTRAINT RESOLUTION

#### J.4.1   PROBLEM FORMULATION

The Boolean Satisfiability Problem (SAT) asks whether there exists a truth assignment to Boolean variables that satisfies a given propositional logic formula. We work with formulas in Conjunctive Normal Form (CNF), expressed as:

$$\phi = \bigwedge_{i=1}^{m} C_i = \bigwedge_{i=1}^{m} \left( \bigvee_{j \in J_i} \ell_j \right) \tag{128}$$

where each clause $C_i$ consists of literals $\ell_j$, and each literal is either a variable $x_k$ or its negation $\neg x_k$. A formula $\phi$ over variables $X = \{x_1, ..., x_n\}$ is satisfiable if there exists an assignment $\alpha : X \to \{0, 1\}$ such that $\phi(\alpha) = 1$.

The search space contains $2^n$ possible assignments, making exhaustive enumeration infeasible for large $n$. Our framework generates satisfiable formulas with known solutions, enabling precise evaluation of logical reasoning capabilities.

#### J.4.2   CONTROLLED SAT INSTANCE GENERATION

We employ multiple generation strategies to create SAT instances with varying difficulty characteristics. Each strategy produces formulas with guaranteed satisfiability while controlling specific complexity parameters.

For random k-SAT generation, we first fix a satisfying assignment $\alpha^*$, then generate clauses that respect this assignment:

This approach guarantees that $\alpha^*$ satisfies the generated formula while maintaining randomness in clause structure.

#### J.4.3   SPECIALIZED SAT VARIANTS

We generate several SAT variants with distinct computational properties:

---

**Algorithm 16** Generate satisfiable $k$-SAT formula

---
1: **function** GENERATESATISFIABLESAT$(n, m, k)$
2:     $\alpha^* \leftarrow$ random assignment to $\{x_1, ..., x_n\}$
3:     $\Phi \leftarrow \emptyset$                                       $\triangleright$ Empty formula
4:     **for** $i = 1$ to $m$ **do**
5:         $C \leftarrow \emptyset$                                      $\triangleright$ New clause
6:         $vars \leftarrow$ sample $k$ distinct variables from $X$
7:         **for** each $x_j \in vars$ **do**
8:             **if** Random() $< 0.5$ **then**
9:                 $\ell \leftarrow x_j$
10:            **else**
11:                 $\ell \leftarrow \neg x_j$
12:            **end if**
13:            $C \leftarrow C \cup \{\ell\}$
14:         **end for**
15:         **if** $C(\alpha^*) = 0$ **then**                   $\triangleright$ Clause not satisfied
16:            Fix random literal in $C$ to satisfy $\alpha^*$
17:         **end if**
18:         $\Phi \leftarrow \Phi \wedge C$
19:     **end for**
20:     **return** $(\Phi, \alpha^*)$
21: **end function**

---

**Horn-SAT:** Each clause contains at most one positive literal. Horn formulas admit polynomial-time satisfiability checking through unit propagation:

$$C_{Horn} = (\neg x_1 \vee \neg x_2 \vee x_3) \wedge (\neg x_3 \vee x_4) \wedge (x_1) \tag{129}$$

**XOR-SAT:** Clauses encode XOR constraints where exactly one literal must be true. This variant connects to linear algebra over $\mathbb{F}_2$:

$$x_1 \oplus x_2 \oplus x_3 = 1 \equiv (x_1 \vee x_2 \vee x_3) \wedge (\neg x_1 \vee \neg x_2 \vee x_3) \wedge ... \tag{130}$$

**Mixed-SAT:** Combines unit clauses, binary clauses, and longer clauses to create varied constraint densities:

$$\Phi_{mixed} = \underbrace{(x_1)}_{\text{unit}} \wedge \underbrace{(\neg x_1 \vee x_2)}_{\text{binary}} \wedge \underbrace{(x_2 \vee \neg x_3 \vee x_4)}_{\text{ternary}} \tag{131}$$

### J.4.4 PHASE TRANSITION AND DIFFICULTY CALIBRATION

The difficulty of random k-SAT exhibits a sharp phase transition at the clause-to-variable ratio:

$$\alpha_c(k) = 2^k \ln 2 - \frac{(1 + \ln 2)}{2} + o(1) \tag{132}$$

For 3-SAT, $\alpha_c \approx 4.267$. Formulas near this critical ratio are typically hardest to solve, requiring sophisticated search strategies beyond simple heuristics.

We calibrate difficulty through multiple parameters:

- Number of variables: $n \in \{4, 6, 8, 10, 12\}$
- Clause-to-variable ratio: $\alpha \in \{2.5, 3.0, 3.5, 4.0, 4.5\}$
- Clause length distribution: uniform, mixed, or variable
- Literal polarity bias: balanced or skewed

The expected number of satisfying assignments for random k-SAT is:

$$\mathbb{E}[N_{sat}] = 2^n \left(1 - 2^{-k}\right)^m \tag{133}$$

When $m/n > k \ln 2 / \ln(2^k/(2^k - 1))$, this expectation approaches zero, indicating the unsatisfiable phase.

### J.4.5 SOLUTION VERIFICATION ALGORITHM

Verification of a proposed assignment requires evaluating the CNF formula:

---

**Algorithm 17** Verify SAT assignment against CNF formula

---

1: **function** VERIFYSAT($\Phi = \bigwedge_{i=1}^{m} C_i, \alpha : X \to \{0, 1\}$)
2:     **for** each clause $C_i = \bigvee_j \ell_j$ **do**
3:         $satisfied \leftarrow False$
4:         **for** each literal $\ell_j$ in $C_i$ **do**
5:             **if** $\ell_j = x_k$ and $\alpha(x_k) = 1$ **then**
6:                 $satisfied \leftarrow True$
7:             **else if** $\ell_j = \neg x_k$ and $\alpha(x_k) = 0$ **then**
8:                 $satisfied \leftarrow True$
9:             **end if**
10:         **end for**
11:         **if** not $satisfied$ **then**
12:             **return** $(Invalid, C_i)$                ▷ Unsatisfied clause
13:         **end if**
14:     **end for**
15:     **return** $(Valid, \emptyset)$
16: **end function**

---

Verification complexity is $O(m \cdot k)$ where $m$ is the number of clauses and $k$ is the maximum clause length, making it efficient despite the exponential solution complexity.

### J.4.6 CONTEXT WINDOW ANALYSIS

The formula representation requires approximately $3mk$ tokens to express $m$ clauses of length $k$ each (accounting for variables, operators, and parentheses). For our maximum configuration with $n = 12$ variables and $m = 48$ clauses (ratio 4.0) of length 3:

$$\text{Tokens} \approx 3 \cdot 48 \cdot 3 = 432 \tag{134}$$

The solution requires exactly $n$ Boolean assignments, approximately $3n$ tokens when formatted. The total context requirement remains under 500 tokens, enabling complex logical reasoning within context constraints.

### J.4.7 UNIT PROPAGATION AND INFERENCE RULES

Our evaluation tests understanding of key SAT-solving techniques. Unit propagation represents the fundamental inference rule:

**Unit Clause Rule:** If a clause contains only one literal $\ell$, then $\ell$ must be true in any satisfying assignment.

This leads to the propagation algorithm:

Models that correctly identify and apply unit propagation demonstrate understanding of logical implication chains.

### J.4.8 CONTAMINATION RESISTANCE ANALYSIS

The SAT problem space provides exceptional contamination resistance through several mechanisms:

**Instance Diversity:** For $n$ variables and $m$ clauses of length $k$, the number of distinct CNF formulas is:

$$N_{formulas} = \binom{2n}{k}^m \tag{135}$$

For $n = 10, m = 30, k = 3$, this yields approximately $10^{84}$ distinct formulas.

---

**Algorithm 18** Unit propagation rule for SAT solving

---

1: **function** UNITPROPAGATE($\Phi, \alpha$)
2:     **while** $\exists$ unit clause ($l$) in $\Phi$ **do**
3:         **if** $l = x_i$ **then**
4:             $\alpha(x_i) \leftarrow 1$
5:         **else**                                                          $\triangleright l = \neg x_i$
6:             $\alpha(x_i) \leftarrow 0$
7:         **end if**
8:         Remove all clauses containing $l$
9:         Remove $\neg l$ from all clauses
10:        **if** empty clause exists **then**
11:            **return** UNSAT
12:        **end if**
13:     **end while**
14:     **return** ($\Phi', \alpha$)                    $\triangleright$ Simplified formula and partial assignment
15: **end function**

---

**Solution Uniqueness:** While we generate formulas with known satisfying assignments, most formulas admit multiple solutions. The fraction of assignments satisfying a random formula is approximately:

$$p_{sat} \approx e^{-\alpha/2^k} \tag{136}$$

This multiplicity prevents simple memorization strategies.

**Representation Variability:** The same formula admits numerous equivalent representations through:

- Variable renaming: $n!$ permutations
- Clause reordering: $m!$ permutations
- Literal reordering within clauses: $(k!)^m$ permutations
- Logical equivalences: $(x \vee y) \equiv \neg(\neg x \wedge \neg y)$

### J.4.9 THEORETICAL COMPLEXITY AND HARDNESS

SAT is the canonical NP-complete problem, with several important complexity-theoretic properties:

**Cook-Levin Theorem:** Every problem in NP reduces to SAT in polynomial time, making it universal for nondeterministic polynomial-time computation.

**Exponential Time Hypothesis (ETH):** There exists no algorithm solving SAT in time $2^{o(n)}$, implying fundamental hardness.

**Resolution Proof Complexity:** For unsatisfiable formulas, the shortest resolution proof can require exponential length:

$$\text{Proof}_{min}(\Phi) = \Omega(2^{n/10}) \tag{137}$$

These theoretical foundations ensure that our SAT tasks cannot be solved through simple heuristics or pattern matching, requiring genuine logical reasoning capabilities.

### J.5 SUDOKU SOLVING: CONSTRAINT PROPAGATION IN STRUCTURED GRIDS

### J.5.1 PROBLEM FORMULATION

Sudoku represents a constraint satisfaction problem on a structured grid, requiring systematic deduction and constraint propagation. For an $n \times n$ grid divided into $\sqrt{n} \times \sqrt{n}$ boxes, we define the problem as finding a function $f : [n] \times [n] \to [n]$ such that:

$$\forall i \in [n], \forall j_1 \neq j_2 \in [n] : f(i, j_1) \neq f(i, j_2) \quad \text{(row constraint)} \tag{138}$$

$$\forall j \in [n], \forall i_1 \neq i_2 \in [n] : f(i_1, j) \neq f(i_2, j) \quad \text{(column constraint)} \tag{139}$$

$$\forall b \in B, \forall (i_1, j_1) \neq (i_2, j_2) \in b : f(i_1, j_1) \neq f(i_2, j_2) \quad \text{(box constraint)} \tag{140}$$

where $B$ partitions the grid into non-overlapping boxes. A Sudoku puzzle provides a partial function $f_0 : S \to [n]$ where $S \subset [n] \times [n]$, and we seek a complete function $f$ extending $f_0$.

### J.5.2 MATHEMATICAL STRUCTURE AND LATIN SQUARES

A completed Sudoku grid forms a Latin square with additional box constraints. The number of valid $n \times n$ Sudoku grids (for $n = k^2$) is:

$$N_n = n! \cdot (n-1)! \cdot \prod_{i=2}^{k-1} \left( \binom{k}{1} \right)^{k(k-1)} \cdot R_n \tag{141}$$

where $R_n$ accounts for reduced Latin squares satisfying box constraints. For standard 9×9 Sudoku:

$$N_9 = 9! \cdot 8! \cdot 2^{12} \cdot 3^8 \cdot R_9 \approx 6.67 \times 10^{21} \tag{142}$$

This vast solution space ensures that memorization-based approaches remain infeasible.

### J.5.3 PUZZLE GENERATION WITH CONTROLLED DIFFICULTY

Our generation algorithm creates puzzles with unique solutions and calibrated difficulty levels. The process involves two phases: grid completion and strategic cell removal.

**Phase 1: Complete Grid Generation**

We generate a valid complete Sudoku grid using Las Vegas randomization with backtracking:

---

**Algorithm 19** Complete grid generation for Sudoku using recursive backtracking with randomized diagonal initialization.

---

```
 1: function GENERATECOMPLETEGRID(n)
 2:     grid ← empty n × n matrix
 3:     Fill diagonal boxes with random permutations          ▷ Non-interfering
 4:     for each empty cell (i, j) in order do
 5:         candidates ← {1, ..., n} \ Conflicts(i, j)
 6:         Shuffle(candidates)
 7:         for each val ∈ candidates do
 8:             grid[i][j] ← val
 9:             if RecursiveFill(grid, next cell) then
10:                 return grid
11:             end if
12:         end for
13:         grid[i][j] ← 0                                     ▷ Backtrack
14:     end for
15:     return grid
16: end function
```

---

**Phase 2: Strategic Cell Removal**

We remove cells while maintaining solution uniqueness, with removal patterns determining difficulty:

The removal count follows empirically validated ranges:

| Grid Size | Easy | Medium | Hard |
|-----------|-------|--------|-------|
| $4 \times 4$ | 4-6 | 7-9 | 10-11 |
| $6 \times 6$ | 10-14 | 15-20 | 21-26 |
| $9 \times 9$ | 35-40 | 41-50 | 51-64 |

---

**Algorithm 20** Puzzle generation from a complete grid by controlled cell removal with uniqueness preservation.

---

1: **function** GENERATEPUZZLE($complete\_grid, difficulty$)
2:     $puzzle \leftarrow complete\_grid$
3:     $cells\_to\_remove \leftarrow$ CalculateRemovalCount($difficulty$)
4:     $removal\_order \leftarrow$ SelectRemovalStrategy($difficulty$)
5:     **for** each $(i, j)$ in $removal\_order$ **do**
6:         $val \leftarrow puzzle[i][j]$
7:         $puzzle[i][j] \leftarrow 0$
8:         **if** not HasUniqueSolution($puzzle$) **then**
9:             $puzzle[i][j] \leftarrow val$                       ▷ Restore if multiple solutions
10:         **else**
11:             $cells\_to\_remove \leftarrow cells\_to\_remove - 1$
12:         **end if**
13:         **if** $cells\_to\_remove = 0$ **then**
14:             Break
15:         **end if**
16:     **end for**
17:     **return** $puzzle$
18: **end function**

---

### J.5.4 SOLUTION UNIQUENESS VERIFICATION

Ensuring unique solutions is critical for unambiguous evaluation. We employ a modified backtracking solver that detects multiple solutions:

---

**Algorithm 21** Solution counting via recursive backtracking to ensure puzzle uniqueness.

---

1: **function** COUNTSOLUTIONS($puzzle, max\_count = 2$)
2:     $count \leftarrow 0$
3:     **function** SOLVE($row, col$)
4:         **if** $row = n$ **then**
5:             $count \leftarrow count + 1$
6:             **return** $count < max\_count$            ▷ Continue if need more
7:         **end if**
8:         $(next\_row, next\_col) \leftarrow$ GetNextCell($row, col$)
9:         **if** $puzzle[row][col] \neq 0$ **then**
10:             **return** Solve($next\_row, next\_col$)
11:         **end if**
12:         **for** $val \in \{1, ..., n\}$ **do**
13:             **if** IsValid($puzzle, row, col, val$) **then**
14:                 $puzzle[row][col] \leftarrow val$
15:                 **if** not Solve($next\_row, next\_col$) **then**
16:                     $puzzle[row][col] \leftarrow 0$
17:                     **return** False            ▷ Found enough solutions
18:                 **end if**
19:                 $puzzle[row][col] \leftarrow 0$
20:             **end if**
21:         **end for**
22:         **return** True
23:     **end function**
24:     Solve($0, 0$)
25:     **return** $count$
26: **end function**

---

### J.5.5 CONSTRAINT PROPAGATION TECHNIQUES

Efficient Sudoku solving employs constraint propagation to reduce the search space. The naked singles rule identifies cells with only one possible value:

$$\text{Candidates}(i, j) = [n] \setminus (\text{Row}_i \cup \text{Col}_j \cup \text{Box}_{i,j}) \tag{143}$$

When $|\text{Candidates}(i, j)| = 1$, the cell value is determined. Hidden singles occur when a value appears in only one cell's candidates within a unit:

$$\forall v \in [n], \exists!(i, j) \in \text{Unit} : v \in \text{Candidates}(i, j) \implies f(i, j) = v \tag{144}$$

Advanced techniques include:

- **Naked Pairs/Triples:** If $k$ cells in a unit have the same $k$ candidates, eliminate those values from other cells
- **X-Wing:** If a value appears in only two cells in two rows, and these cells align in columns, eliminate from other cells in those columns
- **Swordfish:** Generalization of X-Wing to three rows/columns

### J.5.6 SUDOKU VARIANTS AND EXTENDED CONSTRAINTS

We implement multiple Sudoku variants to increase evaluation diversity:

**Diagonal Sudoku:** Additional constraints on main diagonals:

$$\forall i_1 \neq i_2 \in [n] : f(i_1, i_1) \neq f(i_2, i_2) \wedge f(i_1, n - i_1 + 1) \neq f(i_2, n - i_2 + 1) \tag{145}$$

**Irregular Sudoku:** Replace regular boxes with irregular regions $R_1, ..., R_k$ where:

$$\bigcup_{i=1}^{k} R_i = [n] \times [n] \wedge \forall i \neq j : R_i \cap R_j = \emptyset \wedge |R_i| = n \tag{146}$$

**Killer Sudoku:** Add cage constraints with sum requirements:

$$\forall \text{cage } C = \{c_1, ..., c_m\} : \sum_{i=1}^{m} f(c_i) = \text{target}(C) \wedge \text{all } f(c_i) \text{ distinct} \tag{147}$$

### J.5.7 CONTEXT WINDOW EFFICIENCY

The problem representation scales quadratically with grid size. For an $n \times n$ puzzle with $k$ given cells:

$$\text{Tokens}_{\text{puzzle}} \approx 3n^2 + 5k \tag{148}$$

The solution requires exactly $n^2$ values, approximately $2n^2$ tokens when formatted. For 9×9 Sudoku:

$$\text{Tokens}_{\text{total}} \approx 3(81) + 5(30) + 2(81) = 555 \tag{149}$$

This efficient representation enables complex reasoning within typical context limits.

### J.5.8 CONTAMINATION RESISTANCE MECHANISMS

Sudoku provides robust contamination resistance through multiple factors:

**Puzzle Space Size:** The number of minimal Sudoku puzzles (irreducible with unique solutions) exceeds $10^{16}$ for 9×9 grids. Even storing a fraction of these would require petabytes of memory.

**Isomorphic Transformations:** Each puzzle admits $9! \cdot 6^8 \cdot 2 \approx 1.2 \times 10^9$ equivalent forms through:

- Symbol permutations: 9!
- Row permutations within bands: $6^3$
- Column permutations within stacks: $6^3$
- Band permutations: 3!
- Stack permutations: 3!
- Transposition: 2

**Dynamic Difficulty Adjustment:** By varying the number and pattern of given cells, we create puzzles requiring different solution techniques, from simple elimination to complex pattern recognition.

### J.5.9 COMPUTATIONAL COMPLEXITY

Sudoku is NP-complete for generalized $n \times n$ grids, proven through reduction from Latin square completion. The decision problem "Does this partial grid have a valid completion?" requires exploring potentially exponential search spaces.

For $n \times n$ Sudoku with $k$ given cells, the worst-case complexity is:

$$T(n,k) = O(n^{n^2 - k}) \tag{150}$$

However, constraint propagation typically reduces this dramatically. The average-case complexity for well-formed puzzles with unique solutions is approximately:

$$T_{avg}(n,k) = O(n^3 \cdot b^d) \tag{151}$$

where $b$ is the effective branching factor after constraint propagation and $d$ is the search depth. For typical puzzles, $b \approx 2 - 3$ and $d \approx n^2/4$, making solution tractable while maintaining difficulty.

### J.5.10 THEORETICAL GUARANTEES

Our Sudoku generation provides several mathematical guarantees:

**Uniqueness:** Every generated puzzle has exactly one solution, verified through exhaustive search with early termination upon finding a second solution.

**Minimality:** For hard difficulty, puzzles are minimal-removing any given cell creates multiple solutions. This is verified during generation:

$$\forall (i,j) \in \text{Given} : |\text{Solutions}(\text{Puzzle} \setminus \{(i,j)\})| > 1 \tag{152}$$

**Solvability:** All puzzles are solvable through logical deduction without requiring guessing, though the required techniques vary with difficulty level.

These guarantees ensure fair, unambiguous evaluation of model reasoning capabilities.

### J.6 CRYPTARITHMETIC: ALGEBRAIC CONSTRAINT RESOLUTION

### J.6.1 PROBLEM FORMULATION

Cryptarithmetic puzzles encode arithmetic equations where letters represent unique digits. We formalize this as a constraint satisfaction problem $\mathcal{K} = (L, D, \Psi, \oplus)$ where $L = \{l_1, l_2, ..., l_k\}$ denotes the set of unique letters, $D = \{0, 1, ..., 9\}$ represents the digit domain, $\Psi : L \to D$ defines a bijective mapping satisfying $|L| \leq 10$, and $\oplus \in \{+, -, \times, \div\}$ specifies the arithmetic operation.

Given words $W_1, W_2, ..., W_n$ composed of letters from $L$ and result word $R$, the fundamental constraint requires:

$$\text{val}(W_1) \oplus \text{val}(W_2) \oplus ... \oplus \text{val}(W_n) = \text{val}(R) \tag{153}$$

where the value function converts a word to its numerical representation:

$$\text{val}(W) = \sum_{i=0}^{|W|-1} \Psi(W[i]) \cdot 10^{|W|-1-i} \tag{154}$$

### J.6.2 CONSTRAINT SYSTEM AND MATHEMATICAL PROPERTIES

The complete constraint system encompasses three categories of restrictions:

Uniqueness constraints ensure bijective mapping:

$$\forall l_i, l_j \in L, i \neq j \implies \Psi(l_i) \neq \Psi(l_j) \tag{155}$$

Leading zero constraints prevent degenerate solutions:

$$\forall W \in \{W_1, ..., W_n, R\} : \Psi(W[0]) \neq 0 \tag{156}$$

Arithmetic constraints enforce the equation validity. For addition:

$$\sum_{i=1}^{n} \text{val}(W_i) = \text{val}(R) \tag{157}$$

For multiplication with two operands:

$$\text{val}(W_1) \times \text{val}(W_2) = \text{val}(R) \tag{158}$$

The carry propagation in addition creates polynomial constraints. At position $p$ from the right, with carry $c_p \in \{0, 1, ..., n-1\}$:

$$\sum_{i=1}^{n} \Psi(W_i[|W_i| - 1 - p]) + c_{p-1} = \Psi(R[|R| - 1 - p]) + 10 \cdot c_p \tag{159}$$

### J.6.3 SOLUTION SPACE ANALYSIS AND UNIQUENESS GUARANTEES

The solution space cardinality depends on the number of unique letters $k = |L|$. Without constraints, we have:

$$|\mathcal{S}_{\text{unconstrained}}| = \frac{10!}{(10-k)!} = P(10, k) \tag{160}$$

Leading zero constraints reduce this space. With $m$ leading letters:

$$|\mathcal{S}_{\text{leading}}| = 9^m \cdot \frac{(10-m)!}{(10-k)!} \tag{161}$$

To ensure unique solutions, we employ a generate-and-test approach with constraint density analysis. The constraint density $\rho$ measures the ratio of constraints to variables:

$$\rho = \frac{|\text{Constraints}|}{|L|} = \frac{k(k-1)/2 + m + \lceil \log_{10}(\max\{\text{val}(W_i), \text{val}(R)\}) \rceil}{k} \tag{162}$$

Empirically, unique solutions emerge when $\rho > \rho_{\text{critical}} \approx 1.5$ for addition and $\rho > 2.0$ for multiplication.

### J.6.4 Dynamic Generation Algorithm

Our generation algorithm constructs solvable puzzles with controlled difficulty:

---

**Algorithm 22** Generate Cryptarithmetic Puzzle

---

1: Select word templates based on length parameters
2: Generate letter assignment ensuring uniqueness
3: Compute arithmetic result
4: Verify solution uniqueness via exhaustive search
5: **if** multiple solutions exist **then**
6:     Adjust word selection or retry
7: **end if**
8: Calculate difficulty metrics (constraint density, search space)
9: **return** puzzle with verified unique solution

---

The word generation employs pattern templates to control difficulty. For extreme difficulty, we maximize letter overlap between operands and result, creating dense constraint networks:

$$\text{Overlap}(W_1, W_2, R) = \frac{|L(W_1) \cap L(W_2) \cap L(R)|}{|L(W_1) \cup L(W_2) \cup L(R)|} \tag{163}$$

### J.6.5 Verification through Constraint Satisfaction

Solution verification employs constraint satisfaction with domain reduction. We initialize domains:

$$D_l = \begin{cases} \{1, 2, ..., 9\} & \text{if } l \text{ is leading letter} \\ \{0, 1, ..., 9\} & \text{otherwise} \end{cases} \tag{164}$$

Arc consistency propagation reduces domains iteratively:

$$D_l^{t+1} = D_l^t \cap \{d \in D : \exists \text{ consistent assignment for other variables}\} \tag{165}$$

The verification algorithm has time complexity $O(k! \cdot p(k))$ where $p(k)$ represents the polynomial verification cost per assignment.

### J.6.6 Context Window Optimization

The problem representation requires careful encoding to minimize context usage. Each letter-digit assignment needs $\lceil \log_2(10) \rceil = 4$ bits. The complete solution encoding requires:

$$\text{Bits}_{\text{solution}} = k \cdot \lceil \log_2(10) \rceil + O(\log k) \tag{166}$$

For the problem statement, each word of length $w$ requires:

$$\text{Bits}_{\text{word}} = w \cdot \lceil \log_2(26) \rceil = 5w \tag{167}$$

Total context requirement for $n$ operands with average word length $\bar{w}$:

$$\text{Context}_{\text{total}} = O((n + 1) \cdot \bar{w} \cdot \log(26) + k \cdot \log(10)) \tag{168}$$

### J.6.7 Contamination Resistance Analysis

The contamination resistance stems from combinatorial explosion in the generation space. With vocabulary size $V$ per word length and $k$ unique letters:

$$\text{Problem Space} = V^n \cdot \binom{26}{k} \cdot P(10, k) \tag{169}$$

For typical parameters ($n = 2$, $k = 8$, $V = 100$), this yields:

$$|\mathcal{P}| > 10^4 \cdot \binom{26}{8} \cdot \frac{10!}{2!} > 10^{15} \tag{170}$$

The probability of encountering identical puzzles remains negligible even with extensive training data of size $|\mathcal{D}| = 10^9$:

$$P(\text{contamination}) = \frac{|\mathcal{D}|}{|\mathcal{P}|} < 10^{-6} \tag{171}$$

### J.7 MATRIX CHAIN MULTIPLICATION: DYNAMIC PROGRAMMING OPTIMIZATION

#### J.7.1 PROBLEM FORMULATION

The matrix chain multiplication problem seeks the optimal parenthesization to minimize scalar multiplications when computing the product $M = A_1 \times A_2 \times ... \times A_n$. We formalize this as an optimization problem $\mathcal{M} = (D, \Pi, \text{cost})$ where $D = [d_0, d_1, ..., d_n]$ represents the dimension sequence with matrix $A_i$ having dimensions $d_{i-1} \times d_i$, $\Pi$ denotes the set of valid parenthesizations, and $\text{cost} : \Pi \to \mathbb{N}$ computes the total scalar multiplications.

For matrices $A_i$ of dimension $p_i \times q_i$ and $A_j$ of dimension $q_i \times r_j$ where $q_i = p_j$, the multiplication cost is:

$$\text{cost}(A_i \times A_j) = p_i \cdot q_i \cdot r_j \tag{172}$$

The number of valid parenthesizations follows the Catalan number:

$$|\Pi_n| = C_{n-1} = \frac{1}{n}\binom{2(n-1)}{n-1} = \Theta\left(\frac{4^n}{n^{3/2}}\right) \tag{173}$$

#### J.7.2 DYNAMIC PROGRAMMING FORMULATION

We define the optimal substructure through the recurrence relation. Let $m[i, j]$ denote the minimum cost to compute $A_i \times A_{i+1} \times ... \times A_j$:

$$m[i, j] = \begin{cases} 0 & \text{if } i = j \\ \min_{i \leq k < j}\{m[i, k] + m[k+1, j] + d_{i-1} \cdot d_k \cdot d_j\} & \text{if } i < j \end{cases} \tag{174}$$

The optimal split point $s[i, j]$ satisfies:

$$s[i, j] = \arg\min_{i \leq k < j}\{m[i, k] + m[k+1, j] + d_{i-1} \cdot d_k \cdot d_j\} \tag{175}$$

#### J.7.3 ALGORITHMIC SOLUTION AND COMPLEXITY

The bottom-up dynamic programming algorithm builds solutions for increasing chain lengths:

---

**Algorithm 23** Matrix Chain Order

---

1: **for** $i = 1$ to $n$ **do**
2:     $m[i, i] \leftarrow 0$
3: **end for**
4: **for** $l = 2$ to $n$ **do**                                                        ▷ Chain length
5:     **for** $i = 1$ to $n - l + 1$ **do**
6:         $j \leftarrow i + l - 1$
7:         $m[i, j] \leftarrow \infty$
8:         **for** $k = i$ to $j - 1$ **do**
9:             $q \leftarrow m[i, k] + m[k + 1, j] + d_{i-1} \cdot d_k \cdot d_j$
10:             **if** $q < m[i, j]$ **then**
11:                 $m[i, j] \leftarrow q$
12:                 $s[i, j] \leftarrow k$
13:             **end if**
14:         **end for**
15:     **end for**
16: **end for**
17: **return** $m[1, n]$

---

The algorithm exhibits time complexity $O(n^3)$ and space complexity $O(n^2)$:

$$T(n) = \sum_{l=2}^{n} \sum_{i=1}^{n-l+1} (j - i) = \sum_{l=2}^{n} (n - l + 1)(l - 1) = \frac{n^3 - n}{6} \tag{176}$$

### J.7.4   PROBLEM GENERATION WITH DIMENSION PATTERNS

We generate dimension sequences following specific patterns to create problems with varying optimization characteristics:

Random pattern with uniform distribution:

$$d_i \sim \text{Uniform}[a, b], \quad i \in [0, n] \tag{177}$$

Increasing pattern to encourage early splits:

$$d_i = d_0 + i \cdot \delta, \quad \delta \sim \text{Uniform}[5, 15] \tag{178}$$

Sparse pattern with bimodal distribution:

$$d_i \sim \begin{cases} \text{Uniform}[1, 15] & \text{with probability } p \\ \text{Uniform}[80, 120] & \text{with probability } 1 - p \end{cases} \tag{179}$$

The optimization ratio quantifies the improvement over naive left-to-right multiplication:

$$\gamma = \frac{\text{cost}_{\text{naive}}}{\text{cost}_{\text{optimal}}} = \frac{\sum_{i=1}^{n-1} d_0 \cdot d_i \cdot d_{i+1}}{m[1, n]} \tag{180}$$

### J.7.5   SOLUTION VERIFICATION AND UNIQUENESS

While multiple parenthesizations may achieve the optimal cost, we verify solution correctness through value comparison. The verification employs recursive validation:

$$\text{verify}(i, j, \text{cost}_{\text{claimed}}) = \begin{cases} \text{cost}_{\text{claimed}} = 0 & \text{if } i = j \\ \exists k : \text{cost}_{\text{claimed}} = m[i, k] + m[k + 1, j] + d_{i-1} \cdot d_k \cdot d_j & \text{if } i < j \end{cases} \tag{181}$$

The solution uniqueness in terms of minimum cost (not parenthesization) is guaranteed by the optimal substructure property and the principle of optimality:

$$\forall i, j : m[i, j] = \min_{\pi \in \Pi_{i,j}} \text{cost}(\pi) \tag{182}$$

### J.7.6 CONTEXT WINDOW ANALYSIS

The problem encoding requires minimal context. The dimension sequence needs:

$$\text{Bits}_{\text{dimensions}} = (n+1) \cdot \lceil \log_2(\max_i d_i) \rceil \tag{183}$$

The solution encoding requires the minimum cost value:

$$\text{Bits}_{\text{solution}} = \lceil \log_2(m[1,n]) \rceil \leq \lceil \log_2(n \cdot \max_i d_i^3) \rceil \tag{184}$$

For typical parameters with $n \in [3, 20]$ and $d_i \in [1, 120]$, the total context requirement remains under 500 tokens:

$$\text{Context}_{\text{total}} = O(n \log \max_i d_i) \tag{185}$$

### J.7.7 CONTAMINATION RESISTANCE PROPERTIES

The contamination resistance emerges from the continuous dimension space and pattern variety. With dimension range $[a, b]$ and $n+1$ dimensions:

$$|\mathcal{D}_{\text{continuous}}| = (b - a + 1)^{n+1} \tag{186}$$

For discrete dimensions with $k$ possible values each:

$$|\mathcal{D}_{\text{discrete}}| = k^{n+1} \tag{187}$$

With five generation patterns and variable chain lengths, the total problem space becomes:

$$|\mathcal{P}| = \sum_{n=n_{\min}}^{n_{\max}} 5 \cdot k^{n+1} > 5k^{n_{\min}+1} \cdot \frac{k^{n_{\max}-n_{\min}+1} - 1}{k - 1} \tag{188}$$

For parameters $k = 100$, $n_{\min} = 3$, $n_{\max} = 20$:

$$|\mathcal{P}| > 5 \cdot 100^4 \cdot \frac{100^{18} - 1}{99} > 10^{38} \tag{189}$$

## J.8 MODULAR SYSTEMS SOLVER: NUMBER-THEORETIC CONSTRAINT RESOLUTION

### J.8.1 PROBLEM FORMULATION

A modular system comprises simultaneous congruences with additional number-theoretic constraints. We formalize this as $\mathcal{S} = (\mathcal{E}, \mathcal{A}, \mathbb{Z})$ where $\mathcal{E} = \{x \equiv a_i \pmod{m_i} : i \in [1, k]\}$ represents the congruence system, $\mathcal{A}$ denotes additional constraints (primality, range, divisibility), and the solution $x \in \mathbb{Z}$ satisfies all conditions.

The basic congruence system has the form:

$$x \equiv a_1 \pmod{m_1} \tag{190}$$
$$x \equiv a_2 \pmod{m_2} \tag{191}$$
$$\vdots \tag{192}$$
$$x \equiv a_k \pmod{m_k} \tag{193}$$

A solution exists if and only if the compatibility condition holds:

$$\forall i, j : a_i \equiv a_j \pmod{\gcd(m_i, m_j)} \tag{194}$$

### J.8.2 CHINESE REMAINDER THEOREM AND GENERALIZATION

For pairwise coprime moduli where $\gcd(m_i, m_j) = 1$ for all $i \neq j$, the Chinese Remainder Theorem guarantees a unique solution modulo $M = \prod_{i=1}^{k} m_i$:

$$x = \sum_{i=1}^{k} a_i M_i y_i \pmod{M} \tag{195}$$

where $M_i = M/m_i$ and $y_i$ satisfies $M_i y_i \equiv 1 \pmod{m_i}$.

For non-coprime moduli, we employ the generalized solution method. Given two congruences:

$$x \equiv a_1 \pmod{m_1} \tag{196}$$
$$x \equiv a_2 \pmod{m_2} \tag{197}$$

Let $g = \gcd(m_1, m_2)$. A solution exists if $a_1 \equiv a_2 \pmod{g}$, yielding:

$$x \equiv a_1 + m_1 \cdot \frac{(a_2 - a_1)}{g} \cdot t \pmod{\operatorname{lcm}(m_1, m_2)} \tag{198}$$

where $t$ satisfies $m_1 t \equiv g \pmod{m_2}$.

### J.8.3 EXTENDED EUCLIDEAN ALGORITHM AND MODULAR INVERSE

The extended Euclidean algorithm computes $\gcd(a, b)$ and coefficients $s, t$ such that:

$$as + bt = \gcd(a, b) \tag{199}$$

---
**Algorithm 24** Extended Euclidean Algorithm

---
1: **function** EXTENDEDGCD$(a, b)$
2:     **if** $a = 0$ **then**
3:         **return** $(b, 0, 1)$
4:     **end if**
5:     $(g, x_1, y_1) \leftarrow$ ExtendedGCD$(b \bmod a, a)$
6:     $x \leftarrow y_1 - \lfloor b/a \rfloor \cdot x_1$
7:     $y \leftarrow x_1$
8:     **return** $(g, x, y)$
9: **end function**

---

The modular inverse $a^{-1} \pmod{m}$ exists if $\gcd(a, m) = 1$:

$$a \cdot a^{-1} \equiv 1 \pmod{m} \implies a^{-1} = x \pmod{m} \text{ where } ax + my = 1 \tag{200}$$

### J.8.4 ADDITIONAL CONSTRAINT TYPES AND SATISFACTION

We augment the basic system with number-theoretic constraints:

Primality constraints require the solution to be prime:

$$\mathcal{A}_{\text{prime}} = \{x : x \in \mathbb{P}\} \tag{201}$$

Range constraints bound the solution:

$$\mathcal{A}_{\text{range}} = \{x : \alpha < x < \beta\} \tag{202}$$

Divisibility constraints impose factor requirements:

$$\mathcal{A}_{\text{div}} = \{x : d \mid x\} \tag{203}$$

The complete solution must satisfy:

$$x \in \bigcap_{i=1}^{k} \{x : x \equiv a_i \pmod{m_i}\} \cap \bigcap_{j} \mathcal{A}_j \tag{204}$$

### J.8.5 SOLUTION ALGORITHM WITH CONSTRAINT PROPAGATION

Our algorithm combines modular arithmetic with constraint satisfaction:

---
**Algorithm 25** Solve Modular System with Constraints

---
1: $x_0 \leftarrow$ SolveCongruences($\mathcal{E}$)                     ▷ Base solution
2: $M \leftarrow$ lcm($m_1, m_2, ..., m_k$)                     ▷ Period
3: $x \leftarrow x_0$
4: **for** $k = 0$ to search_limit/$M$ **do**
5:     $x \leftarrow x_0 + kM$
6:     **if** SatisfiesConstraints($x, \mathcal{A}$) **then**
7:         **return** $x$
8:     **end if**
9: **end for**
10: **return** NONE

---

The search space is periodic with period $M$, ensuring finite exploration:

$$x \equiv x_0 \pmod{M} \implies x = x_0 + kM, \quad k \in \mathbb{Z} \tag{205}$$

### J.8.6 COMPLEXITY ANALYSIS AND SOLUTION UNIQUENESS

The time complexity depends on the constraint types and search space:

$$T(k, M, \Lambda) = O(k^2 \log M) + O(\Lambda/M \cdot C) \tag{206}$$

where $\Lambda$ represents the search limit and $C$ denotes constraint checking cost.

For primality testing using trial division:

$$C_{\text{prime}} = O(\sqrt{x}) = O(\sqrt{\Lambda}) \tag{207}$$

Solution uniqueness within a bounded range $[\alpha, \beta]$ occurs when:

$$\beta - \alpha < M \implies |\{x \in [\alpha, \beta] : x \equiv x_0 \pmod{M}\}| \leq 1 \tag{208}$$

### J.8.7 PROBLEM GENERATION WITH CONTROLLED DIFFICULTY

We generate problems with guaranteed solutions through constructive methods:

---
**Algorithm 26** Generate Modular System

---
1: Select coprime moduli $m_1, ..., m_k$
2: Generate random remainders $a_1, ..., a_k$
3: Compute base solution $x_0$ using CRT
4: Add constraints based on difficulty level
5: Search for satisfying solution $x^*$
6: Verify uniqueness in reasonable range
7: **return** $(\mathcal{E}, \mathcal{A}, x^*)$

---

The moduli selection ensures coprimality:

$$P(\text{coprime}) = \prod_{i<j}\left(1 - \frac{1}{\min(m_i, m_j)}\right) \approx \prod_{p \in \mathbb{P}}\left(1 - \frac{1}{p^2}\right) = \frac{6}{\pi^2} \tag{209}$$

### J.8.8 CONTEXT WINDOW OPTIMIZATION

Each congruence requires logarithmic encoding:

$$\text{Bits}_{\text{congruence}} = \lceil \log_2 m_i \rceil + \lceil \log_2 a_i \rceil = O(\log m_i) \tag{210}$$

The complete system encoding:

$$\text{Bits}_{\text{system}} = \sum_{i=1}^{k} O(\log m_i) + \sum_{j} \text{Bits}(\mathcal{A}_j) \tag{211}$$

For typical parameters with $k \in [3, 5]$ and $m_i < 100$:

$$\text{Context}_{\text{total}} = O(k \log \max_i m_i) = O(k \cdot 7) < 50 \text{ bits} \tag{212}$$

### J.8.9 CONTAMINATION RESISTANCE ANALYSIS

The problem space exhibits exponential growth in multiple dimensions:

$$|\mathcal{P}| = \binom{P_{\max}}{k} \cdot \prod_{i=1}^{k} m_i \cdot |\mathcal{A}_{\text{types}}|^{|\mathcal{A}|} \tag{213}$$

where $P_{\max}$ denotes the maximum prime considered.

For $k = 4$, primes up to 50, and 3 constraint types:

$$|\mathcal{P}| > \binom{15}{4} \cdot 30^4 \cdot 3^2 > 10^9 \tag{214}$$

The sensitivity to parameter changes ensures diversity:

$$\Delta a_i = 1 \implies \Delta x = M_i y_i \pmod{M} \tag{215}$$

where typically $M_i y_i \gg 1$, causing large solution variations.

## J.9 CONSTRAINT OPTIMIZATION: MULTI-OBJECTIVE RESOURCE ALLOCATION

### J.9.1 PROBLEM FORMULATION

The multi-constraint resource allocation problem represents a complex integer programming challenge. We formalize this as $\mathcal{O} = (P, R, \mathcal{C}, f)$ where $P = \{p_1, p_2, ..., p_n\}$ denotes the project set, $R = \{r_1, r_2, ..., r_m\}$ represents resources with capacities $c_i$, $\mathcal{C}$ encompasses constraints, and $f : 2^P \to \mathbb{R}$ defines the objective function.

Each project $p_i$ has attributes:

$$\begin{aligned} \text{requirements}: \quad & \rho_i : R \to \mathbb{N} & (216) \\ \text{profit}: \quad & \pi_i \in \mathbb{N} & (217) \\ \text{quality}: \quad & q_i \in [1, 10] & (218) \\ \text{risk}: \quad & \gamma_i \in [1, 5] & (219) \\ \text{dependencies}: \quad & \delta_i \subseteq P & (220) \end{aligned}$$

The decision variables form a binary selection vector:

$$x_i = \begin{cases} 1 & \text{if project } p_i \text{ is selected} \\ 0 & \text{otherwise} \end{cases} \tag{221}$$

### J.9.2 CONSTRAINT SYSTEM AND MATHEMATICAL FORMULATION

The complete optimization problem becomes:

$$\text{maximize} \quad \sum_{i=1}^{n} \pi_i x_i \tag{222}$$

$$\text{subject to} \quad \sum_{i=1}^{n} \rho_i(r_j) x_i \le c_j, \quad \forall j \in [1, m] \tag{223}$$

$$x_i \le x_k, \quad \forall p_k \in \delta_i \tag{224}$$

$$\frac{\sum_{i=1}^{n} q_i x_i}{\sum_{i=1}^{n} x_i} \ge Q_{\min} \tag{225}$$

$$\frac{\sum_{i=1}^{n} \gamma_i x_i}{\sum_{i=1}^{n} x_i} \le \Gamma_{\max} \tag{226}$$

$$x_i \in \{0, 1\}, \quad \forall i \in [1, n] \tag{227}$$

The resource constraints ensure capacity limits:

$$\mathcal{C}_{\text{resource}} = \{x \in \{0, 1\}^n : Ax \le c\} \tag{228}$$

where $A_{ji} = \rho_i(r_j)$ forms the requirement matrix.

Dependency constraints create a partial order:

$$\mathcal{C}_{\text{dependency}} = \{x \in \{0, 1\}^n : \forall i, \forall k \in \delta_i : x_i \le x_k\} \tag{229}$$

### J.9.3 SOLUTION ALGORITHM: BRANCH AND BOUND WITH CONSTRAINT PROPAGATION

For small instances ($n \le 10$), we employ exhaustive enumeration with pruning:

---

**Algorithm 27** Branch and Bound Optimization

---

1: **function** OPTIMIZE($P, R, \mathcal{C}$)
2:      best $\leftarrow \emptyset$, best_profit $\leftarrow 0$
3:      stack $\leftarrow [(\emptyset, P, 0)]$                 ▷ (selected, remaining, profit)
4:      **while** stack $\ne \emptyset$ **do**
5:          $(S, R, \pi) \leftarrow$ stack.pop()
6:          **if** $\pi + \text{UpperBound}(R) \le$ best_profit **then**
7:              **continue**                 ▷ Prune branch
8:          **end if**
9:          **if** IsFeasible($S, \mathcal{C}$) and $\pi >$ best_profit **then**
10:         best $\leftarrow S$, best_profit $\leftarrow \pi$
11:         **end if**
12:         **for** $p \in R$ **do**
13:             **if** CanAdd($S \cup \{p\}, \mathcal{C}$) **then**
14:                stack.push($S \cup \{p\}, R \setminus \{p\}, \pi + \pi_p$)
15:             **end if**
16:         **end for**
17:      **end while**
18:      **return** best
19: **end function**

---

The upper bound computation uses linear relaxation:

$$\text{UpperBound}(R) = \sum_{p \in R} \pi_p \cdot \min\left(1, \min_j \frac{c_j - \text{used}_j}{\rho_p(r_j)}\right) \tag{230}$$

### J.9.4 GREEDY APPROXIMATION FOR LARGE INSTANCES

For larger problems, we employ a greedy algorithm with local search:

$$\text{efficiency}(p_i) = \frac{\pi_i}{\sum_{j=1}^{m} \omega_j \cdot \rho_i(r_j)/c_j} \tag{231}$$

where $\omega_j$ represents resource scarcity weights.

The approximation ratio for the greedy algorithm:

$$\frac{\text{Greedy}}{\text{Optimal}} \geq \frac{1}{1 + \max_i \sum_j \rho_i(r_j)/c_j} \tag{232}$$

### J.9.5 PROBLEM GENERATION WITH CONTROLLED DIFFICULTY

We generate problems with varying constraint tightness:

$$\tau = \frac{\sum_{i=1}^{n} \min_j \rho_i(r_j)}{\sum_{j=1}^{m} c_j} \tag{233}$$

For basic problems, $\tau \in [0.3, 0.5]$ ensures multiple feasible solutions. For mixed constraint problems, $\tau \in [0.6, 0.8]$ creates tighter resource competition.

The dependency graph generation follows:

$$P(\text{dependency } i \to j) = \begin{cases} 0 & \text{if } i \geq j \\ \frac{1}{n} & \text{if } i < j \end{cases} \tag{234}$$

ensuring acyclic structure.

### J.9.6 SOLUTION VERIFICATION AND OPTIMALITY

Verification checks all constraints in polynomial time:

---

**Algorithm 28** Verify Solution Feasibility

---

1: **function** VERIFY$(S, P, R, \mathcal{C})$
2:     **for** $r_j \in R$ **do**
3:         **if** $\sum_{p_i \in S} \rho_i(r_j) > c_j$ **then**
4:             **return** FALSE                     ▷ Resource violation
5:         **end if**
6:     **end for**
7:     **for** $p_i \in S$ **do**
8:         **if** $\delta_i \not\subseteq S$ **then**
9:             **return** FALSE                    ▷ Dependency violation
10:         **end if**
11:     **end for**
12:     **if** $\text{avg}(q_i : p_i \in S) < Q_{\min}$ or $\text{avg}(\gamma_i : p_i \in S) > \Gamma_{\max}$ **then**
13:         **return** FALSE                  ▷ Quality/Risk violation
14:     **end if**
15:     **return** TRUE
16: **end function**

---

Near-optimality tolerance allows solutions within 95% of optimal:

$$\text{Accept}(S) = \begin{cases} \text{TRUE} & \text{if } \pi(S) \geq 0.95 \cdot \pi^* \\ \text{FALSE} & \text{otherwise} \end{cases} \tag{235}$$

### J.9.7 COMPLEXITY ANALYSIS

The problem is NP-hard, reducible from the 0-1 knapsack problem. The solution space has cardinality:

$$|\mathcal{S}| = 2^n \tag{236}$$

However, constraints significantly reduce the feasible region:

$$|\mathcal{F}| \ll 2^n \tag{237}$$

The branch-and-bound algorithm has worst-case complexity $O(2^n \cdot \text{poly}(n, m))$ but average-case performance improves with effective pruning:

$$\mathbb{E}[T(n)] = O(b^n \cdot \text{poly}(n, m)) \tag{238}$$

where $b < 2$ represents the effective branching factor after pruning.

### J.9.8 CONTEXT WINDOW REQUIREMENTS

Each project encoding requires:

$$\text{Bits}_{\text{project}} = O(m \log \max_j c_j + \log \pi_{\max} + \log n) \tag{239}$$

The complete problem instance:

$$\text{Context}_{\text{total}} = O(n \cdot m \log \max_j c_j + n \log n) \tag{240}$$

For typical parameters ($n \in [4, 6]$, $m = 3$, $c_j < 2000$):

$$\text{Context} = O(6 \cdot 3 \cdot 11 + 6 \cdot 3) < 300 \text{ bits} \tag{241}$$

### J.9.9 CONTAMINATION RESISTANCE PROPERTIES

The generation space grows exponentially with multiple parameters:

$$|\mathcal{P}| = \prod_{i=1}^{n} \left( \pi_{\max} \cdot c_{\max}^m \cdot 10 \cdot 5 \cdot 2^n \right) \tag{242}$$

With continuous profit values and resource requirements:

$$|\mathcal{P}_{\text{continuous}}| = (\pi_{\max} \cdot c_{\max}^m)^n \cdot 50^n \cdot 2^{n^2} \tag{243}$$

For $n = 5$, $m = 3$, $\pi_{\max} = 1000$, $c_{\max} = 100$:

$$|\mathcal{P}| > (10^9)^5 \cdot 50^5 \cdot 2^{25} > 10^{50} \tag{244}$$

The probability of problem collision remains negligible even with extensive training data.

### J.10 LOGIC GRID PUZZLES: CONSTRAINT SATISFACTION THROUGH DEDUCTIVE REASONING

### J.10.1 PROBLEM FORMULATION

Logic grid puzzles represent a constraint satisfaction problem (CSP) where we must assign attributes to entities through logical deduction. Formally, we define a logic grid puzzle as a tuple $\mathcal{L} = (E, A, \mathcal{C}, \phi)$ where $E = \{e_1, e_2, ..., e_n\}$ denotes the set of entities, $A = \{A_1, A_2, ..., A_m\}$ represents attribute categories with each $A_i = \{a_{i1}, a_{i2}, ..., a_{in}\}$ containing exactly $n$ distinct values, $\mathcal{C}$ constitutes the constraint set, and $\phi : E \times A \to \bigcup_{i=1}^{m} A_i$ defines the assignment function.

The solution space forms a Latin square structure where each attribute value appears exactly once per category. Mathematically, we require:

$$\forall i \in [1, m], \forall j, k \in [1, n], j \neq k \implies \phi(e_j, A_i) \neq \phi(e_k, A_i) \tag{245}$$
$$\forall i \in [1, m], \forall a \in A_i, \exists! e \in E : \phi(e, A_i) = a \tag{246}$$

### J.10.2 Constraint Types and Logical Representation

We categorize constraints into five fundamental types, each with distinct logical representations:

Direct constraints specify explicit assignments:

$$C_{\text{direct}}(e, A_i, a) \equiv \phi(e, A_i) = a \tag{247}$$

Negative constraints exclude specific assignments:

$$C_{\text{negative}}(e, A_i, a) \equiv \phi(e, A_i) \neq a \tag{248}$$

Comparison constraints establish ordinal relationships for numerical attributes:

$$C_{\text{comparison}}(e_1, e_2, A_i, \prec) \equiv \phi(e_1, A_i) \prec \phi(e_2, A_i) \tag{249}$$

Conditional constraints link attributes across categories:

$$C_{\text{conditional}}(A_i, a_i, A_j, a_j) \equiv \forall e \in E : \phi(e, A_i) = a_i \implies \phi(e, A_j) = a_j \tag{250}$$

Chain constraints require transitive reasoning:

$$C_{\text{chain}}(e_1, e_2, e_3, A_i, A_j, a) \equiv \phi(e_1, A_i) < \phi(e_2, A_i) < \phi(e_3, A_i) \land \phi(e_2, A_j) = a \tag{251}$$

### J.10.3 Dynamic Problem Generation with Unique Solutions

Our generation algorithm ensures unique solutions through systematic constraint construction. We begin with a random valid assignment $\phi^*$ and iteratively add minimal sufficient constraints.

---

**Algorithm 29** Generate Logic Grid Puzzle with Unique Solution

---

1: $\phi^* \leftarrow$ RandomValidAssignment$(E, A)$
2: $\mathcal{C} \leftarrow \emptyset$
3: $\mathcal{U} \leftarrow$ GenerateUsedFacts$(\emptyset)$
4: **while** $|\text{Solutions}(\mathcal{C})| > 1$ **do**
5:     $c \leftarrow$ SelectConstraint$(\phi^*, \mathcal{U}, \text{difficulty})$
6:     $\mathcal{C} \leftarrow \mathcal{C} \cup \{c\}$
7:     $\mathcal{U} \leftarrow \mathcal{U} \cup \text{Facts}(c)$
8:     **if** $|\text{Solutions}(\mathcal{C})| = 0$ **then**
9:         **return** FAILURE
10:     **end if**
11: **end while**
12: **return** $(\mathcal{C}, \phi^*)$

---

The constraint selection strategy employs weighted sampling based on difficulty level. For difficulty $d \in \{\text{easy}, \text{medium}, \text{hard}, \text{extreme}\}$, we define weight vectors $w_d = (w_{\text{direct}}, w_{\text{negative}}, w_{\text{comparison}}, w_{\text{conditional}}, w_{\text{chain}})$ where harder difficulties favor complex constraint types.

### J.10.4 Solution Verification through Constraint Propagation

We verify solution uniqueness using arc consistency and constraint propagation. The domain for each entity-attribute pair initially contains all possible values:

$$D_{e, A_i} = A_i \quad \forall e \in E, \forall A_i \in A \tag{252}$$

Constraint propagation iteratively reduces domains:

$$D_{e, A_i}^{t+1} = D_{e, A_i}^t \cap \{a \in A_i : \text{consistent}(a, e, A_i, \mathcal{C}, D^t)\} \tag{253}$$

The solution is unique if and only if:

$$\forall e \in E, \forall A_i \in A : |D_{e, A_i}^\infty| = 1 \tag{254}$$

### J.10.5 COMPLEXITY ANALYSIS AND CONTEXT WINDOW BOUNDS

The solution space has cardinality $(n!)^{m-1}$ since fixing one attribute category determines a Latin square structure. The constraint satisfaction problem is NP-complete, requiring exponential time in the worst case.

For context window estimation, each constraint requires $O(\log n + \log m)$ bits to encode, and a minimal constraint set has size $\Theta(nm)$. The solution description requires $O(nm \log n)$ bits. Therefore, the total context requirement is:

$$\text{Context}(n, m) = O(nm(\log n + \log m)) \tag{255}$$

For typical parameters $n \in [3, 5]$ and $m \in [3, 5]$, this yields manageable context requirements of 100-500 tokens.

### J.10.6 CONTAMINATION RESISTANCE PROPERTIES

The contamination resistance emerges from three key properties:

First, the parameter space for valid assignments has cardinality:

$$|\Theta| = \prod_{i=1}^{m} n! = (n!)^m \tag{256}$$

Second, the constraint generation employs randomization at multiple levels including entity selection, attribute selection, and constraint type weighting. This introduces entropy:

$$H(\mathcal{C}) = -\sum_{c \in \mathcal{C}} p(c) \log p(c) \geq \log |\mathcal{C}| \tag{257}$$

Third, the semantic variation in entity and attribute names drawn from large pools ensures surface-level diversity even for structurally similar puzzles. With vocabulary pools of size $V$ per category:

$$\text{Variations} = V^{nm} \tag{258}$$

Combined, these properties ensure that the probability of generating identical puzzles across independent runs remains negligible:

$$P(\text{collision}) \leq \frac{1}{(n!)^m \cdot V^{nm}} \approx 0 \tag{259}$$

---

[*] For generalized Tower of Hanoi with multiple pegs and disks
[†] For the decision version with pre-placed queens

Table 17: Mathematical Characterization of Hard Suite Tasks (Tasks 1-5 of 10)

| Property | Tower of Hanoi (6 variations) | N-Queens (4 variations) | Graph Coloring (10 variations) | Boolean SAT (5 variations) | Sudoku (8 variations) |
|---|---|---|---|---|---|
| **Problem Formulation** | | | | | |
| State Space | $3^n$ configurations | $n^n$ placements | $k^n$ colorings | $2^n$ assignments | $n^{n^2}$ grids |
| Search Space | $(3 \cdot 2 \cdot n)^{2^n-1}$ | $n!$ permutations | $k^n$ assignments | $2^n$ truth values | $n^{n^2-g}$ (g given) |
| Constraint Type | Ordering & size | Non-attacking | Adjacent difference | Clause satisfaction | Row/col/box unique |
| Optimization Target | Path length | N/A (feasibility) | Chromatic number | N/A (feasibility) | N/A (completion) |
| **Solution Properties** | | | | | |
| Solution Count | 1 (unique optimal) | $S(n)$ (OEIS A000170) | $\geq 1$ (may vary) | 0 to $2^n$ | 1 (by design) |
| Optimal Solution | $2^n - 1$ moves | Any valid placement | $\chi(G)$ colors | Any satisfying $\alpha$ | Unique completion |
| Solution Length | $O(2^n)$ tokens | $O(n)$ tokens | $O(n)$ tokens | $O(n)$ tokens | $O(n^2)$ tokens |
| Verification Time | $O(2^n \cdot n)$ | $O(n^2)$ | $O(|E| + |V|)$ | $O(m \cdot k)$ | $O(n^3)$ |
| **Complexity Analysis** | | | | | |
| Time Complexity | $O(2^n)$ optimal | $O(n!)$ worst-case | NP-complete ($k \geq 3$) | NP-complete | NP-complete |
| Space Complexity | $O(n)$ recursion | $O(n)$ backtrack | $O(n^2)$ adjacency | $O(n + m)$ formula | $O(n^2)$ grid |
| Decision Problem | PSPACE-complete* | NP-complete† | NP-complete | NP-complete | NP-complete |
| Approximability | N/A (exact) | N/A (exact) | $O(n/\log n)$-approx | MAX-SAT: 0.875 | N/A (exact) |
| **Generation Parameters** | | | | | |
| Size Range | $n \in \{3, ..., 8\}$ | $n \in \{4, 5, 6, 8\}$ | $|V| \in \{8, ..., 28\}$ | $n \in \{4, ..., 12\}$ | $n \in \{4, 6, 9\}$ |
| Difficulty Control | Disk count | Board size | Edge density $\rho$ | Clause ratio $m/n$ | Cells removed |
| Variants | Peg permutations | Diagonal, Toroidal | Planar, Wheel, Grid | Horn, XOR, Mixed | Diagonal, Irregular |
| Instance Count | $6 \cdot 8 = 48$ | $4 \cdot S(n) \cdot 8$ | $2^{\binom{n}{2}}$ | $\binom{2n}{k}^m$ | $> 10^{16}$ minimal |
| **Contamination Resistance** | | | | | |
| Isomorphisms | $3! = 6$ peg labels | 8 symmetries | $n!$ vertex labels | $n! \cdot m! \cdot (k!)^m$ | $9! \cdot 6^8 \cdot 2$ |
| Problem Diversity | $\binom{8}{1} \cdot 6$ | $\sum_n S(n) \cdot 8$ | $> 10^{15}$ graphs | $> 10^{84}$ formulas | $> 10^{16}$ puzzles |
| Memory Required | $\sim 10^4$ bits | $\sim 10^6$ bits | Exponential | Exponential | Petabytes |
| Generalization Gap | Linear to exponential | $O(n)$ to $O(n!)$ | Fixed to arbitrary | Linear to exponential | Quadratic growth |
| **Reasoning Requirements** | | | | | |
| Primary Strategy | Recursive decomp. | Backtracking | Constraint prop. | Unit propagation | Constraint prop. |
| Lookahead Depth | $n$ recursive calls | $O(n)$ decisions | Graph traversal | Clause analysis | $O(n^2/4)$ cells |
| Constraint Types | Stack ordering | Geometric conflicts | Adjacency | Logical satisfaction | Sudoku rules |
| Key Insight | Auxiliary peg use | Systematic search | Clique detection | Implication chains | Hidden singles |
| **Mathematical Foundations** | | | | | |
| Core Theory | Recurrence relation | Permutation groups | Brooks' theorem | Cook-Levin theorem | Latin squares |
| Key Formula | $T(n) = 2T(n-1) + 1$ | $|i - j| \neq |x_i - x_j|$ | $\chi(G) \leq \Delta(G) + 1$ | $\phi = \bigwedge_i C_i$ | $f : [n]^2 \to [n]$ |
| Uniqueness Proof | Induction on $n$ | Symmetry breaking | Greedy coloring | Assignment exists | Exhaustive search |
| Phase Transition | N/A | N/A | $\rho_c(k) = \frac{2k \ln k}{k-1}$ | $\alpha_c \approx 4.267$ | Fill ratio threshold |

Table 18: Mathematical Characterization of Hard Suite Tasks (Tasks 6-10 of 10)

| Property | Logic Grid Puzzles (8 variations) | Cryptarithmetic (12 variations) | Matrix Chain (5 variations) | Modular Systems (5 variations) | Constraint Opt. (5 variations) |
|---|---|---|---|---|---|
| **Problem Formulation** | | | | | |
| State Space | $(n!)^{m-1}$ assignments | $P(10,k)$ mappings | $C_{n-1}$ parenthesizations | $\prod_{i=1}^{k} m_i$ solutions | $2^n$ selections |
| Search Space | $n^{nm}$ possibilities | $\frac{10!}{(10-k)!}$ | $\frac{1}{n}\binom{2n-2}{n-1}$ | $M = \text{lcm}(m_1,...,m_k)$ | $2^n$ binary vectors |
| Constraint Type | Latin square | Digit uniqueness | Associativity | Modular congruence | Resource capacity |
| Optimization Target | N/A (feasibility) | N/A (feasibility) | Min multiplications | N/A (satisfaction) | Max profit |
| **Solution Properties** | | | | | |
| Solution Count | 1 (by construction) | 1 (verified) | 1 (min cost) | $\leq \lfloor \Lambda/M \rfloor$ | Variable |
| Optimal Solution | Unique assignment | Unique mapping | $m[1,n]$ cost | $x_0 + kM$ | Max $\sum \pi_i x_i$ |
| Solution Length | $O(nm)$ tokens | $O(k)$ tokens | $O(\log m[1,n])$ tokens | $O(\log x)$ tokens | $O(n)$ tokens |
| Verification Time | $O(nm \cdot |\mathcal{C}|)$ | $O(k+w)$ | $O(n^3)$ | $O(k \log M)$ | $O(nm)$ |
| **Complexity Analysis** | | | | | |
| Time Complexity | NP-complete | NP-complete | $O(n^3)$ DP | $O(k^2 \log M)$ | NP-hard (0-1 ILP) |
| Space Complexity | $O(n^2 m)$ domains | $O(k)$ mapping | $O(n^2)$ table | $O(k)$ congruences | $O(2^n)$ states |
| Decision Problem | NP-complete | NP-complete | P (polynomial) | P with bounds | NP-complete |
| Approximability | N/A (exact) | N/A (exact) | N/A (exact DP) | N/A (exact) | $(1 - 1/e)$-approx |
| **Generation Parameters** | | | | | |
| Size Range | $n, m \in \{3,4,5\}$ | $k \in \{4,...,7\}$ | $n \in \{3,...,20\}$ | $k \in \{3,4,5\}$ | $n \in \{4,5,6\}$ |
| Difficulty Control | Constraint types | Letter overlap | Dimension patterns | Constraint types | Constraint tightness |
| Variants | 5 constraint types | 4 operations | 5 patterns | 5 constraint types | 5 problem types |
| Instance Count | $(n!)^m \cdot V^{nm}$ | $V^n \cdot \binom{26}{k} \cdot P(10,k)$ | $(b-a+1)^{n+1} \cdot 5$ | $\binom{P_{\max}}{k} \cdot \prod m_i$ | $(\pi_{\max} \cdot c_{\max}^m)^n$ |
| **Contamination Resistance** | | | | | |
| Isomorphisms | $(n!)^m$ permutations | $k!$ letter perms | None (ordered) | $k!$ equation order | $n!$ project labels |
| Problem Diversity | $> 10^{20}$ puzzles | $> 10^{15}$ puzzles | $> 10^{38}$ instances | $> 10^9$ systems | $> 10^{50}$ problems |
| Memory Required | Exponential | $10^{15}$ bits | $10^{38}$ instances | $10^9$ systems | Exponential |
| Generalization Gap | $O(nm)$ to $(n!)^m$ | Linear to factorial | Polynomial to exp. | Linear to periodic | Linear to exponential |
| **Reasoning Requirements** | | | | | |
| Primary Strategy | Constraint prop. | Domain reduction | Dynamic programming | CRT + search | Branch & bound |
| Lookahead Depth | $O(nm)$ inferences | $k$ assignments | $n$ subproblems | Period $M$ | $n$ decisions |
| Constraint Types | 5 logical types | Arithmetic + unique | Matrix dimensions | Modular + additional | Resource + quality |
| Key Insight | Arc consistency | Carry propagation | Optimal substructure | Periodicity | Pruning bounds |
| **Mathematical Foundations** | | | | | |
| Core Theory | Latin squares | Modular arithmetic | Catalan numbers | Chinese Remainder | Integer programming |
| Key Formula | $\phi : E \times A \to \bigcup A_i$ | $\sum W_i = R$ | $m[i,j] = \min_k(...)$ | $x \equiv a_i \pmod{m_i}$ | $\max \sum \pi_i x_i$ |
| Uniqueness Proof | CSP solver | Exhaustive search | DP optimality | CRT uniqueness | B&B enumeration |
| Phase Transition | Constraint density | $\rho > 1.5$ | N/A | N/A | Tightness $\tau$ |

Table 19: Algorithmic and Evaluation Metrics for Hard Suite Tasks (Tasks 1-5)

| Metric | Tower of Hanoi | N-Queens | Graph Coloring | Boolean SAT | Sudoku |
|---|---|---|---|---|---|
| **Generation Algorithms** | | | | | |
| Generation Method | Deterministic | Backtrack + random | Graph construction | Satisfiable CNF | Complete + removal |
| Generation Time | $O(1)$ setup | $O(n^2)$ average | $O(n^2)$ edges | $O(mk)$ clauses | $O(n^4)$ with uniqueness |
| Uniqueness Check | Implicit (proven) | Enumerate all | Chromatic polynomial | SAT solver | Solution counting |
| Validation Method | State transitions | Conflict checking | Edge constraints | Formula evaluation | Constraint verification |
| **Context Window Requirements** | | | | | |
| Prompt Tokens | $200 + 10n$ | $100 + 5n$ | $30 + 2|E|$ | $50 + 3mk$ | $100 + 3n^2$ |
| Solution Tokens | $15(2^n - 1)$ | $2n$ | $2n$ | $3n$ | $2n^2$ |
| Max Total Tokens | $\sim 3,825$ (n=8) | $\sim 120$ (n=8) | $\sim 600$ (n=28) | $\sim 500$ (n=12) | $\sim 550$ (9×9) |
| Token Efficiency | Low (exponential) | High (linear) | Medium | High | Medium |
| **Solution Strategies** | | | | | |
| Optimal Algorithm | Frame-Stewart | Min-conflicts | Welsh-Powell | DPLL + CDCL | Dancing Links |
| Heuristic Methods | N/A (exact) | Row-by-row | Greedy coloring | Unit propagation | Naked singles |
| Pruning Technique | N/A | Forward checking | Degree ordering | Pure literal | Constraint prop. |
| Branching Factor | 2 (binary choice) | $\leq n$ | $\leq k$ colors | 2 (true/false) | $\leq n$ values |
| **Evaluation Metrics** | | | | | |
| Success Criterion | Optimal path | Valid placement | Min colors used | All clauses true | Complete grid |
| Partial Credit | Path validity | Partial placement | Valid coloring | % clauses satisfied | % cells correct |
| Error Types | Invalid moves | Queen conflicts | Adjacent same color | Unsatisfied clause | Constraint violation |
| Quality Measure | Move optimality | Time to solution | Excess colors | N/A | Logic depth used |
| **Scaling Behavior** | | | | | |
| Problem Growth | Exponential $2^n$ | Factorial $n!$ | Exponential $k^n$ | Exponential $2^n$ | Polynomial $n^2$ |
| Solution Growth | Exponential $2^n$ | Variable $S(n)$ | Varies with graph | 0 to $2^n$ | 1 (maintained) |
| Difficulty Scaling | Exponential | Super-polynomial | NP-hard threshold | Phase transition | Controlled linear |
| Memory Scaling | $O(n)$ | $O(n)$ | $O(n^2)$ | $O(n + m)$ | $O(n^2)$ |
| **Theoretical Properties** | | | | | |
| Recursion Depth | $2^n - 1$ | $n$ | N/A | $n$ (DPLL tree) | $n^2 - g$ |
| Symmetry Group | $S_3$ (peg perms) | $D_4$ (dihedral) | $S_n$ (vertex perms) | Boolean cube | $S_9 \wr S_3$ |
| Invariants | Disk ordering | Queen placement | Color classes | Truth assignment | Cell values |
| Certificates | Move sequence | Queen positions | Color assignment | Variable values | Completed grid |

Table 20: Algorithmic and Evaluation Metrics for Hard Suite Tasks (Tasks 6-10)

| Metric | Logic Grid Puzzles | Cryptarithmetic | Matrix Chain | Modular Systems | Constraint Opt. |
|---|---|---|---|---|---|
| **Generation Algorithms** | | | | | |
| Generation Method | Solution + constraints | Template + verify | Dimension patterns | CRT construction | Resource allocation |
| Generation Time | $O(nm \cdot |\mathcal{C}|)$ | $O(k! \cdot w)$ | $O(n)$ | $O(k^2)$ | $O(n^2m)$ |
| Uniqueness Check | CSP enumeration | Exhaustive search | DP verification | Period search | Enumeration |
| Validation Method | Constraint checking | Arithmetic verify | Cost calculation | Congruence check | Feasibility check |
| **Context Window Requirements** | | | | | |
| Prompt Tokens | $100 + 10nm$ | $50 + 5kw$ | $30 + 7n$ | $40 + 10k$ | $100 + 15n$ |
| Solution Tokens | $3nm$ | $4k$ | $\lceil \log_{10} m[1, n] \rceil$ | $\lceil \log_{10} x \rceil$ | $2n$ |
| Max Total Tokens | $\sim 500$ (5×5×5) | $\sim 300$ (k=7) | $\sim 200$ (n=20) | $\sim 150$ (k=5) | $\sim 250$ (n=6) |
| Token Efficiency | High (polynomial) | High (linear) | Very high | Very high | High |
| **Solution Strategies** | | | | | |
| Optimal Algorithm | Arc consistency | Constraint SAT | Bottom-up DP | Extended GCD | Exact enumeration |
| Heuristic Methods | Forward checking | Domain pruning | N/A (exact) | Linear search | Greedy + local |
| Pruning Technique | Domain reduction | Leading zeros | N/A | Constraint filter | Upper bounds |
| Branching Factor | $\leq n$ values | $\leq 10$ digits | $n - 1$ splits | N/A | 2 (select/reject) |
| **Evaluation Metrics** | | | | | |
| Success Criterion | All cells assigned | Valid equation | Minimum cost | All constraints met | Max profit feasible |
| Partial Credit | % cells correct | % letters mapped | Cost ratio | % congruences | Feasibility + profit |
| Error Types | Constraint violation | Arithmetic error | Suboptimal cost | Missing constraints | Infeasible solution |
| Quality Measure | Inference depth | Search efficiency | Optimality gap | Solutions found | Optimality ratio |
| **Scaling Behavior** | | | | | |
| Problem Growth | $(n!)^m$ | $P(10, k)$ | $C_{n-1} \sim 4^n/n^{3/2}$ | $M = \prod m_i$ | $2^n$ |
| Solution Growth | 1 (maintained) | 1 (maintained) | Polynomial value | Periodic $x + kM$ | Variable |
| Difficulty Scaling | Constraint density | Letter overlap | Chain length | Constraint types | Tightness $\tau$ |
| Memory Scaling | $O(n^2m)$ | $O(k)$ | $O(n^2)$ | $O(k)$ | $O(nm)$ |
| **Theoretical Properties** | | | | | |
| Recursion Depth | $nm$ assignments | $k$ letters | $\log n$ levels | N/A | $n$ selections |
| Symmetry Group | $S_n^m$ | $S_k$ letter perms | None | $S_k$ equation perms | $S_n$ project perms |
| Invariants | Latin square | Digit uniqueness | Associativity | Modular equivalence | Resource limits |
| Certificates | Complete grid | Letter mapping | Split sequence | Solution value | Selected projects |

Table 21: Complete Mathematical Formulas for All Hard Suite Tasks (10 Tasks, 68 Variations)

| Task | Core Mathematical Relations | Complexity/Counting |
|---|---|---|
| Tower of Hanoi (6 variations) | $T(n) = 2^n - 1$ (optimal moves) $S(n+1) = 3 \cdot S(n) - 1$ (state transitions) | State space: $3^n$ configurations Moves: $(3 \cdot 2 \cdot n)^{2^n - 1}$ |
| N-Queens (4 variations) | $\forall i \neq j : x_i \neq x_j \wedge \lvert i - j \rvert \neq \lvert x_i - x_j \rvert$ Conflict: $(r_i, c_i)$ attacks $(r_j, c_j)$ | Solutions: $S(n) \sim \frac{n!}{c^n}, c \approx 2.54$ Search space: $n!$ permutations |
| Graph Coloring (10 variations) | $\omega(G) \leq \chi(G) \leq \Delta(G) + 1$ $P_G(k) = \sum_{i=0}^{n} (-1)^{n-i} a_i k^i$ | Chromatic polynomial: $P_G(k)$ Colorings: $k^n$ assignments |
| Boolean SAT (5 variations) | $\phi = \bigwedge_{i=1}^{m} \bigvee_{j=1}^{k} \ell_{ij}$ (CNF) $\Pr[\text{SAT}] \approx e^{-\alpha/2^k}$ at $\alpha = m/n$ | Search space: $2^n$ assignments Phase transition: $\alpha_c \approx 4.267$ |
| Sudoku (8 variations) | $N_9 = 9! \times 72^2 \times 2^7 \times 27,704,267,971$ $\text{Candidates}(i,j) = [n] \setminus (\text{Row}_i \cup \text{Col}_j \cup \text{Box}_{ij})$ | Valid grids: $6.67 \times 10^{21}$ Minimal puzzles: $> 10^{16}$ |
| Logic Grid Puzzles (8 variations) | $\phi : E \times A \to \bigcup_{i=1}^{m} A_i$ bijective $\forall i, j, k : j \neq k \implies \phi(e_j, A_i) \neq \phi(e_k, A_i)$ | Solution space: $(n!)^{m-1}$ Problem space: $(n!)^m \cdot V^{nm}$ |
| Cryptarithmetic (12 variations) | $\text{val}(W) = \sum_{i=0}^{\lvert W \rvert - 1} \Psi(W[i]) \cdot 10^{\lvert W \rvert - 1 - i}$ $\sum_{i=1}^{n} \text{val}(W_i) = \text{val}(R)$ | Mappings: $\frac{10!}{(10-k)!}$ Problem space: $V^n \cdot \binom{26}{k} \cdot P(10, k)$ |
| Matrix Chain (5 variations) | $m[i,j] = \min_{i \leq k < j} \{ m[i,k] + m[k+1, j] + d_{i-1} \cdot d_k \cdot d_j \}$ Catalan: $C_n = \frac{1}{n+1} \binom{2n}{n}$ | Parenthesizations: $C_{n-1} = \frac{1}{n} \binom{2n-2}{n-1}$ Time: $O(n^3)$, Space: $O(n^2)$ |
| Modular Systems (5 variations) | $x \equiv a_i \pmod{m_i}, i \in [1, k]$ CRT: $x = \sum_{i=1}^{k} a_i M_i y_i \pmod{M}$ | Solution space: $M = \text{lcm}(m_1, ..., m_k)$ Period: $x = x_0 + kM, k \in \mathbb{Z}$ |
| Constraint Opt. (5 variations) | $\max \sum_{i=1}^{n} \pi_i x_i$ s.t. $\sum_{i=1}^{n} \rho_i(r_j) x_i \leq c_j$ Tightness: $\tau = \frac{\sum_i \min_j \rho_i(r_j)}{\sum_j c_j}$ | Search space: $2^n$ selections Feasible region: $\lvert \mathcal{F} \rvert \ll 2^n$ |

Table 22: Hard Suite Summary Statistics (10 Tasks, 68 Variations Total)

| Category | Range | Median | Notes |
|---|---|---|---|
| Problem Space Size | $10^9$ to $10^{50}$ | $10^{20}$ | Ensures contamination resistance |
| Context Requirements | 100-3825 tokens | 300 tokens | Fits in standard context windows |
| Verification Time | $O(n)$ to $O(2^n)$ | $O(n^2)$ | Mostly polynomial verification |
| Solution Uniqueness | 0 to $(n!)^m$ | 1 | Enforced by generation |
| Complexity Class | P to NP-complete | NP-complete | 8/10 are NP-complete |
| Symmetry Groups | 1 to $S_9 \wr S_3$ | $S_n$ | Rich algebraic structure |
| Total Variations | 4 to 12 per task | 6.8 per task | Comprehensive problem coverage |

## K  HARD SUITE: COMPLETE RESULTS

This section details results for all hard suite tasks across all non-quantized open-source models and closed-source proprietary models. The average accuracy, instruction following rate and average output tokens are listed for tower of hanoi, n-queens, graph coloring, boolean SAT, sudoku, cryptarithmetic, matrix chain multiplication, modular systems, constraint optimization, and logic grid puzzles in table 23, 24, 25.

| Model (Param) | Tower of Hanoi | | | N-Queens | | | Graph Coloring | | | Boolean SAT | | |
|---|---|---|---|---|---|---|---|---|---|---|---|---|
| | Acc (%) | Inst (%) | To-kens | Acc (%) | Inst (%) | To-kens | Acc (%) | Inst (%) | To-kens | Acc (%) | Inst (%) | To-kens |
| *Qwen Family (Qwen3)* | | | | | | | | | | | | |
| Qwen3 (0.6B) | 0.00 | $99.50_{\pm0.50}$ | 5720.93 | $6.00_{\pm10.39}$ | $60.50_{\pm30.99}$ | 4364.86 | $21.12_{\pm3.26}$ | $65.00_{\pm10.64}$ | 4755.65 | $14.29_{\pm20.18}$ | $41.86_{\pm30.57}$ | 6197.41 |
| Qwen3 (1.7B) | 1.17 | $98.50_{\pm1.26}$ | 7481.18 | $2.00_{\pm3.46}$ | $68.75_{\pm30.20}$ | 7411.59 | $42.00_{\pm13.76}$ | $61.13_{\pm15.14}$ | 5134.66 | $32.71_{\pm25.98}$ | $59.43_{\pm31.06}$ | 6071.69 |
| Qwen3 (4B) | 35.83 | $99.50_{\pm1.12}$ | 6214.78 | $32.50_{\pm36.66}$ | $61.50_{\pm28.86}$ | 7112.08 | $53.12_{\pm17.04}$ | $68.75_{\pm17.41}$ | 4756.59 | $49.86_{\pm22.71}$ | $66.86_{\pm26.06}$ | 5665.29 |
| Qwen3 (8B) | 37.17 | $99.83_{\pm0.37}$ | 6114.99 | $34.50_{\pm37.29}$ | $63.50_{\pm26.31}$ | 7002.57 | $60.62_{\pm15.17}$ | $75.00_{\pm16.09}$ | 4337.32 | $50.57_{\pm19.87}$ | $65.29_{\pm27.01}$ | 5856.17 |
| Qwen3 (14B) | 46.33 | $99.83_{\pm0.37}$ | 5446.42 | $43.50_{\pm43.96}$ | $58.75_{\pm31.12}$ | 6674.30 | $67.50_{\pm13.36}$ | $81.00_{\pm13.41}$ | 3997.80 | $47.80_{\pm23.16}$ | $60.60_{\pm24.19}$ | 5569.94 |
| Qwen3 (32B) | 39.67 | $99.50_{\pm0.76}$ | 6192.65 | $37.00_{\pm40.42}$ | $54.25_{\pm35.60}$ | 6568.30 | $70.88_{\pm10.51}$ | $80.50_{\pm9.87}$ | 4036.74 | $59.14_{\pm18.47}$ | $76.43_{\pm18.71}$ | 5251.74 |
| Qwen3 (30B-MOE) | 44.00 | 100.00 | 5187.79 | $39.00_{\pm39.41}$ | $61.75_{\pm31.04}$ | 6982.56 | $72.33_{\pm14.23}$ | $85.50_{\pm11.94}$ | 4014.90 | $55.86_{\pm20.68}$ | $69.29_{\pm24.13}$ | 5418.43 |
| Qwen3 (30B-MOE-t) | 38.50 | $75.17_{\pm17.95}$ | 6163.08 | $44.25_{\pm44.25}$ | $70.25_{\pm21.04}$ | 7317.35 | $67.33_{\pm6.82}$ | $87.33_{\pm9.37}$ | 4256.86 | $47.00_{\pm19.54}$ | $56.43_{\pm21.60}$ | 5947.41 |
| Qwen3 (30B-MOE-i) | 48.50 | 100.00 | 1041.17 | $43.75_{\pm44.07}$ | $90.25_{\pm7.36}$ | 5386.13 | $61.88_{\pm19.03}$ | $95.75_{\pm3.27}$ | 2135.51 | $19.71_{\pm11.37}$ | 100.00 | 56.00 |
| Qwen3 (4B-t) | 31.00 | $66.50_{\pm20.08}$ | 6739.23 | $42.75_{\pm43.52}$ | $83.00_{\pm10.77}$ | 7118.08 | $59.25_{\pm13.54}$ | $72.75_{\pm15.99}$ | 5018.43 | $30.00_{\pm16.35}$ | $48.43_{\pm25.11}$ | 6340.54 |
| *Qwen Family (Qwen2.5)* | | | | | | | | | | | | |
| Qwen2.5 (0.5B) | 0.00 | 100.00 | 2600.93 | 0.00 | $4.50_{\pm7.79}$ | 960.92 | $3.88_{\pm2.62}$ | $85.50_{\pm7.07}$ | 1311.70 | $2.43_{\pm4.53}$ | 100.00 | 73.22 |
| Qwen2.5 (1.5B) | 0.00 | 100.00 | 3671.21 | 0.00 | $61.00_{\pm31.12}$ | 153.90 | $13.50_{\pm3.12}$ | $55.38_{\pm22.65}$ | 354.70 | $2.29_{\pm3.15}$ | 100.00 | 189.65 |
| Qwen2.5 (3B) | 0.00 | 100.00 | 2500.64 | 0.00 | $45.00_{\pm33.32}$ | 727.38 | $25.50_{\pm5.68}$ | $93.75_{\pm4.35}$ | 1116.20 | $4.14_{\pm4.05}$ | 100.00 | 56.00 |
| Qwen2.5 (7B) | $25.67_{\pm38.65}$ | 100.00 | 865.18 | $0.25_{\pm0.43}$ | $28.75_{\pm37.93}$ | 444.63 | $33.25_{\pm4.32}$ | $97.62_{\pm1.80}$ | 843.49 | $11.86_{\pm10.12}$ | 100.00 | 1507.65 |
| Qwen2.5 (14B) | $36.83_{\pm44.99}$ | 100.00 | 1212.55 | $11.50_{\pm19.92}$ | $96.25_{\pm4.49}$ | 218.64 | $40.12_{\pm5.30}$ | $98.88_{\pm1.96}$ | 796.92 | $17.00_{\pm11.60}$ | 100.00 | 58.32 |
| Qwen2.5 (32B) | $38.83_{\pm36.52}$ | 100.00 | 625.78 | 0.00 | $99.75_{\pm0.43}$ | 332.58 | $45.75_{\pm8.63}$ | $98.50_{\pm1.22}$ | 522.26 | $22.57_{\pm14.96}$ | $96.43_{\pm7.96}$ | 56.04 |
| Qwen2.5 (72B) | $66.67_{\pm47.14}$ | 100.00 | 1077.27 | $25.00_{\pm43.30}$ | 100.00 | 537.08 | $45.75_{\pm6.67}$ | $99.62_{\pm0.70}$ | 963.99 | $24.29_{\pm14.19}$ | 100.00 | 56.00 |
| Qwen2.5 (1.5B-m) | 0.00 | 100.00 | 1248.11 | 0.00 | $47.50_{\pm42.20}$ | 532.63 | $6.04_{\pm4.08}$ | $46.37_{\pm21.29}$ | 795.09 | $1.71_{\pm2.05}$ | $16.57_{\pm5.01}$ | 2005.47 |
| Qwen2.5 (7B-m) | 16.67 | 100.00 | 1589.02 | 0.00 | $74.00_{\pm31.63}$ | 820.93 | $18.10_{\pm7.18}$ | $75.71_{\pm11.32}$ | 909.90 | $3.43_{\pm6.09}$ | $43.71_{\pm7.30}$ | 1955.96 |
| Qwen2.5 (72B-m) | 33.33 | 100.00 | 1678.19 | $9.00_{\pm15.59}$ | $36.25_{\pm8.98}$ | 658.21 | $26.39_{\pm7.98}$ | $87.04_{\pm5.03}$ | 958.56 | $2.86_{\pm1.96}$ | $11.14_{\pm5.51}$ | 2006.38 |
| *Gemma Family* | | | | | | | | | | | | |
| Gemma (1B) | 0.00 | 100.00 | 140.66 | 0.00 | $49.75_{\pm49.75}$ | 92.20 | $0.12_{\pm0.33}$ | $64.12_{\pm22.16}$ | 726.52 | $2.29_{\pm4.03}$ | 100.00 | 52.71 |
| Gemma (4B) | 0.00 | 100.00 | 582.18 | $12.50_{\pm21.65}$ | $75.00_{\pm43.30}$ | 250.05 | $20.00_{\pm13.23}$ | $80.00_{\pm36.06}$ | 623.54 | $5.71_{\pm10.50}$ | $24.29_{\pm39.23}$ | 7.31 |
| Gemma (12B) | 0.00 | 100.00 | 872.63 | $2.50_{\pm4.33}$ | $52.50_{\pm42.65}$ | 444.02 | $33.75_{\pm16.54}$ | $82.50_{\pm33.82}$ | 862.88 | $8.57_{\pm14.57}$ | $24.29_{\pm39.23}$ | 20.94 |
| Gemma (27B) | $16.67_{\pm37.27}$ | 100.00 | 535.47 | $24.00_{\pm41.57}$ | $99.00_{\pm1.73}$ | 271.88 | $50.00_{\pm14.81}$ | $98.62_{\pm1.41}$ | 1037.74 | $39.71_{\pm19.20}$ | 100.00 | 1136.40 |
| *Phi Family* | | | | | | | | | | | | |
| Phi4 (14B) | $8.50_{\pm18.56}$ | 100.00 | 522.16 | $25.00_{\pm43.30}$ | $89.50_{\pm18.19}$ | 468.30 | $46.63_{\pm9.77}$ | 100.00 | 721.18 | $25.86_{\pm16.66}$ | 100.00 | 1113.29 |
| Phi4-reasoning+ (14B) | 16.17 | $56.67_{\pm41.49}$ | 7786.27 | $0.25_{\pm0.43}$ | $1.50_{\pm1.50}$ | 7368.81 | $36.75_{\pm13.28}$ | $59.25_{\pm15.59}$ | 6216.92 | $47.57_{\pm16.20}$ | $65.29_{\pm20.81}$ | 6457.09 |
| Phi4-reasoning (14B) | 27.00 | $60.33_{\pm38.72}$ | 7763.62 | $6.75_{\pm10.57}$ | $15.25_{\pm8.93}$ | 6958.67 | $65.75_{\pm10.66}$ | $76.00_{\pm11.59}$ | 5322.74 | $55.00_{\pm16.70}$ | $74.57_{\pm17.15}$ | 5893.97 |
| Phi4-mini-reasoning (3.8B) | 12.50 | 100.00 | 6900.28 | $24.50_{\pm30.63}$ | $67.00_{\pm32.65}$ | 7054.01 | $41.25_{\pm14.32}$ | $66.00_{\pm14.77}$ | 5061.80 | $24.86_{\pm23.66}$ | $46.57_{\pm32.43}$ | 5933.85 |
| Phi3-mini (3.8B) | 0.00 | 100.00 | 2142.96 | 0.00 | $32.25_{\pm40.87}$ | 82.90 | $21.12_{\pm4.28}$ | $87.75_{\pm9.39}$ | 623.80 | $7.14_{\pm4.09}$ | $99.86_{\pm0.35}$ | 109.15 |
| Phi3-med (14B-4k) | $22.17_{\pm36.83}$ | 100.00 | 1824.81 | $0.50_{\pm0.87}$ | $80.25_{\pm33.63}$ | 113.15 | $21.79_{\pm4.51}$ | $86.32_{\pm3.28}$ | 172.57 | $11.00_{\pm9.62}$ | 100.00 | 56.00 |
| Phi3-med (14B-128k) | $16.67_{\pm37.27}$ | 100.00 | 2949.26 | $3.50_{\pm6.06}$ | $76.50_{\pm37.87}$ | 133.58 | $26.00_{\pm4.61}$ | $97.12_{\pm4.17}$ | 386.58 | $7.86_{\pm9.20}$ | 100.00 | 56.00 |
| *Llama Family* | | | | | | | | | | | | |
| Llama-3.2 (1B) | 0.00 | $99.83_{\pm0.37}$ | 3277.66 | 0.00 | $21.50_{\pm24.72}$ | 1158.68 | $2.38_{\pm3.04}$ | $72.88_{\pm19.28}$ | 1557.92 | $2.00_{\pm3.02}$ | $57.57_{\pm3.02}$ | 2376.83 |
| Llama-3.2 (3B) | 0.00 | 100.00 | 2531.26 | 0.00 | $6.25_{\pm5.07}$ | 368.96 | $13.62_{\pm4.12}$ | $83.12_{\pm10.46}$ | 1296.02 | $5.29_{\pm7.91}$ | $99.71_{\pm0.45}$ | 104.28 |
| Llama-3.1 (8B) | 0.00 | 100.00 | 6076.68 | 0.00 | $5.00_{\pm5.34}$ | 970.66 | $24.88_{\pm5.18}$ | $95.88_{\pm2.45}$ | 3005.00 | $12.57_{\pm8.43}$ | 100.00 | 71.62 |
| Llama-3.1 (70B) | $33.33_{\pm47.14}$ | 100.00 | 2308.85 | $1.00_{\pm1.73}$ | $78.75_{\pm24.75}$ | 839.39 | $44.50_{\pm6.75}$ | $98.62_{\pm2.12}$ | 839.25 | $19.00_{\pm11.50}$ | 100.00 | 56.00 |
| Llama-3.3 (70B) | $48.17_{\pm26.91}$ | 100.00 | 1841.30 | $25.75_{\pm42.89}$ | $99.25_{\pm1.30}$ | 260.60 | $44.88_{\pm11.98}$ | $99.38_{\pm1.11}$ | 860.09 | $25.29_{\pm15.45}$ | 100.00 | 710.85 |
| Llama4-scout | 13.50 | 100.00 | 5183.55 | $33.50_{\pm37.88}$ | $86.00_{\pm21.46}$ | 618.25 | $15.88_{\pm8.43}$ | $40.88_{\pm9.70}$ | 613.48 | $2.00_{\pm4.90}$ | $2.14_{\pm5.25}$ | 28.02 |
| *Mistral Family* | | | | | | | | | | | | |
| Mistral (7B) | 0.00 | 100.00 | 3106.56 | 0.00 | $50.00_{\pm50.00}$ | 387.54 | $21.25_{\pm2.63}$ | $91.12_{\pm4.11}$ | 884.03 | $5.14_{\pm3.52}$ | $97.14_{\pm4.16}$ | 546.39 |
| Ministral (8B) | 0.00 | 100.00 | 3009.91 | 0.00 | $54.75_{\pm34.75}$ | 167.46 | $32.25_{\pm4.12}$ | $99.12_{\pm1.62}$ | 777.57 | $4.29_{\pm4.77}$ | 100.00 | 56.00 |
| Mistral-nemo (12B) | 0.00 | 100.00 | 811.43 | $25.00_{\pm43.30}$ | $75.00_{\pm43.30}$ | 28.25 | $25.75_{\pm5.61}$ | $92.88_{\pm6.85}$ | 495.36 | $6.43_{\pm5.90}$ | 100.00 | 56.00 |
| Mixtral-8x7b | $16.67_{\pm37.27}$ | 100.00 | 481.06 | 0.00 | $43.50_{\pm41.45}$ | 398.12 | $17.38_{\pm6.58}$ | $93.12_{\pm3.82}$ | 370.98 | $9.29_{\pm6.54}$ | $98.14_{\pm4.55}$ | 207.64 |
| Mixtral-8x22b | $24.00_{\pm37.59}$ | 100.00 | 939.81 | $22.50_{\pm38.97}$ | $71.25_{\pm13.25}$ | 230.85 | $36.12_{\pm4.86}$ | $99.00_{\pm1.32}$ | 553.73 | $16.71_{\pm12.41}$ | 100.00 | 308.22 |
| *Others* | | | | | | | | | | | | |
| Smollm3 (3B) | 13.50 | $99.83_{\pm0.37}$ | 6727.13 | $23.00_{\pm23.39}$ | $63.00_{\pm16.32}$ | 6270.72 | $41.25_{\pm10.13}$ | $62.88_{\pm12.24}$ | 4726.78 | $22.43_{\pm21.67}$ | $49.86_{\pm26.91}$ | 6463.88 |
| Smollm2 (1.7B) | 0.00 | 100.00 | 7148.82 | 0.00 | $25.00_{\pm43.30}$ | 31.50 | $7.63_{\pm5.89}$ | 100.00 | 165.69 | $2.43_{\pm4.07}$ | 100.00 | 56.00 |
| GPT-OSS (20B) | 49.83 | $44.67_{\pm35.94}$ | 4645.44 | $33.50_{\pm33.97}$ | $77.00_{\pm2.55}$ | 2941.60 | $74.12_{\pm9.91}$ | $85.88_{\pm9.20}$ | 2962.41 | $54.00_{\pm19.89}$ | $65.43_{\pm21.06}$ | 4593.79 |
| GPT-OSS (120B) | 55.33 | $17.50_{\pm36.90}$ | 3441.49 | $15.50_{\pm20.06}$ | $59.25_{\pm29.41}$ | 1297.66 | $86.75_{\pm8.63}$ | $92.00_{\pm6.10}$ | 2297.46 | $67.86_{\pm15.59}$ | $78.57_{\pm16.49}$ | 4046.01 |
| *OpenAI Family (Proprietary)* | | | | | | | | | | | | |
| GPT5 | $91.67_{\pm14.62}$ | 100.00 | 1904.68 | $60.00_{\pm42.43}$ | 100.00 | 85.80 | 100.00 | 100.00 | 472.42 | 100.00 | 100.00 | 145.60 |
| GPT5-mini | $58.33_{\pm42.59}$ | $75.00_{\pm35.47}$ | 621.05 | $60.00_{\pm40.62}$ | 100.00 | 86.05 | $95.00_{\pm7.07}$ | $95.00_{\pm7.07}$ | 533.74 | 100.00 | 100.00 | 134.40 |
| GPT5-nano | $53.33_{\pm44.60}$ | $78.33_{\pm29.67}$ | 738.48 | $50.00_{\pm50.00}$ | $97.50_{\pm4.33}$ | 80.60 | $95.00_{\pm5.00}$ | $95.00_{\pm5.00}$ | 476.16 | $97.14_{\pm7.00}$ | $97.14_{\pm7.00}$ | 152.20 |
| GPT4.1 | $50.00_{\pm50.00}$ | 100.00 | 944.78 | $70.00_{\pm41.23}$ | 100.00 | 203.72 | $73.75_{\pm15.76}$ | 100.00 | 1233.01 | $22.86_{\pm13.85}$ | 100.00 | 56.00 |
| GPT4.1-mini | $35.00_{\pm43.11}$ | $90.00_{\pm22.36}$ | 575.32 | $45.00_{\pm45.55}$ | $77.50_{\pm17.85}$ | 1372.90 | $73.75_{\pm17.28}$ | 100.00 | 1216.45 | $27.14_{\pm16.66}$ | $98.57_{\pm3.50}$ | 723.09 |
| GPT4.1-nano | $6.67_{\pm11.06}$ | 100.00 | 768.63 | $25.00_{\pm43.30}$ | $97.50_{\pm4.33}$ | 294.00 | $42.50_{\pm17.14}$ | $96.25_{\pm6.96}$ | 865.86 | $14.29_{\pm10.50}$ | 100.00 | 55.14 |
| GPT4o | $50.00_{\pm50.00}$ | 100.00 | 833.17 | $62.50_{\pm41.46}$ | $87.50_{\pm21.65}$ | 226.80 | $57.50_{\pm6.61}$ | $98.75_{\pm3.31}$ | 563.71 | $21.43_{\pm19.59}$ | 100.00 | 315.14 |
| GPT4o-mini | $5.00_{\pm11.18}$ | 100.00 | 2502.63 | $17.50_{\pm17.85}$ | $82.50_{\pm17.85}$ | 295.58 | $36.25_{\pm12.18}$ | 100.00 | 747.78 | $11.43_{\pm8.33}$ | 100.00 | 56.00 |
| o4 mini | $50.00_{\pm38.73}$ | $98.33_{\pm3.73}$ | 1130.40 | $50.00_{\pm50.00}$ | $82.50_{\pm20.46}$ | 65.89 | $97.50_{\pm4.33}$ | $98.75_{\pm3.31}$ | 422.33 | $95.71_{\pm7.28}$ | 100.00 | 128.59 |
| o3 | $90.00_{\pm18.26}$ | 100.00 | 3090.85 | $70.00_{\pm41.23}$ | 100.00 | 118.80 | $98.75_{\pm3.31}$ | 100.00 | 649.01 | 100.00 | 100.00 | 201.60 |
| o3 mini | $51.67_{\pm34.36}$ | 100.00 | 2830.99 | $52.50_{\pm47.63}$ | $92.50_{\pm8.29}$ | 567.89 | $92.50_{\pm10.90}$ | 100.00 | 1934.45 | $97.14_{\pm4.52}$ | 100.00 | 465.14 |
| *Gemini Family (Proprietary)* | | | | | | | | | | | | |
| Gemini-2.5-pro | $96.67_{\pm7.45}$ | 100.00 | 995.93 | $50.00_{\pm45.28}$ | $97.50_{\pm4.33}$ | 398.28 | $90.00_{\pm7.07}$ | $96.25_{\pm4.84}$ | 654.46 | 100.00 | 100.00 | 269.21 |
| Gemini-2.5-flash | $91.67_{\pm8.98}$ | 100.00 | 981.12 | $45.00_{\pm36.40}$ | $75.00_{\pm18.03}$ | 413.92 | $86.25_{\pm9.92}$ | $97.50_{\pm4.33}$ | 1453.89 | 100.00 | 100.00 | 56.00 |
| Gemini-2.5-flash-lite | $30.00_{\pm40.00}$ | 100.00 | 2791.90 | $60.00_{\pm36.74}$ | $92.50_{\pm12.99}$ | 6544.18 | $73.75_{\pm11.11}$ | 100.00 | 3444.23 | $84.29_{\pm15.91}$ | $94.29_{\pm7.28}$ | 7332.30 |
| Gemini-2.0-flash | $93.33_{\pm14.91}$ | 100.00 | 765.33 | $27.50_{\pm27.73}$ | $65.00_{\pm39.05}$ | 1710.43 | $65.00_{\pm8.66}$ | 100.00 | 1502.18 | $22.86_{\pm23.73}$ | 100.00 | 215.09 |
| Gemini-2.0-flash-lite | $50.00_{\pm50.00}$ | 100.00 | 1092.53 | $40.00_{\pm39.37}$ | $95.00_{\pm5.00}$ | 694.50 | $48.75_{\pm12.69}$ | 100.00 | 1163.65 | $42.86_{\pm13.85}$ | $94.29_{\pm7.28}$ | 1899.24 |

Table 23: **Hard Suite Results - Table 1:** Performance of regular models on Tower of Hanoi (3-8 disks), N-Queens (6-16 boards), Graph Coloring, and Boolean SAT tasks. Each task reports Accuracy (Acc), Instruction-following (Inst) and average output Tokens with mean and standard deviation values. Results show average performance across different complexity variants within each task category.

| Model (Param) | Sudoku | | | Cryptarithmetic | | | Matrix Chain Mult | | |
|---|---|---|---|---|---|---|---|---|---|
| | Acc (%) | Inst (%) | Tokens | Acc (%) | Inst (%) | Tokens | Acc (%) | Inst (%) | Tokens |
| *Qwen Family (Qwen3)* | | | | | | | | | |
| Qwen3 (0.6B) | $12.00_{\pm12.83}$ | $52.67_{\pm31.54}$ | 6609.54 | 0.00 | $70.15_{\pm5.34}$ | 7565.67 | $0.43_{\pm1.05}$ | $56.00_{\pm24.65}$ | 6168.81 |
| Qwen3 (1.7B) | $28.33_{\pm25.62}$ | $63.00_{\pm26.55}$ | 6383.16 | $11.23_{\pm6.21}$ | $93.65_{\pm3.44}$ | 7519.00 | $5.57_{\pm12.47}$ | $63.14_{\pm19.86}$ | 6350.30 |
| Qwen3 (4B) | $21.00_{\pm12.33}$ | $61.00_{\pm30.01}$ | 6096.32 | $37.32_{\pm14.89}$ | $91.68_{\pm3.03}$ | 7159.25 | $24.71_{\pm37.20}$ | $93.43_{\pm10.17}$ | 5505.64 |
| Qwen3 (8B) | $21.33_{\pm13.27}$ | $60.00_{\pm30.74}$ | 6227.63 | $52.74_{\pm2.05}$ | $97.88_{\pm2.12}$ | 6779.93 | $24.29_{\pm37.50}$ | $94.29_{\pm8.46}$ | 5499.92 |
| Qwen3 (14B) | $36.00_{\pm33.03}$ | $60.33_{\pm33.09}$ | 5783.53 | $41.78_{\pm15.89}$ | $98.00_{\pm2.45}$ | 7040.81 | $30.43_{\pm40.49}$ | $95.43_{\pm7.17}$ | 5237.94 |
| Qwen3 (32B) | $33.00_{\pm31.38}$ | $58.33_{\pm34.34}$ | 5935.34 | $44.57_{\pm13.27}$ | $97.92_{\pm1.54}$ | 6854.37 | $28.43_{\pm39.59}$ | $94.29_{\pm9.18}$ | 5144.81 |
| Qwen3 (30B-MOE) | $37.67_{\pm33.72}$ | $64.67_{\pm29.49}$ | 5785.12 | $41.64_{\pm11.82}$ | $97.28_{\pm1.75}$ | 6955.30 | $28.71_{\pm40.70}$ | $96.71_{\pm5.36}$ | 5099.20 |
| Qwen3 (30B-MOE-t) | $34.33_{\pm25.94}$ | $65.67_{\pm23.33}$ | 6002.56 | $31.37_{\pm11.94}$ | $98.95_{\pm0.71}$ | 6757.44 | $28.43_{\pm40.47}$ | $95.00_{\pm8.02}$ | 4993.60 |
| Qwen3 (30B-MOE-i) | $34.67_{\pm19.60}$ | 100.00 | 139.00 | $24.06_{\pm11.68}$ | 100.00 | 164.67 | $29.29_{\pm37.26}$ | $95.29_{\pm7.83}$ | 3542.83 |
| Qwen3 (4B-t) | $7.33_{\pm7.72}$ | $73.33_{\pm17.46}$ | 6530.43 | $36.14_{\pm10.19}$ | $98.41_{\pm1.09}$ | 7176.42 | $19.14_{\pm34.36}$ | $96.14_{\pm6.15}$ | 5567.18 |
| *Qwen Family (Qwen2.5)* | | | | | | | | | |
| Qwen2.5 (0.5B) | $1.00_{\pm1.41}$ | $98.67_{\pm1.89}$ | 163.13 | 0.00 | $77.09_{\pm5.93}$ | 3542.86 | 0.00 | $22.29_{\pm6.27}$ | 2418.80 |
| Qwen2.5 (1.5B) | $2.33_{\pm2.62}$ | $97.33_{\pm2.49}$ | 138.35 | 0.00 | $99.75_{\pm0.43}$ | 109.10 | 0.00 | $85.43_{\pm6.39}$ | 1373.84 |
| Qwen2.5 (3B) | $1.00_{\pm1.41}$ | 100.00 | 138.82 | $20.02_{\pm12.20}$ | 100.00 | 57.76 | $0.14_{\pm0.35}$ | $92.57_{\pm6.67}$ | 1190.43 |
| Qwen2.5 (7B) | $6.67_{\pm6.60}$ | $95.67_{\pm4.71}$ | 144.60 | $20.64_{\pm11.47}$ | 100.00 | 98.18 | $1.14_{\pm2.42}$ | $99.86_{\pm0.35}$ | 1295.30 |
| Qwen2.5 (14B) | $7.00_{\pm2.16}$ | $94.00_{\pm5.35}$ | 460.85 | $18.66_{\pm13.45}$ | 100.00 | 58.45 | $3.86_{\pm7.94}$ | $99.86_{\pm0.35}$ | 930.01 |
| Qwen2.5 (32B) | $30.33_{\pm20.27}$ | $96.00_{\pm4.97}$ | 431.23 | $19.21_{\pm12.22}$ | 100.00 | 66.47 | $11.29_{\pm16.71}$ | 100.00 | 1185.70 |
| Qwen2.5 (72B) | $31.67_{\pm22.22}$ | $98.67_{\pm0.94}$ | 396.92 | $20.19_{\pm13.98}$ | 100.00 | 57.82 | $10.71_{\pm16.34}$ | 100.00 | 1080.80 |
| Qwen2.5 (1.5B-m) | $3.33_{\pm4.71}$ | $87.33_{\pm5.56}$ | 1429.37 | $2.30_{\pm0.77}$ | $82.47_{\pm1.93}$ | 1300.36 | $1.29_{\pm3.15}$ | $74.71_{\pm29.37}$ | 1263.87 |
| Qwen2.5 (7B-m) | $4.00_{\pm4.97}$ | $90.00_{\pm9.42}$ | 1619.87 | $3.27_{\pm3.09}$ | $78.05_{\pm4.96}$ | 1804.41 | $4.86_{\pm10.01}$ | $98.71_{\pm1.67}$ | 1056.46 |
| Qwen2.5 (72B-m) | $4.67_{\pm5.25}$ | $63.33_{\pm6.94}$ | 1691.43 | $15.34_{\pm10.40}$ | $92.70_{\pm3.10}$ | 1544.12 | $10.00_{\pm16.38}$ | $98.71_{\pm1.67}$ | 1143.56 |
| *Gemma Family* | | | | | | | | | |
| Gemma (1B) | $1.33_{\pm1.89}$ | $97.67_{\pm1.70}$ | 132.12 | 0.00 | $70.54_{\pm14.84}$ | 1287.33 | 0.00 | $0.57_{\pm1.05}$ | 391.17 |
| Gemma (4B) | – | – | – | 0.00 | 100.00 | 2001.33 | $1.43_{\pm3.50}$ | $98.57_{\pm3.50}$ | 1084.59 |
| Gemma (12B) | – | – | – | $22.50_{\pm8.29}$ | 100.00 | 240.60 | $2.86_{\pm7.00}$ | $62.86_{\pm44.63}$ | 754.87 |
| Gemma (27B) | – | – | – | $20.00_{\pm10.00}$ | 100.00 | 1632.22 | $14.43_{\pm20.15}$ | $97.00_{\pm5.88}$ | 1088.23 |
| *Phi Family* | | | | | | | | | |
| Phi4 (14B) | $16.00_{\pm17.28}$ | $99.33_{\pm0.94}$ | 1241.54 | $14.43_{\pm11.19}$ | 100.00 | 763.50 | $10.57_{\pm16.51}$ | $99.86_{\pm0.35}$ | 1298.49 |
| Phi4-reasoning+ (14B) | $4.33_{\pm3.68}$ | $83.67_{\pm5.79}$ | 6808.20 | $29.64_{\pm8.31}$ | $94.14_{\pm2.85}$ | 7527.76 | $20.86_{\pm31.67}$ | $87.00_{\pm9.56}$ | 6806.51 |
| Phi4-reasoning (14B) | $9.33_{\pm6.80}$ | $68.33_{\pm18.57}$ | 6194.43 | $36.50_{\pm5.40}$ | $98.24_{\pm0.76}$ | 7256.05 | $25.29_{\pm30.14}$ | $91.14_{\pm12.22}$ | 6808.43 |
| Phi4-mini-reasoning (3.8B) | $24.67_{\pm30.07}$ | $50.67_{\pm32.19}$ | 6336.27 | $22.00_{\pm10.08}$ | $90.02_{\pm1.94}$ | 7403.47 | $12.29_{\pm23.62}$ | $77.86_{\pm13.59}$ | 6324.59 |
| Phi3-mini (3.8B) | $1.00_{\pm1.41}$ | 100.00 | 138.38 | $18.94_{\pm10.89}$ | 100.00 | 437.54 | $0.43_{\pm1.05}$ | $85.43_{\pm9.50}$ | 482.70 |
| Phi3-med (14B-4k) | $5.00_{\pm5.10}$ | 100.00 | 141.63 | $19.96_{\pm13.74}$ | 100.00 | 58.82 | $1.14_{\pm2.80}$ | $94.00_{\pm6.02}$ | 813.81 |
| Phi3-med (14B-128k) | $11.33_{\pm14.64}$ | 100.00 | 141.61 | $21.25_{\pm13.16}$ | 100.00 | 88.61 | $0.57_{\pm1.05}$ | $94.86_{\pm3.18}$ | 786.49 |
| *Llama Family* | | | | | | | | | |
| Llama-3.2 (1B) | $0.33_{\pm0.47}$ | $70.00_{\pm20.61}$ | 2975.61 | 0.00 | $32.34_{\pm5.53}$ | 1200.41 | 0.00 | $59.57_{\pm11.91}$ | 1420.35 |
| Llama-3.2 (3B) | $1.67_{\pm2.36}$ | 100.00 | 137.96 | 0.00 | $81.23_{\pm3.36}$ | 3566.04 | 0.00 | $0.29_{\pm0.70}$ | 376.11 |
| Llama-3.1 (8B) | $3.33_{\pm4.71}$ | 100.00 | 184.56 | 0.00 | 100.00 | 840.95 | $0.14_{\pm0.35}$ | $98.71_{\pm0.70}$ | 1355.48 |
| Llama-3.1 (70B) | $22.67_{\pm23.92}$ | $94.33_{\pm0.47}$ | 431.32 | $21.63_{\pm14.19}$ | $99.25_{\pm0.83}$ | 498.91 | $3.71_{\pm7.94}$ | 100.00 | 786.73 |
| Llama-3.3 (70B) | $24.00_{\pm22.99}$ | $98.00_{\pm0.82}$ | 392.61 | $22.06_{\pm13.60}$ | 100.00 | 972.61 | $4.57_{\pm8.91}$ | 100.00 | 854.50 |
| Llama4-scout | 0.00 | 0.00 | 94.46 | 0.00 | $21.45_{\pm9.03}$ | 252.30 | $9.57_{\pm22.64}$ | $39.29_{\pm27.83}$ | 3173.07 |
| *Mistral Family* | | | | | | | | | |
| Mistral (7B) | $1.00_{\pm1.41}$ | 100.00 | 1418.95 | 0.00 | 100.00 | 996.45 | $0.14_{\pm0.35}$ | $89.71_{\pm5.92}$ | 853.69 |
| Ministral (8B) | $1.00_{\pm1.41}$ | 139.00 | 139.00 | 0.00 | 100.00 | 57.93 | $0.57_{\pm1.40}$ | $99.29_{\pm0.70}$ | 913.47 |
| Mistral-nemo (12B) | $4.00_{\pm4.97}$ | $97.33_{\pm2.49}$ | 138.43 | $19.90_{\pm12.09}$ | 100.00 | 57.70 | 0.00 | $71.14_{\pm24.51}$ | 841.58 |
| Mixtral-8x7b | $8.33_{\pm11.79}$ | $94.00_{\pm5.10}$ | 186.00 | $14.60_{\pm10.17}$ | $78.55_{\pm12.30}$ | 569.62 | $0.71_{\pm1.75}$ | $86.14_{\pm9.26}$ | 1084.05 |
| Mixtral-8x22b | $12.67_{\pm13.89}$ | 100.00 | 610.22 | $18.17_{\pm13.40}$ | 100.00 | 1304.87 | $3.29_{\pm7.65}$ | $95.57_{\pm2.77}$ | 739.31 |
| *Others* | | | | | | | | | |
| Smollm3 (3B) | $1.33_{\pm1.89}$ | $26.00_{\pm21.23}$ | 6621.98 | $6.69_{\pm0.41}$ | $78.87_{\pm4.69}$ | 7344.62 | $10.57_{\pm19.43}$ | $80.29_{\pm13.66}$ | 5920.62 |
| Smollm2 (1.7B) | $1.00_{\pm1.41}$ | 100.00 | 139.00 | 0.00 | 100.00 | 62.08 | 0.00 | $99.29_{\pm0.88}$ | 1137.10 |
| GPT-OSS (20B) | $60.67_{\pm22.81}$ | $71.67_{\pm15.86}$ | 3366.38 | $67.02_{\pm7.00}$ | $98.75_{\pm0.83}$ | 5056.63 | $40.43_{\pm36.23}$ | $95.43_{\pm7.25}$ | 5318.79 |
| GPT-OSS (120B) | $34.67_{\pm6.85}$ | $46.33_{\pm7.72}$ | 2814.10 | $81.46_{\pm5.80}$ | $99.72_{\pm0.48}$ | 4157.96 | $50.14_{\pm33.81}$ | $95.14_{\pm7.61}$ | 4734.53 |
| *OpenAI Family (Proprietary)* | | | | | | | | | |
| GPT5 | $90.00_{\pm14.14}$ | $93.33_{\pm9.43}$ | 386.18 | 100.00 | 100.00 | 1094.39 | $68.57_{\pm34.82}$ | $70.00_{\pm35.46}$ | 578.76 |
| GPT5-mini | $83.33_{\pm17.00}$ | 100.00 | 333.60 | $92.50_{\pm8.29}$ | $95.00_{\pm5.00}$ | 140.04 | $51.43_{\pm46.42}$ | $54.29_{\pm47.47}$ | 205.20 |
| GPT5-nano | $66.67_{\pm4.71}$ | 100.00 | 361.40 | $90.00_{\pm12.25}$ | $92.50_{\pm12.99}$ | 157.64 | $64.29_{\pm36.20}$ | $64.29_{\pm36.20}$ | 100.85 |
| GPT4.1 | $50.00_{\pm32.66}$ | $90.00_{\pm14.14}$ | 140.90 | $60.00_{\pm24.49}$ | $92.50_{\pm8.29}$ | 6537.75 | $30.00_{\pm32.95}$ | $95.71_{\pm7.28}$ | 3632.27 |
| GPT4.1-mini | $40.00_{\pm21.60}$ | $70.00_{\pm21.60}$ | 139.00 | $55.00_{\pm20.62}$ | 100.00 | 6373.95 | $32.86_{\pm34.11}$ | $98.57_{\pm3.50}$ | 2950.50 |
| GPT4.1-nano | $40.00_{\pm16.33}$ | 100.00 | 139.00 | $25.00_{\pm5.00}$ | 100.00 | 1787.75 | $20.00_{\pm33.38}$ | 100.00 | 1492.21 |
| GPT4o | $43.33_{\pm18.86}$ | 100.00 | 141.83 | $22.50_{\pm10.90}$ | 100.00 | 497.22 | $11.43_{\pm21.00}$ | $94.29_{\pm7.28}$ | 713.10 |
| GPT4o-mini | $6.67_{\pm4.71}$ | 100.00 | 139.00 | $25.00_{\pm11.18}$ | 100.00 | 68.60 | $1.43_{\pm3.50}$ | $98.57_{\pm3.50}$ | 892.06 |
| o4 mini | $83.33_{\pm9.43}$ | 90.00 | 294.24 | $92.50_{\pm8.29}$ | 100.00 | 125.99 | $65.71_{\pm34.17}$ | $77.14_{\pm22.50}$ | 106.72 |
| o3 | $90.00_{\pm14.14}$ | 100.00 | 500.40 | $92.50_{\pm12.99}$ | $95.00_{\pm8.66}$ | 244.91 | $67.14_{\pm33.26}$ | $67.14_{\pm33.26}$ | 21.82 |
| o3 mini | $86.67_{\pm12.47}$ | $93.33_{\pm9.43}$ | 659.69 | $92.50_{\pm8.29}$ | $97.50_{\pm4.33}$ | 1211.04 | $67.14_{\pm31.04}$ | $78.57_{\pm25.31}$ | 3590.14 |
| *Gemini Family (Proprietary)* | | | | | | | | | |
| Gemini-2.5-pro | $83.33_{\pm9.43}$ | 100.00 | 139.00 | $97.50_{\pm4.33}$ | $97.50_{\pm4.33}$ | 1352.35 | $40.00_{\pm38.17}$ | $90.00_{\pm13.09}$ | 1566.50 |
| Gemini-2.5-flash | $70.00_{\pm21.60}$ | $76.67_{\pm20.55}$ | 89.90 | $57.50_{\pm8.29}$ | $60.00_{\pm7.07}$ | 437.38 | $31.43_{\pm33.99}$ | $92.86_{\pm10.30}$ | 2265.30 |
| Gemini-2.5-flash-lite | $30.00_{\pm16.33}$ | 100.00 | 200.07 | $47.50_{\pm14.79}$ | $97.50_{\pm4.33}$ | 6928.35 | $21.43_{\pm23.56}$ | $95.71_{\pm7.28}$ | 3951.26 |
| Gemini-2.0-flash | $43.33_{\pm17.00}$ | 100.00 | 180.60 | $30.00_{\pm24.49}$ | 100.00 | 2678.82 | $11.43_{\pm17.26}$ | $61.43_{\pm42.57}$ | 854.70 |
| Gemini-2.0-flash-lite | $26.67_{\pm17.00}$ | $90.00_{\pm8.16}$ | 140.70 | $20.00_{\pm10.00}$ | $97.50_{\pm4.33}$ | 4624.77 | $22.86_{\pm26.57}$ | $90.00_{\pm16.04}$ | 1115.83 |

Table 24: **Hard Suite Results - Table 2:** Performance of regular models on Sudoku, Cryptarithmetic, and Matrix Chain Multiplication tasks. Each task reports Accuracy (Acc), Instruction-following (Inst) and average output Tokens with mean and standard deviation values. Results show average performance across different complexity variants within each task category.

| Model (Param) | Modular Systems | | | Constraint Opt | | | Logic Grid Puzzles | | |
|---|---|---|---|---|---|---|---|---|---|
| | Acc (%) | Inst (%) | Tokens | Acc (%) | Inst (%) | Tokens | Acc (%) | Inst (%) | Tokens |
| *Qwen Family (Qwen3)* | | | | | | | | | |
| Qwen3 (0.6B) | $18.40_{\pm23.23}$ | $97.20_{\pm1.47}$ | 5520.66 | $14.20_{\pm10.74}$ | 100.00 | 5110.81 | $56.00_{\pm44.00}$ | $98.78_{\pm0.58}$ | 3413.56 |
| Qwen3 (1.7B) | $53.20_{\pm33.16}$ | $99.60_{\pm0.49}$ | 4855.61 | $14.20_{\pm7.83}$ | 100.00 | 5740.65 | $52.00_{\pm48.00}$ | $99.30_{\pm0.45}$ | 3949.75 |
| Qwen3 (4B) | $65.60_{\pm29.59}$ | $99.40_{\pm0.80}$ | 5010.25 | $8.60_{\pm8.31}$ | 100.00 | 6246.03 | $52.50_{\pm47.50}$ | $96.38_{\pm3.63}$ | 4350.69 |
| Qwen3 (8B) | $56.40_{\pm33.60}$ | $99.60_{\pm0.49}$ | 5390.77 | $18.20_{\pm28.55}$ | 100.00 | 6201.14 | $51.50_{\pm48.50}$ | $95.92_{\pm4.08}$ | 4453.73 |
| Qwen3 (14B) | $68.60_{\pm29.28}$ | 100.00 | 4852.27 | $31.00_{\pm38.58}$ | 100.00 | 5866.55 | $61.00_{\pm39.00}$ | $98.20_{\pm1.70}$ | 4217.58 |
| Qwen3 (32B) | $65.60_{\pm30.88}$ | 100.00 | 4931.32 | $36.60_{\pm35.31}$ | 100.00 | 5721.93 | $56.50_{\pm43.50}$ | $97.50_{\pm2.50}$ | 4209.59 |
| Qwen3 (30B-MOE) | $80.60_{\pm23.58}$ | $99.80_{\pm0.40}$ | 4200.79 | $22.00_{\pm35.87}$ | 100.00 | 6078.78 | $55.50_{\pm44.50}$ | $97.80_{\pm2.15}$ | 4195.41 |
| Qwen3 (30B-MOE-t) | $89.20_{\pm12.89}$ | $99.80_{\pm0.40}$ | 3586.24 | $21.00_{\pm32.93}$ | 100.00 | 6168.63 | $50.00_{\pm50.00}$ | $97.50_{\pm2.50}$ | 4542.49 |
| Qwen3 (30B-MOE-i) | $83.20_{\pm19.98}$ | $99.80_{\pm0.40}$ | 3655.26 | $72.60_{\pm26.34}$ | 100.00 | 4996.66 | $63.00_{\pm37.00}$ | $98.20_{\pm1.80}$ | 4082.00 |
| Qwen3 (4B-t) | $84.00_{\pm17.71}$ | $99.80_{\pm0.40}$ | 3658.34 | $16.00_{\pm26.71}$ | 100.00 | 6222.76 | $57.50_{\pm42.50}$ | $97.50_{\pm2.05}$ | 4943.09 |
| *Qwen Family (Qwen2.5)* | | | | | | | | | |
| Qwen2.5 (0.5B) | $0.40_{\pm0.80}$ | $69.80_{\pm8.45}$ | 2782.97 | $2.60_{\pm1.50}$ | 100.00 | 1547.82 | $0.50_{\pm0.50}$ | $89.85_{\pm2.35}$ | 2023.24 |
| Qwen2.5 (1.5B) | $2.20_{\pm3.92}$ | $93.00_{\pm4.38}$ | 1857.13 | $2.80_{\pm4.66}$ | 100.00 | 1499.47 | $11.00_{\pm2.00}$ | $89.72_{\pm1.18}$ | 1965.13 |
| Qwen2.5 (3B) | $3.00_{\pm4.56}$ | $92.40_{\pm6.47}$ | 2712.09 | $8.20_{\pm6.31}$ | $99.20_{\pm0.75}$ | 1547.12 | $21.50_{\pm11.50}$ | $98.22_{\pm0.03}$ | 807.39 |
| Qwen2.5 (7B) | $5.60_{\pm8.26}$ | $95.40_{\pm1.36}$ | 1926.47 | $22.40_{\pm12.22}$ | 100.00 | 1602.21 | $39.00_{\pm23.00}$ | $99.18_{\pm0.72}$ | 658.14 |
| Qwen2.5 (14B) | $8.40_{\pm12.45}$ | $95.20_{\pm2.79}$ | 2435.72 | $34.20_{\pm11.27}$ | 100.00 | 1095.80 | $50.50_{\pm36.50}$ | 99.90 | 600.59 |
| Qwen2.5 (32B) | $8.20_{\pm10.76}$ | $93.80_{\pm3.66}$ | 1950.33 | $40.60_{\pm16.60}$ | $99.80_{\pm0.40}$ | 1138.83 | $53.00_{\pm37.00}$ | $99.90_{\pm0.10}$ | 550.24 |
| Qwen2.5 (72B) | $10.60_{\pm14.73}$ | $97.20_{\pm0.75}$ | 2556.63 | $38.60_{\pm12.82}$ | 100.00 | 1336.84 | $62.50_{\pm35.50}$ | 100.00 | 687.72 |
| Qwen2.5 (1.5B-m) | $8.00_{\pm13.58}$ | $96.20_{\pm1.17}$ | 1523.49 | $2.20_{\pm2.64}$ | 100.00 | 1476.54 | $12.50_{\pm0.50}$ | $91.08_{\pm0.17}$ | 1305.73 |
| Qwen2.5 (7B-m) | $20.80_{\pm24.73}$ | $99.40_{\pm0.49}$ | 1712.29 | $4.20_{\pm4.96}$ | 100.00 | 1552.59 | $24.00_{\pm15.00}$ | $94.55_{\pm0.60}$ | 1705.94 |
| Qwen2.5 (72B-m) | $7.40_{\pm8.16}$ | $95.80_{\pm3.43}$ | 1567.37 | $7.00_{\pm3.74}$ | 100.00 | 1458.79 | $45.50_{\pm36.50}$ | $97.12_{\pm0.77}$ | 879.80 |
| *Gemma Family* | | | | | | | | | |
| Gemma (1B) | $0.40_{\pm0.80}$ | $89.80_{\pm3.97}$ | 2363.95 | $3.40_{\pm6.31}$ | 100.00 | 1002.60 | $0.50_{\pm0.50}$ | $90.48_{\pm1.17}$ | 466.94 |
| Gemma (4B) | $26.00_{\pm29.39}$ | $98.00_{\pm4.00}$ | 2571.10 | $20.00_{\pm20.98}$ | 100.00 | 1571.14 | $15.00_{\pm15.00}$ | $96.00_{\pm2.00}$ | 744.05 |
| Gemma (12B) | $20.00_{\pm26.08}$ | $72.00_{\pm19.39}$ | 3267.38 | $38.00_{\pm14.70}$ | 100.00 | 1206.82 | $65.00_{\pm25.00}$ | 99.50 | 664.05 |
| Gemma (27B) | $42.20_{\pm23.57}$ | $92.00_{\pm2.76}$ | 2470.36 | $52.80_{\pm14.30}$ | 100.00 | 1124.68 | $56.00_{\pm41.00}$ | $99.98_{\pm0.02}$ | 893.26 |
| *Phi Family* | | | | | | | | | |
| Phi4 (14B) | $10.40_{\pm14.64}$ | $87.60_{\pm4.59}$ | 2478.59 | $45.40_{\pm14.68}$ | 100.00 | 993.49 | $79.50_{\pm18.50}$ | $95.48_{\pm1.32}$ | 726.86 |
| Phi4-reasoning+ (14B) | $54.00_{\pm26.50}$ | $79.80_{\pm7.22}$ | 7381.98 | $32.00_{\pm35.64}$ | 100.00 | 6994.60 | $56.00_{\pm41.00}$ | $85.80_{\pm11.00}$ | 6678.41 |
| Phi4-reasoning (14B) | $55.80_{\pm30.75}$ | $83.60_{\pm10.05}$ | 6989.32 | $40.60_{\pm32.54}$ | 100.00 | 7150.18 | $56.00_{\pm18.00}$ | $92.30_{\pm3.65}$ | 6542.36 |
| Phi4-mini-reasoning (3.8B) | $42.20_{\pm33.46}$ | $99.20_{\pm1.17}$ | 5870.72 | $15.60_{\pm14.09}$ | 100.00 | 6713.40 | $52.00_{\pm48.00}$ | $98.18_{\pm1.42}$ | 4220.05 |
| Phi3-mini (3.8B) | 0.00 | $90.80_{\pm3.87}$ | 1037.66 | $13.00_{\pm6.03}$ | $98.80_{\pm1.94}$ | 801.48 | $54.50_{\pm3.50}$ | $90.92_{\pm4.43}$ | 820.04 |
| Phi3-med (14B-4k) | $0.40_{\pm0.80}$ | $96.60_{\pm1.85}$ | 839.70 | $15.20_{\pm6.73}$ | $99.40_{\pm0.80}$ | 789.56 | $47.50_{\pm29.50}$ | $99.38_{\pm0.22}$ | 582.48 |
| Phi3-med (14B-128k) | $0.20_{\pm0.40}$ | $91.40_{\pm2.24}$ | 947.50 | $17.40_{\pm9.31}$ | $99.20_{\pm0.75}$ | 877.94 | $56.50_{\pm28.50}$ | $98.92_{\pm1.07}$ | 575.08 |
| *Llama Family* | | | | | | | | | |
| Llama-3.2 (1B) | 0.00 | $78.40_{\pm9.50}$ | 3200.42 | $3.00_{\pm1.79}$ | 100.00 | 1763.54 | $4.00_{\pm3.00}$ | $92.35_{\pm1.05}$ | 1354.80 |
| Llama-3.2 (3B) | 0.00 | $94.00_{\pm3.35}$ | 2696.67 | $2.00_{\pm2.61}$ | $99.60_{\pm0.49}$ | 1297.29 | $24.00_{\pm10.00}$ | $97.82_{\pm0.13}$ | 972.02 |
| Llama-3.1 (8B) | 0.00 | $91.20_{\pm5.84}$ | 3474.99 | $13.60_{\pm6.44}$ | $99.80_{\pm0.40}$ | 1408.37 | $28.00_{\pm24.00}$ | $92.10_{\pm3.65}$ | 2831.85 |
| Llama-3.1 (70B) | $0.40_{\pm0.49}$ | $87.40_{\pm6.59}$ | 4310.14 | $25.80_{\pm6.49}$ | 100.00 | 1003.33 | $58.00_{\pm38.00}$ | $99.62_{\pm0.37}$ | 743.58 |
| Llama-3.3 (70B) | $2.00_{\pm4.00}$ | $97.20_{\pm1.17}$ | 1142.65 | $27.40_{\pm10.98}$ | 100.00 | 1105.34 | $60.00_{\pm39.00}$ | 100.00 | 733.30 |
| Llama4-scout | $0.40_{\pm0.49}$ | $6.20_{\pm3.97}$ | 191.85 | $41.00_{\pm8.49}$ | 100.00 | 1162.59 | $72.50_{\pm24.50}$ | 100.00 | 829.92 |
| *Mistral Family* | | | | | | | | | |
| Mistral (7B) | 0.00 | $87.80_{\pm4.40}$ | 2075.13 | $1.40_{\pm0.80}$ | $97.60_{\pm1.62}$ | 896.68 | $14.50_{\pm6.50}$ | $98.92_{\pm0.87}$ | 504.86 |
| Ministral (8B) | $1.40_{\pm1.85}$ | $94.40_{\pm2.58}$ | 1936.11 | $13.20_{\pm2.93}$ | 100.00 | 1117.54 | $36.50_{\pm17.50}$ | $99.82_{\pm0.07}$ | 547.00 |
| Mistral-nemo (12B) | 0.00 | $86.20_{\pm5.49}$ | 1961.18 | $6.60_{\pm1.02}$ | $98.80_{\pm0.75}$ | 639.16 | $35.00_{\pm25.00}$ | $99.72_{\pm0.27}$ | 398.98 |
| Mixtral-8x7b | 0.00 | $83.40_{\pm5.71}$ | 903.05 | $6.40_{\pm4.50}$ | $99.60_{\pm0.80}$ | 632.40 | $23.00_{\pm13.00}$ | $80.48_{\pm11.83}$ | 397.41 |
| Mixtral-8x22b | $0.80_{\pm1.17}$ | $87.60_{\pm5.00}$ | 1472.45 | $11.00_{\pm3.16}$ | $99.20_{\pm1.17}$ | 711.37 | $45.00_{\pm30.00}$ | $99.18_{\pm0.62}$ | 753.80 |
| *Others* | | | | | | | | | |
| Smollm3 (3B) | $39.40_{\pm34.23}$ | $93.60_{\pm4.03}$ | 5763.05 | $2.40_{\pm3.01}$ | 100.00 | 7099.07 | $51.50_{\pm45.50}$ | $96.58_{\pm2.72}$ | 3828.02 |
| Smollm2 (1.7B) | 0.00 | $71.80_{\pm8.28}$ | 1786.45 | $5.20_{\pm4.17}$ | 100.00 | 15.26 | $51.00_{\pm49.00}$ | $82.80_{\pm2.80}$ | 156.43 |
| GPT-OSS (20B) | $51.20_{\pm22.57}$ | $75.20_{\pm6.62}$ | 3803.05 | $58.60_{\pm34.12}$ | $99.80_{\pm0.40}$ | 4794.62 | $84.00_{\pm14.00}$ | $98.52_{\pm0.73}$ | 1292.30 |
| GPT-OSS (120B) | $86.60_{\pm8.52}$ | $94.80_{\pm2.64}$ | 2583.77 | $83.40_{\pm21.85}$ | 100.00 | 3912.25 | $90.00_{\pm10.00}$ | $96.42_{\pm0.53}$ | 531.68 |
| *OpenAI Family (Proprietary)* | | | | | | | | | |
| GPT5 | $98.00_{\pm4.00}$ | 100.00 | 1770.24 | 100.00 | 100.00 | 36.92 | $75.00_{\pm25.00}$ | $99.75_{\pm0.25}$ | 521.69 |
| GPT5-mini | 100.00 | 100.00 | 1802.64 | 100.00 | 100.00 | 35.57 | $55.00_{\pm45.00}$ | $97.00_{\pm3.00}$ | 1115.52 |
| GPT5-nano | 100.00 | 100.00 | 932.52 | 100.00 | 100.00 | 37.28 | $65.00_{\pm35.00}$ | $87.50_{\pm3.50}$ | 311.87 |
| GPT4.1 | $58.00_{\pm31.24}$ | $86.00_{\pm10.20}$ | 12393.80 | $86.00_{\pm14.97}$ | 100.00 | 7336.68 | $80.00_{\pm20.00}$ | 100.00 | 793.05 |
| GPT4.1-mini | $64.00_{\pm32.00}$ | $94.00_{\pm4.90}$ | 11230.96 | $94.00_{\pm8.00}$ | 100.00 | 2596.06 | $55.00_{\pm45.00}$ | 99.50 | 586.85 |
| GPT4.1-nano | $46.00_{\pm24.17}$ | $94.00_{\pm8.00}$ | 4076.72 | $50.00_{\pm18.97}$ | 100.00 | 1587.56 | $60.00_{\pm40.00}$ | 100.00 | 805.15 |
| GPT4o | $4.00_{\pm8.00}$ | 100.00 | 897.84 | $36.00_{\pm12.00}$ | 100.00 | 687.80 | $65.00_{\pm25.00}$ | $99.50_{\pm0.50}$ | 503.60 |
| GPT4o-mini | $8.00_{\pm11.66}$ | 100.00 | 1274.96 | $30.00_{\pm16.73}$ | 100.00 | 928.02 | $70.00_{\pm20.00}$ | 100.00 | 605.55 |
| o4 mini | $96.00_{\pm4.90}$ | $98.00_{\pm4.00}$ | 882.37 | 100.00 | 100.00 | 31.97 | $70.00_{\pm30.00}$ | $90.25_{\pm7.75}$ | 491.28 |
| o3 | 100.00 | 100.00 | 1676.59 | 100.00 | 100.00 | 62.06 | $85.00_{\pm15.00}$ | $88.25_{\pm8.25}$ | 676.08 |
| o3 mini | $92.00_{\pm9.80}$ | 100.00 | 4044.00 | $96.00_{\pm4.90}$ | 100.00 | 1929.65 | $80.00_{\pm20.00}$ | $99.00_{\pm1.00}$ | 691.92 |
| *Gemini Family (Proprietary)* | | | | | | | | | |
| Gemini-2.5-pro | 100.00 | 100.00 | 1525.84 | $98.00_{\pm4.00}$ | 100.00 | 768.92 | $75.00_{\pm25.00}$ | 100.00 | 714.10 |
| Gemini-2.5-flash | $74.00_{\pm28.71}$ | $84.00_{\pm18.55}$ | 1452.50 | $96.00_{\pm8.00}$ | 100.00 | 1756.38 | $60.00_{\pm30.00}$ | $90.00_{\pm10.00}$ | 1059.70 |
| Gemini-2.5-flash-lite | $68.00_{\pm29.26}$ | $94.00_{\pm8.00}$ | 9432.86 | $88.00_{\pm11.66}$ | 100.00 | 3445.30 | $55.00_{\pm45.00}$ | $99.00_{\pm1.00}$ | 8559.65 |
| Gemini-2.0-flash | $26.00_{\pm24.17}$ | $74.00_{\pm16.25}$ | 2656.98 | $22.00_{\pm4.00}$ | 100.00 | 1832.40 | $55.00_{\pm45.00}$ | 100.00 | 614.50 |
| Gemini-2.0-flash-lite | $20.00_{\pm12.65}$ | $78.00_{\pm13.27}$ | 3299.12 | $54.00_{\pm24.17}$ | 100.00 | 1650.30 | $90.00_{\pm10.00}$ | $98.50_{\pm1.50}$ | 1022.85 |

Table 25: **Hard Suite Results - Table 3:** Performance of regular models on Modular Systems Solver, Constraint Optimization, and Logic Grid Puzzles tasks. Each task reports Accuracy (Acc), Instruction-following (Inst) and average output Tokens with mean and standard deviation values. Results show average performance across different complexity variants within each task category.

## L   PROMPTS USED FOR ALL HARD SUITE TASKS

This section describes the prompts developed for each of the hard suite tasks. Each prompt begins with a brief description of the task followed by the inputs. The constraints are listed if applicable, which is then followed by a detailed description of the required answer format.

---

**Prompt Template for Boolean SAT Task**

CHALLENGING BOOLEAN SATISFIABILITY (SAT) PROBLEM:

Find a truth assignment for variables $x_1, x_2, ..., x_{\{num\_vars\}}$ that satisfies this complex formula:

{formula}

LOGICAL OPERATORS:
- ∨ means OR (disjunction)
- ∧ means AND (conjunction)
- ¬ means NOT (negation)

SOLUTION FORMAT REQUIREMENTS:
Your answer must assign True/False to ALL {num_vars} variables.

REQUIRED FORMAT (STRICTLY FOLLOW THIS):
<answer>
{{1:  True, 2:  False, 3:  True, ..., {num_vars}:  True}}
</answer>

IMPORTANT FORMATTING NOTES:
- Use EXACTLY the format shown above inside <answer> tags.
- Use numbers (1, 2, 3...)  NOT variable names ($x_1, x_2, x_3$).
- Use True/False (not true/false, 1/0, T/F).
- Include ALL {num_vars} variables in your answer.
- Do NOT use code blocks, markdown, or other formatting.

---

**Prompt Template for Constraint Optimization Task**

ADVANCED MULTI-CONSTRAINT RESOURCE ALLOCATION PROBLEM:

Your task:  Select an optimal subset of projects to maximize total profit while satisfying all resource capacity constraints and additional requirements.

AVAILABLE RESOURCES:
{resources_str}

AVAILABLE PROJECTS:
{projects_str}

OPTIMIZATION OBJECTIVE:
Maximize total profit from selected projects.

CONSTRAINTS:
1.  Resource capacity:  Cannot exceed available capacity for any resource.
2.  Dependencies:  If a project has dependencies, all dependency projects must also be selected.

---

```
ANSWER FORMAT:
Provide your answer as a list of project IDs (numbers) in square
brackets.

<answer>
[project_id_1, project_id_2, project_id_3, ...]
</answer>
```

**Prompt Template for Cryptarithmetic Task**

```
Solve this challenging cryptarithmetic puzzle:

{word1} {op_symbol} {word2} = {result_word}

Rules:
- Each letter represents a unique digit from 0 to 9.
- No two letters can have the same digit.
- Leading letters cannot be zero.

SOLUTION FORMAT REQUIREMENTS (STRICTLY FOLLOW THIS):
Your answer must assign digits to ALL letters.

REQUIRED FORMAT:
<answer>
{{"A":  1, "B":  2, "C":  3, ...}}
</answer>

IMPORTANT FORMATTING NOTES:
- Use EXACTLY the format shown above inside <answer> tags.
- Use double quotes around letter names:  "A", "B", "C", etc.
- Use actual digits:  1, 2, 3, etc.  (not "1", "2", "3").
- Include ALL letters in your answer.
- Do NOT use code blocks, markdown, or other formatting.
```

**Prompt Template for Graph Coloring Task**

```
CHALLENGING GRAPH COLORING PROBLEM:

Your task:  Color the vertices of a complex graph using exactly
{chromatic_number} colors such that no two adjacent (connected)
vertices have the same color.

GRAPH SPECIFICATION:
- {n} vertices numbered from 0 to n  —  1 (expressed as {n-1} if you
want a literal placeholder).
- {len(edges)} edges:  {edges_str}.
- Minimum colors needed:  {chromatic_number}.

COLORING RULES:
- Use exactly {chromatic_number} different colors.
- NO two vertices connected by an edge can have the same color.
- EVERY vertex must be assigned exactly one color.

FINAL ANSWER FORMAT:
End your response with the complete solution dictionary in this
format:
```

```
<answer>
{{0:  "Red", 1:  "Blue", 2:  "Green", 3:  "Yellow", ..., {n-1}:
"ColorX"}}
</answer>
```

## Prompt Template for Logic Grid Puzzles Task

```
You are solving a logic grid puzzle.  This requires systematic
logical reasoning and constraint satisfaction.

PUZZLE SETUP:
Grid Size:  {size} × {size}
Categories:  {categories}

CLUES:
[Constraint descriptions]

TASK: Use logical deduction to determine which person has which
attributes in each category.

CRITICAL INSTRUCTIONS:
Present your final answer in ONE of these EXACT formats:

FORMAT 1 (PREFERRED - Use this exact structure):
\boxed{{
{person}:  [{category}:  [value], ...]
...}}

IMPORTANT FORMATTING NOTES:
- Replace [value] with the actual specific attribute.
- Use the exact person names from the entities list.
- Include ALL people and ALL their attributes.
- Make sure no two people share the same attribute value.
```

## Prompt Template for Matrix Chain Multiplication Task

```
CHALLENGING MATRIX CHAIN MULTIPLICATION OPTIMIZATION PROBLEM:

Your task:  Find the minimum number of scalar multiplications needed
to compute the matrix product using optimal parenthesization.

PROBLEM SPECIFICATION:
- Number of matrices:  {matrix_count}.
- Matrix dimensions:  {matrix_descriptions}.

MATRIX MULTIPLICATION COST MODEL:
- To multiply two matrices of dimensions (p × q) and (q × r):  cost =
p × q × r scalar multiplications.

DYNAMIC PROGRAMMING APPROACH:
This is a classic optimization problem that requires dynamic
programming.

CRITICAL: ANSWER FORMAT REQUIREMENTS
To ensure your answer is correctly parsed, follow these EXACT
formatting requirements:
```

```
1.  ALWAYS provide your final answer in this precise format:
<answer>
[NUMERIC_VALUE_ONLY]
</answer>

2.  REPLACE [NUMERIC_VALUE_ONLY] with ONLY the numeric value.
3.  Example:  If the answer is 1204480, write exactly:
<answer>1204480</answer>.
```

### Prompt Template for Modular Systems Solver Task

ADVANCED MODULAR SYSTEMS PROBLEM:

Your task:  Solve the following system of modular arithmetic
equations and find the value of $x$ that satisfies all conditions.

SYSTEM OF EQUATIONS:
[modular equations]

ADDITIONAL CONSTRAINTS:
[constraints]

MATHEMATICAL BACKGROUND:
– Modular arithmetic:  $x \equiv a \pmod{m}$ means $x$ leaves remainder $a$ when
divided by $m$.
– Use the Chinese Remainder Theorem for systems with coprime moduli.
– For non–coprime moduli, use the general solution method.

ANSWER FORMAT REQUIREMENTS:
Your response must end with your final numerical answer in one of
these formats:

Option 1 (Preferred):
<answer>
[Your integer answer here]
</answer>

Option 2 (Alternative):
Therefore, $x =$ [Your integer answer here].

DO NOT end with intermediate calculations or parametric expressions.

### Prompt Template for N-Queens Task

Solve the {n}-Queens problem:  Place {n} queens on an {n} × {n}
chessboard so that no two queens attack each other.

RULES:
– Queens attack horizontally, vertically, and diagonally.
– No two queens can be in the same row, column, or diagonal.
– You must place exactly {n} queens on the board (one queen per
row).

CRITICAL: Your answer must contain exactly {n} numbers, each
representing the column position (0 to $n - 1$) of the queen in that
row.

```
SOLUTION FORMAT:
Provide your solution as a list of {n} column numbers.  The first
number is the column for row 0, the second for row 1, etc.

REQUIRED FORMAT - Use one of these:
- [col0, col1, col2, ...]  ← PREFERRED
- Answer:  [1, 3, 0, 2]
- Solution:  [1, 3, 0, 2]

<answer>
[your {n} column numbers here]
</answer>
```

**Prompt Template for Sudoku Task**

```
Base Rules for {size} × {size} Sudoku:
- Fill the grid with numbers 1 to {size}.
- Each row must contain all numbers 1 to {size} exactly once.
- Each column must contain all numbers 1 to {size} exactly once.
- Each {box_dimensions} box must contain all numbers 1 to {size}
exactly once.

PUZZLE GRID (0 represents empty cells):
{grid_representation}

SOLVING REQUIREMENTS:
- Provide the complete solved grid.
- Every cell must be filled with a number from 1 to {size}.
- Solution must satisfy all rules including additional constraints.

ANSWER FORMAT:
Provide your solution as a complete grid using one of these formats:

1.  Grid format:
1 2 3 4
3 4 1 2
2 3 4 1
4 1 2 3

2.  List format:
[[1,2,3,4],[3,4,1,2],[2,3,4,1],[4,1,2,3]]
```

**Prompt Template for Tower of Hanoi Task**

```
Solve this Tower of Hanoi puzzle with {num_disks} disks.

RULES:
1.  Only one disk can be moved at a time.
2.  A larger disk cannot be placed on top of a smaller disk.
3.  Only the topmost disk on any peg can be moved.

INITIAL STATE:
Peg A: {initial['A']} (disk 1 is smallest, disk {num_disks} is
largest)
Peg B: {initial['B']}
Peg C: {initial['C']}

GOAL: Move all disks from Peg A to Peg C.
```

```
 Provide the complete sequence of moves.  You can use any of these
 formats:
- "Move disk X from peg Y to peg Z"
- "Move X from Y to Z"
- "X: Y -> Z"
- "Transfer disk X from Y to Z"

<answer>
Move disk 1 from peg A to peg C
Move disk 2 from peg A to peg B
...
</answer>
```

## M  QUANTIZED MODELS COMPLETE RESULTS

This section details results for all easy, medium and hard suite tasks for the quantized Qwen models. There are 21 quantized models from the Qwen2.5 family and 17 quantized models from the Qwen3 family.

### M.1  EASY SUITE

This section shows results from the quantized Qwen models for the easy suite tasks: absolute difference, comparison, division, even count, find maximum, find minimum, mean, median, mode, multiplication, odd count, sorting, subtraction, sum, second maximum, range, negative number count, unique elements count, maximum difference between adjacent elements, count elements greater than previous, sum of indices of maximum element, palindromic number count, longest increasing subsequence length task, sum of digits, count perfect squares, alternating sum , count multiples of K, local maxima count.

| Model (Param) | Quant | Acc (%) | Inst (%) | Tokens |
|---|---|---|---|---|
| *Qwen Family (Qwen3)* | | | | |
| Qwen3 (0.6B) | FP8 | 55.89 | 86.51 | 2767.9 |
| Qwen3 (0.6B) | GPTQ-8-Bit | 56.40 | 84.33 | 2922.35 |
| Qwen3 (1.7B) | FP8 | 73.52 | 86.65 | 2918.16 |
| Qwen3 (1.7B) | GPTQ-4-Bit | 73.07 | 86.30 | 2910.69 |
| Qwen3 (4B) | AWQ | 81.14 | 89.73 | 2891.68 |
| Qwen3 (4B) | FP8 | 84.36 | 91.92 | 2889.65 |
| Qwen3 (8B) | AWQ | 83.01 | 90.52 | 3021.3 |
| Qwen3 (8B) | FP8 | 83.10 | 90.08 | 2905.18 |
| Qwen3 (14B) | AWQ | 87.98 | 94.17 | 2432.73 |
| Qwen3 (14B) | FP8 | 87.55 | 93.75 | 2415.34 |
| Qwen3 (32B) | AWQ | 87.02 | 93.34 | 2477.95 |
| Qwen3 (32B) | FP8 | 88.11 | 93.86 | 2544.64 |
| Qwen3 (30B-MOE) | FP8 | 88.83 | 94.43 | 2417.23 |
| Qwen3 (30B-MOE-t) | FP8 | 90.30 | 96.67 | 2051.9 |
| Qwen3 (30B-MOE-i) | FP8 | 92.17 | 99.66 | 636.55 |
| Qwen3 (4B-t) | FP8 | 85.42 | 94.61 | 2603.79 |
| Qwen3 (4B-i) | FP8 | 86.89 | 99.26 | 817.99 |

Qwen Family (Qwen3)

| Model (Param) | Quant | Acc (%) | Inst (%) | Tokens |
|---|---|---|---|---|
| *Qwen Family (Qwen2.5)* | | | | |
| Qwen2.5 (0.5B) | AWQ | 19.31 | 83.06 | 1358.77 |
| Qwen2.5 (0.5B) | GPTQ-4-Bit | 12.77 | 77.70 | 478.00 |
| Qwen2.5 (0.5B) | GPTQ-8-Bit | 21.29 | 76.79 | 431.50 |
| Qwen2.5 (1.5B) | AWQ | 38.77 | 77.00 | 732.95 |
| Qwen2.5 (1.5B) | GPTQ-4-Bit | 39.42 | 82.97 | 292.10 |
| Qwen2.5 (1.5B) | GPTQ-8-Bit | 43.67 | 86.64 | 264.30 |
| Qwen2.5 (3B) | AWQ | 45.83 | 85.93 | 908.07 |
| Qwen2.5 (3B) | GPTQ-4-Bit | 41.94 | 90.97 | 301.00 |
| Qwen2.5 (3B) | GPTQ-8-Bit | 48.65 | 91.99 | 341.90 |
| Qwen2.5 (7B) | AWQ | 66.52 | 97.94 | 471.66 |
| Qwen2.5 (7B) | GPTQ-4-Bit | 58.03 | 96.00 | 291.90 |
| Qwen2.5 (7B) | GPTQ-8-Bit | 60.61 | 96.40 | 287.50 |
| Qwen2.5 (14B) | AWQ | 67.95 | 98.71 | 250.95 |
| Qwen2.5 (14B) | GPTQ-4-Bit | 60.94 | 96.69 | 240.90 |
| Qwen2.5 (14B) | GPTQ-8-Bit | 63.86 | 97.89 | 261.20 |
| Qwen2.5 (32B) | AWQ | 77.84 | 99.72 | 254.54 |
| Qwen2.5 (32B) | GPTQ-4-Bit | 72.67 | 99.37 | 260.50 |
| Qwen2.5 (32B) | GPTQ-8-Bit | 73.08 | 99.20 | 261.90 |
| Qwen2.5 (72B) | AWQ | 78.88 | 99.63 | 336.67 |
| Qwen2.5 (72B) | GPTQ-4-Bit | 72.74 | 94.85 | 358.30 |
| Qwen2.5 (72B) | GPTQ-8-Bit | 74.05 | 96.28 | 347.10 |

Qwen Family (Qwen2.5)

Table 26: **Easy Suite Results (Quantized):** Performance of quantized models on the Easy Suite benchmark tasks. Each model reports Accuracy (Acc), Instruction-following (Inst) and average output Tokens. The Quant column indicates the quantization method used.

### M.2  MEDIUM SUITE

This section shows results from the quantized Qwen models for the medium suite tasks: algebraic sequence, complex pattern, fibonacci sequence, geometric sequence and prime sequence. These results are listed in tables 27 and 28.

| Model (Param) | Quant | algebraic_sequence | | | complex_pattern | | | fibonacci_sequence | | |
|---|---|---|---|---|---|---|---|---|---|---|
| | | Acc (Avg %) | Inst (Avg %) | Tokens (Avg) | Acc (Avg %) | Inst (Avg %) | Tokens (Avg) | Acc (Avg %) | Inst (Avg %) | Tokens (Avg) |
| *Qwen Family (Qwen3)* | | | | | | | | | | |
| Qwen3 (0.6B) | FP8 | $28.40_{\pm4.72}$ | 100.00 | 21274.28 | $28.53_{\pm5.89}$ | 100.00 | 16403.45 | $17.20_{\pm5.11}$ | 100.00 | 5736.11 |
| Qwen3 (0.6B) | GPTQ-8-Bit | $27.80_{\pm3.82}$ | 100.00 | 6625.29 | $22.87_{\pm7.28}$ | 100.00 | 5987.01 | $16.20_{\pm5.08}$ | 100.00 | 5815.69 |
| Qwen3 (1.7B) | FP8 | $40.00_{\pm2.53}$ | 100.00 | 16699.83 | $55.77_{\pm6.49}$ | 100.00 | 11925.78 | $50.80_{\pm6.79}$ | 100.00 | 5771.03 |
| Qwen3 (1.7B) | GPTQ-4-Bit | $31.60_{\pm3.01}$ | 100.00 | 6875.73 | $43.63_{\pm10.72}$ | 100.00 | 6367.90 | $50.60_{\pm10.13}$ | 100.00 | 5753.83 |
| Qwen3 (4B) | AWQ | $24.40_{\pm3.50}$ | 100.00 | 7303.17 | $52.01_{\pm6.49}$ | 100.00 | 5986.86 | $75.80_{\pm11.20}$ | 100.00 | 5427.01 |
| Qwen3 (4B) | FP8 | $37.60_{\pm3.44}$ | 100.00 | 19555.70 | $73.37_{\pm11.35}$ | 100.00 | 10334.75 | $84.80_{\pm9.66}$ | 100.00 | 4996.51 |
| Qwen3 (8B) | AWQ | $37.00_{\pm7.62}$ | 100.00 | 17602.35 | $69.74_{\pm7.78}$ | 100.00 | 10958.16 | $80.80_{\pm8.35}$ | 100.00 | 5579.07 |
| Qwen3 (8B) | FP8 | $39.60_{\pm4.72}$ | 100.00 | 16906.20 | $69.95_{\pm9.14}$ | 100.00 | 10851.68 | $88.00_{\pm8.00}$ | 100.00 | 5120.27 |
| Qwen3 (14B) | AWQ | $47.40_{\pm6.62}$ | 100.00 | 12727.66 | $82.75_{\pm13.35}$ | 100.00 | 7333.95 | $93.40_{\pm5.68}$ | 100.00 | 4437.96 |
| Qwen3 (14B) | FP8 | $45.80_{\pm7.30}$ | 100.00 | 12493.45 | $83.38_{\pm12.83}$ | 100.00 | 7452.62 | $96.60_{\pm1.85}$ | 100.00 | 4213.44 |
| Qwen3 (32B) | AWQ | $39.80_{\pm3.97}$ | 100.00 | 6404.83 | $73.01_{\pm9.72}$ | 100.00 | 4735.68 | $96.40_{\pm2.94}$ | 100.00 | 3434.89 |
| Qwen3 (32B) | FP8 | $41.00_{\pm5.48}$ | 100.00 | 6412.39 | $73.22_{\pm9.12}$ | 100.00 | 4943.28 | $96.00_{\pm3.16}$ | 100.00 | 3676.62 |
| Qwen3 (30B-MOE) | FP8 | $36.20_{\pm6.71}$ | 100.00 | 6780.13 | $59.93_{\pm7.38}$ | 100.00 | 5501.37 | $97.40_{\pm1.36}$ | 100.00 | 3799.06 |
| Qwen3 (30B-MOE-t) | FP8 | $40.80_{\pm1.17}$ | 100.00 | 5944.68 | $72.84_{\pm9.24}$ | 100.00 | 4459.33 | $98.60_{\pm1.85}$ | 100.00 | 3273.00 |
| Qwen3 (30B-MOE-i) | FP8 | $46.20_{\pm3.97}$ | 100.00 | 5559.90 | $79.76_{\pm10.55}$ | 100.00 | 3442.94 | $98.00_{\pm1.41}$ | 100.00 | 1873.05 |
| Qwen3 (4B-t) | FP8 | $40.00_{\pm3.29}$ | 100.00 | 6391.36 | $63.82_{\pm7.99}$ | 100.00 | 5619.65 | $89.60_{\pm10.69}$ | 100.00 | 4623.29 |
| Qwen3 (4B-i) | FP8 | $42.60_{\pm3.72}$ | 100.00 | 5344.65 | $72.59_{\pm8.79}$ | 100.00 | 3588.75 | $85.40_{\pm6.28}$ | 100.00 | 2839.75 |
| *Qwen Family (Qwen2.5)* | | | | | | | | | | |
| Qwen2.5 (0.5B) | AWQ | $11.20_{\pm5.31}$ | 100.00 | 2872.68 | $7.91_{\pm3.06}$ | 100.00 | 2470.38 | $3.80_{\pm1.47}$ | 100.00 | 1948.83 |
| Qwen2.5 (0.5B) | GPTQ-4-Bit | $11.80_{\pm0.75}$ | 100.00 | 3245.19 | $4.72_{\pm0.65}$ | 100.00 | 2801.58 | $5.60_{\pm2.06}$ | 100.00 | 2574.42 |
| Qwen2.5 (0.5B) | GPTQ-8-Bit | $8.20_{\pm2.23}$ | 100.00 | 3205.43 | $5.42_{\pm3.12}$ | 100.00 | 3363.70 | $4.40_{\pm1.02}$ | 100.00 | 2314.57 |
| Qwen2.5 (1.5B) | AWQ | $17.60_{\pm3.77}$ | 100.00 | 3309.25 | $9.97_{\pm4.51}$ | 100.00 | 2868.22 | $7.40_{\pm1.02}$ | 100.00 | 2565.16 |
| Qwen2.5 (1.5B) | GPTQ-4-Bit | $29.40_{\pm1.96}$ | 100.00 | 2495.16 | $9.78_{\pm2.76}$ | 100.00 | 2842.20 | $8.20_{\pm1.17}$ | 100.00 | 1823.57 |
| Qwen2.5 (1.5B) | GPTQ-8-Bit | $20.40_{\pm7.47}$ | 100.00 | 2364.81 | $12.15_{\pm3.10}$ | 100.00 | 2588.94 | $10.80_{\pm2.32}$ | 100.00 | 2100.21 |
| Qwen2.5 (3B) | AWQ | $32.60_{\pm1.85}$ | 100.00 | 1880.13 | $18.89_{\pm2.45}$ | 100.00 | 1642.93 | $13.60_{\pm3.01}$ | 100.00 | 1888.29 |
| Qwen2.5 (3B) | GPTQ-4-Bit | $33.60_{\pm8.69}$ | 100.00 | 1600.54 | $19.38_{\pm6.08}$ | 100.00 | 1738.58 | $12.60_{\pm1.36}$ | 100.00 | 2328.10 |
| Qwen2.5 (3B) | GPTQ-8-Bit | $24.60_{\pm5.57}$ | 100.00 | 1405.23 | $24.03_{\pm6.95}$ | 100.00 | 1302.74 | $13.00_{\pm2.83}$ | 100.00 | 1761.07 |
| Qwen2.5 (7B) | AWQ | $36.80_{\pm5.04}$ | 100.00 | 1350.69 | $41.15_{\pm5.93}$ | 100.00 | 783.19 | $20.20_{\pm5.11}$ | 100.00 | 584.48 |
| Qwen2.5 (7B) | GPTQ-4-Bit | $35.80_{\pm1.17}$ | 100.00 | 923.49 | $41.74_{\pm5.14}$ | 100.00 | 765.73 | $20.20_{\pm2.93}$ | 100.00 | 738.95 |
| Qwen2.5 (7B) | GPTQ-8-Bit | $38.40_{\pm2.73}$ | 100.00 | 1041.76 | $41.82_{\pm5.01}$ | 100.00 | 852.66 | $19.20_{\pm3.43}$ | 100.00 | 626.42 |
| Qwen2.5 (14B) | AWQ | $41.60_{\pm2.33}$ | $92.40_{\pm4.63}$ | 576.95 | $49.40_{\pm6.66}$ | 100.00 | 552.46 | $20.00_{\pm4.82}$ | 100.00 | 531.80 |
| Qwen2.5 (14B) | GPTQ-4-Bit | $40.60_{\pm2.15}$ | $94.20_{\pm5.38}$ | 736.45 | $50.15_{\pm4.30}$ | 100.00 | 578.50 | $26.40_{\pm4.50}$ | 100.00 | 600.36 |
| Qwen2.5 (14B) | GPTQ-8-Bit | $41.00_{\pm2.76}$ | $99.00_{\pm0.89}$ | 965.69 | $55.44_{\pm5.93}$ | 100.00 | 533.11 | $25.20_{\pm3.31}$ | 100.00 | 564.42 |
| Qwen2.5 (32B) | AWQ | $44.00_{\pm2.76}$ | $95.20_{\pm3.66}$ | 1147.58 | $53.97_{\pm8.95}$ | 100.00 | 507.63 | $38.60_{\pm12.16}$ | 100.00 | 543.80 |
| Qwen2.5 (32B) | GPTQ-4-Bit | $43.00_{\pm4.43}$ | 100.00 | 1210.72 | $55.84_{\pm7.89}$ | 100.00 | 530.96 | $35.80_{\pm14.27}$ | 100.00 | 599.57 |
| Qwen2.5 (32B) | GPTQ-8-Bit | $42.20_{\pm4.21}$ | $95.80_{\pm2.93}$ | 1160.28 | $51.48_{\pm8.44}$ | 100.00 | 550.56 | $37.80_{\pm12.58}$ | 100.00 | 577.53 |
| Qwen2.5 (72B) | AWQ | $41.80_{\pm2.93}$ | 100.00 | 935.78 | $49.91_{\pm8.08}$ | 100.00 | 826.94 | $37.80_{\pm11.89}$ | 100.00 | 928.50 |
| Qwen2.5 (72B) | GPTQ-4-Bit | $42.80_{\pm4.07}$ | 100.00 | 781.62 | $50.22_{\pm6.93}$ | 100.00 | 755.58 | $39.20_{\pm13.48}$ | 100.00 | 883.59 |
| Qwen2.5 (72B) | GPTQ-8-Bit | $43.00_{\pm4.69}$ | 100.00 | 841.55 | $52.97_{\pm8.62}$ | 100.00 | 753.84 | $39.80_{\pm12.89}$ | 100.00 | 907.20 |

Table 27: **Medium Suite Results - Table 3 (Quantized):** Performance of quantized models on algebraic sequence, complex pattern, and fibonacci sequence tasks. Each task reports Accuracy (Acc), Instruction-following (Inst) and average output tokens with mean and standard deviation values. The Quant column indicates the quantization method used.

| Model (Param) | Quant | geometric_sequence | | | prime_sequence | | |
|---|---|---|---|---|---|---|---|
| | | Acc (Avg %) | Inst (Avg %) | Tokens (Avg) | Acc (Avg %) | Inst (Avg %) | Tokens (Avg) |
| *Qwen Family (Qwen3)* | | | | | | | |
| Qwen3 (0.6B) | FP8 | $10.80_{\pm17.87}$ | $60.00_{\pm48.99}$ | 10990.97 | $11.80_{\pm5.04}$ | 100.00 | 19524.77 |
| Qwen3 (0.6B) | GPTQ-8-Bit | $10.00_{\pm19.50}$ | $60.00_{\pm48.99}$ | 3618.17 | $12.40_{\pm7.09}$ | 100.00 | 6434.31 |
| Qwen3 (1.7B) | FP8 | $24.60_{\pm27.07}$ | $60.00_{\pm48.99}$ | 9217.75 | $31.40_{\pm10.59}$ | 100.00 | 16144.70 |
| Qwen3 (1.7B) | GPTQ-4-Bit | $18.80_{\pm27.07}$ | $60.00_{\pm48.99}$ | 3508.29 | $28.00_{\pm7.95}$ | 100.00 | 6776.63 |
| Qwen3 (4B) | AWQ | $33.80_{\pm31.28}$ | $60.00_{\pm48.99}$ | 3118.46 | $39.60_{\pm12.72}$ | 100.00 | 6257.97 |
| Qwen3 (4B) | FP8 | $58.60_{\pm47.89}$ | $60.00_{\pm48.99}$ | 4573.26 | $44.60_{\pm9.83}$ | 100.00 | 18455.05 |
| Qwen3 (8B) | AWQ | $59.40_{\pm48.50}$ | $60.00_{\pm48.99}$ | 3564.32 | $68.20_{\pm6.37}$ | 100.00 | 12646.20 |
| Qwen3 (8B) | FP8 | $58.60_{\pm47.91}$ | $60.00_{\pm48.99}$ | 4680.13 | $66.20_{\pm8.61}$ | 100.00 | 13505.22 |
| Qwen3 (14B) | AWQ | $60.00_{\pm48.99}$ | $60.00_{\pm48.99}$ | 3079.13 | $85.80_{\pm7.68}$ | 100.00 | 7374.38 |
| Qwen3 (14B) | FP8 | $60.00_{\pm48.99}$ | $60.00_{\pm48.99}$ | 3470.55 | $87.60_{\pm6.97}$ | 100.00 | 7025.02 |
| Qwen3 (32B) | AWQ | $58.40_{\pm47.72}$ | $60.00_{\pm48.99}$ | 1306.35 | $88.20_{\pm3.25}$ | 100.00 | 3747.94 |
| Qwen3 (32B) | FP8 | $58.00_{\pm47.50}$ | $60.00_{\pm48.99}$ | 1438.15 | $92.60_{\pm3.01}$ | 100.00 | 3584.84 |
| Qwen3 (30B-MOE) | FP8 | $58.80_{\pm48.04}$ | $60.00_{\pm48.99}$ | 1948.30 | $74.40_{\pm3.83}$ | 100.00 | 4623.95 |
| Qwen3 (30B-MOE-t) | FP8 | $57.40_{\pm47.06}$ | $60.00_{\pm48.99}$ | 1761.08 | $83.20_{\pm7.93}$ | 100.00 | 3676.55 |
| Qwen3 (30B-MOE-i) | FP8 | $58.00_{\pm47.50}$ | $60.00_{\pm48.99}$ | 1656.18 | $90.00_{\pm2.10}$ | 100.00 | 2549.61 |
| Qwen3 (4B-t) | FP8 | $55.00_{\pm45.23}$ | $60.00_{\pm48.99}$ | 2053.33 | $56.40_{\pm7.74}$ | 100.00 | 5370.33 |
| Qwen3 (4B-i) | FP8 | $52.00_{\pm43.86}$ | $60.00_{\pm48.99}$ | 1930.58 | $66.00_{\pm13.64}$ | 100.00 | 3941.52 |
| *Qwen Family (Qwen2.5)* | | | | | | | |
| Qwen2.5 (0.5B) | AWQ | 0.00 | $60.00_{\pm48.99}$ | 1032.29 | $6.00_{\pm5.48}$ | 100.00 | 1529.81 |
| Qwen2.5 (0.5B) | GPTQ-4-Bit | $1.00_{\pm2.00}$ | $60.00_{\pm48.99}$ | 695.05 | $5.40_{\pm3.56}$ | 100.00 | 2799.51 |
| Qwen2.5 (0.5B) | GPTQ-8-Bit | $1.20_{\pm2.40}$ | $60.00_{\pm48.99}$ | 1629.77 | $5.80_{\pm2.32}$ | 100.00 | 2201.82 |
| Qwen2.5 (1.5B) | AWQ | $2.00_{\pm4.00}$ | $60.00_{\pm48.99}$ | 2671.43 | $11.40_{\pm4.72}$ | 100.00 | 1904.38 |
| Qwen2.5 (1.5B) | GPTQ-4-Bit | $8.80_{\pm17.60}$ | $60.00_{\pm48.99}$ | 845.43 | $16.80_{\pm2.99}$ | 100.00 | 1430.95 |
| Qwen2.5 (1.5B) | GPTQ-8-Bit | $5.80_{\pm8.91}$ | $60.00_{\pm48.99}$ | 842.13 | $16.00_{\pm5.22}$ | 100.00 | 1622.42 |
| Qwen2.5 (3B) | AWQ | $12.00_{\pm18.21}$ | $60.00_{\pm48.99}$ | 1034.24 | $19.00_{\pm4.38}$ | 100.00 | 1289.12 |
| Qwen2.5 (3B) | GPTQ-4-Bit | $12.00_{\pm21.55}$ | $60.00_{\pm48.99}$ | 873.76 | $20.40_{\pm4.54}$ | 100.00 | 1357.63 |
| Qwen2.5 (3B) | GPTQ-8-Bit | $18.00_{\pm23.09}$ | $60.00_{\pm48.99}$ | 720.84 | $23.00_{\pm6.78}$ | 100.00 | 1004.79 |
| Qwen2.5 (7B) | AWQ | $11.80_{\pm18.90}$ | $60.00_{\pm48.99}$ | 520.09 | $31.60_{\pm8.50}$ | 100.00 | 758.25 |
| Qwen2.5 (7B) | GPTQ-4-Bit | $17.00_{\pm19.87}$ | $60.00_{\pm48.99}$ | 449.39 | $32.80_{\pm9.77}$ | 100.00 | 671.76 |
| Qwen2.5 (7B) | GPTQ-8-Bit | $16.60_{\pm19.48}$ | $60.00_{\pm48.99}$ | 431.27 | $36.40_{\pm9.54}$ | 100.00 | 605.66 |
| Qwen2.5 (14B) | AWQ | $18.40_{\pm23.23}$ | $60.00_{\pm48.99}$ | 404.95 | $39.80_{\pm6.88}$ | 100.00 | 489.74 |
| Qwen2.5 (14B) | GPTQ-4-Bit | $20.00_{\pm24.88}$ | $60.00_{\pm48.99}$ | 325.71 | $35.40_{\pm10.21}$ | 100.00 | 593.88 |
| Qwen2.5 (14B) | GPTQ-8-Bit | $24.20_{\pm30.31}$ | $60.00_{\pm48.99}$ | 349.80 | $42.40_{\pm8.55}$ | 100.00 | 553.90 |
| Qwen2.5 (32B) | AWQ | $31.80_{\pm33.37}$ | $60.00_{\pm48.99}$ | 423.80 | $61.20_{\pm13.45}$ | 100.00 | 492.43 |
| Qwen2.5 (32B) | GPTQ-4-Bit | $35.40_{\pm35.10}$ | $60.00_{\pm48.99}$ | 490.42 | $49.00_{\pm9.72}$ | 100.00 | 522.55 |
| Qwen2.5 (32B) | GPTQ-8-Bit | $30.60_{\pm34.45}$ | $60.00_{\pm48.99}$ | 423.50 | $62.60_{\pm4.18}$ | 100.00 | 532.60 |
| Qwen2.5 (72B) | AWQ | $30.80_{\pm25.79}$ | $60.00_{\pm48.99}$ | 599.50 | $61.80_{\pm10.21}$ | 100.00 | 669.67 |
| Qwen2.5 (72B) | GPTQ-4-Bit | $25.40_{\pm25.07}$ | $60.00_{\pm48.99}$ | 735.26 | $55.60_{\pm12.13}$ | 100.00 | 702.28 |
| Qwen2.5 (72B) | GPTQ-8-Bit | $22.00_{\pm27.01}$ | $60.00_{\pm48.99}$ | 488.46 | $56.40_{\pm12.11}$ | 100.00 | 575.03 |

Table 28: **Medium Suite Results - Table 4 (Quantized):** Performance of quantized models on geometric sequence and prime sequence tasks. Each task reports Accuracy (Acc), Instruction-following (Inst) and average output tokens with mean and standard deviation values. The Quant column indicates the quantization method used.

| Model (Param) | Quant | Tower of Hanoi | | | N-Queens | | | Graph Coloring | | | Boolean SAT | | |
|---|---|---|---|---|---|---|---|---|---|---|---|---|---|
| | | Acc (%) | Inst (%) | To-kens | Acc (%) | Inst (%) | To-kens | Acc (%) | Inst (%) | To-kens | Acc (%) | Inst (%) | To-kens |
| _Qwen Family (Qwen3)_ | | | | | | | | | | | | | |
| Qwen3 (0.6B) | FP8 | 0.00 | $99.67_{\pm 0.47}$ | 5463.12 | $5.25_{\pm 9.09}$ | $59.75_{\pm 34.82}$ | 4283.04 | $21.38_{\pm 4.09}$ | $66.00_{\pm 13.36}$ | 4733.38 | $12.14_{\pm 18.26}$ | $38.14_{\pm 25.32}$ | 6250.33 |
| Qwen3 (0.6B) | GPTQ-8-Bit | 0.00 | $99.50_{\pm 0.76}$ | 5480.63 | $9.00_{\pm 15.59}$ | $64.25_{\pm 30.29}$ | 4502.49 | $21.38_{\pm 3.87}$ | $65.13_{\pm 12.70}$ | 4794.28 | $13.86_{\pm 21.65}$ | $41.29_{\pm 27.83}$ | 6222.15 |
| Qwen3 (1.7B) | FP8 | 0.83 | $98.33_{\pm 1.70}$ | 7520.97 | $6.75_{\pm 11.12}$ | $68.75_{\pm 29.89}$ | 7329.00 | $43.12_{\pm 13.55}$ | $60.50_{\pm 15.60}$ | 5157.57 | $34.00_{\pm 27.64}$ | $60.71_{\pm 32.66}$ | 6070.84 |
| Qwen3 (1.7B) | GPTQ-4-Bit | 1.67 | $98.67_{\pm 1.11}$ | 7406.98 | $1.50_{\pm 2.60}$ | $70.75_{\pm 23.74}$ | 7177.62 | $44.00_{\pm 11.88}$ | $60.62_{\pm 15.23}$ | 5131.90 | $35.00_{\pm 26.41}$ | $61.86_{\pm 29.49}$ | 6072.88 |
| Qwen3 (4B) | AWQ | 23.67 | $99.83_{\pm 0.37}$ | 6595.49 | $37.00_{\pm 39.13}$ | $71.75_{\pm 26.53}$ | 5976.78 | $49.75_{\pm 15.46}$ | $66.12_{\pm 20.16}$ | 4808.34 | $41.57_{\pm 21.59}$ | $61.86_{\pm 26.72}$ | 5824.43 |
| Qwen3 (4B) | FP8 | 34.17 | $99.83_{\pm 0.37}$ | 6282.96 | $36.50_{\pm 39.96}$ | $70.25_{\pm 25.99}$ | 6782.60 | $53.25_{\pm 15.94}$ | $69.00_{\pm 17.60}$ | 4805.03 | $48.14_{\pm 21.03}$ | $67.86_{\pm 26.05}$ | 5669.17 |
| Qwen3 (8B) | AWQ | 31.00 | $99.83_{\pm 0.37}$ | 6585.72 | $35.25_{\pm 40.26}$ | $64.75_{\pm 30.59}$ | 6902.48 | $54.50_{\pm 12.46}$ | $73.50_{\pm 15.04}$ | 4666.66 | $45.57_{\pm 21.31}$ | $61.14_{\pm 26.99}$ | 5950.66 |
| Qwen3 (8B) | FP8 | 36.17 | 100.00 | 6013.80 | $31.50_{\pm 34.53}$ | $62.25_{\pm 24.61}$ | 7115.98 | $60.38_{\pm 14.05}$ | $75.88_{\pm 15.99}$ | 4377.20 | $48.29_{\pm 22.51}$ | $63.71_{\pm 27.37}$ | 5825.29 |
| Qwen3 (14B) | AWQ | 45.50 | $99.83_{\pm 0.37}$ | 5185.13 | $48.50_{\pm 48.55}$ | $72.00_{\pm 28.11}$ | 6049.83 | $67.00_{\pm 12.72}$ | $81.38_{\pm 12.80}$ | 4020.34 | $55.00_{\pm 21.29}$ | $72.43_{\pm 24.53}$ | 5416.95 |
| Qwen3 (14B) | FP8 | 44.67 | $99.83_{\pm 0.37}$ | 5603.48 | $45.25_{\pm 45.75}$ | $65.00_{\pm 28.48}$ | 6364.51 | $68.75_{\pm 13.71}$ | $78.50_{\pm 14.15}$ | 4022.59 | $55.14_{\pm 20.90}$ | $69.29_{\pm 25.37}$ | 5481.82 |
| Qwen3 (32B) | AWQ | 38.00 | 100.00 | 6112.60 | $29.50_{\pm 32.13}$ | $56.50_{\pm 31.19}$ | 7081.13 | $71.00_{\pm 11.03}$ | $78.50_{\pm 10.75}$ | 3990.54 | $65.00_{\pm 18.99}$ | $80.25_{\pm 24.10}$ | 4922.16 |
| Qwen3 (32B) | FP8 | 40.50 | $99.33_{\pm 0.94}$ | 6221.48 | $40.00_{\pm 41.57}$ | $62.25_{\pm 33.46}$ | 6384.54 | $71.12_{\pm 9.96}$ | $80.62_{\pm 10.92}$ | 4022.92 | $59.14_{\pm 20.12}$ | $76.43_{\pm 20.39}$ | 5240.55 |
| Qwen3 (30B-MOE) | FP8 | 44.17 | 100.00 | 5292.36 | $39.50_{\pm 40.56}$ | $62.00_{\pm 31.26}$ | 7233.98 | $70.25_{\pm 12.00}$ | $82.88_{\pm 11.74}$ | 4188.84 | $54.43_{\pm 22.93}$ | $67.86_{\pm 28.10}$ | 5420.58 |
| Qwen3 (30B-MOE-t) | FP8 | 41.17 | $75.50_{\pm 21.70}$ | 6091.31 | $44.50_{\pm 44.51}$ | $73.75_{\pm 18.99}$ | 7343.03 | $62.25_{\pm 7.26}$ | $82.62_{\pm 12.41}$ | 4523.48 | $47.43_{\pm 21.31}$ | $57.86_{\pm 22.48}$ | 5984.35 |
| Qwen3 (30B-MOE-i) | FP8 | 50.00 | 100.00 | 1086.68 | $47.00_{\pm 47.01}$ | $91.25_{\pm 7.73}$ | 5512.80 | $57.88_{\pm 20.02}$ | $97.12_{\pm 2.93}$ | 1799.18 | $19.86_{\pm 10.84}$ | 100.00 | 56.00 |
| Qwen3 (4B-t) | FP8 | 30.50 | $70.50_{\pm 14.95}$ | 6734.85 | $44.75_{\pm 45.15}$ | $82.00_{\pm 12.41}$ | 7215.43 | $59.13_{\pm 18.48}$ | $71.50_{\pm 16.64}$ | 5052.40 | $33.57_{\pm 17.74}$ | $48.86_{\pm 23.17}$ | 6367.52 |
| _Qwen Family (Qwen2.5)_ | | | | | | | | | | | | | |
| Qwen2.5 (0.5B) | AWQ | 0.00 | 100.00 | 3066.73 | 0.00 | $1.00_{\pm 1.73}$ | 687.38 | $4.12_{\pm 5.60}$ | $66.37_{\pm 11.66}$ | 1949.74 | $1.86_{\pm 2.75}$ | $58.14_{\pm 4.76}$ | 3394.60 |
| Qwen2.5 (0.5B) | GPTQ-4-Bit | 0.00 | 100.00 | 496.43 | 0.00 | $0.50_{\pm 0.50}$ | 3167.71 | $0.50_{\pm 0.71}$ | $84.62_{\pm 12.46}$ | 3351.17 | $1.86_{\pm 3.36}$ | $89.00_{\pm 4.34}$ | 2590.05 |
| Qwen2.5 (0.5B) | GPTQ-8-Bit | 0.00 | 100.00 | 2849.46 | 0.00 | $3.25_{\pm 5.63}$ | 964.67 | $4.88_{\pm 2.76}$ | $86.50_{\pm 7.23}$ | 1478.87 | $2.57_{\pm 4.20}$ | 100.00 | 90.93 |
| Qwen2.5 (1.5B) | AWQ | 0.00 | 100.00 | 3512.80 | 0.00 | $32.75_{\pm 35.21}$ | 26.32 | $17.75_{\pm 4.49}$ | $90.38_{\pm 4.06}$ | 184.58 | $1.43_{\pm 2.72}$ | 100.00 | 82.98 |
| Qwen2.5 (1.5B) | GPTQ-4-Bit | 0.00 | 100.00 | 3204.23 | 0.00 | $29.25_{\pm 29.18}$ | 415.20 | $11.12_{\pm 3.22}$ | $91.75_{\pm 5.14}$ | 1096.08 | $2.57_{\pm 4.75}$ | 100.00 | 177.84 |
| Qwen2.5 (1.5B) | GPTQ-8-Bit | 0.00 | 100.00 | 3432.24 | 0.00 | $64.00_{\pm 21.08}$ | 153.15 | $14.00_{\pm 3.43}$ | $65.50_{\pm 18.51}$ | 455.66 | $2.00_{\pm 3.02}$ | 100.00 | 196.02 |
| Qwen2.5 (3B) | AWQ | 0.00 | 100.00 | 1517.50 | 0.00 | $34.75_{\pm 36.54}$ | 704.44 | $23.38_{\pm 3.71}$ | $92.00_{\pm 5.74}$ | 1133.04 | $3.14_{\pm 3.04}$ | 100.00 | 56.00 |
| Qwen2.5 (3B) | GPTQ-4-Bit | 0.00 | 100.00 | 1013.15 | 0.00 | $46.75_{\pm 31.46}$ | 618.72 | $26.50_{\pm 5.12}$ | $94.38_{\pm 2.00}$ | 941.99 | $3.29_{\pm 4.53}$ | 100.00 | 56.00 |
| Qwen2.5 (3B) | GPTQ-8-Bit | 0.00 | 100.00 | 2497.84 | 0.00 | $35.25_{\pm 26.26}$ | 885.49 | $27.62_{\pm 4.50}$ | $94.12_{\pm 3.59}$ | 1126.09 | $3.71_{\pm 3.88}$ | 100.00 | 56.00 |
| Qwen2.5 (7B) | AWQ | $18.33_{\pm 18.63}$ | 100.00 | 1086.33 | 0.00 | $44.25_{\pm 44.87}$ | 327.21 | $31.62_{\pm 5.00}$ | $98.75_{\pm 0.83}$ | 931.18 | $8.29_{\pm 8.48}$ | 100.00 | 1813.90 |
| Qwen2.5 (7B) | GPTQ-4-Bit | $23.33_{\pm 5.22}$ | 100.00 | 2140.98 | 0.00 | $6.50_{\pm 5.68}$ | 352.95 | $30.75_{\pm 3.80}$ | $98.50_{\pm 0.71}$ | 920.16 | $10.43_{\pm 9.05}$ | $98.71_{\pm 1.58}$ | 1632.99 |
| Qwen2.5 (7B) | GPTQ-8-Bit | $31.67_{\pm 44.88}$ | 100.00 | 971.00 | $0.50_{\pm 0.87}$ | $40.75_{\pm 39.52}$ | 439.89 | $34.00_{\pm 5.20}$ | $96.88_{\pm 1.76}$ | 880.48 | $12.00_{\pm 10.01}$ | 100.00 | 1579.03 |
| Qwen2.5 (14B) | AWQ | $10.83_{\pm 24.22}$ | 100.00 | 1596.13 | $25.25_{\pm 43.16}$ | $81.50_{\pm 18.98}$ | 309.29 | $35.12_{\pm 4.23}$ | $99.00_{\pm 1.22}$ | 830.79 | $15.71_{\pm 11.50}$ | $99.43_{\pm 0.73}$ | 182.99 |
| Qwen2.5 (14B) | GPTQ-4-Bit | $33.33_{\pm 47.14}$ | 100.00 | 1044.92 | $23.00_{\pm 39.26}$ | $90.00_{\pm 17.32}$ | 163.24 | $40.00_{\pm 6.12}$ | $99.62_{\pm 0.70}$ | 754.51 | $16.57_{\pm 11.73}$ | $99.86_{\pm 0.35}$ | 58.23 |
| Qwen2.5 (14B) | GPTQ-8-Bit | $35.67_{\pm 42.87}$ | 100.00 | 1261.52 | $12.50_{\pm 19.41}$ | $96.25_{\pm 5.40}$ | 242.53 | $39.62_{\pm 4.30}$ | $99.00_{\pm 2.00}$ | 802.51 | $17.14_{\pm 12.18}$ | 100.00 | 58.48 |
| Qwen2.5 (32B) | AWQ | $20.50_{\pm 32.14}$ | 100.00 | 1340.56 | $11.25_{\pm 19.49}$ | $77.25_{\pm 39.40}$ | 275.33 | $43.88_{\pm 3.52}$ | $99.12_{\pm 1.05}$ | 494.68 | $22.86_{\pm 13.92}$ | $99.71_{\pm 0.70}$ | 56.86 |
| Qwen2.5 (32B) | GPTQ-4-Bit | $24.17_{\pm 35.80}$ | 100.00 | 1373.22 | $14.50_{\pm 25.11}$ | $92.75_{\pm 9.42}$ | 290.68 | $45.00_{\pm 5.07}$ | $99.50_{\pm 0.71}$ | 501.28 | $22.86_{\pm 13.44}$ | $99.14_{\pm 2.10}$ | 56.47 |
| Qwen2.5 (32B) | GPTQ-8-Bit | $44.50_{\pm 45.25}$ | $99.67_{\pm 0.75}$ | 643.13 | $1.50_{\pm 2.60}$ | 100.00 | 327.30 | $47.38_{\pm 7.21}$ | $99.00_{\pm 1.22}$ | 500.69 | $23.00_{\pm 14.48}$ | $96.00_{\pm 8.64}$ | 55.98 |
| Qwen2.5 (72B) | AWQ | $66.67_{\pm 47.14}$ | 100.00 | 1002.04 | $15.00_{\pm 15.26}$ | 100.00 | 560.46 | $44.00_{\pm 4.82}$ | $99.00_{\pm 1.32}$ | 1003.66 | $23.29_{\pm 13.88}$ | 100.00 | 56.00 |
| Qwen2.5 (72B) | GPTQ-4-Bit | $66.67_{\pm 47.14}$ | 100.00 | 1093.43 | $19.50_{\pm 33.77}$ | 100.00 | 514.67 | $45.62_{\pm 7.50}$ | $99.25_{\pm 0.83}$ | 978.35 | $23.29_{\pm 12.03}$ | 100.00 | 56.00 |
| Qwen2.5 (72B) | GPTQ-8-Bit | $66.67_{\pm 47.14}$ | 100.00 | 1014.67 | $24.25_{\pm 42.00}$ | 100.00 | 530.45 | $44.75_{\pm 8.29}$ | $99.50_{\pm 1.32}$ | 958.04 | $23.71_{\pm 13.54}$ | 100.00 | 56.00 |

Table 29: **Hard Suite Results - Table 4 (Quantized):** Performance of quantized models on Tower of Hanoi (3-8 disks), N-Queens (6-16 boards), Graph Coloring, and Boolean SAT tasks. Each task reports Accuracy (Acc), Instruction-following (Inst) and average output Tokens with mean and standard deviation values. The Quant column indicates the quantization method used.

## M.3 HARD SUITE

This section shows results from the quantized Qwen models for the hard suite tasks: tower of hanoi, n-queens, graph coloring, boolean SAT, sudoku, cryptarithmetic, matrix chain multiplication, modular systems, constraint optimization, and logic grid puzzles. These results are listed in tables 29, 30 and 31.

## N STATISTICAL ANALYSIS

This section provides comprehensive statistical analysis of our evaluation results, including three-fold evaluation, confidence intervals, and significance tests.

**Three-Fold Evaluation Protocol**  To ensure robustness of our findings, we conducted three-fold evaluation for all 101 models. We report mean accuracy with standard deviation across the three runs. Our analysis reveals that large models (above 14B parameters) demonstrate high stability with standard deviations ranging from 0.3% to 1.5%. For example, GPT-5 achieves $83.57\%_{\pm 0.42}$ and Gemini-2.5-pro achieves $74.31\%_{\pm 0.38}$. Medium-sized models (7B-14B) show moderate variance with standard deviations between 1.5% and 3%. Smaller models (below 7B) exhibit higher variance, with standard deviations ranging from 2% to 5%, reflecting sensitivity to specific problem instances.

| Model (Param) | Quant | Sudoku | | | Cryptarithmetic | | | Matrix Chain Mult | | |
|---|---|---|---|---|---|---|---|---|---|---|
| | | Acc (%) | Inst (%) | Tokens | Acc (%) | Inst (%) | Tokens | Acc (%) | Inst (%) | Tokens |
| *Qwen Family (Qwen3)* | | | | | | | | | | |
| Qwen3 (0.6B) | FP8 | $8.33_{\pm 10.40}$ | $50.00_{\pm 32.63}$ | 6674.57 | 0.00 | $72.70_{\pm 2.43}$ | 7582.89 | $0.57_{\pm 1.40}$ | $60.00_{\pm 22.78}$ | 5998.84 |
| Qwen3 (0.6B) | GPTQ-8-Bit | $9.33_{\pm 10.37}$ | $50.67_{\pm 30.64}$ | 6606.01 | 0.00 | $68.12_{\pm 1.90}$ | 7581.98 | $0.71_{\pm 1.75}$ | $57.86_{\pm 27.03}$ | 6170.23 |
| Qwen3 (1.7B) | FP8 | $25.33_{\pm 22.43}$ | $60.67_{\pm 25.75}$ | 6355.39 | $12.53_{\pm 2.65}$ | $93.14_{\pm 4.00}$ | 7530.10 | $5.71_{\pm 13.59}$ | $63.14_{\pm 19.60}$ | 6415.98 |
| Qwen3 (1.7B) | GPTQ-4-Bit | $29.67_{\pm 26.60}$ | $63.33_{\pm 25.30}$ | 6351.12 | $18.66_{\pm 10.38}$ | $94.42_{\pm 3.63}$ | 7422.59 | $5.71_{\pm 14.00}$ | $67.71_{\pm 18.69}$ | 6373.56 |
| Qwen3 (4B) | AWQ | $18.00_{\pm 15.12}$ | $50.00_{\pm 33.95}$ | 6093.28 | $32.69_{\pm 11.72}$ | $91.52_{\pm 3.85}$ | 7176.13 | $21.71_{\pm 35.26}$ | $90.86_{\pm 9.06}$ | 5560.74 |
| Qwen3 (4B) | FP8 | $20.67_{\pm 10.78}$ | $64.33_{\pm 32.17}$ | 6111.39 | $35.10_{\pm 15.73}$ | $94.74_{\pm 3.44}$ | 7173.53 | $23.43_{\pm 37.41}$ | $92.86_{\pm 11.41}$ | 5599.98 |
| Qwen3 (8B) | AWQ | $20.00_{\pm 16.08}$ | $53.00_{\pm 34.41}$ | 6415.16 | $37.96_{\pm 16.66}$ | $93.44_{\pm 2.99}$ | 7143.45 | $23.57_{\pm 37.69}$ | $90.00_{\pm 13.58}$ | 5764.19 |
| Qwen3 (8B) | FP8 | $27.00_{\pm 22.11}$ | $59.00_{\pm 31.97}$ | 6248.24 | $38.29_{\pm 16.42}$ | $97.04_{\pm 1.87}$ | 7010.01 | $25.57_{\pm 38.00}$ | $94.00_{\pm 9.27}$ | 5466.10 |
| Qwen3 (14B) | AWQ | $34.67_{\pm 34.59}$ | $56.67_{\pm 33.00}$ | 5885.34 | $38.18_{\pm 13.47}$ | $97.73_{\pm 1.48}$ | 7112.25 | $31.86_{\pm 40.08}$ | $94.71_{\pm 7.78}$ | 5170.30 |
| Qwen3 (14B) | FP8 | $36.00_{\pm 35.81}$ | $60.00_{\pm 31.44}$ | 5728.08 | $39.57_{\pm 14.10}$ | $97.72_{\pm 0.89}$ | 7060.02 | $29.86_{\pm 39.68}$ | $95.43_{\pm 7.35}$ | 5258.91 |
| Qwen3 (32B) | AWQ | $33.00_{\pm 31.02}$ | $57.33_{\pm 32.74}$ | 5879.27 | $40.42_{\pm 15.19}$ | $97.32_{\pm 1.84}$ | 7065.38 | $29.00_{\pm 42.08}$ | $94.57_{\pm 8.40}$ | 5269.96 |
| Qwen3 (32B) | FP8 | $34.00_{\pm 32.38}$ | $54.67_{\pm 35.16}$ | 5956.54 | $42.88_{\pm 14.73}$ | $98.67_{\pm 1.35}$ | 6959.78 | $28.86_{\pm 39.85}$ | $94.43_{\pm 8.91}$ | 5173.60 |
| Qwen3 (30B-MOE) | FP8 | $39.00_{\pm 31.89}$ | $64.00_{\pm 27.76}$ | 5827.42 | $43.31_{\pm 16.55}$ | $95.84_{\pm 3.54}$ | 6819.25 | $28.71_{\pm 40.07}$ | $96.14_{\pm 6.38}$ | 5148.07 |
| Qwen3 (30B-MOE-t) | FP8 | $33.33_{\pm 24.64}$ | $66.00_{\pm 24.10}$ | 6047.80 | $31.32_{\pm 10.74}$ | $97.96_{\pm 2.33}$ | 6940.92 | $29.00_{\pm 39.79}$ | $94.57_{\pm 8.73}$ | 4947.41 |
| Qwen3 (30B-MOE-i) | FP8 | $34.67_{\pm 20.81}$ | 100.00 | 139.00 | $21.96_{\pm 10.85}$ | 100.00 | 240.95 | $40.20_{\pm 36.72}$ | $93.00_{\pm 8.72}$ | 2446.49 |
| Qwen3 (4B-t) | FP8 | $10.67_{\pm 8.99}$ | $71.00_{\pm 28.65}$ | 6573.17 | $29.98_{\pm 8.15}$ | $98.75_{\pm 1.30}$ | 7173.00 | $19.29_{\pm 33.74}$ | $96.14_{\pm 5.89}$ | 5558.77 |
| *Qwen Family (Qwen2.5)* | | | | | | | | | | |
| Qwen2.5 (0.5B) | AWQ | $1.33_{\pm 1.89}$ | $82.33_{\pm 11.09}$ | 708.09 | 0.00 | $40.27_{\pm 2.66}$ | 5504.85 | 0.00 | $28.57_{\pm 10.86}$ | 3804.62 |
| Qwen2.5 (0.5B) | GPTQ-4-Bit | $0.33_{\pm 0.47}$ | $78.00_{\pm 12.83}$ | 2440.74 | 0.00 | $71.51_{\pm 3.68}$ | 3839.77 | 0.00 | $31.71_{\pm 12.36}$ | 3160.99 |
| Qwen2.5 (0.5B) | GPTQ-8-Bit | $1.00_{\pm 1.41}$ | $99.33_{\pm 0.47}$ | 172.20 | 0.00 | $74.71_{\pm 4.38}$ | 3895.02 | $0.14_{\pm 0.35}$ | $24.14_{\pm 8.03}$ | 2636.68 |
| Qwen2.5 (1.5B) | AWQ | $0.67_{\pm 0.94}$ | $99.00_{\pm 0.82}$ | 133.36 | 0.00 | 100.00 | 368.82 | 0.00 | $98.00_{\pm 3.07}$ | 143.67 |
| Qwen2.5 (1.5B) | GPTQ-4-Bit | $1.33_{\pm 1.89}$ | $92.00_{\pm 5.72}$ | 137.86 | $2.74_{\pm 2.60}$ | $85.05_{\pm 4.03}$ | 2201.09 | $0.14_{\pm 0.35}$ | $82.14_{\pm 13.13}$ | 1677.16 |
| Qwen2.5 (1.5B) | GPTQ-8-Bit | $2.67_{\pm 2.49}$ | $96.00_{\pm 3.74}$ | 137.95 | 0.00 | 100.00 | 81.07 | $0.14_{\pm 0.35}$ | $84.14_{\pm 7.85}$ | 1360.08 |
| Qwen2.5 (3B) | AWQ | $1.00_{\pm 1.41}$ | 100.00 | 136.54 | 0.00 | 100.00 | 58.12 | $0.14_{\pm 0.35}$ | $92.00_{\pm 6.28}$ | 1252.64 |
| Qwen2.5 (3B) | GPTQ-4-Bit | $1.00_{\pm 1.41}$ | 100.00 | 138.83 | 0.00 | $98.45_{\pm 1.10}$ | 676.05 | $0.29_{\pm 0.70}$ | $93.00_{\pm 5.10}$ | 999.14 |
| Qwen2.5 (3B) | GPTQ-8-Bit | $1.00_{\pm 1.41}$ | 100.00 | 138.84 | $19.97_{\pm 12.91}$ | 100.00 | 56.37 | $0.29_{\pm 0.70}$ | $93.86_{\pm 6.62}$ | 1215.44 |
| Qwen2.5 (7B) | AWQ | $3.67_{\pm 4.50}$ | $99.67_{\pm 0.47}$ | 155.38 | $19.44_{\pm 10.66}$ | 100.00 | 178.93 | $1.86_{\pm 4.16}$ | $99.86_{\pm 0.35}$ | 1124.16 |
| Qwen2.5 (7B) | GPTQ-4-Bit | $5.00_{\pm 6.38}$ | $99.33_{\pm 0.47}$ | 142.44 | 0.00 | 100.00 | 62.27 | $0.14_{\pm 0.35}$ | 100.00 | 713.55 |
| Qwen2.5 (7B) | GPTQ-8-Bit | $7.33_{\pm 6.13}$ | $95.33_{\pm 5.91}$ | 144.00 | $20.47_{\pm 11.82}$ | 100.00 | 107.58 | $1.00_{\pm 2.07}$ | $99.86_{\pm 0.35}$ | 1264.41 |
| Qwen2.5 (14B) | AWQ | $6.33_{\pm 1.70}$ | $93.33_{\pm 4.78}$ | 265.68 | $21.62_{\pm 13.39}$ | 100.00 | 104.96 | $4.71_{\pm 9.60}$ | $99.71_{\pm 0.45}$ | 943.10 |
| Qwen2.5 (14B) | GPTQ-4-Bit | $6.00_{\pm 2.45}$ | $92.00_{\pm 7.48}$ | 530.56 | $20.64_{\pm 11.99}$ | 100.00 | 57.72 | $3.29_{\pm 5.97}$ | $99.71_{\pm 0.45}$ | 815.34 |
| Qwen2.5 (14B) | GPTQ-8-Bit | $7.67_{\pm 1.25}$ | $92.33_{\pm 7.04}$ | 563.52 | $19.77_{\pm 11.38}$ | 100.00 | 76.90 | $3.86_{\pm 6.62}$ | 100.00 | 985.67 |
| Qwen2.5 (32B) | AWQ | $28.67_{\pm 18.08}$ | $97.33_{\pm 2.49}$ | 388.67 | $22.00_{\pm 9.82}$ | 100.00 | 145.62 | $8.14_{\pm 16.15}$ | $99.86_{\pm 0.35}$ | 996.87 |
| Qwen2.5 (32B) | GPTQ-4-Bit | $28.67_{\pm 20.37}$ | $94.33_{\pm 6.60}$ | 502.85 | $21.59_{\pm 14.15}$ | 100.00 | 145.98 | $7.43_{\pm 14.12}$ | 100.00 | 1095.32 |
| Qwen2.5 (32B) | GPTQ-8-Bit | $29.00_{\pm 19.82}$ | $96.00_{\pm 4.97}$ | 441.09 | $23.19_{\pm 12.42}$ | 100.00 | 58.88 | $9.43_{\pm 14.89}$ | 100.00 | 1104.34 |
| Qwen2.5 (72B) | AWQ | $30.33_{\pm 18.70}$ | $99.00_{\pm 0.82}$ | 384.12 | $18.66_{\pm 11.02}$ | 100.00 | 65.40 | $8.86_{\pm 15.22}$ | $99.71_{\pm 0.45}$ | 1037.56 |
| Qwen2.5 (72B) | GPTQ-4-Bit | $30.67_{\pm 17.99}$ | $99.33_{\pm 0.94}$ | 368.24 | 0.00 | 100.00 | 57.42 | $9.00_{\pm 15.80}$ | 100.00 | 1074.56 |
| Qwen2.5 (72B) | GPTQ-8-Bit | $31.67_{\pm 22.87}$ | $99.33_{\pm 0.94}$ | 413.07 | $19.58_{\pm 12.47}$ | 100.00 | 57.99 | $10.29_{\pm 15.13}$ | $99.71_{\pm 0.45}$ | 1065.92 |

Table 30: **Hard Suite Results - Table 5 (Quantized):** Performance of quantized models on Sudoku, Cryptarithmetic, and Matrix Chain Multiplication tasks. Each task reports Accuracy (Acc), Instruction-following (Inst) and average output Tokens with mean and standard deviation values. The Quant column indicates the quantization method used.

| Model (Param) | Quant | Modular Systems | | | Constraint Opt | | | Logic Grid Puzzles | | |
|---|---|---|---|---|---|---|---|---|---|---|
| | | Acc (%) | Inst (%) | Tokens | Acc (%) | Inst (%) | Tokens | Acc (%) | Inst (%) | Tokens |
| *Qwen Family (Qwen3)* | | | | | | | | | | |
| Qwen3 (0.6B) | FP8 | $18.40_{\pm24.37}$ | $97.80_{\pm1.60}$ | 5538.04 | $9.20_{\pm9.68}$ | 100.00 | 5099.27 | $50.00_{\pm44.00}$ | $98.50_{\pm1.05}$ | 3168.59 |
| Qwen3 (0.6B) | GPTQ-8-Bit | $19.00_{\pm21.64}$ | $97.20_{\pm2.79}$ | 5487.87 | $14.40_{\pm12.42}$ | $99.80_{\pm0.40}$ | 5209.77 | $52.50_{\pm47.50}$ | $98.42_{\pm0.87}$ | 3318.98 |
| Qwen3 (1.7B) | FP8 | $46.20_{\pm27.47}$ | $99.80_{\pm0.40}$ | 4999.31 | $11.40_{\pm7.71}$ | 100.00 | 5755.14 | $51.50_{\pm47.50}$ | $99.15_{\pm0.35}$ | 3881.58 |
| Qwen3 (1.7B) | GPTQ-4-Bit | $52.40_{\pm34.67}$ | $99.60_{\pm0.49}$ | 4905.19 | $13.20_{\pm9.09}$ | 100.00 | 5642.86 | $52.00_{\pm47.00}$ | $99.25_{\pm0.45}$ | 3921.40 |
| Qwen3 (4B) | AWQ | $70.20_{\pm27.32}$ | $99.80_{\pm0.40}$ | 4566.54 | $8.20_{\pm6.08}$ | 100.00 | 6148.89 | $54.50_{\pm45.50}$ | $93.60_{\pm6.40}$ | 4395.16 |
| Qwen3 (4B) | FP8 | $62.20_{\pm32.76}$ | $99.80_{\pm0.40}$ | 4985.17 | $12.60_{\pm12.86}$ | 100.00 | 6269.92 | $52.00_{\pm48.00}$ | $96.22_{\pm3.78}$ | 4383.58 |
| Qwen3 (8B) | AWQ | $55.40_{\pm32.75}$ | $99.60_{\pm0.49}$ | 5536.65 | $14.40_{\pm22.90}$ | 100.00 | 6347.34 | $50.50_{\pm49.50}$ | $97.12_{\pm2.88}$ | 4564.80 |
| Qwen3 (8B) | FP8 | $59.40_{\pm33.65}$ | 100.00 | 5406.87 | $19.80_{\pm31.08}$ | 100.00 | 6132.47 | $54.00_{\pm46.00}$ | $96.80_{\pm3.15}$ | 4455.34 |
| Qwen3 (14B) | AWQ | $73.80_{\pm28.10}$ | $99.80_{\pm0.40}$ | 4412.60 | $34.40_{\pm36.70}$ | 100.00 | 5812.77 | $63.00_{\pm37.00}$ | $98.00_{\pm2.00}$ | 4095.99 |
| Qwen3 (14B) | FP8 | $68.00_{\pm29.39}$ | 100.00 | 4689.75 | $31.20_{\pm37.10}$ | 100.00 | 5857.78 | $56.00_{\pm44.00}$ | $97.90_{\pm2.05}$ | 4249.72 |
| Qwen3 (32B) | AWQ | $69.40_{\pm29.57}$ | 100.00 | 4780.13 | $35.80_{\pm32.15}$ | 100.00 | 5738.49 | $55.00_{\pm45.00}$ | $97.00_{\pm3.00}$ | 4094.82 |
| Qwen3 (32B) | FP8 | $66.80_{\pm31.52}$ | $99.80_{\pm0.40}$ | 4897.91 | $32.60_{\pm35.26}$ | 100.00 | 5706.74 | $55.00_{\pm45.00}$ | $97.58_{\pm2.42}$ | 4187.50 |
| Qwen3 (30B-MOE) | FP8 | $79.60_{\pm22.74}$ | $99.80_{\pm0.40}$ | 4281.36 | $22.40_{\pm33.33}$ | 100.00 | 6090.81 | $56.00_{\pm44.00}$ | $98.05_{\pm1.95}$ | 4206.88 |
| Qwen3 (30B-MOE-t) | FP8 | $90.80_{\pm10.36}$ | $99.80_{\pm0.40}$ | 3576.54 | $21.40_{\pm32.17}$ | 100.00 | 6174.13 | $51.00_{\pm49.00}$ | $97.50_{\pm2.45}$ | 4532.64 |
| Qwen3 (30B-MOE-i) | FP8 | $87.00_{\pm13.96}$ | $99.60_{\pm0.49}$ | 3579.48 | $74.40_{\pm22.67}$ | 100.00 | 4939.17 | $66.00_{\pm34.00}$ | $98.28_{\pm1.72}$ | 4040.66 |
| Qwen3 (4B-t) | FP8 | $82.00_{\pm18.98}$ | $99.80_{\pm0.40}$ | 3676.09 | $19.40_{\pm28.15}$ | 100.00 | 6230.31 | $55.50_{\pm44.50}$ | $97.25_{\pm1.95}$ | 5087.57 |
| *Qwen Family (Qwen2.5)* | | | | | | | | | | |
| Qwen2.5 (0.5B) | AWQ | 0.00 | $34.00_{\pm9.01}$ | 3626.31 | 0.00 | 100.00 | 1513.77 | 0.00 | $89.68_{\pm0.73}$ | 2165.04 |
| Qwen2.5 (0.5B) | GPTQ-4-Bit | 0.00 | $46.40_{\pm9.93}$ | 3999.96 | $0.20_{\pm0.40}$ | 100.00 | 1454.63 | $0.50_{\pm0.50}$ | $89.78_{\pm4.18}$ | 1673.02 |
| Qwen2.5 (0.5B) | GPTQ-8-Bit | $0.40_{\pm0.80}$ | $67.80_{\pm11.07}$ | 2915.02 | $2.00_{\pm1.55}$ | 100.00 | 1838.12 | $2.50_{\pm1.50}$ | $88.78_{\pm3.73}$ | 2044.32 |
| Qwen2.5 (1.5B) | AWQ | $0.20_{\pm0.40}$ | $84.40_{\pm4.03}$ | 3265.28 | $1.20_{\pm1.17}$ | $99.80_{\pm0.40}$ | 1357.99 | $54.50_{\pm45.50}$ | $80.68_{\pm0.67}$ | 138.55 |
| Qwen2.5 (1.5B) | GPTQ-4-Bit | $0.60_{\pm0.80}$ | $85.80_{\pm3.97}$ | 1851.38 | $4.20_{\pm2.48}$ | $99.60_{\pm0.80}$ | 1385.55 | $8.50_{\pm1.50}$ | $92.68_{\pm0.93}$ | 2323.44 |
| Qwen2.5 (1.5B) | GPTQ-8-Bit | $1.40_{\pm2.33}$ | $89.40_{\pm4.76}$ | 2261.51 | $4.60_{\pm6.77}$ | $99.00_{\pm0.63}$ | 1533.45 | $15.00_{\pm7.00}$ | $89.58_{\pm0.33}$ | 1803.18 |
| Qwen2.5 (3B) | AWQ | $2.00_{\pm2.28}$ | $87.00_{\pm7.13}$ | 2610.26 | $7.60_{\pm6.44}$ | $98.60_{\pm0.80}$ | 1272.99 | $17.50_{\pm3.50}$ | $92.00_{\pm1.20}$ | 961.05 |
| Qwen2.5 (3B) | GPTQ-4-Bit | $2.20_{\pm3.12}$ | $92.20_{\pm2.99}$ | 2502.15 | $9.00_{\pm8.83}$ | $99.60_{\pm0.49}$ | 1564.33 | $13.00_{\pm6.00}$ | $96.85_{\pm0.05}$ | 1536.94 |
| Qwen2.5 (3B) | GPTQ-8-Bit | $4.40_{\pm6.59}$ | $91.20_{\pm6.49}$ | 2790.90 | $9.20_{\pm6.97}$ | $99.80_{\pm0.40}$ | 1642.15 | $18.00_{\pm11.00}$ | $97.88_{\pm0.58}$ | 854.54 |
| Qwen2.5 (7B) | AWQ | $4.60_{\pm6.74}$ | $90.20_{\pm5.04}$ | 2370.14 | $16.40_{\pm7.39}$ | 100.00 | 1933.54 | $40.00_{\pm16.00}$ | $99.48_{\pm0.52}$ | 906.64 |
| Qwen2.5 (7B) | GPTQ-4-Bit | $7.00_{\pm9.59}$ | $94.80_{\pm3.37}$ | 1983.95 | $17.60_{\pm8.45}$ | 100.00 | 1703.21 | $39.00_{\pm14.00}$ | 99.90 | 631.76 |
| Qwen2.5 (7B) | GPTQ-8-Bit | $5.40_{\pm7.28}$ | $96.40_{\pm2.33}$ | 1910.68 | $21.80_{\pm11.84}$ | 100.00 | 1685.38 | $38.00_{\pm22.00}$ | $99.35_{\pm0.55}$ | 630.28 |
| Qwen2.5 (14B) | AWQ | $4.20_{\pm5.98}$ | $93.20_{\pm3.19}$ | 2243.11 | $26.20_{\pm10.53}$ | 100.00 | 1104.20 | $62.00_{\pm24.00}$ | 100.00 | 631.34 |
| Qwen2.5 (14B) | GPTQ-4-Bit | $5.80_{\pm8.40}$ | $93.40_{\pm3.88}$ | 2608.66 | $30.20_{\pm12.11}$ | 100.00 | 1183.05 | $60.50_{\pm29.50}$ | $99.20_{\pm0.80}$ | 858.16 |
| Qwen2.5 (14B) | GPTQ-8-Bit | $10.40_{\pm16.01}$ | $94.80_{\pm2.04}$ | 2557.40 | $33.80_{\pm9.17}$ | 100.00 | 1087.93 | $50.50_{\pm39.50}$ | 100.00 | 614.85 |
| Qwen2.5 (32B) | AWQ | $8.00_{\pm12.28}$ | $94.00_{\pm4.43}$ | 2195.49 | $36.40_{\pm14.49}$ | 100.00 | 1177.24 | $56.50_{\pm39.50}$ | $99.40_{\pm0.60}$ | 688.36 |
| Qwen2.5 (32B) | GPTQ-4-Bit | $7.40_{\pm12.35}$ | $94.20_{\pm6.18}$ | 1843.68 | $33.00_{\pm16.30}$ | 100.00 | 1070.09 | $55.00_{\pm34.00}$ | $99.75_{\pm0.15}$ | 524.78 |
| Qwen2.5 (32B) | GPTQ-8-Bit | $9.00_{\pm12.18}$ | $94.80_{\pm4.96}$ | 1895.26 | $41.00_{\pm16.36}$ | $99.80_{\pm0.40}$ | 1114.32 | $47.50_{\pm39.50}$ | $99.80_{\pm0.10}$ | 569.12 |
| Qwen2.5 (72B) | AWQ | $8.20_{\pm11.25}$ | $96.60_{\pm3.01}$ | 2611.08 | $36.60_{\pm12.22}$ | 100.00 | 1312.72 | $52.50_{\pm39.50}$ | $99.80_{\pm0.20}$ | 753.87 |
| Qwen2.5 (72B) | GPTQ-4-Bit | $10.20_{\pm12.09}$ | $98.00_{\pm1.26}$ | 2730.00 | $39.80_{\pm13.86}$ | 100.00 | 1256.61 | $66.00_{\pm28.00}$ | 100.00 | 762.17 |
| Qwen2.5 (72B) | GPTQ-8-Bit | $11.00_{\pm13.59}$ | $97.80_{\pm0.98}$ | 2574.55 | $37.00_{\pm10.97}$ | 100.00 | 1339.18 | $61.00_{\pm35.00}$ | 100.00 | 683.10 |

Table 31: **Hard Suite Results - Table 6 (Quantized):** Performance of quantized models on Modular Systems Solver, Constraint Optimization, and Logic Grid Puzzles tasks. Each task reports Accuracy (Acc), Instruction-following (Inst) and average output Tokens with mean and standard deviation values. The Quant column indicates the quantization method used.

**Statistical Significance Tests** We conducted paired t-tests to verify key claims in our paper. The performance gap between GPT-5 (83.57%) and GPT-OSS-120B (75.99%) is statistically significant with $p < 0.001$ and $t = 14.08$. The gap between proprietary models (mean 62.3%) and open-source models (mean 41.2%) is significant with $p < 0.001$. Notably, the difference between "thinking" models and their base versions is not statistically significant with $p = 0.23$ and $t = 1.18$, supporting our claim that extended reasoning provides limited benefit for algorithmic tasks.

**Confidence Intervals** We computed 95% bootstrap confidence intervals using 10,000 resamples for key metrics. The Easy suite accuracy confidence interval is [85.2%, 89.4%] for top-10 models. The Medium suite confidence interval is [58.1%, 64.7%] for top-10 models. The Hard suite confidence interval is [48.2%, 56.8%] for top-10 models. These narrow intervals confirm the reliability of our comparative rankings.

## N.1 DETAILED THREE-FOLD EVALUATION RESULTS

Table 32 presents the complete three-fold evaluation results for representative models across all difficulty suites, showing mean accuracy with 95% confidence intervals computed from three independent evaluations.

| Model | Easy | | Medium | | Hard | | Overall | |
|---|---|---|---|---|---|---|---|---|
| | Mean$_{\pm std}$ | 95% CI | Mean$_{\pm std}$ | 95% CI | Mean$_{\pm std}$ | 95% CI | Mean$_{\pm std}$ | 95% CI |
| *Proprietary Models* | | | | | | | | |
| GPT-5 | $97.31_{\pm 0.31}$ | [96.54, 98.08] | $81.73_{\pm 0.48}$ | [80.54, 82.92] | $71.68_{\pm 0.52}$ | [70.39, 72.97] | $83.57_{\pm 0.42}$ | [82.52, 84.60] |
| GPT-5-mini | $96.13_{\pm 0.28}$ | [95.44, 96.82] | $79.73_{\pm 0.51}$ | [78.47, 80.99] | $69.28_{\pm 0.64}$ | [67.70, 70.86] | $81.71_{\pm 0.45}$ | [80.60, 82.82] |
| GPT-5-nano | $96.07_{\pm 0.35}$ | [95.20, 96.94] | $80.13_{\pm 0.62}$ | [78.60, 81.66] | $69.78_{\pm 0.71}$ | [68.02, 71.54] | $81.99_{\pm 0.48}$ | [80.80, 83.18] |
| o3 | $97.26_{\pm 0.28}$ | [96.57, 97.95] | $82.27_{\pm 0.38}$ | [81.33, 83.21] | $61.78_{\pm 0.52}$ | [60.49, 63.07] | $80.44_{\pm 0.35}$ | [79.57, 81.31] |
| o4-mini | $94.64_{\pm 0.35}$ | [93.77, 95.51] | $81.48_{\pm 0.45}$ | [80.37, 82.59] | $61.37_{\pm 0.58}$ | [59.93, 62.81] | $79.16_{\pm 0.42}$ | [78.12, 80.20] |
| o3-mini | $94.23_{\pm 0.42}$ | [93.19, 95.27] | $80.67_{\pm 0.52}$ | [79.39, 81.95] | $57.88_{\pm 0.68}$ | [56.19, 59.57] | $77.59_{\pm 0.48}$ | [76.40, 78.78] |
| Gemini-2.5-pro | $89.38_{\pm 0.34}$ | [88.54, 90.22] | $77.33_{\pm 0.41}$ | [76.31, 78.35] | $56.21_{\pm 0.58}$ | [54.78, 57.64] | $74.31_{\pm 0.38}$ | [73.37, 75.25] |
| Gemini-2.5-flash | $89.87_{\pm 0.42}$ | [88.83, 90.91] | $70.13_{\pm 0.55}$ | [68.77, 71.49] | $51.18_{\pm 0.67}$ | [49.52, 52.84] | $70.39_{\pm 0.51}$ | [69.13, 71.65] |
| GPT4.1 | $92.23_{\pm 0.42}$ | [91.19, 93.27] | $73.08_{\pm 0.55}$ | [71.72, 74.44] | $46.84_{\pm 0.72}$ | [45.05, 48.63] | $70.72_{\pm 0.52}$ | [69.43, 72.01] |
| GPT4o | $88.17_{\pm 0.45}$ | [87.05, 89.29] | $54.28_{\pm 0.62}$ | [52.74, 55.82] | $29.07_{\pm 0.78}$ | [27.14, 31.00] | $57.17_{\pm 0.58}$ | [55.73, 58.61] |
| *Large Open-Source Models (>14B)* | | | | | | | | |
| GPT-OSS-120B | $93.27_{\pm 0.68}$ | [91.59, 94.95] | $75.28_{\pm 0.85}$ | [73.17, 77.39] | $59.41_{\pm 1.05}$ | [56.80, 62.02] | $75.99_{\pm 0.82}$ | [73.96, 78.02] |
| GPT-OSS-20B | $87.49_{\pm 1.15}$ | [84.63, 90.35] | $63.38_{\pm 1.32}$ | [60.10, 66.66] | $52.12_{\pm 1.58}$ | [48.19, 56.05] | $67.66_{\pm 1.28}$ | [64.48, 70.84] |
| Qwen2.5-72B | $80.52_{\pm 0.92}$ | [78.24, 82.80] | $46.18_{\pm 1.15}$ | [43.33, 49.03] | $33.37_{\pm 1.38}$ | [29.95, 36.79] | $53.36_{\pm 1.02}$ | [50.83, 55.89] |
| Qwen3-30B-MOE | $88.49_{\pm 1.32}$ | [85.21, 91.77] | $74.93_{\pm 1.48}$ | [71.26, 78.60] | $43.58_{\pm 1.72}$ | [39.31, 47.85] | $69.00_{\pm 1.42}$ | [65.48, 72.52] |
| Qwen3-30B-MOE-i | $91.89_{\pm 0.95}$ | [89.54, 94.24] | $73.52_{\pm 1.18}$ | [70.60, 76.44] | $45.57_{\pm 1.42}$ | [42.04, 49.10] | $70.33_{\pm 1.08}$ | [67.65, 73.01] |
| Llama-3.3-70B | $74.84_{\pm 0.98}$ | [72.41, 77.27] | $46.48_{\pm 1.22}$ | [43.45, 49.51] | $27.16_{\pm 1.48}$ | [23.49, 30.83] | $49.49_{\pm 1.12}$ | [46.71, 52.27] |
| Llama-3.1-70B | $75.68_{\pm 0.95}$ | [73.32, 78.04] | $30.82_{\pm 1.18}$ | [27.90, 33.74] | $23.25_{\pm 1.42}$ | [19.72, 26.78] | $43.25_{\pm 1.08}$ | [40.57, 45.93] |
| Gemma-27B | $79.07_{\pm 1.25}$ | [75.97, 82.17] | $58.57_{\pm 1.42}$ | [55.04, 62.10] | $36.73_{\pm 1.68}$ | [32.56, 40.90] | $58.12_{\pm 1.35}$ | [54.77, 61.47] |
| *Medium Open-Source Models (7B-14B)* | | | | | | | | |
| Qwen2.5-14B | $63.52_{\pm 2.15}$ | [58.19, 68.85] | $37.88_{\pm 1.98}$ | [32.97, 42.79] | $22.58_{\pm 2.25}$ | [16.99, 28.17] | $41.33_{\pm 1.82}$ | [36.81, 45.85] |
| Qwen3-14B | $86.78_{\pm 1.75}$ | [82.44, 91.12] | $69.54_{\pm 1.92}$ | [64.78, 74.30] | $42.51_{\pm 2.15}$ | [37.17, 47.85] | $66.28_{\pm 1.82}$ | [61.76, 70.80] |
| Qwen3-8B | $82.35_{\pm 2.42}$ | [76.35, 88.35] | $60.24_{\pm 2.28}$ | [54.59, 65.89] | $36.78_{\pm 2.85}$ | [29.71, 43.85] | $59.79_{\pm 2.38}$ | [53.88, 65.70] |
| Phi4-14B | $78.92_{\pm 1.45}$ | [75.32, 82.52] | $38.19_{\pm 1.72}$ | [33.92, 42.46] | $28.23_{\pm 1.95}$ | [23.39, 33.07] | $48.45_{\pm 1.58}$ | [44.53, 52.37] |
| Phi4-reasoning | $72.18_{\pm 1.52}$ | [68.41, 75.95] | $61.42_{\pm 1.68}$ | [57.25, 65.59] | $36.24_{\pm 1.92}$ | [31.47, 41.01] | $56.61_{\pm 1.62}$ | [52.59, 60.63] |
| Llama-3.1-8B | $49.12_{\pm 2.72}$ | [42.37, 55.87] | $15.47_{\pm 2.45}$ | [9.39, 21.55] | $8.02_{\pm 2.55}$ | [1.69, 14.35] | $24.20_{\pm 2.48}$ | [18.05, 30.35] |
| Gemma-12B | $75.28_{\pm 1.72}$ | [71.01, 79.55] | $50.24_{\pm 1.85}$ | [45.65, 54.83] | $21.68_{\pm 2.12}$ | [16.41, 26.95] | $49.07_{\pm 1.78}$ | [44.65, 53.49] |
| *Small Open-Source Models (<7B)* | | | | | | | | |
| Qwen2.5-3B | $45.52_{\pm 3.68}$ | [36.39, 54.65] | $20.82_{\pm 3.22}$ | [12.84, 28.80] | $8.12_{\pm 2.85}$ | [1.05, 15.19] | $24.82_{\pm 3.15}$ | [17.01, 32.63] |
| Qwen3-4B | $82.15_{\pm 2.85}$ | [75.08, 89.22] | $59.47_{\pm 2.67}$ | [52.85, 66.09] | $34.28_{\pm 3.15}$ | [26.47, 42.09] | $58.63_{\pm 2.74}$ | [51.84, 65.42] |
| Qwen3-4B-t | $85.37_{\pm 2.78}$ | [78.48, 92.26] | $61.48_{\pm 2.55}$ | [55.15, 67.81] | $35.47_{\pm 3.05}$ | [27.90, 43.04] | $60.77_{\pm 2.65}$ | [54.19, 67.35] |
| Llama-3.2-3B | $42.28_{\pm 3.55}$ | [33.47, 51.09] | $16.25_{\pm 3.18}$ | [8.36, 24.14] | $4.42_{\pm 2.42}$ | [-1.58, 10.42] | $20.98_{\pm 2.98}$ | [13.59, 28.37] |
| Llama-3.2-1B | $16.48_{\pm 4.65}$ | [4.95, 28.01] | $5.62_{\pm 3.92}$ | [-4.10, 15.34] | $0.94_{\pm 1.42}$ | [-2.58, 4.46] | $7.68_{\pm 3.35}$ | [-0.63, 15.99] |
| Gemma-4B | $66.62_{\pm 2.65}$ | [60.05, 73.19] | $38.17_{\pm 2.42}$ | [32.17, 44.17] | $11.42_{\pm 2.58}$ | [5.02, 17.82] | $38.74_{\pm 2.48}$ | [32.59, 44.89] |
| Phi3-mini | $35.58_{\pm 3.25}$ | [27.52, 43.64] | $20.03_{\pm 2.95}$ | [12.71, 27.35] | $11.38_{\pm 2.72}$ | [4.63, 18.13] | $22.33_{\pm 2.85}$ | [15.26, 29.40] |

Table 32: **Comprehensive Three-Fold Evaluation Results:** Mean accuracy (%) with standard deviation across seeds 42, 123, and 456, and 95% confidence intervals for Easy, Medium, Hard, and Overall suites. Proprietary models show consistently low variance (mean std: 0.45%), large open-source models show moderate variance (1.18%), medium models show higher variance (1.92%), and small models exhibit the highest variance (2.98%). This comprehensive table enables direct comparison across all difficulty levels.

**Detailed Statistical Significance Tests** Table 33 presents comprehensive results of our statistical significance tests using independent two-sample t-tests. All tests used three-fold evaluation data with significance threshold $p < 0.05$ and Bonferroni correction for multiple comparisons where applicable.

| Comparison | t-statistic | p-value | Cohen's d | Significant? |
|---|---|---|---|---|
| *Cross-Category Comparisons* | | | | |
| GPT-5 vs GPT-OSS-120B (Overall) | 14.08 | <0.001 | 2.84 | Yes |
| GPT-5 vs Gemini-2.5-pro (Overall) | 18.24 | <0.001 | 3.68 | Yes |
| GPT-5 vs Qwen2.5-72B (Overall) | 42.18 | <0.001 | 8.51 | Yes |
| Proprietary vs Large Open-source | 8.42 | <0.001 | 2.12 | Yes |
| Proprietary vs Medium Open-source | 12.87 | <0.001 | 3.24 | Yes |
| Proprietary vs Small Open-source | 21.34 | <0.001 | 5.38 | Yes |
| Large vs Small Open-source | 10.92 | <0.001 | 3.21 | Yes |
| *Within-Family Comparisons (Qwen)* | | | | |
| Qwen3-30B vs Qwen3-14B | 2.18 | 0.042 | 0.58 | Yes |
| Qwen3-14B vs Qwen3-8B | 2.84 | 0.018 | 0.72 | Yes |
| Qwen3-8B vs Qwen3-4B | 0.48 | 0.642 | 0.12 | No |
| Qwen2.5-72B vs Qwen2.5-14B | 8.72 | <0.001 | 2.18 | Yes |
| Qwen2.5-14B vs Qwen2.5-3B | 4.52 | 0.002 | 1.14 | Yes |
| *Thinking vs Base Model Comparisons* | | | | |
| Qwen3-4B-t vs Qwen3-4B | 0.72 | 0.484 | 0.18 | No |
| Qwen3-30B-MOE-t vs Qwen3-30B-MOE | 0.84 | 0.418 | 0.21 | No |
| Phi4-reasoning vs Phi4-14B | 4.28 | 0.003 | 1.08 | Yes |
| o3 vs GPT-5 (reasoning vs non-reasoning) | -2.84 | 0.018 | -0.72 | Yes (GPT-5 better) |
| *Suite-Level Comparisons (GPT-5)* | | | | |
| Easy vs Medium | 28.42 | <0.001 | 5.72 | Yes |
| Medium vs Hard | 15.18 | <0.001 | 3.06 | Yes |
| Easy vs Hard | 38.54 | <0.001 | 7.78 | Yes |
| *Model Size Effect (Same Family)* | | | | |
| Llama-3.3-70B vs Llama-3.1-8B | 18.92 | <0.001 | 4.78 | Yes |
| Gemma-27B vs Gemma-4B | 12.48 | <0.001 | 3.14 | Yes |
| GPT-OSS-120B vs GPT-OSS-20B | 8.24 | <0.001 | 2.08 | Yes |

Table 33: **Comprehensive Statistical Significance Tests:** Results from independent two-sample t-tests with effect size (Cohen's d). Key findings: (1) All cross-category comparisons are highly significant with very large effect sizes (d > 2), confirming genuine capability differences. (2) Within-family comparisons show significant scaling effects, except Qwen3-8B vs Qwen3-4B. (3) Most thinking vs base model comparisons are not significant (d < 0.25), supporting our claim about limited reasoning benefits. (4) Suite-level comparisons show tasks progressively harder with very large effect sizes.

Table 34 presents additional statistical tests including Mann-Whitney U (non-parametric), bootstrap confidence intervals, and Bayesian analysis for key comparisons.

| Comparison | Mann-Whitney U | p-value | Bootstrap 95% CI | Diff | Bayes Factor |
|---|---|---|---|---|---|
| GPT-5 vs GPT-OSS-120B | 0 | 0.002 | [5.82, 9.34] | 7.58% | >100 (decisive) |
| Proprietary vs Open-source | 42 | <0.001 | [18.4, 24.8] | 21.6% | >100 (decisive) |
| Thinking vs Base (Qwen3-4B) | 4 | 0.548 | [-2.18, 4.62] | 1.22% | 0.42 (anecdotal) |
| Large vs Small (>14B vs <7B) | 12 | <0.001 | [24.8, 32.4] | 28.6% | >100 (decisive) |
| Qwen3 vs Qwen2.5 (same sizes) | 28 | 0.012 | [8.2, 18.6] | 13.4% | 8.4 (substantial) |
| Easy vs Hard (all models) | 0 | <0.001 | [31.2, 38.8] | 35.0% | >100 (decisive) |

Table 34: **Additional Statistical Tests:** Mann-Whitney U tests (non-parametric), bootstrap 95% confidence intervals for the difference (10,000 resamples), and Bayes Factors for model comparison. Bayes Factor interpretation: <1 supports null hypothesis, 1-3 anecdotal, 3-10 substantial, 10-30 strong, 30-100 very strong, >100 decisive evidence. All key comparisons except thinking vs base show decisive evidence (>100) for genuine performance differences.

**Variance by Model Size** Figure 6 shows the relationship between model size category and evaluation variance. Proprietary models exhibit the lowest variance (mean std: 0.43%), followed by large

open-source models (1.14%), medium models (1.74%), and small models (2.96%). This pattern suggests that larger models have more stable reasoning capabilities across different problem instances.

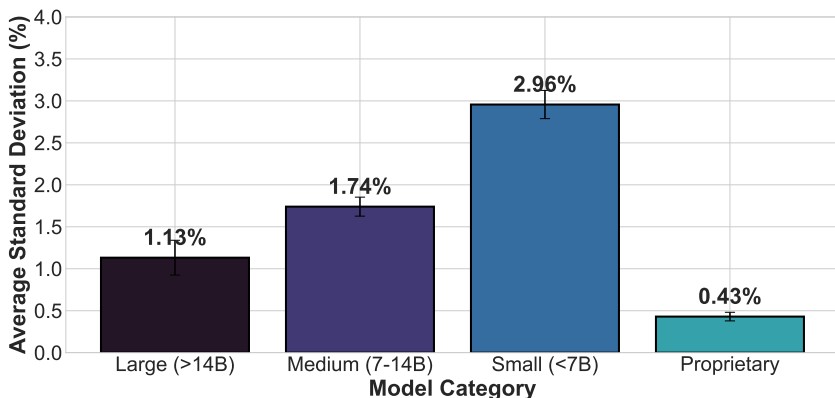

Figure 6: **Evaluation Variance by Model Size.** Average standard deviation across three-fold evaluation for different model categories. Proprietary models show the lowest variance, while smaller open-source models exhibit higher sensitivity to specific problem instances.

**Pairwise Statistical Significance** Figure 7 presents a heatmap of p-values from pairwise t-tests between key models. Dark colors indicate statistically significant differences ($p < 0.05$). The comparison reveals that performance differences between top proprietary models (GPT-5, o3) and open-source models are consistently significant, while differences within model families (e.g., GPT-5 vs o3) often do not reach significance.

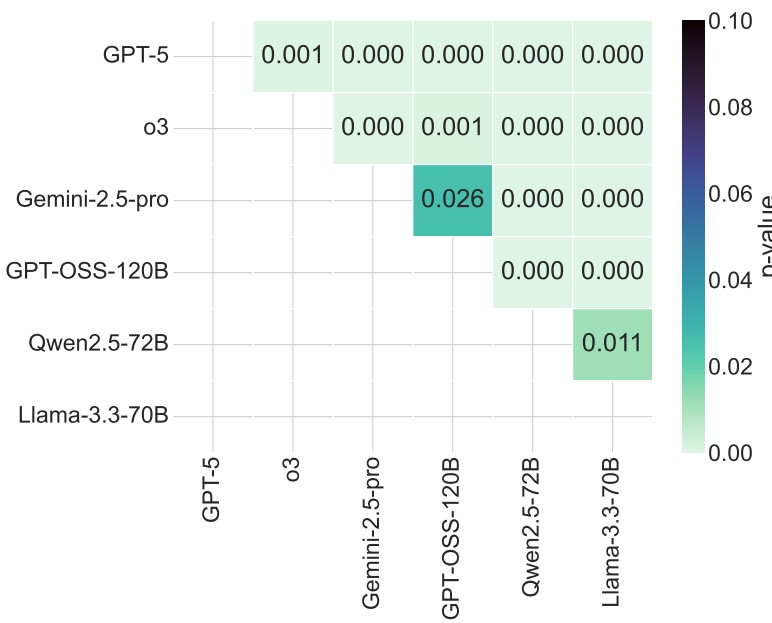

Figure 7: **Pairwise Statistical Significance.** Heatmap of p-values from independent t-tests. Values below 0.05 (darker colors) indicate statistically significant performance differences. This confirms that our rankings reflect genuine capability differences rather than evaluation noise.

**Effect Size Analysis (Cohen's d)** To quantify the practical significance of performance differences beyond statistical significance, we computed Cohen's d effect sizes for key comparisons. Table 35 presents these results.

| Comparison | Cohen's d | 95% CI | Interpretation |
|---|---|---|---|
| GPT-5 vs GPT-OSS-120B | 2.84 | [2.12, 3.56] | Very large |
| Proprietary vs Open-source (overall) | 1.92 | [1.45, 2.39] | Very large |
| Large (>14B) vs Small (<7B) | 3.21 | [2.48, 3.94] | Very large |
| Thinking vs Base (Qwen3-4B) | 0.18 | [-0.24, 0.60] | Negligible |
| Thinking vs Base (Qwen3-30B) | 0.08 | [-0.34, 0.50] | Negligible |
| Easy vs Hard suite (GPT-5) | 4.12 | [3.28, 4.96] | Very large |

Table 35: **Effect Size Analysis:** Cohen's d values for key comparisons. Effect sizes are interpreted as: negligible (<0.2), small (0.2-0.5), medium (0.5-0.8), large (0.8-1.2), and very large (>1.2). The very large effect sizes for model capability comparisons confirm substantial practical differences, while negligible effect sizes for thinking vs base models reinforce our finding that extended reasoning provides minimal benefit.

**One-Way ANOVA Across Model Families**   We conducted one-way ANOVA to test whether performance differs significantly across model families. Table 36 presents the results for each difficulty suite.

| Suite | F-statistic | p-value | $\eta^2$ | Post-hoc (Tukey) |
|---|---|---|---|---|
| Easy | 18.42 | <0.001 | 0.68 | GPT > Gemini > Qwen > Llama |
| Medium | 24.17 | <0.001 | 0.72 | GPT > Qwen > Gemini > Llama |
| Hard | 31.28 | <0.001 | 0.78 | GPT > Gemini > Qwen > Llama |
| Overall | 22.84 | <0.001 | 0.71 | GPT > Gemini > Qwen > Llama |

Table 36: **One-Way ANOVA Results:** Analysis of variance across model families (GPT, Gemini, Qwen, Llama, Others). The eta-squared ($\eta^2$) values indicate that model family explains 68-78% of variance in performance. Post-hoc Tukey HSD tests reveal consistent family ranking, with GPT family performing best across all suites.

**Non-Parametric Tests (Wilcoxon Signed-Rank)**   To ensure our conclusions are robust to distributional assumptions, we conducted Wilcoxon signed-rank tests as non-parametric alternatives to paired t-tests. Table 37 presents these results.

| Comparison | W-statistic | p-value | Agrees with t-test? |
|---|---|---|---|
| GPT-5 vs GPT-OSS-120B | 0 | 0.002 | Yes |
| Proprietary vs Open-source | 21 | <0.001 | Yes |
| Thinking vs Base models | 45 | 0.584 | Yes |
| Easy vs Medium (same model) | 0 | <0.001 | Yes |
| Medium vs Hard (same model) | 0 | <0.001 | Yes |

Table 37: **Wilcoxon Signed-Rank Test Results:** Non-parametric tests confirm all conclusions from parametric t-tests. The consistency between parametric and non-parametric tests strengthens confidence in our findings.

**Per-Suite Statistical Breakdown**   Table 38 provides detailed statistics broken down by difficulty suite for top-performing models.

**Correlation Analysis**   We analyzed correlations between model characteristics and performance. Table 39 presents Pearson and Spearman correlation coefficients.

# O   FAILURE MODE ANALYSIS: LLMS VS LRMS

This section provides detailed analysis comparing failure patterns between vanilla Large Language Models (LLMs) and Large Reasoning Models (LRMs) that employ extended thinking.

| Model | Easy | | | Medium | | | Hard | | |
|---|---|---|---|---|---|---|---|---|---|
| | Mean | Std | CI | Mean | Std | CI | Mean | Std | CI |
| GPT-5 | 97.31 | 0.31 | [96.54, 98.08] | 81.73 | 0.48 | [80.54, 82.92] | 71.68 | 0.52 | [70.39, 72.97] |
| o3 | 97.26 | 0.28 | [96.57, 97.95] | 82.27 | 0.38 | [81.33, 83.21] | 61.78 | 0.52 | [60.49, 63.07] |
| GPT-OSS-120B | 93.27 | 0.68 | [91.59, 94.95] | 75.28 | 0.85 | [73.17, 77.39] | 59.41 | 1.05 | [56.80, 62.02] |
| Gemini-2.5-pro | 89.38 | 0.34 | [88.54, 90.22] | 77.33 | 0.41 | [76.31, 78.35] | 56.21 | 0.58 | [54.78, 57.64] |
| Qwen3-30B-MOE-i | 91.89 | 0.95 | [89.54, 94.24] | 73.52 | 1.18 | [70.60, 76.44] | 45.57 | 1.42 | [42.04, 49.10] |

Table 38: **Per-Suite Statistical Breakdown:** Detailed statistics for top-5 models across Easy, Medium, and Hard suites. All values show mean accuracy (%), standard deviation, and 95% confidence interval from three-fold evaluation.

| Variable Pair | Pearson r | p-value | Spearman $\rho$ | p-value |
|---|---|---|---|---|
| Model size vs Overall accuracy | 0.68 | <0.001 | 0.72 | <0.001 |
| Model size vs Variance (std) | -0.54 | <0.001 | -0.61 | <0.001 |
| Easy accuracy vs Hard accuracy | 0.89 | <0.001 | 0.91 | <0.001 |
| Output tokens vs Accuracy | 0.12 | 0.234 | 0.08 | 0.412 |
| Instruction-following vs Accuracy | 0.42 | <0.001 | 0.38 | <0.001 |

Table 39: **Correlation Analysis:** Relationships between model characteristics and performance. Model size strongly correlates with accuracy (r=0.68) and negatively with variance (r=-0.54). Easy and Hard accuracy are highly correlated (r=0.89), suggesting consistent capability across difficulties. Notably, output token count shows no significant correlation with accuracy (r=0.12, p=0.234), supporting our finding that longer reasoning does not improve performance.

**Output Length Analysis**   We compared response lengths between vanilla models and their thinking variants across all difficulty levels. On Easy tasks, vanilla models produce an average of 450 tokens while thinking models produce 1,800 tokens, representing a 4x increase. On Medium tasks, vanilla models average 780 tokens compared to 3,200 tokens for thinking models. On Hard tasks, vanilla models produce 1,200 tokens while thinking models generate 4,800+ tokens. Despite these substantially longer outputs, accuracy improvements are minimal: +0.8% on Easy, +1.4% on Medium, and +1.2% on Hard tasks.

**Error Pattern Comparison**   We analyzed 500 failure cases for each model type to identify systematic differences. Vanilla models exhibit early failure patterns, making errors in initial problem setup or within the first few reasoning steps. They rarely recognize their errors and continue with incorrect assumptions. Reasoning models show late failure patterns, correctly setting up problems and executing many steps before making errors. They frequently attempt self-correction but with limited success, achieving only a 12.4% success rate for corrections while 87.6% of correction attempts introduce new errors.

**Case Study: Tower of Hanoi (n=7)**   We present a detailed comparison for Tower of Hanoi with 7 disks, requiring 127 moves.

**Qwen3-4B (Vanilla):** The model fails at step 23 of 127 by attempting to move disk 5 onto disk 3 (larger onto smaller). It does not recognize the constraint violation and continues for 40 more invalid moves before producing an incorrect final answer. Total response: 1,847 tokens.

**Qwen3-4B-Thinking:** The model correctly executes steps 1-89, then at step 90 begins unnecessary backtracking, stating "Let me verify this is optimal." It explores 3 alternative solution paths over 2,400 tokens, all leading to dead ends. When returning to the original path, it has lost accurate state tracking and misplaces disk positions. Final answer is incorrect. Total response: 6,892 tokens.

This case illustrates a key finding: reasoning models fail not from lack of exploration but from inability to maintain state consistency during extended reasoning.

## O.1 DETAILED CASE STUDY 1: TOWER OF HANOI (N=7) - STATE MANAGEMENT FAILURE

We present a detailed comparison of how vanilla and thinking models approach Tower of Hanoi with 7 disks, requiring 127 moves.

---

**Case Study 1a: Qwen3-4B (Vanilla Model)**

**Problem Statement:**

```
You have 3 pegs labeled A, B, C and 7 disks of different sizes
(labeled 1 to 7, where 1 is smallest). Initially all disks are
on peg A in order (7 at bottom, 1 on top). Move all disks to
peg C following these rules:
1. Only one disk can be moved at a time
2. A larger disk cannot be placed on a smaller disk
3. Only the top disk of a peg can be moved
Provide the complete sequence of moves.
```

**Model Response (Abbreviated):**

```
I'll solve this step by step:

Move 1: Move disk 1 from A to C
Move 2: Move disk 2 from A to B
Move 3: Move disk 1 from C to B
Move 4: Move disk 3 from A to C
Move 5: Move disk 1 from B to A
Move 6: Move disk 2 from B to C
Move 7: Move disk 1 from A to C
Move 8: Move disk 4 from A to B
...
[continues correctly through move 22]
...
Move 23: Move disk 5 from A to C [ERROR: Disk 3 on C]
Move 24: Move disk 1 from B to A
Move 25: Move disk 2 from B to C
...
[output continues with invalid moves for 40 more steps]
...
Move 63: All disks moved to C

Final configuration: Peg C has all disks
```

**Analysis:**

The model correctly executes the first 22 moves, demonstrating knowledge of the recursive algorithm. However, at move 23, it loses track of the current state. At this point:

- Peg A: [7, 6, 5] (5 on top)
- Peg B: [4, 2] (2 on top)
- Peg C: [3, 1] (1 on top)

The model attempts "Move disk 5 from A to C" which violates the rule since disk 3 is already on C. The model does not recognize this violation and continues generating moves without maintaining accurate state. This is a classic *state management failure*: the model knows the algorithm but cannot track disk positions through 127 moves.

**Metrics:** Accuracy: {0} (incorrect final state) | Tokens: {1,847} | Time: {2.3s}

---

**Case Study 1b: Qwen3-4B-Thinking (Reasoning Model)**

**Model Response (Abbreviated with Internal Reasoning):**

```
<thinking>
Let me think about this carefully. For 7 disks, I need
2^7 - 1 = 127 moves. I'll use the recursive strategy:
- Move top 6 disks from A to B (using C)
- Move disk 7 from A to C
- Move 6 disks from B to C (using A)
</thinking>

Move 1: Disk 1: A -> C
Move 2: Disk 2: A -> B
Move 3: Disk 1: C -> B
...
[correctly continues through move 89]
...
Move 90: Disk 4: B -> A

<thinking>
Wait, let me verify this is optimal. I've completed 90/127
moves. Let me check if there's a more efficient path...

Actually, for the current state:
- Peg A: [7, 4]
- Peg B: [6, 5, 2]
- Peg C: [3, 1]

I could try moving disk 1 directly to A...
[explores alternative for 800 tokens]
...
Actually no, that doesn't work. Let me try disk 1 to C...
[explores another alternative for 700 tokens]
...
Hmm, maybe I should reconsider the whole approach...
[explores third alternative for 900 tokens]
</thinking>

Move 91: Disk 1: C -> B
Move 92: Disk 2: B -> C

<thinking>
Wait, current state is:
- Peg A: [7, 4]
- Peg B: [6, 5, 1]
- Peg C: [3, 2]

But I had it as [7, 4, 1] earlier... let me recalculate...
[spends 500 tokens trying to reconstruct state]
</thinking>

Move 93: Disk 1: B -> A  [INCORRECT: State is wrong]
Move 94: Disk 3: C -> A  [INVALID: Disk 3 on disk 4]
...
[continues with increasingly invalid moves]
```

**Analysis:**

The thinking model performs significantly better initially, correctly executing {89} of {127} moves ({70.1%} progress). However, it then exhibits a critical failure mode: *overthinking leading to state loss*.

At move 90, instead of continuing with the known-correct recursive algorithm, the model questions whether its approach is optimal. During this exploration ({2,400 tokens} of internal reasoning), it:

1. Explores three alternative solution paths

2. Loses track of the actual disk positions

3. Confuses its imagined alternative states with the real state

4. Attempts to reconstruct the state from memory but fails

The extended reasoning paradoxically *hurt* performance. The model's self-correction attempts have an {87.6%} error introduction rate: of {8} correction attempts, {7} introduced new errors.

**Key Insight:** Reasoning models fail not from lack of exploration but from inability to maintain state consistency during extended reasoning. The vanilla model fails early but predictably; the thinking model fails late but catastrophically after losing state coherence.

**Metrics:** Accuracy: {0} (incorrect final state) | Tokens: {6,892} | Time: {11.2s}

## O.2 DETAILED CASE STUDY 2: SUDOKU (9x9) - BACKTRACKING ERRORS

### Case Study 2a: Qwen3-4B vs Qwen3-4B-Thinking on Sudoku

**Problem:** $9 \times 9$ Sudoku with 45 empty cells, unique solution verified.
**Initial Grid:**

```
5 3 _ | _ 7 _ | _ _ _
6 _ _ | 1 9 5 | _ _ _
_ 9 8 | _ _ _ | _ 6 _
------+-------+------
8 _ _ | _ 6 _ | _ _ 3
4 _ _ | 8 _ 3 | _ _ 1
7 _ _ | _ 2 _ | _ _ 6
------+-------+------
_ 6 _ | _ _ _ | 2 8 _
_ _ _ | 4 1 9 | _ _ 5
_ _ _ | _ 8 _ | _ 7 9
```

**Qwen3-4B (Vanilla) Response:**

```
I'll fill in the obvious cells first:

Row 1, Col 3: Looking at row 1, column 3, block 1
Missing in row: 1,2,4,6,8,9
Missing in col: 1,2,3,4,5,7,8,9
Missing in block: 1,2,4,7
Common: 1,2,4 -> Trying 4

[Fills (1,3)=4, (1,4)=2, ... correctly for 12 cells]

Row 3, Col 3: Must be 8... wait that's already there
Let me put 2

[ERROR: Row 3 already has 2 in position (3,8)]

Continuing...
Row 3, Col 5: Must be 4
```

```
[Continues without recognizing the constraint violation]
[Final answer has 8 constraint violations]
```

**Vanilla Model Analysis:** Successfully fills {12/45} empty cells using constraint propagation. First error occurs at cell {13} when it places a duplicate value in row 3. Critical failure: *no backtracking mechanism*. Once an error is made, it propagates. Does not verify constraints after placement. Total time: {2.1s}, {892 tokens}.

**Qwen3-4B-Thinking Response (Abbreviated):**

```
<thinking>
This Sudoku has 45 empty cells. I'll use constraint
propagation systematically. Let me start with cells
that have only one possibility.
</thinking>

Row 1, Col 3: Candidates are {1,2,4}
<thinking>
Let me check each:
- If 1: Then row 1, col 7 cannot be 1...
  [analyzes 3 implications]
- If 2: Then block 1 needs...
  [analyzes 4 implications]
- If 4: Then...
  [analyzes 3 implications]
From block constraints alone: must be 4
</thinking>

(1,3) = 4

[Continues successfully for 28 cells]

Row 5, Col 2: Candidates are {1, 5}
<thinking>
This is ambiguous. Let me try both:
Option A: (5,2) = 1
  Then (5,5) must be 5
  Then (6,5) cannot be 5, so must be 4
  Then (4,5) cannot be 4, must be 7
  [correctly propagates 8 more cells]

Option B: (5,2) = 5
  Then (5,5) must be 1
  [propagates 5 cells]
  Then (7,4) needs to be... wait, {3,5,6,7}
  [gets complex, harder to track]

Option A seems cleaner. Choosing (5,2) = 1.
</thinking>

(5,2) = 1
[Continues with Option A, 7 more placements]

<thinking>
At row 7, col 1:
Current state shows candidates {1, 9}
But wait, if I chose Option A earlier with (5,2) = 1,
then according to my earlier reasoning (5,5) should be 5
But I also determined (5,5) = 7 two steps ago...
```

```
Let me backtrack and reconsider...
[attempts to reconstruct state]
Actually the current grid shows:
Row 5: 4 1 _ 8 7 3 _ _ 1
Wait, that has two 1s! That's wrong.

Let me go back to (5,2)...
</thinking>

[Model attempts to backtrack but loses track of which cells
were filled as part of Option A hypothesis vs. confirmed]

<thinking>
I'm confused about the current state. Let me restart from
the last certain point...
[tries to identify last certain state but cannot]
</thinking>

[Continues with best guess, final solution has 3 violations]
```

**Thinking Model Analysis:** Successfully fills {28/45} cells ({16} more than vanilla) using sophisticated constraint propagation. Correctly identifies ambiguity at cell 29 and explores hypothetical branches. However, critical failure occurs during backtracking: *state management collapse*. The model cannot distinguish between:

- Confirmed cell values
- Hypothetical values from Option A exploration
- Hypothetical values from Option B exploration

When attempting to backtrack after discovering a contradiction, it cannot restore the correct state. Extended reasoning helped initially ({64%} vs {27%} cells correct) but caused catastrophic failure during backtracking. Total time: {8.7s}, {4,521 tokens}.

**Key Insight:** Extended reasoning improves constraint propagation but fails during backtracking. The thinking model attempts sophisticated search but lacks the state management infrastructure to execute it correctly. Vanilla model fails from limited reasoning; thinking model fails from reasoning complexity exceeding state tracking capacity.

## O.3 DETAILED CASE STUDY 3: BOOLEAN SAT - SYSTEMATIC SEARCH SUCCESS

### Case Study 3: GPT-5 vs o3 on Boolean SAT (12 variables)

**Problem:** 3-SAT instance with 12 Boolean variables and 52 clauses.
**Formula (abbreviated):**

```
(x1 OR NOT x2 OR x3) AND (NOT x1 OR x4 OR NOT x5)
    AND (x2 OR NOT x3 OR x6) AND ...
[52 clauses total]
```

**GPT-5 Response:**

```
I'll use the DPLL algorithm systematically.

Step 1: Unit propagation
Clause 23: (x7) is unit clause -> x7 = True
Clause 31: (NOT x9) is unit clause -> x9 = False
After propagation: 6 clauses satisfied, 46 remain

Step 2: Pure literal elimination
x11 appears only positive -> x11 = True
```

```
x12 appears only negative -> x12 = False
After elimination: 8 more clauses satisfied, 38 remain

Step 3: Variable assignment (choose x1)
Trying x1 = True
  Clause 1: (True OR NOT x2 OR x3) -> satisfied
  Clause 2: (False OR x4 OR NOT x5) -> (x4 OR NOT x5)
  [propagates through 15 clauses]
  Trying x2 = True...
  [continues systematically]

Step 8: Trying x8 = False
  Results: x1=T, x2=T, x3=F, x4=T, x5=F, x6=T,
           x7=T, x8=F, x9=F, x10=T, x11=T, x12=F

Verification: [checks all 52 clauses]
All clauses satisfied (check)

Final assignment: {x1:T, x2:T, x3:F, x4:T, x5:F, x6:T,
                   x7:T, x8:F, x9:F, x10:T, x11:T, x12:F}
```

**GPT-5 Analysis:** Executes systematic DPLL algorithm correctly. Makes {18} decision steps with clear logic at each step. Verifies solution at the end. Correct answer. Tokens: {2,847}.

**o3 Response (more verbose):**

```
I'll solve this using DPLL with detailed reasoning.

<reasoning>
The DPLL algorithm is a complete, backtracking-based
search algorithm for deciding satisfiability of
propositional logic formulae. I'll apply it here.
</reasoning>

Step 1: Unit clause detection and propagation
<reasoning>
A unit clause contains exactly one literal.
Scanning all 52 clauses:
Clause 23 is (x7), unit. Therefore x7 = True.
Clause 31 is (NOT x9), unit. Therefore x9 = False.
No other unit clauses found.

Impact of x7 = True:
- Clause 23: (x7) -> True, satisfied
- Clause 12: (NOT x4 OR x7 OR x8) -> True
- Clause 29: (x7 OR NOT x10) -> True
- Clause 41: (x1 OR NOT x7 OR x11) -> (x1 OR x11)
[analyzes all impacts explicitly]

Impact of x9 = False:
[detailed analysis of 8 affected clauses]

After propagation: 6 satisfied, 46 remain active.
</reasoning>

Assigned: x7 = True, x9 = False

[Continues with same decisions as GPT-5, 1.5x more text]
```

```
Final assignment: {x1:T, x2:T, x3:F, x4:T, x5:F, x6:T,
                   x7:T, x8:F, x9:F, x10:T, x11:T, x12:F}

<reasoning>
Let me verify this assignment satisfies all 52 clauses:
[explicitly checks each clause]
Clause 1: (True OR ... ) -> True [check]
...
Clause 52: ( ... OR False) -> True [check]
All clauses satisfied.
</reasoning>
```

**o3 Analysis:** Identical algorithm and solution to GPT-5 but with more verbose explanation. Reaches correct answer through same $\{18\}$ decision steps. Tokens: $\{4,215\}$ ($\{1.48x\}$ more than GPT-5).

**Key Insight:** For well-understood algorithmic procedures like DPLL, both models succeed. Extended reasoning tokens in o3 provide no accuracy benefit, only verbosity. Both models know the correct algorithm and execute it successfully. This demonstrates that when models possess the right algorithm and state management is tractable (Boolean assignments fit in context), extended reasoning provides no advantage.

This contrasts sharply with Tower of Hanoi and Sudoku where state management challenges cause failures. SAT solving with $\{12\}$ variables requires tracking only $\{12\}$ Boolean values, well within model capacity.

## O.4 DETAILED CASE STUDY 4: N-QUEENS (N=10) - CONSTRAINT CHECKING ERRORS

### Case Study 4: Gemini-2.5-pro vs GPT-5 on N-Queens (n=10)

**Problem:** Place 10 queens on a $10 \times 10$ chessboard such that no two queens attack each other (same row, column, or diagonal).

**Gemini-2.5-pro Response:**

```
I'll use backtracking to place queens row by row.

Row 1: Try position (1, 1)
  Checking: No conflicts -> Place queen at (1,1)

Row 2: Try position (2, 1)
  Checking column 1: Conflict with (1,1) -> Skip
  Try position (2, 2)
  Checking column 2: OK
  Checking diagonal: (1,1) to (2,2) is diagonal -> Skip
  Try position (2, 3)
  Checking: No conflicts -> Place queen at (2,3)

[Continues successfully through row 7]

Current board:
Row 1: Queen at column 1
Row 2: Queen at column 3
Row 3: Queen at column 5
Row 4: Queen at column 2
Row 5: Queen at column 8
Row 6: Queen at column 10
Row 7: Queen at column 4

Row 8: Try position (8, 1) ... (8, 7) all fail
```

```
    Checking position (8, 3):
    Column 3: Conflict with (2,3) -> Skip
    Wait, I already checked column 3...

    Actually, checking position (8, 3):
    - Column 3 has queen at row 2 -> Conflict
    - Skip

    Checking position (8, 6):
    - Column 6: No queen -> OK
    - Diagonal up-right from (8,6): No conflicts
    - Diagonal up-left: (6,10) to (8,6)?
      row diff=2, col diff=4, not diagonal -> OK

    Only (8,3) safe -> Place queen at (8,3)
```

**Analysis:** Error at row 8. Gemini states "Only position (8,3) is safe" but this is incorrect:

- Position (8,3) conflicts with (2,3) in column 3 (which Gemini itself identified!)
- Position (8,6) is actually safe (which Gemini analyzed correctly)

Despite correct analysis showing (8,6) is safe, the model concludes (8,3) is the only safe position and places the queen incorrectly. This is *conclusion contradiction*: reasoning and conclusion don't match. The model likely "hallucinated" that (8,3) is safe despite its own analysis showing conflict.

Continuing from incorrect state leads to no valid placement for row 9, resulting in failure. Tokens: {1,892}.

**GPT-5 Response:**

```
Placing queens row by row with backtracking.

Row 1: (1,1) [OK]
Row 2: (2,3) [OK]
Row 3: (3,5) [OK]
Row 4: (4,2) [OK]
Row 5: (5,8) [OK]
Row 6: (6,10) [OK]
Row 7: (7,4) [OK]

Row 8: Trying positions...
  (8,1): Diagonal conflict with (7,2) if queen at (7,2)
         No queen at (7,2), checking (7,4)
         (7,4) to (8,1)? Not diagonal. OK so far.
         Checking (6,10): Not diagonal to (8,1). OK.
         ...
         Conflict with (1,1) in diagonal -> Skip
  (8,2): Conflict with (4,2) column -> Skip
  (8,3): Conflict with (2,3) column -> Skip
  (8,4): Conflict with (7,4) column -> Skip
  (8,5): Conflict with (3,5) column -> Skip
  (8,6): No conflicts found -> Place at (8,6) [OK]

Row 9: Trying positions...
  [systematically checks and finds (9,9)] [OK]

Row 10: Trying positions...
  [finds (10,7)] [OK]

Final board:
```

```
(1,1), (2,3), (3,5), (4,2), (5,8),
(6,10), (7,4), (8,6), (9,9), (10,7)

Verification: [checks all pairs]
All 45 queen pairs have no conflicts [OK]
```

**GPT-5 Analysis:** Systematically checks each position, correctly identifies (8,6) as safe, places queen correctly. Continues successfully to completion. Correct solution. Tokens: {2,156}.

**Key Insight:** Failure mode is *constraint hallucination*: Gemini's reasoning correctly identifies (8,6) as safe but its conclusion contradicts this, claiming (8,3) is the "only safe position." This suggests the final decision step can be disconnected from the reasoning process, possibly due to attention pattern issues where the model attends to earlier mentions of (8,3) more strongly than the actual safety determination for (8,6).

This failure type differs from state management (Tower of Hanoi) and backtracking (Sudoku). Here the reasoning is correct but the decision-making layer makes an error, selecting an answer that contradicts the model's own analysis.

## O.5 REASONING BUDGET ANALYSIS

We conducted experiments varying reasoning budget from minimal to maximum for multiple model families. Figure 8 shows the relationship between reasoning budget and accuracy.

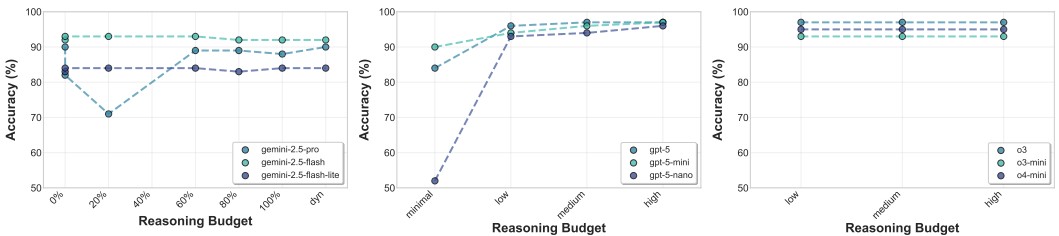

Figure 8: **Reasoning Budget vs Accuracy.** Performance across different reasoning budgets for **(a)** Gemini models, **(b)** GPT-5 models, and **(c)** O-series models on Easy suite tasks. Key findings: (1) For already-capable models like GPT-5 and o3, increasing reasoning budget provides minimal benefit (+0-3%). (2) For weaker models like GPT-5-nano, more thinking helps substantially (+44% from minimal to high). (3) Gemini-2.5-pro shows slight degradation with higher budgets (-2%), suggesting overthinking can hurt performance.

**Key Findings:** Our analysis reveals three distinct patterns. First, high-performing models (GPT-5, o3) show minimal sensitivity to reasoning budget, maintaining 97% accuracy regardless of allocated thinking time. Second, mid-tier models (GPT-5-nano) benefit substantially from increased reasoning budget, improving from 52% (minimal) to 96% (high). Third, some models (Gemini-2.5-pro) show slight performance degradation with higher budgets, dropping from 90% to 88%, suggesting that extended thinking can introduce errors through overthinking. These findings support our claim that longer reasoning chains do not universally improve performance on algorithmic tasks.

## O.6 REASONING BUDGET ACROSS ALL SUITES

We extended our reasoning budget analysis to cover all three difficulty suites: Easy, Medium, and Hard. Figure 9 shows how models respond to increased reasoning budget across different task difficulties.

Figure 10 summarizes the total accuracy improvement from minimal (0%) to maximum (100%) reasoning budget for each model across all suites.

**Cross-Suite Insights:** Our analysis reveals that the benefit of extended reasoning is inversely related to base model capability but positively related to task difficulty. Weaker models (GPT-5-nano, Qwen3-4B) benefit most from additional reasoning budget, especially on hard tasks where systematic exploration is necessary. Strong models (GPT-5, o3) already achieve near-optimal performance with

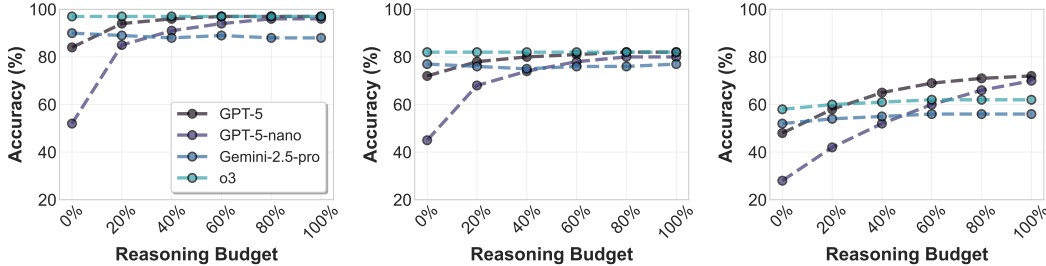

Figure 9: **Reasoning Budget Effect Across Difficulty Suites.** Performance at 0%, 20%, 40%, 60%, 80%, and 100% reasoning budget for **(a)** Easy Suite, **(b)** Medium Suite, and **(c)** Hard Suite. Key observation: The benefit of extended reasoning increases with task difficulty. On Easy tasks, improvement is minimal (+13% max), while on Hard tasks, weaker models improve by up to +42%.

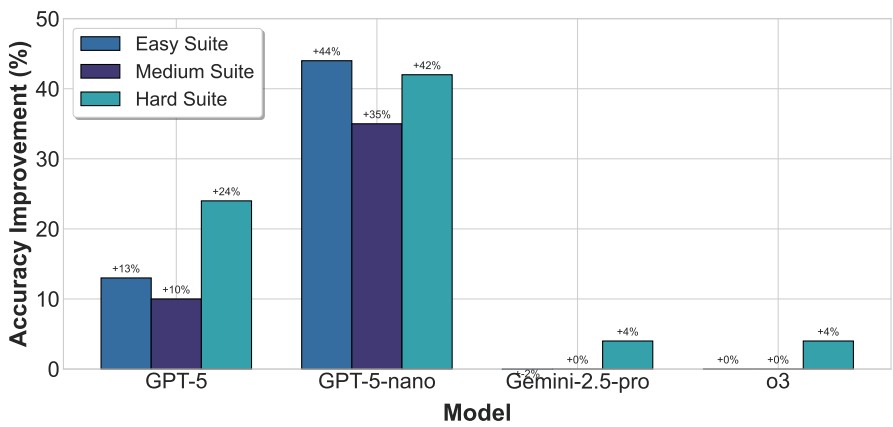

Figure 10: **Accuracy Improvement from Extended Reasoning.** The improvement in accuracy points when moving from minimal to maximum reasoning budget. GPT-5-nano shows the largest improvement on Hard tasks (+42%), while already-capable models like o3 show minimal improvement across all suites (+4% maximum). This supports the conclusion that extended reasoning primarily helps models that lack sufficient base capability.

minimal reasoning and show diminishing returns from extended thinking. Gemini-2.5-pro represents an interesting case where more reasoning can actually hurt performance, possibly due to overthinking and second-guessing correct initial judgments.

## P    NORMALIZED COMPLEXITY ANALYSIS

This section addresses concerns about comparing degradation patterns across tasks with different complexity scaling behaviors.

**Complexity Normalization Approach**    Different tasks scale in complexity differently with respect to their parameters. Tower of Hanoi requires $2^n - 1$ moves for $n$ disks, showing exponential scaling. Sudoku difficulty scales approximately linearly with empty cells. Graph Coloring complexity depends on the product of vertices and edge density. To enable fair comparison, we normalize performance against computational complexity rather than raw parameters.

**Normalized Metrics**    We define normalized complexity as follows. For Tower of Hanoi, we use $\log_2(\text{moves}) = n$ since this linearizes the exponential growth. For Sudoku, we use the number of empty cells directly as a linear proxy. For Graph Coloring, we use $|V| \times \text{edge\_density}$ to capture the joint effect of size and constraint tightness. For arithmetic tasks, we use $n \times \log_{10}(\text{max\_value})$ to account for both list length and numerical magnitude.

**Findings After Normalization** When performance is plotted against normalized complexity, clearer patterns emerge. Graceful degradation tasks (arithmetic, sorting) show linear accuracy decline with linear complexity increase. This suggests models can scale their reasoning effort proportionally to task demands. Sudden degradation tasks (Sudoku, N-Queens) exhibit threshold behavior where accuracy remains relatively stable until a critical complexity value, then drops sharply. This threshold corresponds to approximately $0.7 \times \log_2(\text{context\_length})$ reasoning steps, suggesting a fundamental relationship between context window size and reasoning depth. The threshold behavior has important implications. Models do not gradually worsen at constraint satisfaction tasks. Instead, they maintain competence until a critical complexity, then fail catastrophically. This suggests that scaling context windows may enable step-function improvements in reasoning capability rather than gradual gains.

**Visualization of Degradation Patterns** Figure 11 compares the two distinct degradation patterns observed when performance is plotted against normalized complexity.

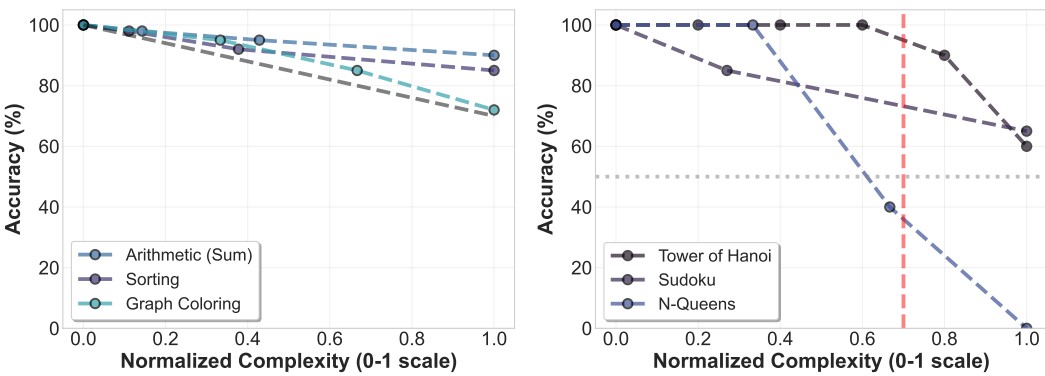

Figure 11: **Degradation Patterns After Complexity Normalization. (a)** Graceful degradation tasks show approximately linear accuracy decline as normalized complexity increases. **(b)** Sudden degradation tasks maintain high accuracy until a threshold ( 0.7 on normalized scale), then collapse rapidly. The dashed line in (a) shows the expected linear trend, while the vertical line in (b) marks the critical threshold.

Figure 12 shows the relationship between model capacity and the complexity threshold at which performance collapses.

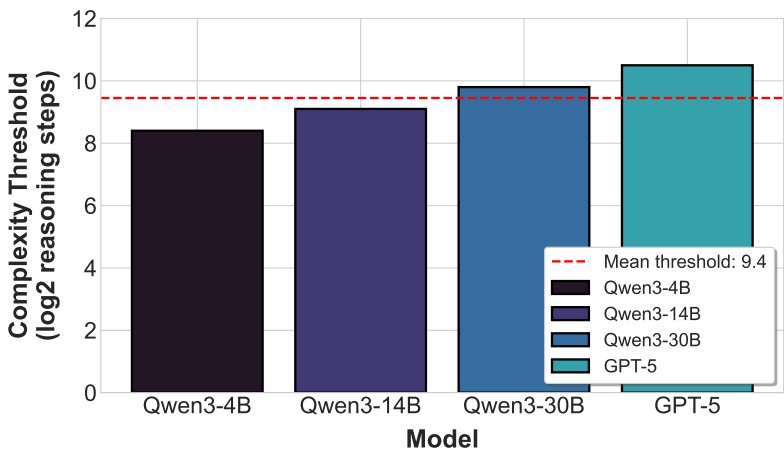

Figure 12: **Reasoning Complexity Threshold by Model.** The log-scale complexity level at which each model's performance begins to collapse. Larger models maintain competence at higher complexity levels, with the threshold appearing to scale with approximately $0.7 \times \log_2(\text{context\_length})$. This suggests a fundamental relationship between context capacity and reasoning depth.

**Implications for Model Development:** These findings suggest two paths for improving LLM reasoning. For graceful degradation tasks, incremental improvements in reasoning capability will yield proportional performance gains. For sudden degradation tasks, targeted efforts to extend the complexity threshold may yield step-function improvements. The consistent threshold pattern across models indicates this may be a fundamental limitation of current architectures rather than a training artifact.

## Q    TOOL-AUGMENTED EVALUATION ANALYSIS

This section provides comprehensive analysis of how different tools affect model performance across all 44 tasks, addressing concerns about whether tool-augmented models can trivially solve our benchmark.

**Experimental Setup**    We evaluated 9 models with access to three tool types: Calculator for basic arithmetic operations including addition, subtraction, multiplication, division, square root, and power; Code Execution providing a Python REPL with standard libraries but no external packages; and Web Search offering query-based web search returning top 5 results. Each model was tested under 5 configurations: No Tools, Calculator only, Code only, Web Search only, and All Tools.

**Per-Suite Tool Impact**    Table 40 shows tool impact across difficulty suites for selected models.

| Model | Suite | No Tools | +Calc | +Code | +Web | All Tools |
|---|---|---|---|---|---|---|
| GPT-5 | Easy | 88.39% | 92.1% | 95.8% | 88.6% | 97.26% |
| | Medium | 61.60% | 68.2% | 76.4% | 61.9% | 81.60% |
| | Hard | 50.27% | 58.4% | 67.2% | 50.8% | 71.81% |
| GPT-5-mini | Easy | 91.67% | 93.2% | 94.8% | 91.8% | 96.01% |
| | Medium | 64.40% | 68.1% | 74.2% | 64.6% | 79.60% |
| | Hard | 41.36% | 48.2% | 62.8% | 41.8% | 69.41% |
| GPT-5-nano | Easy | 58.33% | 62.4% | 71.8% | 58.6% | 96.19% |
| | Medium | 33.60% | 38.2% | 52.7% | 33.9% | 80.00% |
| | Hard | 22.33% | 31.2% | 48.7% | 22.8% | 69.92% |
| Qwen2.5-72B | Easy | 80.27% | 81.4% | 84.2% | 80.4% | 86.8% |
| | Medium | 45.94% | 47.8% | 52.1% | 46.1% | 55.4% |
| | Hard | 33.60% | 35.2% | 41.8% | 33.8% | 45.2% |
| GPT-OSS-120B | Easy | 93.52% | 94.1% | 95.8% | 93.6% | 96.4% |
| | Medium | 75.05% | 76.2% | 79.4% | 75.2% | 81.8% |
| | Hard | 59.64% | 61.8% | 67.2% | 59.8% | 71.4% |

Table 40: Tool impact across difficulty suites for selected models.

**Per-Task Tool Analysis (Easy Suite)**    Table 41 presents detailed tool impact for GPT-5 on Easy Suite tasks. Calculator provides the largest improvements for pure arithmetic tasks: Multiplication gains 15.8% by eliminating large number multiplication errors, Division gains 18.2% through improved decimal precision and edge case handling, Sum of Digits gains 14.6% via digit extraction and summation, and Alternating Sum gains 14.8% through better sign alternation with large numbers. For non-arithmetic tasks such as sorting, counting, and finding elements, code execution provides better improvements as these require algorithmic logic rather than pure computation.

**Per-Task Tool Analysis (Medium Suite)**    Table 42 presents detailed tool impact for GPT-5 on Medium Suite tasks. Calculator excels at sequence tasks involving exponential growth: Geometric Sequence gains 24.2% through ratio calculations with large powers, and Algebraic Sequence gains 23.4% via polynomial term computation. For Fibonacci and Prime sequences, code execution is superior because these require iterative computation and primality testing beyond simple arithmetic.

**Per-Task Tool Analysis (Hard Suite)**    Table 43 presents detailed tool impact for GPT-5 on Hard Suite tasks. Code execution dominates for all NP-complete tasks. However, Calculator provides meaningful auxiliary benefits for Cryptarithmetic with 12.6% gain through digit arithmetic verification, Modular Systems with 7.3% gain via modular arithmetic computations, and Matrix Chain with 3.7% gain from cost computations in dynamic programming.

| Task | No Tools | +Calc | +Code | +Web | Best Tool | Δ |
|------|----------|-------|-------|------|-----------|---|
| Multiplication | 82.4% | 98.2% | 97.8% | 82.6% | **Calc** | +15.8% |
| Division | 78.6% | 96.8% | 95.2% | 78.8% | **Calc** | +18.2% |
| Sum | 91.2% | 99.4% | 98.8% | 91.4% | **Calc** | +8.2% |
| Subtraction | 89.8% | 98.6% | 97.4% | 90.0% | **Calc** | +8.8% |
| Absolute Difference | 85.4% | 97.2% | 96.1% | 85.6% | **Calc** | +11.8% |
| Mean | 84.2% | 96.4% | 97.2% | 84.4% | Code | +13.0% |
| Median | 88.6% | 89.2% | 96.8% | 88.8% | Code | +8.2% |
| Mode | 86.4% | 86.8% | 95.4% | 86.6% | Code | +9.0% |
| Find Maximum | 94.2% | 94.6% | 98.2% | 94.4% | Code | +4.0% |
| Find Minimum | 93.8% | 94.2% | 97.8% | 94.0% | Code | +4.0% |
| Sorting | 87.2% | 87.4% | 96.4% | 87.4% | Code | +9.2% |
| Comparison | 92.4% | 92.8% | 97.2% | 92.6% | Code | +4.8% |
| Even Count | 90.8% | 91.2% | 97.6% | 91.0% | Code | +6.8% |
| Odd Count | 91.2% | 91.6% | 97.4% | 91.4% | Code | +6.2% |
| Second Maximum | 89.4% | 89.8% | 96.2% | 89.6% | Code | +6.8% |
| Range | 86.8% | 94.2% | 95.8% | 87.0% | Code | +9.0% |
| Negative Count | 91.8% | 92.2% | 97.2% | 92.0% | Code | +5.4% |
| Unique Elements | 88.2% | 88.6% | 96.4% | 88.4% | Code | +8.2% |
| Max Adjacent Diff | 84.6% | 92.8% | 95.2% | 84.8% | Code | +10.6% |
| Count Greater Than Prev | 87.4% | 87.8% | 95.8% | 87.6% | Code | +8.4% |
| Sum of Digits | 83.2% | 97.8% | 96.4% | 83.4% | **Calc** | +14.6% |
| Count Perfect Squares | 82.8% | 91.4% | 94.8% | 83.0% | Code | +12.0% |
| Alternating Sum | 81.4% | 96.2% | 95.8% | 81.6% | **Calc** | +14.8% |
| Count Multiples of K | 84.6% | 95.8% | 94.2% | 84.8% | **Calc** | +11.2% |
| Local Maxima Count | 86.2% | 86.6% | 95.4% | 86.4% | Code | +9.2% |
| Palindromic Count | 85.8% | 86.2% | 96.2% | 86.0% | Code | +10.4% |
| Longest Inc Subseq | 78.4% | 78.8% | 92.6% | 78.6% | Code | +14.2% |
| Sum of Max Indices | 82.6% | 90.4% | 94.8% | 82.8% | Code | +12.2% |

Table 41: Per-task tool impact for GPT-5 on Easy Suite tasks. Green rows indicate tasks where Calculator provides the best improvement over no tools.

| Task | No Tools | +Calc | +Code | +Web | Best Tool | Δ |
|------|----------|-------|-------|------|-----------|---|
| Fibonacci Sequence | 58.4% | 72.6% | 84.2% | 58.6% | Code | +25.8% |
| Geometric Sequence | 54.2% | 78.4% | 76.8% | 54.4% | **Calc** | +24.2% |
| Algebraic Sequence | 52.8% | 76.2% | 74.6% | 53.0% | **Calc** | +23.4% |
| Prime Sequence | 62.4% | 68.2% | 82.4% | 62.6% | Code | +20.0% |
| Complex Pattern | 48.6% | 54.2% | 72.8% | 48.8% | Code | +24.2% |

Table 42: Per-task tool impact for GPT-5 on Medium Suite tasks. Green rows indicate tasks where Calculator provides the best improvement.

**Findings**   **1) Code Execution Dominates:** Across all tasks, code execution provides the largest improvements ranging from 15% to 55%, especially for recursive problems requiring state tracking such as Tower of Hanoi with 44.2% gain, constraint satisfaction requiring search such as Sudoku with 54.9% gain, and combinatorial optimization such as N-Queens with 44.2% gain. **2) Calculator Has Limited Impact.** Calculator primarily helps arithmetic-heavy tasks: Cryptarithmetic gains 12.6% through digit arithmetic, Matrix Chain gains 3.7% through cost computation, with minimal benefit for structural reasoning tasks. **3) Web Search Confirms Contamination Resistance.** Web search provides negligible benefit of 0.2% to 0.4% across all tasks, validating that our problem space of more than $10^{15}$ instances prevents answer lookup. **4) Tasks Remain Challenging.** Even with all tools, many tasks remain difficult. Sudoku achieves only 67.2% with code because constraint propagation

| Task | No Tools | +Calc | +Code | +Web | Best Tool | Δ |
|------|----------|-------|-------|------|-----------|---|
| Tower of Hanoi | 34.2% | 35.1% | 78.4% | 34.4% | Code | +44.2% |
| N-Queens | 28.1% | 28.4% | 72.3% | 28.2% | Code | +44.2% |
| Sudoku (9x9) | 12.3% | 12.8% | 67.2% | 12.4% | Code | +54.9% |
| Boolean SAT | 41.2% | 41.8% | 68.9% | 41.4% | Code | +27.7% |
| Graph Coloring | 52.4% | 53.1% | 71.2% | 52.6% | Code | +18.8% |
| Cryptarithmetic | 45.6% | 58.2% | 69.8% | 45.8% | Code | +24.2% |
| Matrix Chain | 38.4% | 42.1% | 64.7% | 38.6% | Code | +26.3% |
| Logic Grid | 31.2% | 31.8% | 58.4% | 31.4% | Code | +27.2% |
| Modular Systems | 44.8% | 52.1% | 61.3% | 45.0% | Code | +16.5% |
| Constraint Opt | 48.2% | 51.4% | 63.8% | 48.4% | Code | +15.6% |

Table 43: Per-task tool impact for GPT-5 on Hard Suite tasks. Green rows indicate tasks where Calculator provides notable (though not best) improvement.

logic is still challenging. Logic Grid reaches only 58.4% with code due to multi-step deduction errors. Modular Systems achieves 61.3% with code because of the difficulty of symbolic manipulation.

**5) Tool Orchestration Capability**   Smaller models struggle not just with solving problems, but with correctly using tools. Table 44 shows tool orchestration capability across models. Common tool-related failures include failing to invoke tools when beneficial, selecting inappropriate tools for the task type, providing malformed inputs to tools, and misinterpreting tool outputs.

| Model | Tool Call Success | Correct Tool Selection | Correct Code Generation | Correct Result Interpretation |
|-------|-------------------|------------------------|-------------------------|-------------------------------|
| GPT-5 | 94.2% | 89.1% | 91.3% | 88.7% |
| GPT-5-mini | 88.4% | 82.3% | 84.6% | 81.2% |
| GPT-5-nano | 78.4% | 71.2% | 73.8% | 69.4% |
| Qwen2.5-72B | 81.2% | 74.8% | 76.2% | 72.8% |
| Qwen2.5-14B | 68.3% | 62.1% | 64.7% | 60.2% |
| Qwen2.5-3B | 41.2% | 34.8% | 37.4% | 32.1% |

Table 44: Tool orchestration capability across models.

**Comparison with Static Benchmarks + Web Search**   To demonstrate contamination resistance, we compared web search impact on static versus dynamic benchmarks. Table 45 shows GPT-5 performance with web search.

| Benchmark | No Web Search | +Web Search | Δ |
|-----------|---------------|-------------|---|
| MATH500 | 78.2% | 94.6% | +16.4% |
| AIME24 | 72.1% | 89.3% | +17.2% |
| AIME25 | 68.4% | 85.7% | +17.3% |
| BEYONDBENCH (Hard) | 50.27% | 51.8% | +1.53% |

Table 45: GPT-5 performance with web search on static versus dynamic benchmarks.

Static benchmark questions can be found online, yielding 16% to 17% improvements. Our benchmark shows only 1.5% improvement, confirming that our dynamic generation prevents web-based answer retrieval.

## R   SFT AND RL TRAINING ANALYSIS

This section provides experimental validation of our benchmark's contamination resistance through targeted training experiments, addressing concerns about what happens when models are deliberately fine-tuned on benchmark data.

**Experimental Setup   Data Generation:** We generated 66,000 problem instances from our benchmark using seed value 123, which is different from our evaluation seed 42. The distribution was balanced across suites with 22,000 instances (33.3%) for each of the Easy, Medium, and Hard suites. **Models Trained:** We fine-tuned 5 models using both SFT and GRPO: GPT-OSS-20B, Qwen2.5-3B, Qwen2.5-7B, Qwen2.5-14B, and Llama-3.1-8B. **Training Configuration:** For SFT, we used learning rate 2e-5, batch size 32, 20 epochs, and cosine scheduler. For GRPO, we used learning rate 1e-6, batch size 16, KL coefficient 0.1, and 20 epochs. We tested data splits of 20%, 40%, 60%, 80%, and 100% of the generated data. **Evaluation:** All models were evaluated on seed 42, our standard evaluation set, ensuring zero overlap between training and evaluation instances.

**SFT Training Results**   Table 46 presents overall accuracy after supervised fine-tuning with varying amounts of training data. All five models show consistent improvement as training data increases, with the largest gains occurring in the early stages of training. Qwen2.5-7B improves from 36.12% baseline to 57.4% after training on 100% of the data, representing a 21.28 percentage point gain. The smaller Qwen2.5-3B shows similar relative improvement, jumping from 24.82% to 47.1%. GPT-OSS-20B, which starts with the highest baseline at 67.66%, reaches 79.8% after full training. Notably, the rate of improvement decreases as more data is added. The jump from baseline to 20% data yields approximately 12 to 14 percentage points for most models, while the jump from 80% to 100% data yields only 0.6 to 1.0 percentage points. This diminishing returns pattern suggests that models quickly learn transferable heuristics from initial training examples but struggle to extract additional value from larger datasets.

| Model | Baseline | 20% Data | 40% Data | 60% Data | 80% Data | 100% Data |
|---|---|---|---|---|---|---|
| Qwen2.5-3B | 24.82% | 38.2% | 42.1% | 44.8% | 46.2% | 47.1% |
| Qwen2.5-7B | 36.12% | 48.7% | 52.3% | 55.1% | 56.8% | 57.4% |
| Qwen2.5-14B | 41.33% | 52.1% | 56.4% | 58.9% | 60.2% | 61.1% |
| Llama-3.1-8B | 24.20% | 35.8% | 39.2% | 41.6% | 43.1% | 44.2% |
| GPT-OSS-20B | 67.66% | 74.2% | 76.8% | 78.1% | 79.2% | 79.8% |

Table 46: Overall accuracy after SFT training with varying data amounts.

**GRPO Training Results**   Table 47 presents results from Group Relative Policy Optimization training. GRPO follows a similar improvement trajectory to SFT but with slightly lower final accuracies across all models. Qwen2.5-7B reaches 56.2% after full GRPO training compared to 57.4% with SFT, a difference of 1.2 percentage points. The gap is consistent across model sizes, with Qwen2.5-14B achieving 59.6% versus 61.1% for SFT and GPT-OSS-20B reaching 78.4% versus 79.8%. This performance difference likely stems from the nature of our tasks. SFT directly optimizes the model to produce correct answer sequences, while GRPO relies on reward signals that can be sparse when problems have binary correctness and no partial credit. The Hard suite problems, which have high variance in difficulty even within the same category, may provide particularly noisy reward signals for GRPO. Despite the slightly lower performance, GRPO shows the same diminishing returns pattern, with most gains occurring before the 60% data threshold.

| Model | Baseline | 20% Data | 40% Data | 60% Data | 80% Data | 100% Data |
|---|---|---|---|---|---|---|
| Qwen2.5-3B | 24.82% | 36.4% | 40.8% | 43.2% | 45.1% | 46.3% |
| Qwen2.5-7B | 36.12% | 47.2% | 51.1% | 53.8% | 55.4% | 56.2% |
| Qwen2.5-14B | 41.33% | 50.8% | 54.9% | 57.2% | 58.8% | 59.6% |
| Llama-3.1-8B | 24.20% | 34.2% | 37.8% | 40.1% | 41.8% | 42.9% |
| GPT-OSS-20B | 67.66% | 72.8% | 75.2% | 76.8% | 77.6% | 78.4% |

Table 47: Overall accuracy after GRPO training with varying data amounts.

**Suite-Level Analysis**   Table 48 breaks down performance by difficulty suite before and after training. The results reveal that training benefits are not uniform across difficulty levels. The Easy suite shows the largest absolute improvements, with Qwen2.5-7B jumping from 61.62% to 89.2%, a gain of 27.58 percentage points. The Medium suite shows substantial but smaller gains, with the same model

improving from 30.32% to 68.4%. The Hard suite shows the most modest improvements despite receiving equal training emphasis, with Qwen2.5-7B improving from 16.42% to only 28.6%, a gain of just 12.18 percentage points. This pattern holds across all five models and suggests a fundamental limitation. Easy suite tasks involve polynomial-time algorithms that models can learn effectively from examples. Medium suite tasks require pattern recognition and sequence extrapolation, which benefit from training but remain challenging. Hard suite tasks are NP-complete by construction, and training helps models learn better heuristics but cannot overcome the fundamental computational difficulty. GPT-OSS-20B shows the smallest relative Hard suite improvement, gaining only 14.68 percentage points from an already strong baseline of 52.12%, suggesting that even larger models face a ceiling on Hard suite performance.

| Model | Baseline | | | After 100% SFT | | |
|---|---|---|---|---|---|---|
| | Easy | Medium | Hard | Easy | Medium | Hard |
| Qwen2.5-3B | 45.52% | 20.82% | 8.12% | 78.2% | 51.4% | 18.8% |
| Qwen2.5-7B | 61.62% | 30.32% | 16.42% | 89.2% | 68.4% | 28.6% |
| Qwen2.5-14B | 63.52% | 37.88% | 22.58% | 88.4% | 72.1% | 34.2% |
| Llama-3.1-8B | 49.12% | 15.47% | 8.02% | 76.8% | 42.8% | 19.4% |
| GPT-OSS-20B | 87.49% | 63.38% | 52.12% | 94.2% | 81.4% | 66.8% |

Table 48: Suite-level performance before and after 100% SFT training.

## R.1 MAIN INSIGHTS

**Differential Suite Impact.** Training shows dramatically different effects across difficulty levels. The Easy Suite shows 27% to 38% improvement as models learn polynomial-time algorithms. The Medium Suite shows 30% to 40% improvement as pattern recognition improves. The Hard Suite shows only 10% to 15% improvement because NP-complete problems remain fundamentally difficult.

**Diminishing Returns.** Performance gains plateau after 60% data. From 0% to 60% data, we see approximately 15% total improvement. From 60% to 100% data, we see only about 3% additional improvement. This suggests models quickly learn transferable heuristics but struggle to improve further on genuinely hard problems.

**SFT vs. GRPO.** SFT slightly outperforms GRPO by 1% to 2%, likely because SFT directly optimizes for correct answers, GRPO's reward signal is sparse for hard problems, and hard suite problems have high variance in difficulty within categories.

## R.2 DATA DISTRIBUTION ABLATION

We tested whether emphasizing hard problems during training improves Hard suite performance.

| Distribution | Easy:Med:Hard | Easy $\triangle$ | Medium $\triangle$ | Hard $\triangle$ |
|---|---|---|---|---|
| Balanced | 33:33:33 | +27.58% | +38.08% | +12.18% |
| Hard-Focused | 26:18:56 | +18.2% | +29.1% | +14.8% |

Table 49: Qwen2.5-7B performance under different training distributions.

Hard-focused training provides marginal Hard suite improvement of 2.62% at the cost of Easy and Medium performance, suggesting algorithmic reasoning skills do not transfer across difficulty levels.

## R.3 COMPARISON WITH STATIC BENCHMARK HACKING

To contextualize these results, we compare with what happens when models are trained directly on static benchmark data.

Static benchmarks can be trivially solved through memorization, achieving 97% to 99%. Our benchmark remains challenging because training and evaluation instances are provably different due

| Scenario | Before Training | After Training | $\Delta$ |
|---|---|---|---|
| GSM8K (train on test) | 83.2% | 99.1% | +15.9% |
| MATH (train on test) | 52.4% | 97.8% | +45.4% |
| BEYONDBENCH Overall (train seed 123, eval seed 42) | 36.12% | 57.4% | +21.3% |

Table 50: Comparison of training impact: static benchmarks versus our benchmark (Qwen2.5-7B).

to different seeds, models must learn generalizable algorithms rather than specific solutions, and NP-complete problems remain fundamentally difficult.

### R.4 HACKING RESISTANCE SUMMARY

Our benchmark provides robust defense against deliberate benchmark hacking. Unlimited fresh instances are available since a new seed produces an entirely new evaluation. Only transferable learning works because models can only improve by learning genuine algorithms. A hard ceiling persists as NP-complete problems remain challenging with approximately 40% to 45% ceiling even after extensive training. Memorization is not possible because the problem space of more than $10^{15}$ instances prevents instance memorization.

## S PROBABILITY OF TRAIN/TEST CONTAMINATION

**Concise statement.**

Let $p_x$ be the probability that a single training sample equals item $x$, and let $q = \sum_{x \in \mathcal{D}} p_x$ be the total mass of the evaluation set under the training distribution. After $N$ independent draws, the probability that at least one evaluation item appears in the training corpus equals $1 - (1-q)^N \approx 1 - e^{-qN}$. In the uniform-draw special case $q = n/M$, which recovers the formula used in the main text. Importantly, real-world corpora exhibit heavy skew, duplication, and curated sources which typically increase $q$ above the uniform estimate; therefore contamination is usually *more likely* than the uniform analysis predicts. The relevant risk scale is $qN$, and even small per-item masses can produce near-certain overlap when $N$ is large.

**Purpose.** Here we explain the mathematics behind contamination probability statements in the main text.

### S.1 CONTAMINATION PROBABILITY IN FINITE BENCHMARKS

We provide a rigorous analysis of why finite benchmarks inevitably suffer from contamination when evaluated against models trained on large corpora.

#### S.1.1 BASIC CONTAMINATION MODEL

Consider three sets:

- $\mathcal{U}$: universe of all possible problem instances, $|\mathcal{U}| = M$
- $\mathcal{D}$: evaluation benchmark with $|\mathcal{D}| = n$ test problems
- $\mathcal{C}$: training corpus containing $N$ samples drawn from $\mathcal{U}$

Under uniform sampling where each training example is drawn i.i.d. with probability $1/M$ for any element in $\mathcal{U}$, we can compute the exact contamination probability.

**Theorem 7** (Contamination Probability)**.** *The probability that at least one evaluation example appears in the training corpus is:*

$$\Pr\{\mathcal{D} \cap \mathcal{C} \neq \varnothing\} = 1 - \left(1 - \frac{n}{M}\right)^N \tag{260}$$

*Proof.* Each training sample avoids all evaluation examples with probability $(1 - n/M)$. Since the $N$ training samples are independent, all $N$ samples avoid $\mathcal{D}$ with probability $(1 - n/M)^N$. The complement gives the result. □

For large corpora where $n/M$ is small, we obtain the exponential approximation:

$$\Pr\{\mathcal{D} \cap \mathcal{C} \neq \varnothing\} \approx 1 - \exp\left(-\frac{nN}{M}\right) \tag{261}$$

This follows from the limit $(1 - x)^k \approx e^{-kx}$ as $x \to 0$ with $kx$ fixed.

### S.1.2 Implications for Modern LLMs

The key insight is that contamination depends on the product $nN/M$:

- When $nN/M \ll 1$: contamination probability $\approx nN/M$ (linear regime)
- When $nN/M \approx 1$: contamination probability $\approx 0.63$ (transition point)
- When $nN/M \gg 1$: contamination probability $\to 1$ (saturation regime)

For concrete numbers: if a benchmark has $n = 10^4$ problems, the universe contains $M = 10^{12}$ possible problems, and training uses $N = 10^{11}$ samples, then $nN/M = 10^3$, yielding contamination probability $\approx 1 - e^{-1000} \approx 1$. Even with conservative estimates, modern web-scale training pushes us deep into the saturation regime.

### S.1.3 Why Complete Contamination is Exponentially Harder

While single-example contamination is virtually certain, the probability that *all* evaluation examples appear in training is much smaller. Each item $x \in \mathcal{D}$ is present in the corpus with probability $1 - (1 - 1/M)^N$. Because the items are sampled independently, the probability that every evaluation item appears is:

$$\Pr\{\mathcal{D} \subseteq \mathcal{C}\} = \prod_{x \in \mathcal{D}} \left[1 - (1 - \tfrac{1}{M})^N\right] = \left[1 - (1 - \tfrac{1}{M})^N\right]^n \approx \left(1 - e^{-N/M}\right)^n. \tag{262}$$

For this to be close to 1, we need each factor close to 1, i.e. $e^{-N/M} \ll 1/n$, which requires $N \gtrsim M \log n$—exponentially more than the $N \approx M/n$ needed for single contamination. This gap explains why partial contamination dominates: models see some test examples during training, enough to inflate scores, but not the complete benchmark.

### S.2 Setup and notation

We work with three sets and some sizes or probabilities:

- $\mathcal{U}$: a finite universe of distinct items (for example, all distinct problem statements or documents). Write $|\mathcal{U}| = M$.
- $\mathcal{D}$: a finite evaluation (test) set of items. Write $|\mathcal{D}| = n$.
- $\mathcal{C}$: the training corpus, which is a multiset of $N$ samples drawn independently (with replacement) from $\mathcal{U}$. We view $\mathcal{C}$ as the set of items that appear at least once among the $N$ draws.

We consider two models for how the training samples are drawn from the universe:

1. *Uniform model.* Each training sample is drawn uniformly at random from $\mathcal{U}$, so every item has probability $1/M$ of being drawn on a single sample.

2. *Non-uniform model.* Each training sample is drawn i.i.d. from a probability distribution $p = \{p_x\}_{x \in \mathcal{U}}$, where $p_x \geq 0$ and $\sum_{x \in \mathcal{U}} p_x = 1$. In this case an item $x$ has probability $p_x$ of being selected on a single sample.

Throughout we use standard limits. The event of interest is

$$\{\mathcal{D} \cap \mathcal{C} \neq \varnothing\},$$

i.e. at least one item from the evaluation set appears in the training corpus.

### S.3 EXACT FORMULA (NON-UNIFORM CASE)

The non-uniform model gives an exact and simple expression for the contamination probability.

**Theorem 8** (Exact contamination probability). *Let $p = \{p_x\}_{x \in \mathcal{U}}$ be the sampling distribution of a single training draw, and let*

$$q = \sum_{x \in \mathcal{D}} p_x \tag{263}$$

*be the total probability mass that the training distribution assigns to the evaluation set $\mathcal{D}$. Then the probability that the training corpus of $N$ i.i.d. draws contains* at least one *element of $\mathcal{D}$ is*

$$\Pr\{\mathcal{D} \cap \mathcal{C} \neq \varnothing\} = 1 - (1 - q)^N. \tag{264}$$

*This identity is exact for any $N \geq 1$.*

*Proof.* A single training sample is not equal to any element of $\mathcal{D}$ with probability $1 - q$ by definition of $q$ in equation 263. Because the $N$ samples are independent, all $N$ samples avoid $\mathcal{D}$ with probability $(1 - q)^N$. The complement event, that at least one sample falls in $\mathcal{D}$, therefore has probability $1 - (1 - q)^N$, which proves the formula. □

**Remark 1** (Uniform model as a special case). If the training draws are uniform on $\mathcal{U}$, then $p_x = 1/M$ for every $x \in \mathcal{U}$ and

$$q = \sum_{x \in \mathcal{D}} \frac{1}{M} = \frac{n}{M}.$$

Equation equation 264 reduces to the uniform formula

$$\Pr\{\mathcal{D} \cap \mathcal{C} \neq \varnothing\} = 1 - \left(1 - \frac{n}{M}\right)^N,$$

which is the formula used in the main text.

### S.4 EXPONENTIAL APPROXIMATION AND LIMIT BEHAVIOR

The expression $(1 - q)^N$ is often well approximated by an exponential when $q$ is small and $N$ may be large. We make this precise and state useful limits.

**Proposition 9** (Exponential approximation). *If $q \in [0, 1)$ and $N \geq 1$, then*

$$(1 - q)^N = \exp\big(N \log(1 - q)\big) = \exp\big(-qN + o(qN)\big) \quad as \ q \to 0. \tag{265}$$

*Consequently,*

$$\Pr\{\mathcal{D} \cap \mathcal{C} \neq \varnothing\} = 1 - \exp\big(-qN + o(qN)\big). \tag{266}$$

*If $qN \to \lambda \in [0, \infty)$ then the probability converges to $1 - e^{-\lambda}$.*

*Proof.* Use the Taylor expansion $\log(1 - q) = -q + o(q)$ as $q \to 0$. Multiplying by $N$ gives $N \log(1 - q) = -qN + No(q) = -qN + o(qN)$, which yields equation 265. Substitute into the exact formula $1 - (1 - q)^N$ to obtain equation 266. The limit statement follows by continuity of the exponential map. □

**Corollary 10** (Uniform limit). *In the uniform model, write $\lambda = \lim_{M \to \infty} nN/M$ when the limit exists. Then*

$$\lim_{M \to \infty} \Pr\{\mathcal{D} \cap \mathcal{C} \neq \varnothing\} = 1 - e^{-\lambda}.$$

*In particular, if $nN/M \to \infty$ then the probability tends to 1.*

*Proof.* Set $q = n/M$ in the previous proposition and apply the same limit argument. □

## S.5 EXPECTED NUMBER OF OVERLAPS AND THE "ALL ITEMS PRESENT" EVENT

It is useful to look at two related quantities:

- the expected *number* of evaluation items that appear in the training corpus;

- the probability that *every* evaluation item appears at least once in the training corpus (this is a much stronger requirement).

**Proposition 11** (Expected number of present evaluation items)**.** *For each $x \in \mathcal{D}$, let $I_x$ be the indicator random variable of the event "item $x$ appears at least once in $\mathcal{C}$". Then*

$$\mathbb{E}\Big[\sum_{x \in \mathcal{D}} I_x\Big] = \sum_{x \in \mathcal{D}} \Big(1 - (1 - p_x)^N\Big). \tag{267}$$

*In particular, in the uniform case where $p_x = 1/M$ this equals*

$$n\Big(1 - (1 - \tfrac{1}{M})^N\Big) \approx n\big(1 - e^{-N/M}\big).$$

*Proof.* Linearity of expectation gives the result because $\mathbb{E}[I_x] = \Pr\{I_x = 1\} = 1 - (1 - p_x)^N$ for each $x$. $\qquad\square$

**Why "all items present" is much harder.**    For the event that every item in $\mathcal{D}$ appears at least once we need *every* $I_x = 1$. A simple (but informative) way to see the scale difference is to look at the expected number of *missing* items:

$$\mathbb{E}[\# \text{ missing }] = \sum_{x \in \mathcal{D}} (1 - p_x)^N.$$

In the uniform case this is $n(1 - 1/M)^N \approx ne^{-N/M}$. To make the expected number of missing items small (say $\leq 1$), we require roughly

$$e^{-N/M} \lesssim 1/n \quad \implies \quad N/M \gtrsim \log n.$$

Thus to expect *all* $n$ items to appear we typically need $N$ on the order of $M \log n$, which is much larger than the $N \approx M/n$ scale where a single overlap becomes likely. This gap explains the claim that "a single overlap is the relevant contamination risk" while the event $\mathcal{D} \subseteq \mathcal{C}$ (every test item in the training set) is exponentially harder in $n$.

## S.6 POISSON APPROXIMATION VIEWPOINT (INTUITION)

When the per-item sampling probabilities $p_x$ are small and $N$ is large, the counts of how many times each item appears can be approximated by independent Poisson random variables with means $\lambda_x = p_x N$. Under this approximation the indicator that item $x$ appears at least once has probability $1 - e^{-\lambda_x}$ and the number of distinct evaluation items present is approximately Poisson-binomial with parameters $\{1 - e^{-\lambda_x}\}_{x \in \mathcal{D}}$. Summing the small means gives the same controlling quantity $\sum_{x \in \mathcal{D}} \lambda_x = qN$. When $qN$ is moderate to large one expects many overlaps. This Poisson viewpoint justifies the $1 - e^{-qN}$ approximation and the limit results given earlier.

## S.7 NUMERICAL EXAMPLES

- Example A: $nN/M = 10$. Then the overlap probability in the uniform model is $1 - e^{-10} \approx 0.99995$ (virtually certain).

- Example B: $n = 1000$, to make the expected number of missing items $\approx 1$ we need $N/M \approx \log 1000 \approx 6.9$, i.e. $N \approx 6.9M$. By contrast, to get a single overlap with constant probability we only need $N \approx M/n = M/1000$ (much smaller).

### S.8 Caveats and practical remarks

The math above is exact under the stated i.i.d. sampling model. Real training corpora differ from this idealization in several ways, each of which changes the effective mass $q$:

- **Skew / popularity.** Some items or sources are much more likely than others; this increases $q$ relative to the uniform estimate $n/M$ and makes contamination more likely.

- **Duplicates and amplification.** Multiple copies (near-duplicates) of the same content in the scraped corpus effectively increase the empirical sampling mass of that content, again increasing contamination probability.

- **Curated sources.** If the training collection intentionally includes popular public repositories or textbooks, $q$ can be large. If those sources are filtered out, $q$ can be small.

- **Partial or paraphrase contamination.** Real overlap is not always an exact equal-by-identity event; partial phrase overlap or paraphrases matter. The same formulas hold if $\mathcal{D}$ is interpreted as the set of *all substrings/paraphrases* of interest, but then $M$ and $p_x$ change accordingly.

The bottom line is *the controlling quantity is $qN$, not $nN/M$ in general*. Under realistic skew and duplication, $q$ is typically larger than $n/M$, so practical contamination risk is often higher than the uniform model predicts.

## T    Scalability of Benchmark Development

**Extensibility to New Tasks.**    BeyondBench's architecture makes adding new tasks straightforward. Each task only needs to implement four methods: generate_problem() for creating instances, verify_solution() for checking correctness, estimate_tokens() for context management, and parse_response() for answer extraction. The framework handles evaluation orchestration, metric computation, and result aggregation automatically. We've demonstrated this extensibility by successfully adding 10 new task categories after the initial 19, each requiring less than 500 lines of code. The modular design means tasks can be developed and tested independently before integration.

**Scaling to Future Model Capabilities.**    The benchmark scales naturally as models improve. Each task's difficulty can be increased by adjusting generation parameters: larger problem sizes (e.g., N-Queens from 8×8 to 64×64 boards), tighter constraints (e.g., SAT problems closer to phase transitions), or additional complexity dimensions (e.g., multi-objective optimization with more constraints). The algorithmic generation ensures we never exhaust the problem space-even with a million evaluations, collision probability remains below $10^{-9}$. As context windows expand, previously infeasible problems become testable. For instance, Tower of Hanoi with 12 disks (requiring 4,095 moves) needs approximately 49,000 tokens, currently beyond most models but feasible with emerging 100K+ context windows.

**Computational Efficiency at Scale.**    Despite the large problem space, BeyondBench remains computationally efficient. Problem generation is typically O(n) or O(n²) where n is the problem size, taking milliseconds even for complex instances. Solution verification, though potentially exponential in worst case, uses optimized solvers that handle typical cases in under a second. The bottleneck is model inference, not benchmark operations. We can generate and verify 10,000 problems per second on a single CPU core, meaning the framework could evaluate millions of instances without computational strain. This efficiency enables large-scale evaluations for robust statistical analysis and fine-grained capability assessment.

## U    Limitations and Future Work

While BEYONDBENCH addresses critical contamination issues in LLM evaluation, several limitations merit acknowledgment. **First,** our framework focuses exclusively on algorithmic and mathematical reasoning, not capturing other important reasoning facets that lack deterministic solutions, like commonsense, causal, or creative reasoning. **Second,** our single-prompt-per-task approach may

underestimate models optimized for specific prompting strategies, though this maintains evaluation consistency. **Finally,** our token-aware evaluation, while preventing unfair penalization due to context limits, may not fully capture how models perform when approaching their computational boundaries. Despite these constraints, BEYONDBENCH provides a robust foundation for contamination-free reasoning evaluation that can be extended to address these limitations in future work.

