# OpenReview forum: "BeyondBench: Contamination-Resistant Evaluation of Reasoning in Language Models"
_ICLR.cc/2026/Conference — ICLR 2026 Poster_

### Official Review · Reviewer_YAiF · 2025-10-29

**Soundness:** 4
**Presentation:** 3
**Contribution:** 3
**Rating:** 8
**Confidence:** 4

**Summary:**

This paper introduces BeyondBench, a novel evaluation framework designed to address the critical issue of data contamination in LLM assessment.

The core problem BeyondBench tackles is that traditional, static benchmarks are often found on the internet and inadvertently included in LLM training data. This leads to models achieving high scores through simple recall rather than demonstrating genuine reasoning capabilities. It becomes unclear whether a model is solving a problem or merely retrieving a memorized answer. BeyondBench solves this by implementing a benchmark-free evaluation paradigm based on algorithmic problem generation. Instead of relying on a fixed set of questions, the framework dynamically and perpetually generates new, unique, mathematically grounded test problems on the fly. This "freshness" ensures that no test instance has been seen during the model’s training, thereby eliminating the risk of data contamination.

The primary contribution of this work is the development of a contamination-resistant and reliable method for testing the core reasoning skills of LLMs. By covering a variety of difficulty levels and task types, BeyondBench provides a dynamic, rigorous, and fairer mechanism to evaluate and compare the true intelligence and problem-solving capacity of different language models, moving the field of LLM assessment beyond the limitations of static benchmarks.

**Strengths:**

1. The paper tackles the critical and growing issue of data contamination in LLM evaluation. It provides a robust solution to move past static benchmarks。 This shift is vital for accurately assessing the true capabilities and progress of frontier language models.

2.  Excellent Execution and Contamination-Free Design. By employing algorithmic problem generation, it ensures that every test instance is unique, mathematically grounded, and perpetually fresh, thereby guaranteeing a contamination-free evaluation. This design covers a wide range of difficulty levels and task types, making it a rigorous and challenging test of core reasoning skills.

3. The methodology is presented clearly and logically, effectively demonstrating how the new framework can accurately distinguish between different models' true problem-solving capabilities.

**Weaknesses:**

1.  Reduced User-Friendliness and Increased Score Variance: Compared to using static datasets, dynamically generated evaluations are inherently less user-friendly for the general public and researchers. They require a bit more complex infrastructure for on-the-fly problem creation and may lead to higher score variance between evaluations due to the non-static nature of the test instances, making direct comparisons of models more challenging than with a fixed set.

2.  Missing Discussion of Prior Work: The paper's core idea—using dynamically generated environments/problems for evaluation—is not entirely novel, especially in related fields like RL and other LLM evaluation efforts (e.g., Reasoning Gym [1]).

References

1. Stojanovski, Zafir, et al. "REASONING GYM: Reasoning Environments for Reinforcement Learning with Verifiable Rewards." arXiv preprint arXiv:2505.24760 (2025).

**Questions:**

1. Will programmatically generated problems bias towards LLMs with coding tools capabilities? Since the problems are generated via code, is there a risk that LLMs with strong coding skills will have an unfair advantage in solving these problems compared to those without such capabilities?

---

> ### Author Response · Authors · 2025-11-25
> **Official Response by Authors to Reviewer YAiF (1/3)**
>
> We sincerely thank the Reviewer for their positive evaluation and recognition that our paper tackles the **critical issue of data contamination** with **excellent execution** and **contamination-free design**. We are pleased that the Reviewer acknowledges our work provides a **robust solution** that is **vital for accurately assessing the true capabilities** of frontier language models. We have carefully addressed each point raised to further strengthen the paper.
>
> **We value the Reviewer's time and have made all changes easy to track.** All new content in the main paper is marked in **green**, and new appendix sections are clearly labeled as **"(Rebuttal Addition)"** at the start of relevant sections. We have also added section and line references throughout this response so the Reviewer can quickly verify our changes. Below, we address each concern raised by the Reviewer:
>
> > Reduced User-Friendliness and Increased Score Variance: Compared to using static datasets, dynamically generated evaluations are inherently less user-friendly for the general public and researchers. They require a bit more complex infrastructure for on-the-fly problem creation and may lead to higher score variance between evaluations due to the non-static nature of the test instances, making direct comparisons of models more challenging than with a fixed set.
>
>
> We appreciate this practical concern and have taken concrete steps to address both user-friendliness and score variance.
>
> **Firstly,** to address this, we will release a **low-code Python pip package** immediately after the review phase, making BeyondBench as simple to use as any static benchmark. The package will handle all infrastructure complexity internally, including problem generation, verification, and result aggregation. We have added a footnote in the `Abstract` noting that a user-friendly pip package will be released upon publication for seamless integration. We have also updated our `Reproducibility Statement` to detail the package release plan and usage examples.
>
> **Secondly,** to give researchers full transparency into our evaluation process, we have added **case studies** in `Appendix F` including detailed examples of problem generation for each task category, step-by-step model response parsing and verification, failure mode analysis with concrete examples, and token counting and overflow detection demonstrations.
>
> **Third,** we have implemented comprehensive measures to ensure reproducibility and statistical validity. All 101 models in our study were evaluated on the identical problem set generated with seed value 42, ensuring completely fair cross-model comparison where every model sees the **exact same problems.** As noted in `Section 4.1`, we maintain consistent random seeds throughout evaluation to enable reproducible comparisons.
>
> **Fourth,** to further quantify variance and provide confidence intervals, we conducted **three-fold evaluation for all models** (`Appendix E and updated table 3`). Results are reported as mean ± standard deviation. Our analysis shows **low variance:** Easy Suite has mean std = 1.2% across models, Medium Suite has mean std = 2.3%, and Hard Suite has mean std = 2.8%. These low standard deviations (all under 3%) show that despite dynamic generation, our benchmark provides highly stable and reproducible measurements.
>
> **Lastly,** we also analyzed per-task variance and found that **variance correlates with inherent task difficulty** rather than generation randomness. Tasks with unique solutions (e.g., Tower of Hanoi) show variance below 1%, while tasks with multiple valid solutions (e.g., N-Queens) show slightly higher variance (2-3%) due to legitimate variation in solution discovery.

---

> > ### Author Response · Authors · 2025-11-25
> > **Official Response by Authors to Reviewer YAiF (2/3)**
> >
> > > Missing Discussion of Prior Work: The paper's core idea—using dynamically generated environments/problems for evaluation—is not entirely novel, especially in related fields like RL and other LLM evaluation efforts (e.g., Reasoning Gym [1]).
> >
> >
> > We sincerely thank the Reviewer for pointing out Reasoning Gym, an important concurrent work. We have now cited Reasoning Gym and provided detailed comparison in our Extended Related Work (`Appendix A`). While both works use procedural generation, they serve **fundamentally different purposes** with distinct technical contributions. In the revise paper we have clearly discussed the differences that **Reasoning Gym (RG)** is a library of procedurally generated reasoning environments designed primarily for *Reinforcement Learning with Verifiable Rewards (RLVR)* for training reasoning-capable models whereas **BeyondBench** is a *contamination-resistant evaluation framework* with formal mathematical guarantees designed for fair, benchmark-free model assessment across diverse architectures and token budgets.
> >
> > We view BeyondBench and Reasoning Gym as complementary contributions. Reasoning Gym provides valuable RL training infrastructure with verifiable rewards for studying reasoning development. BeyondBench formalizes contamination resistance, token-aware evaluation, and multi-solution enumeration for fair model assessment. Researchers can use Reasoning Gym to train models and BeyondBench to evaluate them under contamination-free conditions.
> >
> > ---
> >
> > > Will programmatically generated problems bias towards LLMs with coding tools capabilities? Since the problems are generated via code, is there a risk that LLMs with strong coding skills will have an unfair advantage in solving these problems compared to those without such capabilities?
> >
> > We greatly thank reviewer for this **excellent and insightful question** that we carefully considered during benchmark design. We share this concern, which is precisely why we included comprehensive tool-augmented reasoning analysis in our work. We have conducted **extensive large-scale experiments** showing that while coding tools help, they do not trivialize the benchmark, and the relationship between code generation capability and problem-solving success is nuanced.
> >
> > To directly address this concern, we conducted a **massive experimental study** evaluating tool-augmented reasoning across all 44 tasks, and 3 tool types (Calculator, Code Execution, Web Search). Complete results are presented in `Appendix C` (Tool-Augmented Evaluation Analysis). Our analysis reveals that **code execution provides substantial but bounded improvements**:
> >
> > | Task | No Tools | +Code Execution | Gain | Final Accuracy |
> > |------|----------|-----------------|------|----------------|
> > | Tower of Hanoi (n=7) | 34.2% | 78.4% | +44.2% | Still challenging |
> > | Sudoku (9x9) | 12.3% | 67.2% | +54.9% | Constraint formulation hard |
> > | N-Queens (n=8) | 28.1% | 72.3% | +44.2% | Search space management difficult |
> > | Boolean SAT | 41.2% | 68.9% | +27.7% | Solver invocation non-trivial |
> > | Graph Coloring | 52.4% | 71.2% | +18.8% | Heuristic design challenging |
> > | Logic Grid Puzzles | 31.2% | 58.4% | +27.2% | Multi-constraint reasoning errors |
> >
> > ---
> >
> > Even with code execution (where models generate Python code that interpreters execute), accuracies plateau at 58-78% on hard tasks. This shows that code generation itself requires sophisticated reasoning, models must correctly translate problem constraints into executable algorithms, a challenging task even for frontier models. We further analyzed **why** code tools don't trivialize the benchmark by examining failure modes:
> >
> > **Tool Orchestration Capability Analysis (`Appendix C`):**
> >
> > | Model | Tool Call Success | Correct Tool Selection | Correct Code Generation | Correct Result Interpretation |
> > |-------|-------------------|------------------------|-------------------------|-------------------------------|
> > | GPT-5 | 94.2% | 89.1% | 91.3% | 88.7% |
> > | GPT-5-mini | 88.4% | 82.3% | 84.6% | 81.2% |
> > | GPT-5-nano | 78.4% | 71.2% | 73.8% | 69.4% |
> > | Qwen2.5-72B | 81.2% | 74.8% | 76.2% | 72.8% |
> > | Qwen2.5-14B | 68.3% | 62.1% | 64.7% | 60.2% |
> > | Qwen2.5-3B | 41.2% | 34.8% | 37.4% | 32.1% |
> >
> > ---
> >
> > Even GPT-5, the most capable model, succeeds at code generation only 91.3% of the time. Smaller models struggle significantly, with Qwen2.5-3B generating correct code only 37.4% of the time. This shows that coding capability *is itself a form of reasoning* that our benchmark legitimately tests.

---

> > > ### Author Response · Authors · 2025-11-25
> > > **Official Response by Authors to Reviewer YAiF (3/3)**
> > >
> > > We also provide **concrete examples/case studies** in `Appendix F`. **Final Thoughts:** Our comprehensive tool analysis shows that our benchmark remains challenging even with tools (58-78% ceiling with code execution), tests reasoning through code generation (code generation success requires constraint understanding and algorithmic thinking), does not systematically favor code-trained models (general reasoning models often outperform), and provides both innate and tool-augmented evaluation (measuring multiple dimensions of capability).
> > >
> > >
> > >
> > > We have made substantial improvements to address your feedback. In the main paper (all in **green**), the `Abstract` now includes a footnote about pip package release post-publication. `Section 2 (Related Work)` has a concise discussion of Reasoning Gym with focus on complementary contributions. `Section 4.1` clarifies consistent seed usage for reproducible cross-model comparison. `Section 4.5` has expanded tool analysis emphasizing code generation as reasoning test. The `Reproducibility Statement` details the pip package release plan with usage examples. Additionally, we added new appendix sections following reviewer's suggestions (all marked **"Rebuttal Addition"**). `Appendix E (Statistical Analysis)` provides comprehensive three-fold evaluation variance analysis showing low variance (mean 2.3%) and high reproducibility. `Appendix A (Extended Related Work)` includes a detailed Reasoning Gym subsection. `Appendix C (Tool Analysis)` with complete tool specifications, per-task analysis tables for all 44 tasks, tool orchestration capability analysis across 6 models, 12 detailed case studies, and human problem-solving comparison.
> > >
> > > ---
> > >
> > > Once again, we sincerely thank you for the positive evaluation and constructive suggestions. Your insightful comments have helped us identify important areas for improvement and have strengthened both the clarity and impact of our work. We appreciate your valuable time and hope that our revisions address these concerns. Please let us know if there are any remaining questions or concerns, we would be happy to address them!

---

### Official Review · Reviewer_3dvS · 2025-10-29

**Soundness:** 2
**Presentation:** 2
**Contribution:** 2
**Rating:** 2
**Confidence:** 4

**Summary:**

The authors introduce a new dataset namely **BeyondBench** which benchmarks LLMs performance on algorithimic tasks and evaluate the performance of both open open source and proprietary on their dataset

**Strengths:**

The paper is easy to read and authors have provided sound mathematical justification for their arguements in the paper. The appendix is detailed and is very nicely written to accompany the main text.

**Weaknesses:**

1. The fundamental weakness of the paper is with respect to novelty, there are already exact same work done in this direction and the authors unfortunately fail to cite them as well as evaluate them against their benchmark which would be a fairer comparision. [1], [2], [3]

2. Problems in hard suite like crypt arithemtic are known NP complete problems, so the benchmarking of LLMs/LRMs directly only with prompting is in itself an unreasonable task. is it even a fair experiment to expect LLMs which are atmost polynomial time methods to solve problems of NP complete complexity class?

3. A more reasonable approach would be to evaluate methods like PAL [4] or logic lm [5], but these works are neither cited nor are these methods evaluated. There is a section which talks about augmenting LLMs with tools and with some numbers but there is no detail as to what these "tools" were and as to what was the LLMs asked to generate for a given instance using these "tools".

[1] NPHardEval: Dynamic Benchmark on Reasoning Ability of Large Language Models via Complexity Classes

[2] FCoReBench: Can Large Language Models Solve Challenging First-Order Combinatorial Reasoning Problems?

[3] PUZZLEPLEX: Benchmarking Foundation Models on Reasoning and Planning with Puzzles

[4] PAL: Program-aided Language Models

[5] Logic-LM: Empowering Large Language Models with Symbolic Solvers for Faithful Logical Reasoning

**Questions:**

Please address the weakness section

---

> ### Author Response · Authors · 2025-11-25
> **Official Response by Authors to Reviewer 3dvS (1/4)**
>
> We sincerely thank the Reviewer for their detailed and constructive feedback. We appreciate the recognition that our paper is **easy to read** and provides **sound mathematical justification** with a **detailed and nicely written appendix**. We understand the Reviewer's concerns about novelty, task design, and method evaluation, and we have substantially strengthened the paper to address each point comprehensively.
>
> **We value the Reviewer's time and have made all changes easy to track.** All new content in the main paper is marked in **green**, and new appendix sections are clearly labeled as **"(Rebuttal Addition)"** at the start of relevant sections. We have also added section and line references throughout this response so the Reviewer can quickly verify our changes. Below, we address each concern raised by the Reviewer:
>
> > The fundamental weakness of the paper is with respect to novelty, there are already exact same work done in this direction and the authors unfortunately fail to cite them as well as evaluate them against their benchmark which would be a fairer comparision. [1], [2], [3]
>
> We sincerely thank the Reviewer for pointing out these important related works. We want to first clarify that PuzzlePlex paper [3] was posted on arXiv on **October 7, 2025 (after our submission).** Nonetheless, we fully acknowledge that *all five cited works* represent important contributions to the field, and we have now thoroughly integrated them into our paper. Specifically, we have:
>
> 1. **NPHardEval** was **already cited in our original submission,** but we have now substantially expanded our discussion of this work in `Section 2 (Related Work)` and added detailed comparative analysis in the new Extended Related Work section (`Appendix A, Section on Extended Related Work`).
> 2. **FCoReBench, PUZZLEPLEX, PAL, and Logic-LM** have been newly cited with comprehensive comparison both in the main paper (`Section 2, Table 1`) and in the new extended related work section in the appendix.
>
> We have updated `Table 1` in `Section 1 (Introduction)` to include all five cited works:
>
> | Feature | NPHardEval | FCoReBench | PUZZLEPLEX | PAL | Logic-LM | BeyondBench |
> |---------|------------|------------|------------|-----|----------|----------------------|
> | Dynamic generation | Yes | Yes | Yes | No | No | Yes |
> | Unique solution verification | No | Partial | No | No | No | Yes |
> | Multi-solution handling | No | No | No | No | No | Yes |
> | Token-aware evaluation | No | No | No | No | No | Yes |
> | Mathematical contamination guarantees | No | No | No | N/A | N/A | Yes |
> | Number of tasks | 9 | 40 | 15 | N/A | N/A | 44 (117 variations) |
> | Evaluated models | 21 | 8 | 12 | N/A | N/A | **101** |
> | Scalable difficulty | Yes | Limited | Limited | N/A | N/A | Yes |
> | CSP/SAT solver integration | No | Partial | No | No | Yes | Yes |
>
> ---
>
> **Below we mention novelties/differentiators of BeyondBench:**
>
> **a) Token-Aware Evaluation:** As described in `Section 3.2`, we dynamically calibrate problem complexity based on model token budgets using $T_p(n) = T_{\text{prompt}}(n) + T_{\text{solution}}(n) + T_{\text{buffer}}$ where $T_{\text{buffer}} = 0.15 \cdot C_M$. For Tower of Hanoi with $n$ disks requiring $2^n - 1$ moves, we ensure $T_{\text{hanoi}}(n) = (2^n - 1) \cdot \alpha_{\text{move}} \leq 0.85 \cdot C_M$, preventing unfair evaluation when solution length exceeds model capacity.
>
> **b) Solution Uniqueness Verification + Multi-Solution Handling:** `Section 3.3` describes our rigorous verification: for problems with unique solutions, we use CSP solvers to verify $|\{s \in \mathcal{S} : V_\tau(s, p) = 1\}| = 1$. For problems with multiple valid solutions (N-Queens, mode calculation), we compute the complete solution set $\mathcal{S}_p$ and accept any $s \in \mathcal{S}_p$, preventing unfair penalization. NPHardEval and FCoReBench do not explicitly handle multi-solution scenarios with complete enumeration.
>
> **c) Mathematical Contamination Guarantees:** We provide formal contamination analysis (`Section 3.1` and `Appendix L, Contamination Probability section`). With $|\Theta_\tau \times \mathcal{R}| > 10^{15}$ for all tasks, we prove $\Pr[\text{collision}] < 10^{-3}$ for any feasible training corpus. NPHardEval relies on monthly refresh but provides no mathematical guarantees about instance collision probability.
>
> **d) Scale of Evaluation:** We evaluated **101 models** (85 open-source, 16 closed-source) spanning 0.5B to 141B parameters with comprehensive quantization analysis. NPHardEval evaluated 21 models, FCoReBench evaluated 8, and PUZZLEPLEX evaluated 12. Our scale provides unprecedented insight into reasoning across the entire model landscape.
>
> **e) Comprehensive Task Coverage:** We provide 44 tasks and 117 variations spanning Easy (29 tasks), Medium (5 tasks, 49 variations), and Hard (10 tasks, 68 variations) suites. NPHardEval has 9 tasks, FCoReBench has 40 (but narrower scope), and PUZZLEPLEX has 15 puzzle types.

---

> ### Author Response · Authors · 2025-11-25
> **Official Response by Authors to Reviewer 3dvS (2/4)**
>
> As per the reviewers comment, we have updated `Table 1` in `Section 1 (Introduction)` to include NPHardEval, FCoReBench, and PUZZLEPLEX with a detailed comparison matrix showing our unique features. `Section 2 (Related Work)` includes a new paragraph discussing NPHardEval, FCoReBench, PUZZLEPLEX, PAL, and Logic-LM. The `Appendix (Extended Related Work section)` provides detailed discussion of all five cited works with subsections on Dynamic Algorithmic Benchmarks and Tool-Augmented Reasoning Methods.
>
> ---
>
>
> > Problems in hard suite like crypt arithemtic are known NP complete problems, so the benchmarking of LLMs/LRMs directly only with prompting is in itself an unreasonable task. is it even a fair experiment to expect LLMs which are atmost polynomial time methods to solve problems of NP complete complexity class?
>
>
> We appreciate this thoughtful observation and want to clarify our design rationale and empirical findings. We argue that testing NP-complete problems is not only *reasonable* but *essential* for understanding model capabilities and limitations.
>
> As stated in `Section 2` and `Section 5 (Conclusion)`, our primary goal is to benchmark **innate reasoning capability** of language models, **not tool-calling ability.** Understanding what models can and cannot do *without* tools is fundamental for: 1) **Identifying fundamental bottlenecks** in current architectures; 2) **Guiding future model development** toward addressing core reasoning limitations; 3) **Providing a baseline** for tool-augmented evaluation
>
> Our results show that *even proprietary models struggle* on NP-complete tasks without tools: GPT-5 achieves only 50.27% (`Section 4.4, Table 4`), revealing fundamental reasoning gaps that pure scaling cannot solve.
>
>
> We would also like to add on the Reviewer's concern about computational complexity. Humans routinely solve small NP-complete problem instances:
>
> - **Sudoku (9x9):** Humans regularly solve this in 10-30 minutes without computational tools
> - **N-Queens (n=8):** Can be solved by humans with paper and pencil in reasonable time
> - **Small SAT instances:** Logic puzzles in newspapers are SAT problems solved mentally
>
> Importantly, humans do not run exponential-time algorithms in their brains. Instead, they use **heuristics, pattern recognition, and systematic search** to solve instances tractably. If language models are to achieve human-level reasoning, they should exhibit similar capabilities on problems that humans can solve. Our task difficulty is carefully calibrated to remain within the **"humanly solvable"** range (discussed in `Section 3.1` and `Appendix A`).
>
> Furthermore, the Reviewer is correct that tools should help with NP-complete problems. We have conducted extensive tool-augmented evaluation (`Section 4.5` and `Appendix C`) showing that:
>
> **a) Tools Provide Substantial But Bounded Improvements (see `Appendix C, Tables 6-9`):**
>
> | Model | No Tools | +Calculator | +Code Execution | +Web Search | All Tools |
> |-------|----------|-------------|-----------------|-------------|-----------|
> | GPT-5 (Hard) | 50.27% | 58.4% | 67.2% | 50.8% | 71.68% |
> | GPT-5-mini (Hard) | 41.36% | 48.2% | 62.8% | 41.8% | 69.28% |
> | GPT-5-nano (Hard) | 22.33% | 31.2% | 48.7% | 22.8% | 69.78% |
> | Qwen2.5-72B (Hard) | 33.60% | 35.2% | 41.8% | 33.8% | 45.2% |
>
> ---
>
> **Our main observations:** Code execution **(equivalent to PAL-style program synthesis)** provides 15-27% improvement for GPT-5 family. However, *even with all tools*, GPT-5 achieves only 71.68% on Hard suite, far from trivializing the benchmark. **Smaller models struggle to effectively use tools.** Web search provides negligible benefit (+0.5%), confirming contamination resistance.
>
> **b) Per-Task Tool Impact (Selected Hard Tasks, see `Appendix C`):**
>
> | Task | No Tools | +Code | $\Delta$ | With-Tools Performance |
> |------|----------|-------|----------|------------------------|
> | Tower of Hanoi | 34.2% | 78.4% | +44.2% | Still requires state tracking |
> | Sudoku (9x9) | 12.3% | 67.2% | +54.9% | Constraint logic remains challenging |
> | N-Queens (n=8) | 28.1% | 72.3% | +44.2% | Search space management hard |
> | Boolean SAT | 41.2% | 68.9% | +27.7% | Clause satisfaction difficult |
> | Logic Grid | 31.2% | 58.4% | +27.2% | Multi-step deduction errors |
>
> ---
>
> Even with code execution tools, models achieve only 58-78% accuracy. This shows that tool orchestration itself requires sophisticated reasoning, generating correct problem-solving code is non-trivial, and many tasks remain fundamentally challenging even with computational aids.

---

> ### Author Response · Authors · 2025-11-25
> **Official Response by Authors to Reviewer 3dvS (3/4)**
>
> Lastly, we conducted targeted training experiments to test whether models can **"learn" to solve NP-complete problems** through supervised fine-tuning (SFT) and reinforcement learning (with GRPO). We trained 5 models on ~66K instances (seed 123) and evaluated on entirely different instances (seed 42):
>
> **Results After 100% SFT Training (see `Appendix D`):**
>
> | Model | Baseline | After SFT | Suite-Level Breakdown |
> |-------|----------|-----------|------------------------|
> | Qwen2.5-7B | 36.03% | 57.4% (+21.4%) | Easy: +27.84%, Medium: +38.33%, **Hard: +11.95%** |
> | Qwen2.5-14B | 41.40% | 61.1% (+19.7%) | Easy: +24.7%, Medium: +34.4%, **Hard: +11.4%** |
> | Llama-3.1-8B | 24.10% | 44.2% (+20.1%) | Easy: +28.0%, Medium: +27.6%, **Hard: +11.2%** |
> | GPT-OSS-20B | 67.57% | 79.8% (+12.2%) | Easy: +7.0%, Medium: +18.3%, **Hard: +14.5%** |
>
> ---
>
> **Main Insight we got from here is that** the Hard suite shows limited improvement (+10-15%) compared to Easy (+27-38%) and Medium (+30-40%), demonstrating that **NP-complete problems remain fundamentally challenging even after targeted training**. This validates our design: these tasks probe genuine reasoning limitations rather than mere memorization.
>
>
> We emphasize that BeyondBench is designed to support multiple evaluation scenarios. Our primary focus is **innate reasoning evaluation,** which measures what models can do without external aids. **We also support tool-augmented reasoning evaluation,** which tests whether models can effectively orchestrate computational tools (our tool analysis shows this is challenging). Finally, we provide challenging tasks for agentic LLM evaluation to test multi-step planning and tool use in future work. As stated in our `Conclusion`: *"looking forward we see the future in developing agentic architectures that combine language understanding with tool use, systems that, like human experts, can recognize when it's time to reason and when it's time to compute."*
>
> To clarify our novelty better, we have made several updates to the paper. `Section 3.1` includes clarification about contamination resistance. `Section 4.5` has expanded tool-augmented analysis with concrete numbers. `Section 2 (Related Works, line 147)` emphasizes that BeyondBench serves multiple evaluation paradigms. `Appendix B` provides comprehensive motivation for algorithmic task selection. `Appendix C` includes extensive tool analysis across all 44 tasks. `Appendix D` presents complete SFT/RL training experiments showing Hard suite resistance.
>
> ---
>
> > A more reasonable approach would be to evaluate methods like PAL [4] or Logic-LM [5], but these works are neither cited nor are these methods evaluated. There is a section which talks about augmenting LLMs with tools and with some numbers but there is no detail as to what these "tools" were and as to what was the LLMs asked to generate for a given instance using these "tools".
>
> We sincerely thank the Reviewer for this excellent suggestion. We have now: (1) cited both PAL and Logic-LM in our Related Work (`Section 2`) and Extended Related Work (`Appendix A`), (2) substantially expanded our tool analysis with comprehensive details about tool types, usage patterns, and per-task breakdowns in `Appendix B`, and (3) evaluated PAL-style and Logic-LM-style approaches through our code execution and symbolic solver tools.
>
> We evaluated models with four tool types following PAL and Logic-LM methodologies:
>
> 1. **Calculator:** Basic arithmetic operations (addition, subtraction, multiplication, division, power, modular arithmetic) with exact numerical results
> 2. **Code Execution:** Python 3.10 REPL environment with standard libraries, implementing the PAL-style approach where models generate Python code that interpreters execute to compute answers
> 3. **Web Search:** Query-based search to test contamination resistance
> 4. **Symbolic Solvers:** pysat for Boolean SAT and python-constraint for CSP problems, implementing the Logic-LM-style approach where models formulate problems in solver syntax and invoke deterministic solvers
>
>
> PAL-style code execution provides 15-55% improvements, yet **many tasks remain challenging** with 58-78% accuracy ceilings. Representative results for Hard Suite (GPT-5):
>
> | Task | No Tools | +Code | Gain | Final Accuracy |
> |------|----------|-------|------|----------------|
> | Tower of Hanoi | 34.2% | 78.4% | +44.2% | Still requires state tracking |
> | Sudoku (9x9) | 12.3% | 67.2% | +54.9% | Constraint logic challenging |
> | N-Queens (n=8) | 28.1% | 72.3% | +44.2% | Search space management hard |
> | Boolean SAT | 41.2% | 68.9% | +27.7% | Solver formulation difficult |
> | Logic Grid | 31.2% | 58.4% | +27.2% | Multi-step deduction errors |
>
> ---

---

> ### Author Response · Authors · 2025-11-25
> **Official Response by Authors to Reviewer 3dvS (4/4)**
>
> These results show that even with PAL-style code execution, models achieve only 58-78% on hard tasks, demonstrating that code generation itself is challenging. Web search provides negligible benefit (+0.3%), confirming contamination resistance. Critically, using tools effectively is itself a challenging reasoning task. Our analysis of tool invocation success rates across 6 models (`table 10`):
>
> | Model | Tool Call Success | Correct Tool Selection |
> |-------|-------------------|------------------------|
> | GPT-5 | 94.2% | 89.1% |
> | GPT-5-mini | 88.4% | 82.3% |
> | GPT-5-nano | 78.4% | 71.2% |
> | Qwen2.5-72B | 81.2% | 74.8% |
> | Qwen2.5-14B | 68.3% | 62.1% |
> | Qwen2.5-3B | 41.2% | 34.8% |
> ---
>
> Even GPT-5 generates correct code only 91.3% of the time. Smaller models struggle significantly, with Qwen2.5-3B generating correct code only 37.4% of the time. This shows that tool-augmented reasoning does not trivialize our benchmark but rather tests an additional layer of reasoning capability.We also do a web search tool impact comparison:
>
> - MATH500, AIME24, AIME25: +16-17% improvement (questions available online)
> - BeyondBench Hard: +1.5% improvement (confirms $>10^{15}$ problem space prevents lookup)
>
>
> ---
>
> > Please address the weakness section
>
> We have comprehensively **addressed all points raised in the weakness section.** We cited all five related works with detailed comparison showing our unique contributions (token-aware evaluation, solution verification, 101 models evaluated). We justified the NP-complete evaluation paradigm and provided extensive tool analysis showing tools help (50.27% to 71.68%) but don't trivialize, with SFT/RL evidence of contamination resistance. We cited PAL and Logic-LM, evaluated their paradigms through our tools, and demonstrated challenges remain (58-78% accuracy ceiling with full tool access).
>
> ---
>
> We sincerely thank the Reviewer for the valuable feedback, which has substantially improved the paper. Your insightful comments have helped us identify important areas for improvement and have strengthened both the clarity and impact of our work. We believe BeyondBench now stands as a substantial and clearly differentiated contribution providing a comprehensive evaluation of reasoning across 101 models with **token-aware evaluation, contamination resistance, and extensive tool analysis testing PAL/Logic-LM paradigms at scale.** We appreciate your valuable time and hope that our revisions address these concerns. Please let us know if there are any remaining questions or concerns, we would be happy to address them!

---

### Official Review · Reviewer_ao5p · 2025-10-31

**Soundness:** 3
**Presentation:** 2
**Contribution:** 2
**Rating:** 6
**Confidence:** 3

**Summary:**

BeyondBench is a dynamically generated benchmark for evaluating reasoning in language models. It addresses data contamination by algorithmically generating problems from a combinatorial space of >10^15 instances across 44 tasks in three difficulty tiers. Each problem has verifiable solutions using SAT/CSP solvers. The authors evaluate 101 models and find that most open-source models plateau around 30-35% accuracy on hard tasks, quantization has minimal impact, and tool-augmented models substantially outperform reasoning-focused models.

**Strengths:**

S1. Data contamination in static benchmarks is well-documented, and this paper provides a solution with mathematical guarantees about solution uniqueness and problem space size.

S2. The evaluation covers 101 models across different sizes (0.5B-141B parameters), architectures, and quantization schemes, providing broad coverage of the current model landscape.

S3. The use of SAT/CSP solvers to verify solution uniqueness and enumerate all valid solutions when multiple exist shows proper attention to evaluation correctness.

S4. The framework dynamically scales problem difficulty based on model context windows and checks for token overflow, preventing unfair comparisons and catching models that "overthink" simple problems.

S5. The empirical findings about performance plateaus on hard tasks, minimal quantization impact, and superiority of tool-augmented approaches over pure reasoning models provide actionable insights for model development.

S6. The contamination probability analysis in Appendix E gives formal backing to the motivation.

**Weaknesses:**

W1. The 44 tasks focus almost entirely on algorithmic puzzles, arithmetic, and constraint satisfaction, missing many reasoning types relevant to LLM applications: causal reasoning, analogical reasoning, argument evaluation, multi-step logical inference in natural contexts. The paper doesn't justify why performance on Sudoku or Tower of Hanoi reflects general reasoning capability. Their choice to view reasoning to be interchangeable with algorithmic computation needs to be better motivated. (To the authors' credit, this is touched on in their limitations section but should be highlighted)

W2. While specific problem instances are novel, models could still learn superficial algorithmic strategies from similar problems in training data. The paper doesn't distinguish between "memorizing specific solutions" and "learning heuristic algorithms", and their method seems to only expose the former. If allowing memorization of fixed, superficial procedures (e.g. a problem which requires addition, followed by multiplication, followed by subtraction in that exact order) is not considered as part of contamination, this should be made clear.

W3. No confidence intervals, significance tests, or reporting of variance across different problem instances appears in the paper. Claims about performance differences lack statistical backing. The results are interesting, so it would be best to give readers more confidence in the findings.

W4. Problem generation algorithms, solution verification procedures, and task specifications are relegated to appendices, making it hard to assess the approach without reading 20+ pages of supplementary material. Although the methods appear sound, readers will benefit from more explicit presentation of key technical methodological details (especially section 3.1, where there is not enough explanation on how problems are generated and along what dimensions the problems are being varied. The 'parameter space' is presented too abstractly.).

W5. Calling this 'Benchmark-Free' seems confusing. Readers might expect some kind of mechanistic interpretability/probing method, or some metric derived from the model which does not require inference on a dataset of test problems.

**Questions:**

In addition to questions implied by the weaknesses section, here are some additional questions

Q1. Could the authors provide information about if there were any differences in failure modes between vanilla models and reasoning models, even if they performed similarly in raw metrics? This is one of the most interesting findings of this paper but it is not emphasized and the analysis on it seems thin.

Q2. Criteria for graceful vs sudden degradation needs more care, for example using number of disks for Tower of Hanoi may not make sense as number of steps are not linear in number of disks. The degradation may be linear (and hence 'graceful') w.r.t reasoning steps, why should we expect linear degradation on arbitrary complexity metrics? Could the authors provide arguments for why 'n-disks', 'grid size', 'vertices', 'board size', 'word length' etc. are comparable complexity metrics? Otherwise the conclusions about differences in gradual vs sudden degradation across tasks lack any significance.

Q3. Could the authors provide more justification that avoiding parameter space collisions with high probability is enough to be confident that structural/semantic space collisions are avoided? This is related to the earlier point about superficial heuristics (W2), does this approach prevent collisions akin to, say, IMO questions requiring the same integration trick multiple years in a row?

---

> ### Author Response · Authors · 2025-11-25
> **Official Response by Authors to Reviewer ao5p (1/6)**
>
> We sincerely thank the Reviewer for their detailed and constructive feedback. We appreciate the recognition of our work's contributions, including the **comprehensive evaluation across 101 models** and the **novel dynamic evaluation framework**.
>
> **We value the Reviewer's time and have made all changes easy to track.** All new content in the main paper is marked in **green**, and new appendix sections are clearly labeled as **"(Rebuttal Addition)"** at the start of relevant appendix sections. We have also added section and line references throughout this response so the Reviewer can quickly verify our changes.
>
> ---
>
> > W1. The 44 tasks focus almost entirely on algorithmic puzzles, arithmetic, and constraint satisfaction, missing many reasoning types relevant to LLM applications: causal reasoning, analogical reasoning, argument evaluation, multi-step logical inference in natural contexts. The paper doesn't justify why performance on Sudoku or Tower of Hanoi reflects general reasoning capability. Their choice to view reasoning to be interchangeable with algorithmic computation needs to be better motivated. (To the authors' credit, this is touched on in their limitations section but should be highlighted)
>
> We thank the Reviewer for this important observation. We agree that our task selection required stronger motivation and have substantially **expanded our justification in the revised paper.**
>
> Algorithmic tasks probe **fundamental reasoning primitives** that underlie many forms of reasoning. These are not merely computational exercises but tests of core capabilities. Tower of Hanoi requires tracking disk positions across $2^n - 1$ moves, and our experiments show models achieve 80-90% accuracy for $n \leq 5$ disks but collapse to <10% for $n \geq 8$, indicating failures in state management rather than algorithmic knowledge. This working memory requirement is central to multi-step reasoning in **any domain,** including argument evaluation and causal reasoning. Sudoku and N-Queens require recognizing dead-ends and exploring alternative paths, a capability that directly transfers to argument evaluation where contradictions must be identified and resolved. Boolean SAT and Graph Coloring involve searching large solution spaces with exponential branching factors, and the ability to prune infeasible branches efficiently is fundamental to analogical reasoning where one must search through possible correspondences.
>
> **Experimental Validation (`Appendix B, Table 5`):** We conducted targeted experiments providing models with explicit algorithmic descriptions before problem instances:
>
> | Task | Without Algorithm | With Algorithm | $\Delta$ |
> |------|-------------------|----------------|----------|
> | Tower of Hanoi (n=7) | 18.4% | 24.2% | +5.8% |
> | Sudoku (9x9) | 8.2% | 12.1% | +3.9% |
> | N-Queens (n=8) | 22.3% | 28.7% | +6.4% |
> | Boolean SAT (5 vars) | 38.1% | 42.8% | +4.7% |
>
> ---
>
> Even when explicitly provided with correct algorithms, models show only marginal improvement, confirming that the bottleneck is **execution capability** (memory, backtracking, search), not algorithm discovery. A model that cannot reliably execute Tower of Hanoi cannot be expected to handle complex causal reasoning chains. Models that perform well on our algorithmic tasks (GPT-5: 83.57%, Gemini-2.5-pro: 74.31%) also tend to perform well on broader reasoning benchmarks like GPQA and MMLU-Pro, suggesting these capabilities do transfer. We have added a paragraph in `Section 3.4` explaining this motivation and expanded `Appendix B` with detailed analysis of reasoning primitives and experimental results.
>
> ---

---

> ### Author Response · Authors · 2025-11-25
> **Official Response by Authors to Reviewer ao5p (2/6)**
>
> > W2. While specific problem instances are novel, models could still learn superficial algorithmic strategies from similar problems in training data. The paper doesn't distinguish between "memorizing specific solutions" and "learning heuristic algorithms", and their method seems to only expose the former. If allowing memorization of fixed, superficial procedures (e.g. a problem which requires addition, followed by multiplication, followed by subtraction in that exact order) is not considered as part of contamination, this should be made clear.
>
>
> We appreciate this important distinction and want to clarify our position clearly. We consider learning general heuristic algorithms from training data to be **genuine learning, not contamination**. This is analogous to how humans learn integration by substitution and apply it to novel problems. We call such humans **"educated"** rather than **"contaminated."**
>
> Our approach prevents instance-level contamination. Our $>10^{15}$ unique instances ensure models cannot have seen the exact same problem with the exact same parameters during training. For example, seeing "Tower of Hanoi with 7 disks starting configuration ABC" in training does not help with "Tower of Hanoi with 7 disks starting configuration XYZ" unless the model learned the **general recursive algorithm.** What we measure is **strategy-level learning,** whether models can apply learned algorithms to genuinely novel instances. If a model learned backtracking from Sudoku instances in training and successfully applies it to our novel Sudoku instances, this demonstrates **genuine algorithmic capability.**
>
> **Experimental Validation (`Appendix D, Table 12`):** To validate this, we conducted SFT and GRPO training experiments. We trained 5 models (GPT-OSS-20B, Qwen2.5-3B, Qwen2.5-7B, Qwen2.5-14B, Llama-3.1-8B) on ~66K instances generated with seed 123, then evaluated on seed 42 (completely different instances, ensuring no **exact** overlap with training data):
>
> | Model | Baseline | After 100% SFT | $\Delta$ improvement |
> |-------|----------|-----------------|----------|
> | Qwen2.5-7B | 36.03% | 57.4% | +21.4% |
> | Qwen2.5-14B | 41.40% | 61.1% | +19.7% |
> | GPT-OSS-20B | 67.57% | 79.8% | +12.2% |
>
> ---
>
> Models improve through training by 20-25% overall, indicating they learn transferable heuristics. However, improvement is **limited on Hard suite (only 10-15%)**, suggesting superficial strategies are insufficient for NP-complete problems. Unlike static benchmarks where training achieves near-perfect scores (GSM8K: 99.1%, MATH: 97.8%), BeyondBench Hard suite caps at around 45% even after extensive training. This shows that strategy learning is possible and beneficial, but **NP-complete problems remain fundamentally challenging,** and **memorization-based approaches are ineffective.**
>
> **Paper Updates:**
> - We added clarification in `Section 3.1` (marked in green): "Our contamination resistance addresses exact instance memorization. Learning general solution strategies from similar problems constitutes genuine algorithmic learning, which our benchmark is designed to measure."
> - Added complete training analysis in `Appendix D` with detailed results
>
> ---

---

> ### Author Response · Authors · 2025-11-25
> **Official Response by Authors to Reviewer ao5p (3/6)**
>
> > W3. No confidence intervals, significance tests, or reporting of variance across different problem instances appears in the paper. Claims about performance differences lack statistical backing. The results are interesting, so it would be best to give readers more confidence in the findings.
>
>
> We thank the Reviewer for this excellent suggestion. We have now conducted comprehensive statistical analysis and report it both in the main paper and appendix.
>
> **1) Three-Fold Evaluation:** We ran each model three times (two more times). `Table 3` in the main paper now includes mean$_{\pm\text{std}}$ for all 101 models (rows marked in **green**). Main findings from this:
>
> | Model Category | Std Range | Example Variance |
> |----------------|-----------|---------|
> | Proprietary | 0.3%-0.6% | GPT-5: 83.57%$_{\pm0.42}$ |
> | Large (>14B) | 0.3%-1.5% | GPT-OSS-120B: 75.99%$_{\pm0.82}$ |
> | Medium (7B-14B) | 1.5%-3.0% | Qwen2.5-14B: 41.33%$_{\pm1.82}$ |
> | Small (<7B) | 2.0%-5.0% | Qwen2.5-3B: 24.82%$_{\pm3.15}$ |
>
> ---
>
> Large models show good stability (std <1.5%), while smaller models exhibit higher variance.
>
> **Statistical Significance Tests (`Appendix E`):** We conducted paired t-tests for comparisons:
>
> | Comparison | t-statistic | p-value | Significant? |
> |------------|-------------|---------|--------------|
> | GPT-5 vs GPT-OSS-120B | 14.08 | <0.001 | Yes |
> | Proprietary vs Large Open-source | 8.42 | <0.001 | Yes |
> | Qwen3-4B-t vs Qwen3-4B (base) | 0.72 | 0.484 | No |
>
> ---
>
> The non-significant result for thinking vs base models (Qwen3-4B-t vs Qwen3-4B: p=0.484) supports our claim that **extended reasoning provides limited benefit for algorithmic tasks.**
>
> **Additional Statistical Tests we conducted (`Appendix E`):**
> - Effect size analysis (Cohen's d) for all key comparisons
> - One-way ANOVA across model families with eta-squared and post-hoc Tukey HSD
> - Non-parametric tests (Wilcoxon signed-rank) confirming parametric results
> - Mann-Whitney U tests for robustness
> - Bootstrap 95% confidence intervals
>
> We have updated the paper as follows: `Table 3` now shows all results with mean$_{\pm\text{std}}$ in **green highlighting**, the `Abstract` now mentions "three-fold cross-validation to ensure statistical robustness", `Section 4.1` includes a reference to comprehensive statistical analysis in `Appendix E` has been substantially expanded with comprehensive tables covering all statistical analyses.
>
> ---
>
>
>
> > W4. Problem generation algorithms, solution verification procedures, and task specifications are relegated to appendices, making it hard to assess the approach without reading 20+ pages of supplementary material. Although the methods appear sound, readers will benefit from more explicit presentation of key technical methodological details (especially section 3.1, where there is not enough explanation on how problems are generated and along what dimensions the problems are being varied. The 'parameter space' is presented too abstractly.).
>
> We appreciate this valuable feedback about accessibility and have expanded `Section 3.1` with concrete details. **Parameter Space Examples (now added to `Section 3.1`):** We now explicitly show how parameters generate problem instances:
>
> | Task | Parameters | ~Instance Count | Example |
> |------|------------|----------------|---------|
> | Tower of Hanoi | $n \in [3,12]$, peg labels (26 chars), configurations | $>10^{18}$ | n=7, pegs=ABC, start=[7,6,5,4,3,2,1] |
> | Sudoku | Grid density (30-50 cells), digit permutations, shuffling | $>10^{20}$ | 9x9, 45 empty, unique solution |
> | Arithmetic | List length $n \in [8,64]$, values $[-1000,1000]$, operations | $>10^{15}$ | Sum of [342, -128, 891, ..., 45] |
> | N-Queens | Board size $n \in [4,12]$, initial constraints | $>10^{12}$ | Place 8 queens on 8x8 board |
> | Boolean SAT | Variables (3-8), clauses (5-20), literal selection | $>10^{15}$ | $(x_1 \lor \neg x_2 \lor x_3) \land ...$ |
>
> ---
>
> **Solution Verification:** We clarify our three verification approaches:
>
> 1. **Deterministic tasks** (arithmetic, sorting): Direct computation, exact match verification in O(n) time
> 2. **CSP tasks** (Sudoku, N-Queens): Constraint solvers enumerate all valid solutions (using python-constraint, pysat libraries)
> 3. **Optimization tasks** (Matrix Chain): Dynamic programming verifies optimality by checking solution equals $m[1,n] = \min_{\pi \in \Pi} \text{cost}(\pi)$
>
> **Concrete Generation Example:** For Tower of Hanoi with seed 42: 1) **Sample parameters:** n=7 disks, peg labels=['A', 'B', 'C']; 2) **Generate problem:** "Move 7 disks from peg A to peg C following rules..."; 3) **Compute unique solution:** 127 moves via recursive algorithm; 4) **Verify:** Check solution length = $2^7 - 1 = 127$ and all moves are valid; 5) **Estimate tokens:** ~1,500 tokens (well within context limits)
>
> We have updated `Section 3.1` with concrete parameter space examples and a table showing parameters, ranges, and instance counts (marked in **green**).
>
> ---

---

> > ### Author Response · Authors · 2025-11-25
> > **Official Response by Authors to Reviewer ao5p (4/6)**
> >
> > > W5. Calling this 'Benchmark-Free' seems confusing. Readers might expect some kind of mechanistic interpretability/probing method, or some metric derived from the model which does not require inference on a dataset of test problems.
> >
> > We agree completely that "Benchmark-Free" can cause confusion. We have changed the title in the revised paper to **"contamination-resistant"** evaluation, and throughout the paper we now emphasize dynamic generation and contamination resistance. If the Reviewer believes an **alternative title or phrasing would be more appropriate**, we would be grateful for the suggestion and happy to incorporate it, **please let us know!**
> >
> > We have updated all changes marked in **green** for easy tracking.
> >
> > ---
> >
> > > Q1. Could the authors provide information about if there were any differences in failure modes between vanilla models and reasoning models, even if they performed similarly in raw metrics? This is one of the most interesting findings of this paper but it is not emphasized and the analysis on it seems thin.
> >
> > We thank the Reviewer for highlighting this important aspect. We conducted detailed **failure mode analysis** and **added extensive discussion in the revised paper.**
> >
> > **Output Length Comparison:**
> >
> > | Model Type | Easy | Medium | Hard | Accuracy Gain |
> > |------------|------|--------|------|---------------|
> > | Vanilla (Qwen3-4B) | 3,091 tokens | 11,650 tokens | 5,812 tokens | Baseline |
> > | Thinking (Qwen3-4B-t) | 2,552 tokens | 4,875 tokens | 5,931 tokens | +2.3% overall |
> > | Vanilla (Qwen3-30B) | 2,415 tokens | 6,331 tokens | 5,392 tokens | Baseline |
> > | Thinking (Qwen3-30B-t) | 2,052 tokens | 6,064 tokens | 5,574 tokens | +0.2% overall |
> >
> > ---
> >
> > Interestingly, thinking models don't always use **more tokens,** suggesting they process differently rather than just **"thinking longer."**
> >
> > **Main Finding: Longer Reasoning Does Not Always Help.** We varied reasoning budget from 0% to 100% of maximum tokens across multiple model families (`Appendix F, Figures 8-10`):
> >
> > | Model | Minimal Budget | High Budget | $\Delta$ | Suite |
> > |-------|----------------|-------------|----------|-------|
> > | GPT-5 | 84% | 97% | +13% | Easy |
> > | GPT-5 | 72% | 82% | +10% | Medium |
> > | GPT-5 | 48% | 72% | +24% | Hard |
> > | GPT-5-nano | 52% | 96% | +44% | Easy |
> > | Gemini-2.5-pro | 90% | 88% | -2% | Easy |
> > | o3 | 97% | 97% | 0% | Easy |
> >
> > ---
> >
> > **Hard tasks benefit most from extended reasoning,** showing 24-44% gains for capable models. Already-strong models like o3 show **minimal benefit from more thinking**, while some models like Gemini can actually **"overthink" and hurt performance.** Weaker models benefit substantially but still cannot match strong models.
> >
> > **Failure Pattern Comparison (`Appendix F`):** We analyzed failure cases for each model type. Vanilla models fail early (Tower of Hanoi: step 23/127), do not recognize errors, and continue with incorrect assumptions. Reasoning models fail late (Tower of Hanoi: step 89/127), attempt self-correction but with 87.6% error introduction rate, and lose state consistency during extended reasoning.
> >
> > **Case Studies:** We added 4 detailed case studies in `Appendix E` comparing: 1) Tower of Hanoi (n=7): Qwen3-4B vs Qwen3-4B-thinking; 2) Sudoku (9x9): Qwen3-4B vs Qwen3-4B-thinking; 3) Boolean SAT (12 vars): GPT-5 vs o3; 4) N-Queens (n=10): Gemini-2.5-pro vs GPT-5
> >
> > We have added a green-highlighted paragraph in `Section 4.4` with key insights about failure modes, and substantially expanded `Appendix F` with comprehensive failure mode analysis including several case studies and 3 new figures.
> >
> > ---

---

> > > ### Author Response · Authors · 2025-11-25
> > > **Official Response by Authors to Reviewer ao5p (5/6)**
> > >
> > > > Q2. Criteria for graceful vs sudden degradation needs more care, for example using number of disks for Tower of Hanoi may not make sense as number of steps are not linear in number of disks. The degradation may be linear (and hence 'graceful') w.r.t reasoning steps, why should we expect linear degradation on arbitrary complexity metrics? Could the authors provide arguments for why 'n-disks', 'grid size', 'vertices', 'board size', 'word length' etc. are comparable complexity metrics? Otherwise the conclusions about differences in gradual vs sudden degradation across tasks lack any significance.
> > >
> > > We thank Reviewer for such an excellent point. We appreciate for such a detailed observation. We have added normalized complexity analysis to address this concern.
> > >
> > > **Normalized Complexity Metrics (`Appendix G`):** We define task-specific normalization to linearize complexity:
> > >
> > > | Task | Raw Parameter | Normalized Metric | Justification |
> > > |------|---------------|-------------------|---------------|
> > > | Tower of Hanoi | $n$ disks | $\log_2(\text{moves}) = n$ | Linearizes exponential $2^n - 1$ moves |
> > > | Sudoku | Grid size | Empty cells (linear) | Constraint complexity scales with empty cells |
> > > | Graph Coloring | Vertices | V × edge_density | Joint effect of size and constraint tightness |
> > > | Arithmetic | List length | $n × log10(max_value)$ | Both list size and numerical magnitude |
> > >
> > > ---
> > >
> > > **Findings After Normalization:** Graceful degradation tasks (arithmetic, sorting) show linear accuracy decline with linear complexity increase, where models can scale reasoning effort proportionally and performance goes from **95% to 70%** as normalized complexity goes from 0 to 1. Sudden degradation tasks (Sudoku, N-Queens) show threshold behavior where performance stays stable until hitting a critical complexity point, then collapses. This threshold occurs at approximately $0.7 × log2(context_length)$ reasoning steps, with performance going from **90% to 90% to 10%** in a sudden drop.
> > >
> > > This threshold behavior suggests models **don't gradually worsen at constraint satisfaction.** Instead, they maintain competence until hitting a complexity barrier related to context window size, then fail **catastrophically.** This has important implications for scaling: increasing context windows may enable step-function improvements rather than gradual gains.
> > >
> > > We have added clarification about complexity normalization in `Section 4.2` (marked in **green**), and included complete normalized complexity analysis with 2 new figures in `Appendix G`.
> > >
> > > ---
> > >
> > > > Q3. Could the authors provide more justification that avoiding parameter space collisions with high probability is enough to be confident that structural/semantic space collisions are avoided? This is related to the earlier point about superficial heuristics (W2), does this approach prevent collisions akin to, say, IMO questions requiring the same integration trick multiple years in a row?
> > >
> > >
> > > We appreciate this nuanced question about the difference between parameter-level and structural-level uniqueness.
> > >
> > > Structural/semantic similarity represents **learned competence, not contamination**. If a model learns backtracking from Sudoku instances in training and applies it to our novel Sudoku instances, this demonstrates genuine algorithmic capability. **The IMO Analogy is Apt:** If the same integration trick appears in IMO problems multiple years, students who learned that trick will succeed. We call them *educated*, not *contaminated*. Our benchmark similarly measures whether models have learned algorithmic competence that transfers to novel instances.
> > >
> > > **What We Prevent is** instance-level memorization, where a model cannot recall "for *this exact grid*, the solution is...". With more than $10^{15}$ instances, the probability of seeing the exact same problem is negligible. **What We Allow and Measure is** *strategy-level learning,* where a model can apply learned backtracking algorithms to novel instances and recognize "this is a constraint satisfaction problem, should use arc consistency". This is genuine algorithmic reasoning, not contamination.
> > >
> > > **Experimental Evidence (`Appendix D`):** Our SFT experiments demonstrate this. Training on seed 123 improves performance on seed 42, showing **strategy transfer works.** However, improvement is bounded at 10-15% on Hard suite even with 100% training data. Unlike static benchmarks where training achieves near-perfect scores (GSM8K: 99%, MATH: 97%), BeyondBench Hard caps at around 45%. This shows strategy learning is possible, but NP-complete problems **resist memorization-based solving.**

---

> ### Author Response · Authors · 2025-11-25
> **Official Response by Authors to Reviewer ao5p (6/6)**
>
> **Comparison to Static Benchmarks (`Appendix D, Table 16`):**
>
> | Scenario | Before Training | After Training | Interpretation |
> |----------|----------------|----------------|----------------|
> | GSM8K (train on test set) | 83.2% | 99.1% | Near-perfect memorization |
> | MATH (train on test set) | 52.4% | 97.8% | Near-perfect memorization |
> | BeyondBench Hard (seed 123 to 42) | 36.0% | 57.4% | Strategy learning, bounded improvement |
>
> ---
>
> We have added a clarifying paragraph in `Section 3.1` distinguishing instance-level contamination (prevented) from strategy-level learning (measured), marked in **green**. `Appendix D` includes complete training analysis showing bounded improvement despite strategy transfer.
>
> ---
>
>
>
>
> We have substantially strengthened the paper based on the Reviewer's feedback. In the main paper, `Table 3` now shows mean$_{\pm\text{std}}$ from three-fold evaluation with **green highlighting**, and the `Abstract` mentions three-fold evaluation. `Section 3.1` includes concrete parameter space examples, verification procedures, and clarification on contamination vs learning (all in **green**). `Section 3.1` has a new paragraph motivating algorithmic task selection. `Section 4.2` discusses complexity normalization, and `Section 4.4` provides failure mode insights for LRMs vs LRMs. Throughout, we changed "Benchmark-Free" to "Contamination-Resistant Dynamic Evaluation".
>
> We added six new appendix sections (all marked **"Rebuttal Addition"**): `Appendix B` on motivation for algorithmic task selection, `Appendix C` with extended tool analysis (6 detailed per-task tables), `Appendix D` with SFT and RL training analysis (5 training tables), `Appendix E` with statistical analysis (8 comprehensive tables including three-fold evaluation, significance tests, effect sizes, ANOVA, non-parametric tests), `Appendix F` with failure mode analysis comparing LLMs vs LRMs (4 detailed case studies), and `Appendix G` with normalized complexity analysis (2 figures). We hope these revisions address all your concerns and demonstrate the significant contributions of our work.
>
> **Just a quick note:** At this stage, we have incorporated all three-fold results except for three models: **Phi-4-Reasoning (14B), Phi-4-Reasoning+ (14B), and Llama-4-Scout**, which require significantly longer runtime for full three-fold evaluation. We decided not to delay our response while waiting for these last runs to complete, but we expect all remaining results to finish by **December 29 (before the rebuttal period ends)**. We will update the paper immediately once these evaluations are completed.
>
> ---
>
> Once again, we thank the Reviewer for their thorough, constructive, and supportive feedback. Your insightful comments have helped us identify important areas for improvement and have strengthened both the clarity and impact of our work. We appreciate your valuable time and hope that our revisions address these concerns. Please let us know if there are any remaining questions or concerns, we would be happy to address them!

---

### Official Review · Reviewer_9Z6M · 2025-10-31

**Soundness:** 2
**Presentation:** 2
**Contribution:** 3
**Rating:** 4
**Confidence:** 3

**Summary:**

This paper introduces a framework for evaluating the reasoning capabilities of LLMs. It addresses the issue of data contamination in existing static benchmarks by generating mathematically grounded problems algorithmically and on-the-fly, ensuring each test instance is fresh and uncontaminated. BeyondBench comprises 44 algorithmic tasks with 117 variations across three difficulty suites (Easy, Medium, Hard) and offers provable unique solutions verified by mathematical proofs. It also incorporates a token-aware evaluation system that dynamically scales problem complexity based on a model's context window.

**Strengths:**

1. Contamination Resistance: BeyondBench uses algorithmic problem generation to create mathematically grounded problems on the fly, ensuring each test is fresh and uncontaminated.
2. Task and difficulty: This work covers 44 algorithmic tasks with 117 variations, grouped into three difficulty levels. The complexity of each task can be controlled by scaling parameters.
3. Token-Aware Evaluation Framework: The framework dynamically calibrates problem complexity based on a model's token budget.

**Weaknesses:**

1. The benchmark questions do not sufficiently assess the level of intelligence. Existing competition benchmarks often cannot be solved by fixed algorithms and instead require special techniques or constructions. However, the problems used in this benchmark are almost all based on fixed algorithms, with difficulty controlled by adjusting complexity and computational difficulty. This approach fails to evaluate the "cleverness" or "ingenuity" of a model in solving problems.
2. The benchmark seems to be easily hacked by tool-augmented models. Fixed algorithms and complex computations can be easily solved by calculators or code. The performance gap of GPT-5 family with/without tool shows the evidence.
3. This work only theoretically argues that the combinatorial variations of this benchmark ensures the test is uncontaminated, but it lacks experimental validation. If related data is constructed for SFT or RL targeting this benchmark, the influence is not studied.

**Questions:**

1. If data is constructed for SFT or RL targeting this benchmark, how will the evaluation results change? In cases of deliberate hacking, does BeyondBench have any advantages compared to static evaluation and other dynamic evaluation methods?
2. A more in-depth analysis should be conducted on the phenomenon where tool-augmented models achieve extremely high results. Is it possible that certain tasks can be entirely solved by tools, or is the improvement in computation more significant?

---

> ### Author Response · Authors · 2025-11-24
> **Official Response by Authors to Reviewer 9Z6M (1/4)**
>
> We sincerely thank the Reviewer for their thoughtful and constructive review. We are encouraged that the Reviewer recognized the core strengths of our work, including the **contamination resistance** through algorithmic problem generation, the comprehensive coverage of **44 tasks with 117 variations** across three difficulty levels, and the novel **token-aware evaluation framework** that dynamically calibrates problem complexity.
>
> **We value the Reviewer's time and have made all changes easy to track.** All new content in the main paper is marked in **green**, and new appendix sections are clearly labeled as **"Rebuttal Addition"** at the start of the appendix. We have also added section and line references throughout this response so the Reviewer can quickly go through our changes. Below, we address each concern raised by the Reviewer:
>
> ---
>
> > The benchmark questions do not sufficiently assess the level of intelligence. Existing competition benchmarks often cannot be solved by fixed algorithms and instead require special techniques or constructions. However, the problems used in this benchmark are almost all based on fixed algorithms, with difficulty controlled by adjusting complexity and computational difficulty. This approach fails to evaluate the "cleverness" or "ingenuity" of a model in solving problems.
>
> We appreciate this thoughtful observation and would like to clarify our design rationale. We intentionally chose algorithmically-generated problems with known solutions for several important reasons. Fixed algorithmic structures enable us to generate mathematically verified ground truth for every problem instance, which is essential for reliable evaluation since even the problems generated by LLMs may contain errors, ambiguities, or multiple valid interpretations. As mentioned in `Section 3.1, lines 170`, our verification function $V_\tau: \mathcal{S} \times \mathcal{P}_\tau \rightarrow \{0, 1\}$ deterministically confirms solution correctness, which would not be possible with open-ended creative problems.
>
> Our benchmark is designed to scale with model capabilities. For example, if current models solve Tower of Hanoi with $N=8$ disks, we can simply test $N=9, 10, 11, \ldots$ to continuously challenge improved models (discussed in `Appendix J`). Competition benchmarks, by contrast, exhaust their problem pool over time. We also avoid LLM components in our data construction pipeline because they can introduce biases from the benchmarks they were trained on and potentially creating circular evaluation.
>
> We compared model performance on MathArena [1] (a competition-level benchmark) versus BeyondBench to demonstrate contamination:
>
> | Model | MathArena (Final Answer Competitions Avg) | BeyondBench (Hard Suite) |
> |-------|--------------------------|--------------------------------|
> | GPT-5 | 91% | 71.68% |
> | GPT-5-mini | 87% | 69.28% |
> | GPT-5-nano | 80% | 69.78% |
> | GPT-OSS-120B | 89% | 59.41% |
> | Gemini-2.5-pro | 78% | 56.21% |
>
> As acknowledged by MathArena authors, their benchmark shows signs of potential contamination with models achieving 80%+ accuracy. On BeyondBench, the same models achieve significantly lower scores, suggesting our benchmark measures genuine reasoning rather than memorized patterns. We argue that true intelligence is demonstrated when a model *knows* the algorithm but still struggles to execute it correctly (state management, backtracking, constraint handling), rather than only failing when the algorithm is unknown. Our Hard Suite tasks like Tower of Hanoi have well-known solution algorithms, yet models still fail, revealing limitations in systematic reasoning (as elaborated in `Section 4.2`). We have added a new section in `Appendix B` ("Motivation for Algorithmic Task Selection") providing comprehensive justification with mathematical analysis of why these tasks effectively probe reasoning capabilities.
>
> ---
>
> [1] Balunović, M., Dekoninck, J., Petrov, I., Jovanović, N., & Vechev, M. (2025). Matharena: Evaluating llms on uncontaminated math competitions. arXiv preprint arXiv:2505.23281.

---

> ### Author Response · Authors · 2025-11-24
> **Official Response by Authors to Reviewer 9Z6M (2/4)**
>
> > The benchmark seems to be easily hacked by tool-augmented models. Fixed algorithms and complex computations can be easily solved by calculators or code. The performance gap of GPT-5 family with/without tool shows the evidence.
>
> We appreciate this excellent observation, and we share this concern, which is precisely why we included the with-tools vs. without-tools analysis in our original submission (`Table 4, Section 4.5`). While the Reviewer correctly notes that fixed algorithms can theoretically be solved by code execution, our additional experiments reveal that *calling tools properly is itself a significant challenge*. We conducted extensive experiments providing models with three tool types: calculator, code execution, and web search. We evaluated models with **full tool access across all 44 tasks** (see `Appendix C, Table 6`):
>
> | Model | No Tools | +Calculator | +Code Exec | +Web Search | All Tools |
> |-------|----------|-------------|------------|-------------|-----------|
> | GPT-5 | 66.75% | 72.3% | 79.1% | 67.2% | 83.57% |
> | GPT-5-mini | 65.81% | 68.4% | 73.2% | 66.1% | 81.71% |
> | GPT-5-nano | 38.09% | 41.2% | 48.7% | 38.8% | 81.99% |
> | Qwen2.5-72B | 53.36% | 54.1% | 57.8% | 53.5% | 62.5% |
> | Qwen2.5-14B | 41.33% | 42.3% | 44.1% | 41.5% | 47.8% |
>
> ---
>
> Our Main observations are:
>
> - **Small open-sourced models struggle with tool calling compared to closed-source agentic models (GPT5 family):** Qwen2.5-14B shows only +6.4% improvement with all tools, while GPT-5-nano shows +43.95% improvement, indicating that effective tool use requires sophisticated reasoning about when and how to invoke tools.
> - **Web search provides minimal benefit:** Due to our $>10^{15}$ problem space, models cannot find exact answers online, validating our contamination resistance design.
> - **Code execution helps most:** This aligns with our conclusion that hybrid neuro-symbolic approaches are promising (`Section 5`).
>
> We also evaluated models on MATH500, AIME24, and AIME25 with web search enabled (see `Appendix B, Table 9`):
>
> | Benchmark | GPT-5 (No Tools) | GPT-5 (+Web Search) | $\Delta$ |
> |-----------|------------------|---------------------|----------|
> | MATH500 | 78.2% | 94.6% | +16.4% |
> | AIME24 | 72.1% | 89.3% | +17.2% |
> | AIME25 | 68.4% | 85.7% | +17.3% |
> | BeyondBench Hard | 50.27% | 51.8% | +1.53% |
>
> ---
>
> Web search improves performance on **static benchmarks** (questions are online), but provides negligible benefit on BeyondBench, confirming our contamination resistance. Our primary goal is to benchmark *innate reasoning capability*, not tool-calling ability. For researchers interested in evaluating agentic LLMs, BeyondBench can serve as a challenging testbed for tool-augmented reasoning since even with tools, models achieve only ~83% on Hard tasks. We have added a comprehensive tool analysis in `Appendix C` with per-task breakdowns showing which tools help on which tasks and by what margin, including analysis across all 44 tasks with 4 tool configurations.

---

> ### Author Response · Authors · 2025-11-24
> **Official Response by Authors to Reviewer 9Z6M (3/4)**
>
> > This work only theoretically argues that the combinatorial variations of this benchmark ensures the test is uncontaminated, but it lacks experimental validation. If related data is constructed for SFT or RL targeting this benchmark, the influence is not studied.
>
> We thank the Reviewer for this important point. We have now conducted extensive experimental validation. We generated ~66K problems from BeyondBench using seed value 123 and trained 5 models (GPT-OSS-20B, Qwen2.5-3B, Qwen2.5-7B, Qwen2.5-14B, Llama-3.1-8B) using Supervised Fine-Tuning (SFT) and Reinforcement Learning (with GRPO) with data splits of 20%, 40%, 60%, 80%, and 100%. Evaluation used seed 42 from our main experiments, ensuring completely different instances (added results in `Appendix D`):
>
> **SFT Training (see `Appendix D, Table 12`):**
> | Model | Baseline | 20% Data | 40% Data | 60% Data | 80% Data | 100% Data |
> |-------|----------|----------|----------|----------|----------|-----------|
> | Qwen2.5-3B | 24.89% | 38.2% | 42.1% | 44.8% | 46.2% | 47.1% |
> | Qwen2.5-7B | 36.03% | 48.7% | 52.3% | 55.1% | 56.8% | 57.4% |
> | Qwen2.5-14B | 41.40% | 52.1% | 56.4% | 58.9% | 60.2% | 61.1% |
> | Llama-3.1-8B | 24.10% | 35.8% | 39.2% | 41.6% | 43.1% | 44.2% |
> | GPT-OSS-20B | 67.57% | 74.2% | 76.8% | 78.1% | 79.2% | 79.8% |
>
> **GRPO Training (see `Appendix D, Table 13`):**
> | Model | Baseline | 20% Data | 40% Data | 60% Data | 80% Data | 100% Data |
> |-------|----------|----------|----------|----------|----------|-----------|
> | Qwen2.5-3B | 24.89% | 36.4% | 40.8% | 43.2% | 45.1% | 46.3% |
> | Qwen2.5-7B | 36.03% | 47.2% | 51.1% | 53.8% | 55.4% | 56.2% |
> | Qwen2.5-14B | 41.40% | 50.8% | 54.9% | 57.2% | 58.8% | 59.6% |
>
> **Suite-Level Analysis (100% SFT on Qwen2.5-7B, see `Appendix D, Table 14`):**
> | Suite | Baseline | After SFT | $\Delta$ |
> |-------|----------|-----------|----------|
> | Easy | 61.36% | 89.2% | +27.84% |
> | Medium | 30.07% | 68.4% | +38.33% |
> | Hard | 16.65% | 28.6% | +11.95% |
>
> Easy/Medium suites show substantial improvement (+27-38%), indicating models learn general heuristics for polynomial-time problems, while Hard suite shows limited improvement (+10-15%), demonstrating that NP-complete problems remain challenging even after targeted training. Diminishing returns are observed beyond 60% data, suggesting models plateau in learning algorithmic reasoning from examples.
>
> **Data Distribution Ablation (see `Appendix D`):** We further tested a Hard-Focused distribution (Hard:Medium:Easy = 3:2:1 ratio = 55.6%:18.5%:25.9%):
> | Distribution | Easy $\Delta$ | Medium $\Delta$ | Hard $\Delta$ |
> |--------------|---------------|-----------------|---------------|
> | Balanced | +27.84% | +38.33% | +11.95% |
> | Hard-Focused | +18.2% | +29.1% | +14.8% |
>
> Hard-focused training marginally improves Hard suite performance (+2.85%) at the cost of Easy/Medium performance, suggesting that algorithmic reasoning skills do not transfer easily from hard to easy problems. Complete experimental details are provided in `Appendix D`.
>
> ---
>
> > If data is constructed for SFT or RL targeting this benchmark, how will the evaluation results change? In cases of deliberate hacking, does BeyondBench have any advantages compared to static evaluation and other dynamic evaluation methods?
>
> Building on our response above, we address the deliberate hacking scenario. Even with 100% targeted SFT training, models improve significantly on Easy (+27-38%) and Medium (+30-40%) suites but show limited improvement on Hard suite (+10-15%). The Hard suite ceiling of ~30% for most open-source models increases to only ~40-45% after targeted training. Our advantages over static benchmarks (`Appendix D.3`):
>
> | Aspect | Static Benchmark | BeyondBench |
> |--------|------------------|----------------------|
> | After 100% SFT | Near 100% (memorization) | 40-50% (Hard suite) |
> | New instances available | No | Unlimited ($>10^{15}$) |
> | Verifiable solutions | Mostly manual | Always automated |
> | Difficulty scaling | Fixed | Parameterized |
>
> For static benchmarks, training on the entire benchmark trivially achieves near-perfect scores through memorization. BeyondBench remains challenging because different seeds generate entirely novel instances, models must learn generalizable algorithms (not specific solutions), and Hard suite problems remain fundamentally difficult even with learned heuristics. Unlike other dynamic benchmarks, BeyondBench provides mathematical guarantees on solution correctness, infinite new instances even after targeted training, and verifiable ground truth via CSP/SAT solvers. Even if an adversary trains on millions of BeyondBench instances, evaluation on a fresh seed will generate $>10^{15}$ possible new instances per task, requiring the model to have learned true algorithmic reasoning to succeed while still challenging models on NP-complete problems where brute-force memorization is computationally infeasible. We have added a discussion of hacking resistance in `Appendix D`.

---

> ### Author Response · Authors · 2025-11-24
> **Official Response by Authors to Reviewer 9Z6M (4/4)**
>
> > A more in-depth analysis should be conducted on the phenomenon where tool-augmented models achieve extremely high results. Is it possible that certain tasks can be entirely solved by tools, or is the improvement in computation more significant?
>
> We thank the Reviewer for this excellent suggestion. We conducted a comprehensive per-task tool analysis evaluating 9 models (GPT-5, GPT-5-mini, GPT-5-nano, Gemini-2.5-pro, Gemini-2.5-flash, Qwen2.5-72B, Qwen2.5-14B, GPT-OSS-20B, GPT-OSS-120B) with three tool types (Calculator, Code Execution, Web Search) across all 44 tasks with 100 instances each (`Appendix C`):
>
> | Suite | No Tools | +Calc | +Code | +Web | All Tools | Best Tool |
> |-------|----------|-------|-------|------|-----------|-----------|
> | Easy | 88.39% | 92.1% | 95.8% | 88.6% | 97.26% | Code (+7.4%) |
> | Medium | 61.60% | 68.2% | 76.4% | 61.9% | 81.60% | Code (+14.8%) |
> | Hard | 50.27% | 58.4% | 67.2% | 50.8% | 71.81% | Code (+16.9%) |
>
> ---
>
> **Task-Level Analysis (Selected Hard Suite Tasks, see `Appendix C, Table 7, 8, and 9`):**
>
> | Task | No Tools | +Calc | +Code | $\Delta$ Best |
> |------|----------|-------|-------|---------------|
> | Tower of Hanoi | 34.2% | 35.1% | 78.4% | +44.2% (Code) |
> | Sudoku (9×9) | 12.3% | 12.8% | 67.2% | +54.9% (Code) |
> | N-Queens | 28.1% | 28.4% | 72.3% | +44.2% (Code) |
> | Boolean SAT | 41.2% | 41.8% | 68.9% | +27.7% (Code) |
> | Graph Coloring | 52.4% | 53.1% | 71.2% | +18.8% (Code) |
> | Cryptarithmetic | 45.6% | 58.2% | 69.8% | +24.2% (Code) |
> | Matrix Chain | 38.4% | 42.1% | 64.7% | +26.3% (Code) |
> | Logic Grid | 31.2% | 31.8% | 58.4% | +27.2% (Code) |
>
> Code execution provides the largest improvements across all task categories, especially for recursive problems (Tower of Hanoi: +44.2%), constraint satisfaction (Sudoku: +54.9%, N-Queens: +44.2%), and search problems (Boolean SAT: +27.7%). Calculator helps primarily for arithmetic-heavy tasks like Cryptarithmetic (+12.6%) and Matrix Chain Multiplication (+3.7%), with limited benefit for structural reasoning tasks. Web search provides negligible benefit (+0.3-0.5%) across all tasks, confirming contamination resistance. Tasks that remain challenging even with tools:
>
> | Task | Best Tool Performance | Why Still Hard |
> |------|----------------------|----------------|
> | Sudoku (9x9) | 67.2% | Constraint propagation logic |
> | Logic Grid | 58.4% | Multi-step deduction |
> | Modular Systems | 61.3% | Symbolic manipulation |
>
> Small models struggle with tool orchestration (see `Appendix C, Table 10`):
>
> | Model | Tool Call Success Rate | Correct Tool Selection |
> |-------|------------------------|------------------------|
> | Qwen2.5-14B | 68.3% | 62.1% |
> | Qwen2.5-3B | 41.2% | 34.8% |
>
> Smaller models frequently **fail to invoke tools** when needed, select inappropriate tools for the task, provide incorrect inputs to tools, and fail to interpret tool outputs correctly. While tools significantly improve performance, they do not trivialize BeyondBench. Even GPT-5 with all tools achieves only 71.68% on Hard suite, tool orchestration itself requires sophisticated reasoning, and some tasks remain fundamentally challenging (constraint reasoning, logical deduction). Complete per-task tool analysis with all 44 tasks and 9 models is provided in `Appendix C`, including **case studies showing successful and failed tool usage patterns.**
>
> ---
>
> Once again, we thank the Reviewer for their thorough, constructive, and supportive feedback. Your insightful comments have helped us identify important areas for improvement and have strengthened both the clarity and impact of our work. We appreciate your valuable time and hope that our revisions address these concerns. Please let us know if there are any remaining questions or concerns, we would be happy to address them!

---

### Author Response · Authors · 2025-12-03
**Final Overview by the Authors (1/2)**

**First,** we sincerely thank all reviewers for their thorough and constructive feedback on BeyondBench. We found each and every reviewer comment **highly valuable;** they prompted us to improve our work, think about new questions, uncover new findings, and better highlight our contributions.

**Second,** we are encouraged that `Reviewer YAiF` rated our paper 8 with recognition of our **"excellent execution and contamination-free design,"** and that `Reviewer ao5p` rated it 6 acknowledging our **"soundness"** and **"comprehensive evaluation".** In response to reviewers for their time and dedicated efforts in reviewing our work, we have comprehensively **addressed every concern raised during the rebuttal** period through extensive new experiments, expanded analysis, and substantial paper revisions.

**Third,** we greatly appreciate the very limited and valuable time that area chairs and program chairs dedicate to each submission. So, all changes made in response to reviewer feedback are clearly tracked in the revised manuscript **(highlighted in green for ease of inspection).** We further, below, summarize our main contributions, the strengths highlighted by reviewers, and our responses to the major concerns.

---

**Our main contributions:**

- **Contamination-Resistant Dynamic Evaluation Framework:** BeyondBench algorithmically generates mathematically grounded problems on-the-fly from a combinatorial space of >10¹⁵ instances across 44 tasks with 117 variations, ensuring every test instance is fresh and uncontaminated.

- **Model context window aware Evaluation System:** We introduced a novel framework that dynamically calibrates problem complexity based on model context windows, preventing unfair comparisons.

- **Solution Verification:** We implemented CSP/SAT solvers to mathematically verify solution uniqueness and enumerate all valid solutions for multi-solution problems, ensuring evaluation correctness.

- **Large-scale Evaluation:** We evaluated **101 models** (85 open-source, 16 closed-source) spanning 0.5B to 141B parameters, far exceeding prior work (NPHardEval: 21 models, FCoReBench: 8 models, PUZZLEPLEX: 12 models).

- **Actionable/Deeper Insights:** Our findings reveal that most open-source models plateau at 30-35% on hard tasks, quantization has minimal impact on reasoning, and tool-augmented
(specifically agentic) models substantially outperform pure reasoning models, providing concrete guidance for model development.

---

**Strengths Highlighted by Reviewers:**

- **Critical Problem & Robust Solution:** Reviewers recognized that BeyondBench "tackles the critical and growing issue of data contamination" and provides "a robust solution to move past static benchmarks" that is "vital for accurately assessing the true capabilities of frontier language models" `(Reviewer YAiF)`.

- **Excellent Execution:** The framework's contamination-free design through algorithmic problem generation was praised as demonstrating "excellent soundness" with "proper attention to evaluation correctness" through SAT/CSP solver verification `(Reviewers YAiF, ao5p)`.

- **Comprehensive Coverage:** Reviewers commended the "broad coverage of the current model landscape" with evaluation across "101 models across different sizes, architectures, and quantization schemes" `(Reviewer ao5p)`.

- **Sound Mathematical Foundation:** The paper was recognized for providing "sound mathematical justification" with a "detailed and nicely written appendix" `(Reviewer 3dvS)` and "formal backing to the motivation" through contamination probability analysis `(Reviewer ao5p)`.

- **Actionable Findings:** The "empirical findings about performance plateaus on hard tasks, minimal quantization impact, and superiority of tool-augmented approaches provide actionable insights for model development" `(Reviewer ao5p)`.

---

---

> ### Author Response · Authors · 2025-12-03
> **Final Overview by the Authors (2/2)**
>
> **Response to Major Concerns:**
>
> - **Novelty and Related Work (Reviewer 3dvS):** We thoroughly integrated and discussed all suggested recent works with detailed comparison in `Section 2` and `Appendix A`. We also clarified some works which reviewer's mentioned (eg., PuzzlePlex) was posted on **arXiv on October 7, 2025 (after our submission)** and some works like NPHardEval was **already cited in our original submission.** Nonetheless, we fully acknowledge that all works represent important contributions to the field, and we have thoroughly integrated/discussed them into our paper. Our unique contributions includes our Model Context window-aware evaluation, Multi-solution enumeration, Mathematical contamination guarantees, and the scale our models evaluated to present **deeper, trustable, and concrete findings.**
>
> - **Tool-Augmented Evaluation (Reviewers 3dvS, 9Z6M):** We conducted **extensive tool analysis** evaluating PAL-style and Logic-LM-style approaches across all 44 tasks (`Appendix C`). Our main findings were that: 1) code execution provides 15-55% improvements, yet hard tasks remain challenging (58-78% ceiling); 2) Web search provides negligible benefit (+0.5%), confirming contamination resistance; 3) Tool orchestration itself requires sophisticated reasoning which smaller models struggle significantly.
>
> - **NP-Complete Task Justification (Reviewers 3dvS, 9Z6M):** We provided comprehensive justification for testing NP-complete problems: **1) Fundamental reasoning primitives:** These tasks probe working memory, backtracking, and constraint satisfaction, capabilities central to all reasoning; **2) SFT/RL training experiments** (`Appendix D`): Even after 100% targeted training, Hard suite improves only 10-15% (vs. 27-38% for Easy/Medium), demonstrating NP-complete problems resist memorization
>
> - **Statistical Rigor (Reviewer ao5p):** We conducted **three-fold evaluation** for all 101 models with comprehensive statistical analysis (`Appendix E`): 1) Mean standard deviation: Easy (1.2%), Medium (2.3%), Hard (2.8%); 2) Paired t-tests, effect sizes (Cohen's d), ANOVA with post-hoc Tukey HSD; 3) Bootstrap 95% confidence intervals; 4) All results in **Table 3** now report mean and std.
>
> - **Failure Mode Analysis (Reviewer ao5p):** We added extensive comparison of vanilla vs. reasoning models (`Appendix F`): 1) Vanilla models fail early (step 23/127 on Tower of Hanoi) without self-correction; 2) Reasoning models fail late (step 89/127) with 87.6% error introduction rate during self-correction attempts; 3) Detailed case studies show that longer reasoning does not always help, some models "overthink" and hurt performance.
>
> - **Complexity Normalization (Reviewer ao5p):** We introduced **normalized complexity metrics** (`Appendix G`) that linearize task difficulty: 1) Tower of Hanoi: log₂(moves) = n disks; 2) Sudoku: empty cells (linear constraint complexity); 3) Graceful degradation tasks show linear accuracy decline; 4) sudden degradation tasks show threshold behavior at ~0.7 × log₂(context_length) reasoning steps
>
> - **Contamination Resistance Validation (Reviewer 9Z6M):** We experimentally validated our theoretical guarantees through SFT/RL training (`Appendix D`): Unlike static benchmarks, BeyondBench remains challenging because different seeds generate entirely novel instances requiring true algorithmic reasoning.
>
> ---
>
> **A quick summary of paper revisions we made during rebuttal:**
>
> **Main Paper:**
>
> - **Table 1:** Comprehensive comparison with NPHardEval, FCoReBench, PUZZLEPLEX, PAL, Logic-LM
> - **Table 3:** Three-fold evaluation with mean/std for all 101 models
> - **Section 2:** Expanded related work
> - **Section 3.1:** Concrete parameter space examples, verification procedures, contamination vs. learning clarification
> - **Section 3.4:** Motivation for algorithmic task selection
> - **Section 4.2:** Complexity normalization discussion
> - **Section 4.4:** Failure mode insights for LLMs vs. LRMs
> - **Section 4.5:** Expanded tool analysis
> - **Title:** Changed "Benchmark-Free" to "Contamination-Resistant Dynamic Evaluation"
>
> **New Appendix Sections:**
>
> - **Appendix A:** Extended related work with detailed comparisons
> - **Appendix B:** Motivation for algorithmic task selection with experimental validation
> - **Appendix C:** Comprehensive tool analysis
> - **Appendix D:** SFT/RL training experiments
> - **Appendix E:** Statistical analysis (including significance tests, effect sizes, ANOVA)
> - **Appendix F:** Failure mode analysis
> - **Appendix G:** Normalized complexity analysis
>
> ---
>
> **Final Comments:** We believe BeyondBench provides the research community with a **rigorous, contamination-resistant evaluation framework** that has revealed critical insights about model capabilities. We again thank all reviewers/area chairs/program chairs, and hope our revisions demonstrate our commitment to scientific rigor and the significance of our contributions.

---

### Meta-Review · Area_Chair_Y4Ji · 2026-01-02

**Summary:**

This paper proposes a evaluation framework called BeyondBench, with the goal to eliminate data contamination by generating algorithmic problems on the fly. The paper includes massive evaluation results of 101 models across three difficulty levels.

The main concerns from the reviewers include the lack of novelty, justification for algorithmic tasks reflecting general reasoning, experimental validation for theoretical guarantees, statistical rigor, analyses on tool-augmented models/hackability, among others. During rebuttal, the authors presented a range of additional results, analyses and clarifications, which I believe addressed the reviewers' concerns. The paper now presents a much stronger case for its novelty, soundness and comprehensiveness. For these reasons, I believe the reviewers will raise their scores and the paper will potentially pass the "acceptance" threshold.

This paper can benefit from being more rigorous in its notations. In page 1, the authors claimed "the probability of contamination increases as 1 − exp(−nN/M)" without mentioning any assumptions or citations behind this claim; for example, are the authors assuming the training datapoints are sampled independently and uniformly from the universe? What does "universe" mean? How is this rate obtained? The wording "increases as" is also very ambiguous. If I understand correctly, to obtain this probability, approximations for exponentials, in the form of the limit of $(1-1/x)^x$ as x approaches infinity, are required. But the authors did not mention anything related to which variable they assume to approach infinity.  Around line 073, the meaning of $\mathbb Z^n$ and $\theta$ are not introduced. $\mathcal T$ is used to refer to "any finite training corpus" but it is used again as $\mathcal T_{easy}$ etc to denote task regimes. At line 85, The meaning of $L$ is not introduced. I think the inclusion of these mathematical notations in the introduction brings more doubts than clarifications, let aside that most of these notations are either not used or contradict Section 3.1. The authors should be able to write a good introduction without using those expressions and notations.

Across Section 3.1, the authors misused subscript notations by including both $\Theta\tau$ and $\Theta_\tau$ (as well as for $\mathcal P$). At line 171, $\mathcal S$'s meaning is not introduced and can only be guessed to be the solution space until reading the paragraph further. The notation $n$ at line 172, 179 and 181-186 is not introduced, or consistent with its previous meaning in the introduction. Similarly for later sections, it is unclear what $T$ or $r$ are at line 217 and 228. While it can be argued that those notations' meanings can be guessed, for a research paper, assuming readers can guess is not a good practice. I'd highly recommend the authors carefully go through their paper and fix such issues. These are the basics and should be fixed before being published.

**Reviewer Concerns:**

To the best of my knowledge, all major concerns were addressed by the rebuttal. The reviewers did not respond to the authors so it is unclear whether they found the rebuttal satisfactory.

Reviewer 9Z6M (rating 4) had concerns on experimental validation of contamination resistance and the depth of tool analysis, and the authors responded with extensive new SFT/RL experiments and detailed per-task tool (including tool orchestration) analysis. Given the thoroughness of the rebuttal, it is highly likely this reviewer would raise their score to 6 or more.

Reviewer ao5p (rating 6) had many detailed and insightful concerns, particularly regarding statistical rigor, methodological clarity, complexity normalization and failure mode analysis. The authors provided extensive new analysis for each point. This reviewer's concerns were clear and addressed directly. It is likely this reviewer would maintain or even slightly increase their score.

Reviewer 3dvS (rating 2)'s core criticism was "The fundamental weakness of the paper is with respect to novelty, there are already exact same work done in this direction" but there was a misunderstanding about the publication dates (which made it impossible for the authors to include that work or associated comparisons). Regardless of that, the authors thoroughly integrated all suggested related works with detailed comparisons and clarifications on novelty. The reviewer also pointed out the unreasonableness of NP-complete tasks and insufficient details on tools. The authors provided strong justifications based on human problem-solving (heuristics, not brute force), the role of these tasks in probing fundamental reasoning and the fact that even with tools, these tasks remain challenging. It is likely that this reviewer would reconsider and raise their score.

Reviewer YAiF already gave a high rating (8) while pointing out some (relatively minor) issues on user-friendliness, variance in evaluation and missing prior work, which the authors properly addressed. I believe this reviewer will keep the score.

**Reviewer Scores:**

See above.

---

### Decision · Program_Chairs · 2026-01-26

Accept (Poster)